

# Cirque-like alcoves in the northern mid-latitudes of Mars as evidence of glacial erosion

An Y. Li[1], Michelle R. Koutnik[1], Stephen Brough[2], Matteo Spagnolo[3], Iestyn Barr[4]

[1]Department of Earth and Space Sciences and Astrobiology Program, University of Washington, Seattle, 98195, USA
[2]Department of Geography and Planning, School of Environmental Sciences, University of Liverpool, Liverpool, L69 7ZT, UK
[3]School of Geosciences, University of Aberdeen, Aberdeen, AB243UF, UK
[4]Department of Natural Sciences, Manchester Metropolitan University, Manchester, M15 6BH UK, UK

*Correspondence to*: An Y. Li (anli7@uw.edu)

**Abstract.** While glacial remnants in the form of viscous flow features in the mid-latitudes of Mars are considered to be cold-based in the present-day, an increasing amount of geomorphic evidence suggests that at least some flow features were previously wet-based or had a mixed thermal state (polythermal) at during their evolution. Many of the viscous flow features known as glacier-like forms have been observed to emerge from alcoves that appear similar to cirques on Earth. Terrestrial cirques are typically characterized by a concave basin connected to a steep backwall. Cirques are expected to form from depressions in mountainsides that fill with snow/ice and over time support active glaciers that deepen the depressions by wet-based glacial erosion. To assess which alcoves on Mars are most "cirque-like", we mapped a population of ~2000 alcoves in Deuteronilus Mensae, a region in the mid-latitudes of Mars characterized by mesas encompassed by glacial remnants. Based on visual characteristics and morphometrics, we refined our dataset to 386 "cirque-like alcoves", which is five times the amount of glacier-like forms in the region, and used this to assess the past extent and style of glaciation on Mars. Using high resolution imagery, we find geomorphic evidence for glacial occupation associated with the cirque-like alcoves, including crevasse-like features, surface lineations, polygonal terrain, and moraine-like ridges. We propose that the cirque-like alcoves with icy remnants similar to rock glaciers on Earth represent a late stage of glacier-like form evolution. We also outline stages of cirque-like alcove evolution, linking a potential early stage of cirque-like alcoves to gully activity. On a population-wide scale, the cirque-like alcoves have a south to southeastward aspect bias, which may indicate a requirement for increased insolation for melting to occur and a connection to gullies on Mars. While the alcoves also have similarities to other features such as landslide scarps and amphitheater-headed valleys, the cirque-like alcoves have unique morphologies and morphometrics that differentiate their origin. Assuming warm-based erosion rates, the cirque-like alcoves have timescales consistent with both glacier-like forms and other viscous flow features like lobate debris aprons, whereas cold-based erosion rates would only allow the older timescales of lobate debris aprons. We propose that based on the geomorphic features and southward aspect, cirque-like alcove formation is more consistent with warm-based glaciation.





## 1 Introduction

The surface morphology of the mid-latitudes of Mars (especially between 30 and 60°, north and south) is characterized by glacial remnants in the form of subsurface ice, debris-covered ice (Fig. 1) and icy mantling deposits. Extending from the cold and dry conditions of present-day Mars, the climate of the past 3 Gyr (Amazonian Epoch; Michael, 2013) is presumed to have led to limited to no liquid water on the planet's surface (e.g., Kite, 2019). Glacial remnants in the mid-latitudes of Mars, referred to as viscous flow features, are typically considered to been frozen to their beds with limited subglacial erosion (cold-based) and ice flow only by internal deformation and gravity-driven viscous creep throughout their evolution (e.g., Mangold and Allemand, 2001; Head and Marchant, 2003; Shean et al., 2005). However, the presence of glacial landforms such as moraines and lineations observed in tandem with at least select viscous flow features suggests subglacial erosion could have occurred and that the ice-flow regime could have been formerly wet-based or at least a mixed thermal state as polythermal (e.g., Arfstrom and Hartmann, 2005; Morgan et al., 2009; Hubbard et al., 2011; Hubbard et al., 2014). In addition, recent work proposed that some depositional and erosional evidence of wet-based glaciation within the last 1 Gyr (Middle to Late Amazonian) exists, especially in the form of eskers, which would indicate warmer subglacial conditions at these sites (e.g., Gallagher and Balme, 2015; Butcher et al., 2017; Butcher et al., 2021; Gallagher et al., 2021, Woodley et al., 2022). Eskers are ridges left behind by ice retreat, where prior to ice retreat the sediment forming the ridge was deposited by meltwater flowing through subglacial or englacial tunnels. In some cases, the evidence for eskers is found in association with certain types of viscous flow features across the mid-latitudes of Mars (Butcher et al., 2017).

Viscous flow features include the landform classifications of glacier-like forms (Souness et al., 2012), lobate debris aprons, lineated valley fill, and concentric crater fill (e.g., Squyres, 1979; Milliken et al., 2003; Levy et al., 2014). In the cases where it can be observed, lobate debris aprons consist of up to ~90% of ice (Holt et al., 2008; Plaut et al., 2009), and they account for ~63% of the total volume of ice contained within all viscous flow features (Levy et al., 2014). Lobate debris aprons can be a few to tens of kilometers long and up to one kilometer thick (Holt et al., 2008; Plaut et al., 2009). In comparison, glacier-like forms are smaller, on average ~4.66 km long, ~1.27 km wide (Souness et al., 2012), and ~130 m thick (Brough et al., 2019). All viscous flow features are believed to have been deposited during Amazonian orbital and axial excursions (Madeleine et al., 2009). Lobate debris aprons, lineated valley fills, and concentric crater fills are estimated to range from ~10 Myr to 1.2 Gyr in age (Morgan et al., 2009; Berman et al., 2015), whereas glacier-like forms can superpose lobate debris aprons or lineated valley fills, indicating polyphase glaciation with age clusters estimated to be around 2-20 Myr and 45-65 Myr (Hepburn et al., 2020). In addition to viscous flow features, there is a separate icy mantling deposit over the mid-latitudes originating from airfall deposits of ice nucleated on dust, known as the latitude-dependent mantle (e.g., Mustard et al., 2001; Kreslavsky and Head, 2002; Schon et al., 2009; Conway et al., 2018). The latitude-dependent mantle consists of different layers rich in water ice and dust that were deposited during different variations in climate (Schon et al., 2009). Although the mantling unit covers >23% of the surface of Mars (Kreslavsky and Head, 2002), with estimates ranging from 1-30 m thick (Mustard et al., 2001; Conway and Balme, 2014), the mantling unit represents a small contribution ($10^3$-$10^4$ km$^3$) to the overall ice volume (Conway and Balme, 2014). In some locations the mantling unit has been mapped as "pasted-on terrain", though it remains unclear whether the pasted-on terrain is a thicker mantling layer that is separate from an overlying mantle (Conway et al., 2018). The icy mantling unit is estimated to be younger than glacier-like forms at 0.15 to 10 Myr in age (Mustard et al., 2001; Conway et al., 2018).

69



70

**Figure 1: Legend in the bottom left applies to panels (a) and (c). (a) Standalone mesa in Deuteronilus Mensae, centered at 26.3°E, 45.5°N. Black filled polygons are alcoves mapped as part of this study, yellow filled polygons are previously mapped glacier-like forms (Brough et al., 2019), blue represents the previously mapped lobate debris apron (Levy et al., 2014), and pink represents an updated map of lobate debris apron (Baker and Head, 2015). Note that there is some overlap between what previous studies generally classified as lobate debris apron and Brough et al. (2019) later specifically defined as glacier-like forms. (b) The same mesa as in (a) but without mapped units delineated. The basemap is the CTX mosaic (Dickson et al., 2023a) overlaid on a High Resolution Stereo Camera (HRSC) (Neukum et al., 2004) digital elevation model (DEM) that was mosaicked from 29 frames. (c) A zoomed out view of alcoves mapped in this study. The area is centered at 34.0°E, 41.5°N. Green filled polygons represent lineated valley fill mapped by Levy et**



**al. (2014). (d) Same area as in (c) but without mapped units delineated. CTX data credit: Caltech/NASA/JPL/MSSS.**
**HRSC data credit: ESA/DLR/FU Berlin.**
Since glacier-like forms extend out of cirque-like alcoves (Fig. 1), it has been hypothesized that these alcoves may be
analogous to glacial cirques on Earth (Hubbard et al., 2014; Fig. 2). Herein, we use the term "alcove" loosely to describe any
hollow with an arcuate headwall and opening downslope, on the scale of hundreds of meters to a few kilometers in width and
length. From terrestrial studies we know that cirques are formed by glacial erosion, which generally requires liquid water at
the base of a wet-based/warm-based glacier (Glasser and Bennett, 2004). On the other hand, cold-based glaciers are minimally
erosive (Table 1) and are therefore not typically associated with large glacial erosion features such as glacial valleys, troughs
or cirques. Glacial cirques on Earth are characterized by a concave basin connected to a steep headwall, often with a threshold
or lip of higher topography at the lower end of the basin (Fig. 2). They develop from incipient depressions in mountain and
plateau sides that fill with snow/ice and over time support active, wet-based glaciers that deepen the depressions by glacial
erosion (Evans and Cox, 1974; Glasser and Bennett 2004). Due to their presence at high topographic locations on Earth and
due to their concave shape, cirques trap snow and ice and are often the first sites to glaciate and the last sites to deglaciate
(Graf, 1976). On Earth, with over 10,000 glacial cirques mapped globally, landform morphometrics are used to reveal regional
climatic trends and the extent of glaciation in the past (e.g., Mîndrescu et al., 2010; Evans, 2006; Barr and Spagnolo, 2015).
Putative cirques on Mars have been identified in the mid-latitudes (Gallagher et al., 2021) and equatorial regions
(Davila et al., 2013; Bouquety, et al., 2019; Williams et al., 2023). While the cirque-like alcoves in areas such as Deuteronilus
Mensae have been interpreted as potentially connected to past glaciation (e.g., Head et al., 2006; Morgan et al., 2009, Hubbard
et al., 2011; Souness and Hubbard, 2013), there have not been any in-depth studies dedicated to these cirque-like alcoves on a
population scale. If these martian alcoves are analogous to terrestrial glacial cirques, then they may have formed either during
an earlier wet-based phase of glacier-like form activity, or formed during a prior glacial cycle separate from the glacier-like
forms. In this study, we mapped ~2000 alcoves in Deuteronilus Mensae in the northern mid-latitudes of Mars and conducted
a morphometric geomorphological analysis to determine which of these are most likely glacial cirques.

**Table 1: Comparison of published erosion rates for cold-based glaciers, wet-based glaciers, and glacial cirques**
**(wet-based) on Earth.**

| Type of glacier | Erosion rate (m/Myr) | Reference |
|---|---|---|
| Cold-based and debris-covered on Mars | 0.1–10 | Levy et al., (2016) |
| Cold-based on Earth | 0.2–3 | Balco and Shuster, (2009); Cuffey, (1999a) |
| Wet-based on Earth | 10–10,000 | Hallet et al., (1996) |
| Cirque (wet-based) on Earth | 8–5,900 | Reviewed by Barr and Spagnolo, (2015) |
















**Figure 2: (a)(i) Example of a cirque-like alcove on Mars (40.24°N, 34.48°E) (CTX mosaic; Dickson et al., 2023a) (a)(ii) a cirque on Earth in the Uinta Mountains (40.71°N, 110.11°E). (b)-(d) Examples of cirques on Earth incised into mesa topography, along with an example of a cirque profile in each. Part (i) of (b)-(d) provides an overview of the cirques in that location with an inset of the location of part (ii). Part (ii) of (b)-(d) offers a zoomed-in view of an individual cirque. Part (iii) of (b)-(d) shows the profile of the individual cirques in part (ii). (b) Kamchatka Peninsula, Russa (58.48°N, 160.70°E). (c) Uinta Mountains, Utah, USA (40.74°N, 110.05°W); (d) Transantarctic Mountains, Antarctica (80.01°S, 156.35°E). CTX data credit: Caltech/NASA/JPL/MSSS. Earth imagery is from © Google Earth including Landsat/Copernicus/U.S. Geological Survey coverage. North is toward the top of the page, unless otherwise indicated.**



## 2 Study Area: Deuteronilus Mensae

Our study region covers ~600,000 km² of Deuteronilus Mensae in the northern mid-latitudes of Mars (40-48°N, 16-35°E) (Fig. 3). While we also observe alcoves in other regions in the mid-latitudes of Mars, we focus on Deuteronilus Mensae as a study region for identifying cirque candidates due to its high density of icy viscous flow features (e.g., Levy et al., 2014; Baker and Carter, 2019; Brough et al., 2019). In addition to lobate debris aprons observed by the Mars Reconnaissance Orbiter (MRO) SHAllow RADar (SHARAD) instrument (e.g., Plaut et al., 2009; Baker and Carter, 2017), recently, the Mars Subsurface Water Ice Mapping (SWIM) project identified Deuteronilus Mensae as one of the candidates with the most shallow subsurface ice, thus making it a location of interest for future human missions to Mars (e.g., Morgan et al., 2021). Deuteronilus Mensae is characterized by fretted mesa terrain of disputed origin encompassed by remnants from previous glaciations (Sharp, 1973; Squyres, 1978; Carr, 2001; Morgan et al., 2009). The geologic history of Mars is divided into three main epochs: the Noachian around 4.0 to 3.85 Ga, the Hesperian around 3.56 to 3.24 Ga, and the Amazonian around 3.24 Ga to present-day (e.g., Hartmann, 2005; Michael, 2013; Kite, 2019). Previous geomorphic mapping estimated that the mesas date back to the ancient Noachian and the plains to the Hesperian, while younger Amazonian sedimentary deposits and a mantling unit overlay the mesas (e.g., Baker and Carter, 2019).

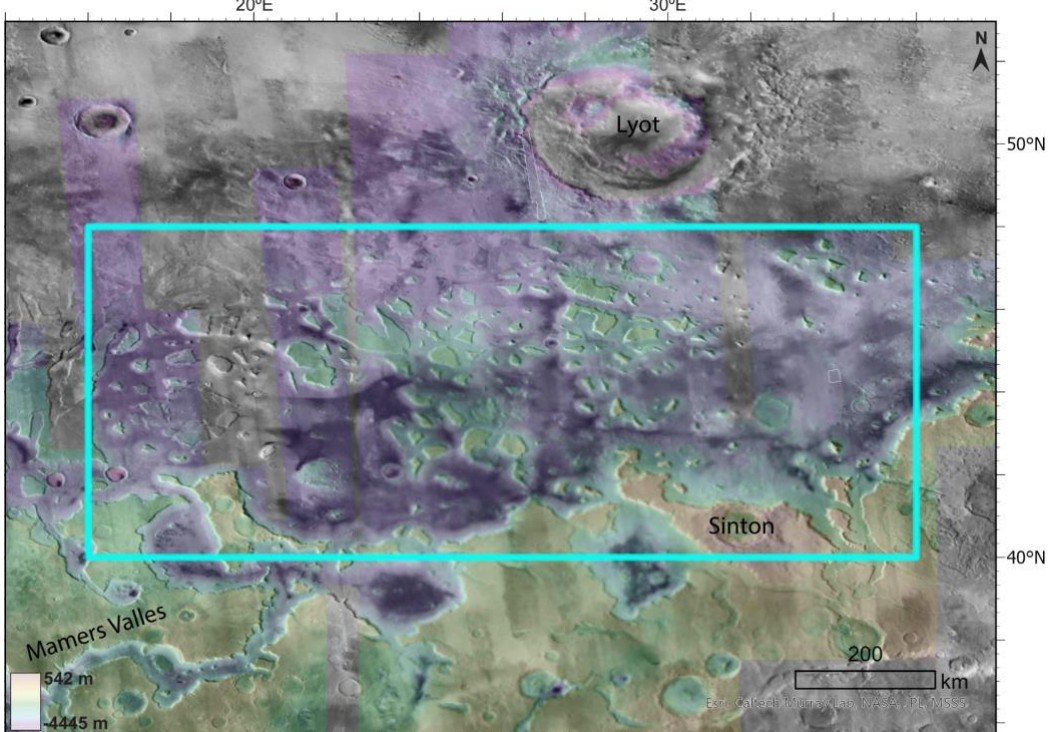

**Figure 3: The study region Deuteronilus Mensae in the northern mid-latitudes of Mars is within the teal box (16-35°E, 40-48°N). Colors represent HRSC DEM data where red corresponds to a maximum of 542 m and purple corresponds to a minimum of -4445 m in the study region. White rectangles show where the CTX mosaic did not have coverage and gray components show where the**



**mosaicked HRSC DEM did not have coverage. Major surface features including Lyot Crater, Sinton Crater, and Mamers Valles**
**are noted. CTX data credit: Caltech/NASA/JPL/MSSS. HRSC data credit: ESA/DLR/FU Berlin.**
**3 Methods and Data**
**3.1 Alcove Mapping**
We mapped ~2000 alcoves at a 1:30,000 scale using mosaicked ~6m/pixel Context Camera imagery (Malin et al.,
2007; Dickson et al., 2023a). We only map alcoves that do not contain previously mapped glacier-like forms or lobate debris
apron (Fig. 1). We digitized the outlines of the alcoves using ArcGIS software. For morphometric analyses, we used a 50-100
m/pixel High Resolution Stereo Camera (HRSC; Neukum et al., 2004) digital elevation model (DEM) of 29 frames mosaicked
together (see Data availability for exact frames). Where available, we used ~25 cm/pixel High Resolution Imaging Science
Experiment (HiRISE; McEwen et al., 2007) images to examine glacial geomorphic features within and next to the alcoves.

**3.2 Criteria for Identification of Cirque-like Alcoves**
**3.2.1 Seven Classes of Alcoves**
Cirques on Earth are categorized into five grades ranging from a "classic" cirque that contains "textbook" attributes
to a "marginal" cirque, where the cirque status is doubtful (Evans and Cox 1995). In addition, there are also numerous cirque
types including simple cirques, compound cirques, cirque complexes, staircase cirques, and cirque troughs (Benn and Evans
2010), which we drew upon to inform our preliminary alcove classes for our Deuteronilus Mensae analyses. Based on their
kilometer-scale physical characteristics including shape, size, and associated landforms, we classified our population of
mapped alcoves into seven broad classes: a) simple, b) joined, c) interiorly ridged, d) staircase, e) crater-like, f) channel-related,
g) branching (Fig. 4). Descriptions and interpretations of each class are provided in Table 2. Note that the joined and staircase
alcoves were mapped as one alcove, but due to their larger scale, branching alcoves offshooting from the same valley were
mapped as individual alcoves. As such, smaller simple alcoves that reside within the larger branching alcoves would fall into
both classes. Thus, ~4% of the alcoves were classified as two or more types. Both crater-like alcoves and channel-related
alcoves suggest that a different erosional mechanism other than glaciation may have dominated their formation. Although
terrestrial glacial cirques may also fall into different categories, for our study of martian alcoves that are considered most
analogous to terrestrial cirques, we focus on the alcoves classified as simple alcoves (and that do not belong to any other class).
By definition, simple alcoves have morphometrics consistent with simple cirques on Earth. Herein, we use the term "cirque-
like alcove" for these martian alcoves that are the most likely candidate cirques.



## (a) Simple

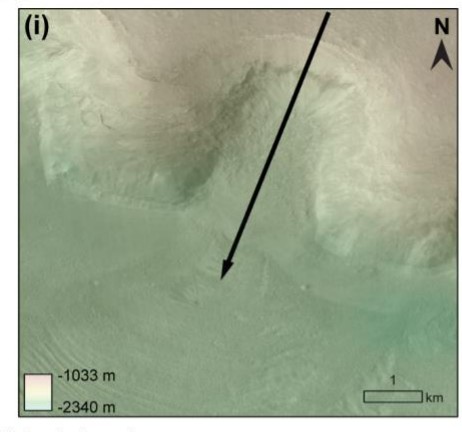

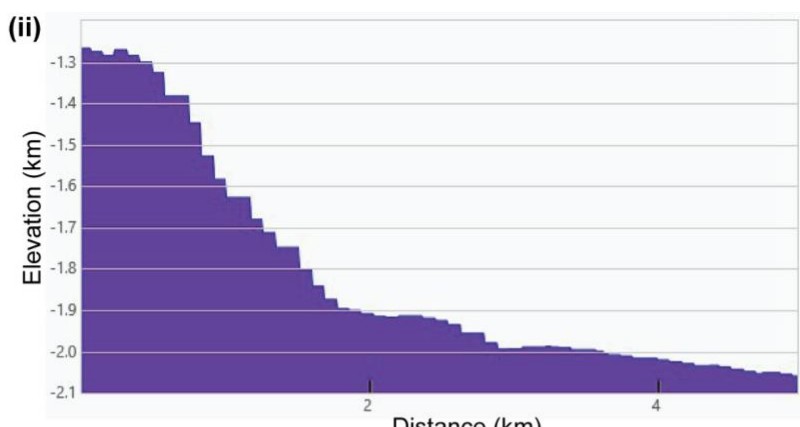

## (b) Joined

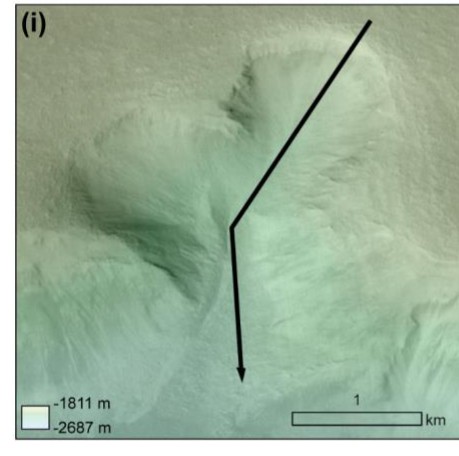

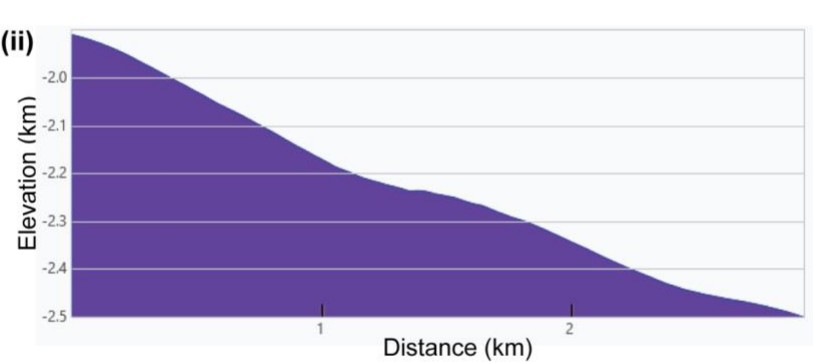

## (c) Interiorly ridged

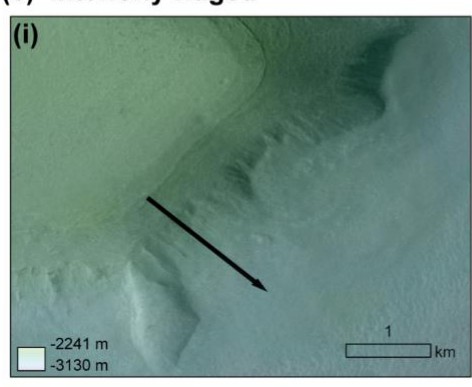

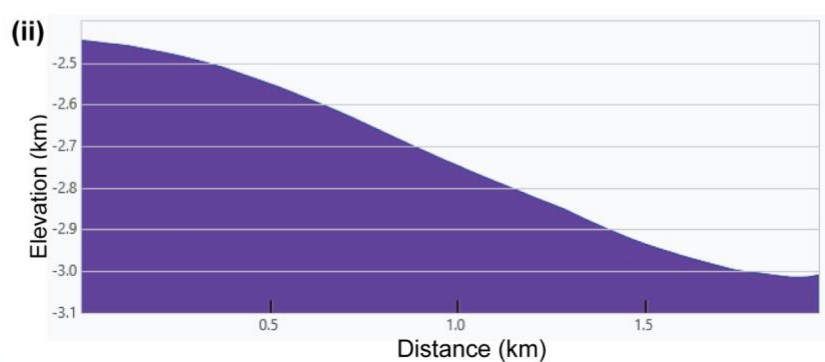




**(d) Staircase**

**(e) Crater-like**

**(f) Channel-related**

**(g) Branching**



**Figure 4: Our preliminary classification of these alcoves assigns seven classes. (a) Simple: characterized by its armchair shape, a defined headwall, two sidewalls, and an opening downslope (40.24°N, 34.48°E). (b) Joined: two simple alcoves adjacent to one another that join together downslope (37.72°N, 20.35°E). (b)(i) represents the profile on the left while (b)(ii) represents the profile on the right. (c) Interiorly ridged: An alcove that has ridges within it rather than a clean headwall (44.62°N, 24.95°E). (d) Staircase: A simple or crater-like alcove that has a step up to another simple or crater-like alcove alcove (37.62°N, 19.59°E). (e) Crater-like: very circular or semicircular with a clean-cut headwall with depth over diameter ratios ~0.1-0.3 and have a wide opening downslope (45.08°N, 21.59°E). (f) Channel-related: a channel is adjacent to if not feeding into this class of alcove (42.34°N, 18.30°E). (g) Branching: a large alcove that is much longer than it is wide with multiple offshoots of smaller simple alcoves (37.97°N, 19.58°E). (g)(i) represents the long profile while (g)(ii) represents the short profile. All images are using the CTX mosaic (Dickson et al., 2023a) overlaid on HRSC (Neukum et al., 2004) elevation data. HRSC values are included in the elevation profiles. Arrows all point downslope. CTX data credit: Caltech/NASA/JPL/MSSS. HRSC data credit: ESA/DLR/FU Berlin.**

**Table 2: Seven broad classes of alcoves identified in this study.**

| Feature classification | Description of feature on Mars | Number of alcoves with this classification (and that fit in multiple classes) | Percent of alcoves with only this classification | Evaluation |
|---|---|---|---|---|
| Simple alcove | Simple alcoves are characterized by an armchair shape with a defined headwall, two sidewalls, and are open downslope. | 1273 (58) | 63% | Morphologies of simple alcoves are most similar to simple cirques on Earth (e.g., Barr and Spagnolo, 2015). |
| Joined alcove | Joined alcoves consist of two adjacent simple alcoves that join together downslope. | 307 (4) | 15% | Joined alcoves are most similar to compound cirques on Earth since compound cirques have two simple cirques in the headwall (e.g., Barr and Spagnolo, 2015). |



| Interiorly ridged alcove | Interiorly ridged alcoves do not have a well-defined single headwall with an armchair shape, but instead contain ridges within the headwall. | 289 (16) | 14% | These alcoves may represent a prior stage of simple alcove formation and are discussed further in Section 5.2. |
|---|---|---|---|---|
| Crater-like alcove | Crater-like alcoves are circular, or at least semicircular, with a clean-cut headwall, and with a wide opening downslope. | 12 (20) | 0.6% | Craters are an important component of the geologic record of Mars (e.g., Michael, 2013, Li et al., 2022), and we use the term "crater-like" to acknowledge that this class of alcoves has retained morphologies of craters though do not have a complete crater rim. |
| Staircase alcove | Staircase alcoves include a simple or crater-like alcove that has a step up to another simple or crater-like alcove. | 34 (1) | 2% | Morphologies of staircase alcoves are most similar to staircase cirques on Earth (e.g., Barr and Spagnolo, 2015). |
| Channel-related alcove | Channel-related alcoves have channels near, or feeding into, the headwall of the alcove. | 7 (13) | 0.3% | Some of these channels are near impact craters and may have arisen from melting induced by the impact (e.g., Morgan et al., 2009). Since the channels connect or nearly connect with the headwall of this class of alcoves, it is possible that the erosion of these alcoves was initiated by–if not heavily influenced by–channels. |
| Branching alcove | Branching alcoves consist of an alcove that is much longer than wide, with multiple tributaries to smaller simple alcoves. We chose to map the individual alcoves in the | 24 (7) | 1% | These branching alcoves appear qualitatively most similar to theater-headed valleys that have been hypothesized to have originated from groundwater sapping or outburst |





| | system that defines a branching alcove. Smaller simple alcoves that reside within the larger branching alcoves would fall into both classifications. | | | flooding in previous work (e.g., Lapotre and Lamb; 2018). We discuss these further in Section 6.4. |
|---|---|---|---|---|


### 3.2.2 Alcove Morphometric Calculations

We applied the Automated Cirque Metric Extraction (ACME; Spagnolo et al., 2017) tool in ArcMap to calculate
alcove morphometrics, specifically, length (L), width (W), elongation (L/W), altitudinal range or height (H; difference between
maximum and minimum elevation), area, slope, elevation, and aspect (Table 3). However, we note that the ACME tool is
designed for classic cirques on Earth and while the tool works with complex shapes, it should not be relied on for curving,
elongated features (Spagnolo et al., 2017). To use the ACME tool, we provided the mapped shape of the alcove, a threshold
midpoint, which is defined as the midpoint of the down-valley lip of the cirque, and the HRSC DEM as inputs (Fig. 5). ACME
outputs the morphometrics into the alcove's feature class attribute table. On Earth, typical L/W ratios are 0.5-4.25 (Derbyshire
and Evans, 1976), and based on 10,362 globally distributed cirques, both L/H and W/H ratios typically range between 1.5 to
4.0 (Barr and Spagnolo, 2015). We combine the simple alcove class (based on morphology) with these morphometric values
for L/W, L/H, and W/H as constraints to further narrow down our population to the most cirque-like alcoves. By applying
these constraints, we were able to identify 386 cirque-like alcoves from our initial mapping and classification of over 2000
alcoves.
**Table 3:** Alcove morphometrics as outputted by the Automated Cirque Metric Extraction (ACME) tool. The content
was modified from Spagnolo et al. (2017) to fit a table format.

| Name | Unit | ACME's output name | Definition |
|---|---|---|---|
| Length | Meters | L | Length of the line within the alcove polygon that intersects the alcove threshold midpoint and splits the polygon into two equal halves (Fig. 5) |
| Width | Meters | W | Length of the line perpendicular to the length line and intersecting the length line midpoint (Fig. 5) |



| Elongation | Dimensionless | L/W | Derived from dividing length by width |
|---|---|---|---|
| Altitudinal range or height | Meters | Z_range (H in this paper) | Range of elevations found by subtracting max elevation minus minimum elevation (Fig. 5) |
| Elevation | Meters | Z_mean | Mean elevation |
| Area | Meters$^2$ | Area_2D | Area of the polygon |
| Slope | Degrees | Slope_mean | Mean value of slope for all DEM pixels included in the alcove polygon. |
| Aspect | Degrees north, within the 0-360° interval | Aspect_mean | Mean of all pixel aspects across the entire surface of alcove by converting these into radians, extracting the mean sine and cosine of these values, calculating the arctangent of the ratio between mean sine and mean cosine, then finally converting this back into degrees. This is the direction that the headwall faces and can provide insights into paleowind directions and slopes where snow and ice accumulation is promoted on Earth (Barr and Spagnolo, 2015). |






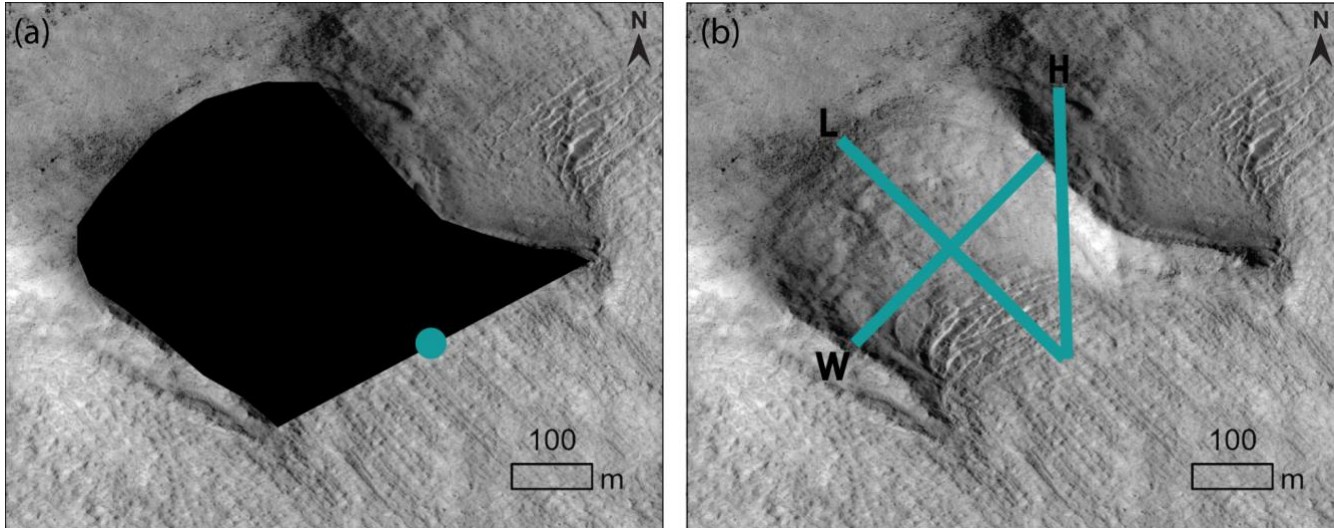


**Figure 5: (a) Inputs for the Automatic Cirque Metric Extraction (ACME) tool (Spagnolo et al., 2017) include a shapefile**
**for the alcove, a point for the alcove threshold, as pictured here, and a DEM. (b) Outputs from ACME include**
**morphometrics such as the length, width, and height of the alcove. CTX data credit: Caltech/NASA/JPL/MSSS.**

**3.2.3 Uncertainty in alcove longitudinal profile**

Longitudinal profiles of cirques on Earth are typically characterized by a concave bowl-shape with a steep headwall,

flatter floor, and a lip or threshold at the end of the profile that separates the cirque from the valley below (e.g., Barr and
Spagnolo, 2015). In Deuteronilus Mensae, an alcove threshold may be visible with the HRSC DEM resolution of 50-100
m/pixel (Fig. 6). However, in some cases, we do not see the threshold because of low DEM resolution or because the feature
may be covered by material (debris or ice; Fig. 6). Not all glacial cirques on Earth have thresholds either (e.g., Fig. 2), nor is
having a threshold a definitional requirement of terrestrial cirques (Barr and Spagnolo, 2015). As a result, we do not use the
existence of an observable threshold as a requirement for identification as a cirque-like alcove on Mars.





**Figure 6: Arrows represent the path of the profiles from higher to lower elevations for both (a) and (b). (a) Example of the longitudinal profile of a mapped alcove using the HRSC DEM that includes the overdeepening from glacial erosion then threshold or lip in the profile. The alcove is centered at 40.22°N, 34.57°E. in the CTX mosaic (Dickson et al., 2023a). (b) Comparison between the HRSC and CTX DEMs. The alcove is centered at 46.55°N, 22.08°E in HiRISE image ESP_019214_2270_RED. The CTX longitudinal profile contains a threshold, but the same feature in the HRSC DEM is**



**not resolvable. CTX data credit: Caltech/NASA/JPL/MSSS. The CTX DEM was constructed by Dr. Mackenzie Day's**
**GALE lab at UCLA. HRSC data credit: ESA/DLR/FU Berlin.**

**4 Results: Morphometric observations of cirque-like alcoves in Deuteronilus Mensae**

**4.1 Comparison of length, width, height of cirque-like alcoves on Mars with cirques on Earth**

Focusing on cirque-like alcoves only because they are the most likely candidate cirques, Table 4 compares the length,
width, and height for the cirque-like alcoves in Deuteronilus Mensae (as defined by morphometrics and the simple alcove class
in Section 3.2.2) to a global population of 10,362 cirques on Earth as compiled in a review by Barr and Spagnolo (2015).
Similar to Earth, both length and width are on average over twice the value for height in all cirques (Table 4). On average,
cirque-like alcoves have length, width, and height values that are larger than cirques on Earth.
**Table 4**. A comparison of the length (L), width (W), and height (H) for cirque-like alcoves mapped in this study in
Deuteronilus Mensae, Mars, and a population of 10,362 cirques on Earth (Barr and Spagnolo, 2015).

| | Mean L (m) | Range in L (m) | Mean W (m) | Range in W (m) | Mean H (m) | Range in H (m) |
|---|---|---|---|---|---|---|
| **Cirque-like alcoves in Deuteronilus Mensae (386 alcoves)** | 1260 _Median:_ 982 | 240–5636 | 1281 _Median:_ 999 | 231–5945 | 454 _Median:_ 353 | 92–1850 |
| **Cirques on Earth (10,362 cirques)** | 744 | 53–4584[1] _Typical:_ 100–1500 | 749 | 99–3240[1] _Typical:_ 100–1500 | 309 | 20–1328 _Typical:_ 150–600 |

[1]_Both 4584 m for the maximum length and 3240 m for the maximum width are specific to cirques in the Dry Valleys of_
_Antarctica (Aniya and Welch, 1981)._

**4.2 Trends in aspect, size, area, latitude, slope, and elevation**

By examining the aspect of the population of 386 cirque-like alcoves, we observe a south to southeast bias with an
average of 156.33° (Fig. 7). The largest mean size and area for cirque-like alcoves correspond to lower latitudes (Fig. 8a). The
largest mean size and area correspond to slopes of 25-30° (Fig. 8b), and alcoves have an average slope of ~19.3°. The mean
size and area of the alcoves increase with elevation (Fig. 8c). Above 46.5° in latitude, most alcoves cluster between 125-240°
in aspect (Fig. 9a). We also notice a lower density of alcoves facing 250°-360° at all latitudes (Fig. 8a). Alcove elevation
decreases as latitude increases (Fig. 9b). Similarly, alcove height also decreases as latitude increases (Fig. 9c).




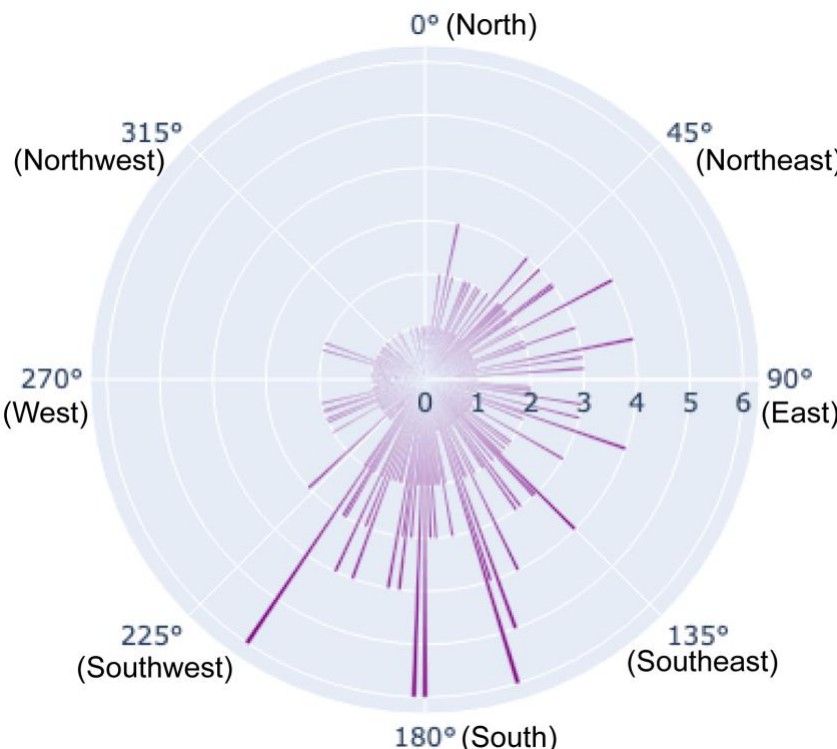


**Figure 7: Rose diagram showing the aspect of cirque-like alcoves. Cirque-like alcove aspect averages 156.33° between**

**the south and southeast directions.**



**Figure 8: Mean cirque-like alcove size (LWH)$^{1/3}$ and mean area vs. a) latitude, b) slope, and c) elevation for only cirque-like alcoves in Deuteronilus Mensae.**



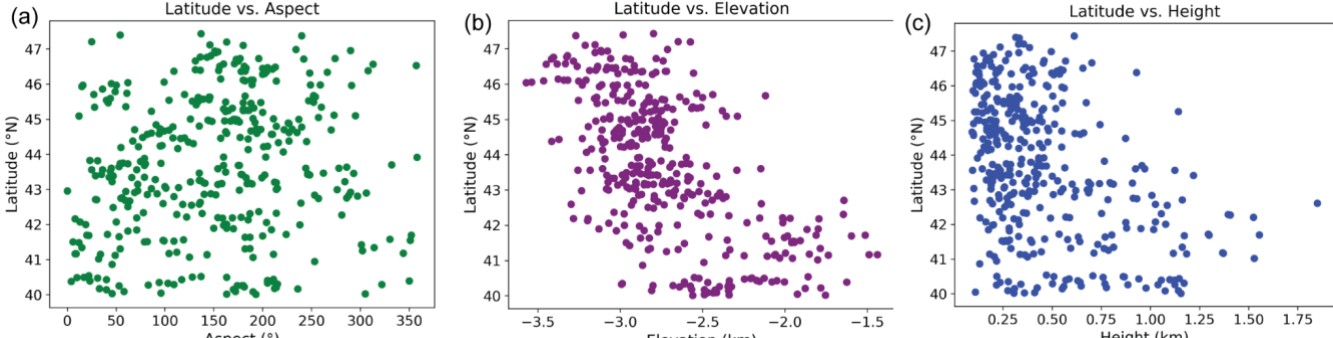

**Figure 9:** Cirque-like alcoves in Deuteronilus Mensae plotted by latitude versus a) aspect, b) elevation, and c) height.

## 4.3 Comparison between cirque-like alcoves and glacier-like forms mapped in Deuteronilus Mensae

Both the largest cirque-like alcoves and glacier-like forms are located in the southeast part of the study region (Fig. 10). While the average area of an alcove is smaller than a glacier-like form, the total area of all the cirque-like alcoves is larger than the total area of the glacier-like forms (Table 5).There are 74 mapped glacier-like forms in Deuteronilus Mensae (Brough et al., 2019), which is only about 19% of the total 386 cirque-like alcoves in this study area. As a result, the aggregate total area and aggregate total volume for the alcoves are larger than for the glacier-like forms. In addition, the average volume of an alcove is larger than that for a glacier-like form because the alcoves have a greater height than the typical estimated thickness of a glacier-like form. 18% of all alcoves are within 10 km of a glacier-like form and all alcoves are within 146 km of a glacier-like form.

**Table 5:** Area and volume statistics of cirque-like alcoves versus glacier-like forms. Statistics for the cirque-like alcoves come from the topographic expression of the alcove, whereas the statistics for the glacier-like forms are from the present-day ice-rich form,

| | Average Area (km²) | Total Area (km²) | Average Volume (km³) | Total Volume (km³) |
|---|---|---|---|---|
| **Cirque-like alcoves (386)** | 2.22 | 856.14 | 2.07 | 800.95 |
| **Glacier-like forms (74)** | 7.79 | 576.82 | 1.14 | 84.01 |



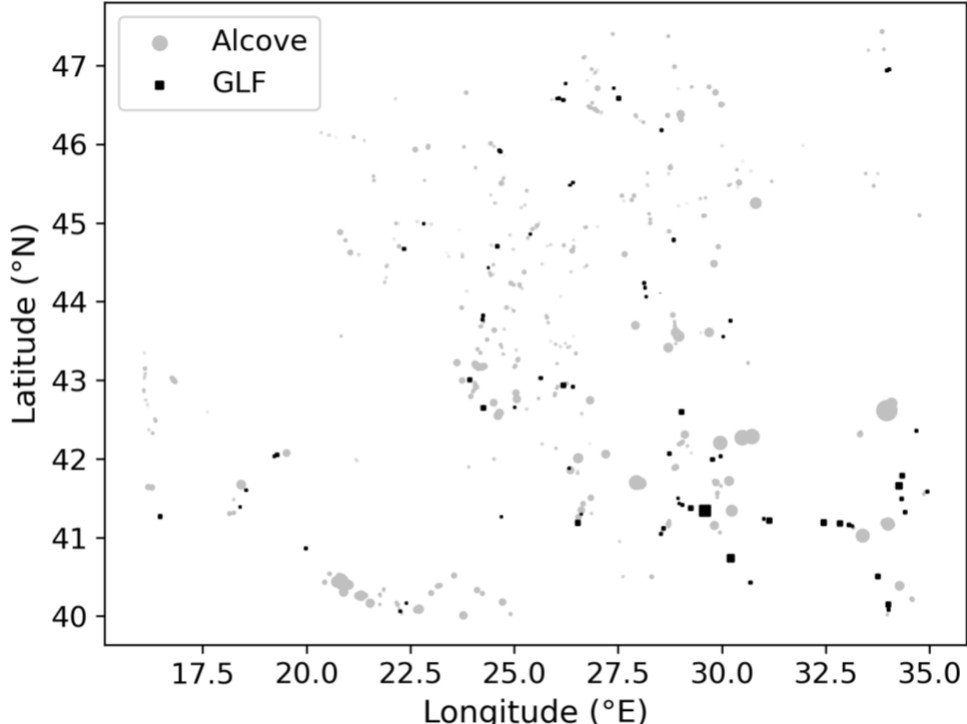


**Figure 10: Distribution of 386 simple alcoves (gray circles) and 74 glacier-like forms (glacier-like form; black squares) in the study
region Deuteronilus Mensae. Polygons show relative size differences.**

While the highest percentage (22%) of glacier-like forms have a northeast orientation, the highest percentage (26%)

of cirque-like alcoves have a southward orientation (Fig. 11a). For both glacier-like forms and cirque-like alcoves, the west
and northwest aspects have relatively low numbers ranging from 3% to 8% of the entire population, though unlike glacier-like
forms (15%), cirque-like alcoves also had a low proportion of 6% for the north-facing aspect. For mean glacier-like form
volume grouped by aspect, the largest glacier-like forms face southwards in Deuteronilus Mensae, whereas the largest cirque-



like alcoves by volume face the north (average = 5.24 km$^3$) and southwest (average = 3.97 km$^3$; Fig. 11b).

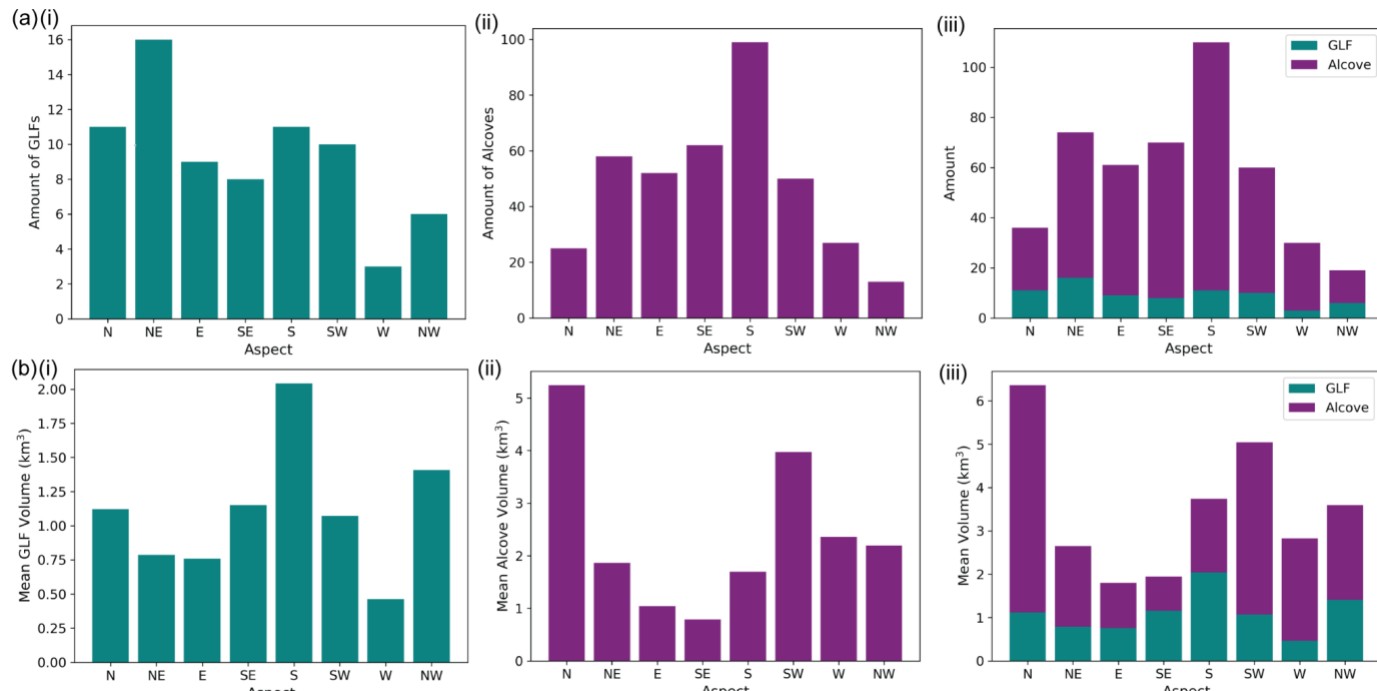


**Figure 11: a) Bar plots of the aspect compared to the quantity of i) cirque-like alcoves, ii) glacier-like forms, and iii) both cirque-like**
**alcoves and glacier-like forms. b) Bar plots of the aspect compared to the average area in each aspect direction for i) cirque-like**
**alcoves, ii) glacier-like forms, and iii) both cirque-like alcoves and glacier-like forms.**

**5 Discussion**
**5.1 Geologic context of morphometrics**
**5.1.1 Comparison of length, width, and height of cirque-like alcoves on Mars with cirques on Earth**
The median cirque-like alcove in Deuteronilus Mensae is ~32% larger than the average cirque on Earth. This suggests
that in comparison with Earth, either more episodes of glaciation occurred on Mars and lasted a longer amount of time to erode
the cirque-like alcoves in Deuteronilus Mensae, the erosion rates on Mars were much more rapid, or the initial hollow for snow
to accumulate in was larger on Mars. Future modeling may better investigate which is the most likely cause.

**5.1.2 Trends in aspect**
The eastward bias for cirque-like alcove aspect is similar to the trend of cirques in the mid-latitudes on Earth, where
cirque aspect commonly faces eastward because glaciers are more likely to grow on the lee side of westerly winds present at
these latitudes (Barr and Spagnolo, 2015). On Mars a slight easterly bias has also been identified in the overall glacier-like
form population (Souness et al., 2012). Climate modeling shows that both westerly winds and ice deposition are expected in
Deuteronilus Mensae during the northern winter (Madeleine et al., 2009). Both alcoves and glacier-like forms have an easterly





bias that might be consistent with atmospheric control, but further work is needed to understand this. The southern bias is less
intuitive. Cirques in the northern hemisphere on Earth are generally biased toward having a northerly (poleward) orientation,
where total solar radiation is lowest and lower air temperatures allow for glaciers to persist for longer (Barr and Spagnolo,
2015).

This pattern is seen for glacier-like forms too: for example, Souness et al. (2012) found glacier-like forms to have a
poleward bias, although in Deuteronilus Mensae, the largest glacier-like forms face southwards, though the largest cirque-like
alcoves by volume face north and southwest. However, this may be due to a localized topographic effect for glacier-like forms
in Deuteronilus Mensae because overall for the northern hemisphere, glacier-like forms flowing northward are larger than
those flowing southward by about 20% (Brough et al., 2019). For both glacier-like forms and cirque-like alcoves, the aspect
with the highest percentage of the population does not correspond to the aspect with the largest mean volume. However, in all
cases, the aspect south or southwest does correspond to one of the maxima in each plot for the amount and volume of glacier-
like forms and cirque-like alcoves (Fig. 11). To explain the southward bias of cirque-like alcoves, we propose two possible
reasons:

1) As described in Brough et al. (2016), during periods of high obliquity >45°, poleward facing slopes receive higher
   insolation and summer day temperatures (Costard et al., 2002). This would make a southward facing alcove in the
   northern mid-latitudes more favorable for ice accumulation.
2) Previous work on gullies on Mars demonstrate that for regions poleward of 40°, like Deuteronilus Mensae, gullies
   were primarily on equator-facing slopes (Harrison et al., 2015; Conway et al., 2018). This might suggest a relationship
   between gullies and the cirque-like alcoves. For example, gullies could provide the initial concavity for a later cirque-
   like alcove to develop due to glaciation (Section 5.2.2), which is consistent with gully heads that have been proposed
   as initiation points for cirques on Earth (Derbyshire and Evans; 1976). This may suggest that cirque-like alcoves
   prefer to reside on equator-facing slopes because this would allow for increased insolation and the chance for
   increased meltwater or sublimation of $CO_2$ as temperatures increase (e.g., Pilorget and Forget; 2016; Dundas et al.,
   2022; Dickson et al., 2023b). Modeling found that temperatures above freezing for meltwater and gully formation are
   possible during high obliquity excursions in the mid-latitudes (Costard et al., 2002; Williams et al., 2008; Williams
   et al., 2009; Dickson et al., 2023b). According to Dickson et al. (2023), at high obliquities reaching 35°, meltwater is
   possible during the Amazonian because pressures exceed the triple point of water. Since increased surface meltwater
   has been linked to increased subglacial flow at the bed of cold polythermal glaciers (e.g., Bingham et al., 2006;
   Copland et al., 2017), increased meltwater for glaciers would also likely lead to more wet-based glacial conditions
   and erosion (see Section 5.4 for arguments in favor of wet-based glacier erosion).

### 5.1.3 Trends between size, area, latitude, slope, and elevation

Relationships between size, area, latitude, mean elevation, and height of the cirque-like alcoves are likely due to the
nature of topography. At lower latitudes, the mesas are at a higher elevation relative to the basin at lower latitude (Fig. 3), and



the overall elevation decreases toward the north. These two factors combined mean that at lower latitudes, the cirque-like alcoves have a higher mean value for elevation and height. The steep topography also provides rockfall as a debris source to shield the ice and shaded slopes to allow for cooler microclimates for ice preservation (Dickson et al., 2012). The larger height corresponds to a larger size, where size is calculated as $\sqrt[3]{LWH}$. Height also scales with length and width for cirque-like alcoves (Section 4.1), which is why both larger heights and areas of cirque-like correspond to lower latitudes.

**5.1.4 Comparison between cirque-like alcoves and glacier-like forms mapped in Deuteronilus Mensae**

Since both the largest cirque-like alcoves and glacier-like forms are located in southeast Deuteronilus Mensae, this suggests that glacier-like form size is proportional to alcove size. For example, larger cirque-like alcoves may only erode when glacier-like forms become large enough. Alternatively, the size of glacier-like forms may be limited to the initial size of the alcove that it occupies. Nevertheless, the average glacier-like form still has a larger area than the average cirque-like alcove because glacier-like forms typically extend beyond the cirque-like alcoves that they emerge from. While we do not distinguish between these two hypotheses in this study, we recommend future work to investigate the direct cause of the size correlation between glacier-like forms and cirque-like alcoves.

Although the average area of a cirque-like alcove is smaller than a glacier-like form, the total area of all the cirque-like alcoves is larger than the total area of the glacier-like forms (Table 5). If the simple cirque-like alcoves that we identify here are in fact representative of glacial erosion, then we extend our previous knowledge of the areal extent of past glaciation in Deuteronilus Mensae by at least 48%.

While the largest glacier-like forms face southwards in Deuteronilus Mensae and the largest cirque-like alcoves face the north, this may be due to a localized topographic effect for glacier-like forms in Deuteronilus Mensae because overall for the northern hemisphere, glacier-like forms flowing northward are larger than those flowing southward by about 20% (Brough et al., 2019).

**5.2 Geomorphic interpretations of cirque-like alcoves and associated features**

**5.2.1 Geomorphic associations with remnant ice or other ice-associated landforms**

Out of 386 cirque-like alcoves, 6 cirque-like alcoves (1.6%) had partial coverage, and 38 cirque-like alcoves (9.8%) had complete coverage within an available HiRISE frame. While we designed the study so that none of the cirque-like alcoves that we mapped included mapped glacier-like forms, using the available inventory of HiRISE images we observed other features associated with the cirque-like alcoves that appear consistent with the presence of ice or ice loss. For example, the cirque-like alcove shown in Fig. 12 hosts remnant material that appears similar to a degraded glacier-like form and exhibits crevasse-like features (Fig. 12c). On Earth, crevasses in glaciers occur because of high strain rates, and on Mars, crevasse-like forms have been observed in select glacier-like forms (e.g., Hubbard et al., 2014). Here, we observe a potential remnant of what may have been a glacier-like form as it degraded in the alcove (Fig. 12c), a potential analog to rock glaciers on Earth (Section 56.21). Of the 38 cirque-like alcoves with full coverage in HiRISE imagery, ~42% contained crevasse-like features.



Previous work focusing on landforms that appear similar to transverse crevasses at the base of crater walls has referred to these landforms as "washboard terrain", interpreted as a paraglacial dominated landform (Jawin et al., 2018; Jawin and Head, 2021). As the ice sublimates downslope and debuttresses, this causes remnant ice upslope to lose its basal support and begin to flow downslope, thus causing the ice to stretch and form transverse crevasses. In geomorphology, paraglacial activity is defined as nonglacial, but is conditioned by glaciation including during and after glacier retreat (Church and Ryder, 1972; Ballantyne et al, 2017). Based on visual comparisons, we observe that the crevasse-like features in some of the cirque-like alcoves incised into the sides of mesas in Deuteronilus Mensae have been observed in other regions as washboard terrain and suggested to form from deglaciation.

In addition, ~74% of cirque-like alcoves contain observable surface lineations consisting of parallel raised ridges and mounds (Fig. 12d) extending out from the crevasse-like features at the base of the cirque-like alcoves (Fig. 12). These surface lineations appear similar to the lineated "pasted-on terrain" identified by previous studies (e.g., Conway et al., 2018), as well as the "linear terrain" and "mound-and-tail-terrain" described by Hubbard et al. (2011) in association with glacier-like forms. While different origins of these terrains are proposed, they may provide further indication of the past presence of ice in these locations. In particular, as their leading hypothesis, both Conway et al. (2018) and Hubbard et al. (2011) interpreted the landforms that they observed to be subglacial in origin that required ice flow over water-lubricated sediment. Hubbard et al. (2011) offered megalineations as a terrestrial analog for the martian linear terrain and drumlins as a terrestrial analog for the martian mound-and-tail terrain, where both terrestrial analogs are large-scale features formed by wet-based glacial erosion.

~15% of cirque-like alcoves with HiRISE coverage contained "polygonal" or "polygonized" terrain, two terms that we use synonymously (see Fig. 12f). As described in association with the glacier-like forms in Hubbard et al. (2011) as "polygonized terrain," we see this type of polygonal terrain between the surface lineations or linear terrain (Fig. 12f). In addition, the polygonal terrain is observed farther downslope of the arcuate ridges and troughs, consisting of individually raised polygons surrounded by troughs (Fig. 12f). On Mars, polygonal terrain has been attributed to a combination of thermal contraction cracking and sublimation of ice (Levy et al., 2009b). Similarly, on Earth, polygonal terrain results from periglacial processes such as contraction cracking and frost heave (French, 2018), though sublimation-type polygons that arise from thermal contraction and sublimation of ice have also been observed in the Antarctic Dry Valleys (Marchant and Head, 2007).

~45% of cirque-like alcoves with HiRISE coverage include landforms that have irregular wavy textures expressed as arcuate ridges and troughs in the transverse direction located downslope of the crevasse-like terrain and surface lineations (Fig. 12e). The lobate shape terminating at the ridges and troughs appears similar to the spoon-shaped depressions that form from the melting of ice near the terminus of a rock glacier (e.g., Janke et al., 2013) or moraine-like ridges on Mars (Arfstrom and Harmann, 2005). This is similar to the spoon-shaped depression at the terminus of lobate rock glaciers on Earth, which typically occurs from subsidence (Janke et al., 2013) as the surface deflates due to the loss of internal ice, usually from melting on Earth, though sublimation is more likely for Mars. The texture of the ridges and troughs appear similar to brain terrain—a complex texture resembling the surface of a brain formed by a combination of glacial flow, thermal contraction cracking, and differential sublimation—that has been previously identified on Mars (Levy et al., 2009a), perhaps suggesting that differential



sublimation of buried ice may have contributed to the formation of these arcuate ridges. In the context of a proglacial environment, the arcuate ridges and troughs appear most similar to the "rectilinear-ridge terrain" identified for glacier-like forms (Hubbard et al., 2011). The rectilinear-ridge terrain is interpreted as similar to two types of proglacial moraines on Earth: thrust-block/push moraines and moraine-mound complexes. However, the rectilinear-ridge terrain appears more similar in both its scale and its less regular geometries to moraine-mound complexes, which has a disputed origin but likely requires liquid water to form (Hubbard et al., 2011). Here we suggest that due to the similarities between brain terrain and the ridges seen here, as well as the presence of polygonal terrain nearby, this rectilinear-ridge terrain or moraine-like ridges may have formed from a combination of wet-based glacial processes that form moraine-mound complexes and sublimation as a former glacier-like form lost its ice.

~21% of cirque-like alcoves with HiRISE coverage have a terminal moraine-like ridge, while ~31% have either terminal or lateral moraine-like ridges (e.g. Fig. 13a). While not all alcoves have moraine-like ridges, many cirque-like alcoves do exhibit surface foliation, suggesting that down-slope flow is present. When raised ridges akin to moraines are not evident (Fig. 13b-c), the ice deposit may have never reached the stage of activity to develop terminal landforms or the moraine-like ridges were not preserved.

At a potentially earlier stage of evolution of the glacier-like forms, we might also see forms with defined moraine-like ridges that lack clear elongation (Fig. 14a), potentially similar to a terrestrial cirque glacier sitting within the cirque basin instead of extending into the valley below. In Fig. 14a, we interpret the sinuous ridge as a moraine-like ridge due to its lobate form. It is likely that the most lobate part of the moraine-like ridge also reflects where the bulk of the ice or remnant material is located. This is consistent with the present-day dune fields toward the bottom right of the alcove, which indicate a stronger signature of eolian erosion.

Fig. 14b also shows additional examples of moraine-like ridges downslope of alcoves (that are not all cirque-like), with along-flow surface lineations between the alcove headwall and the moraine-like ridge. As in Fig. 12, crevasse-like features, surface lineations, and polygonal terrain are all present. Previous work has observed moraine-like ridges downslope of both alcoves and gullies incised into crater walls, where gullies consist of an alcove feeding into a channel and then depositing into an apron (Arfstrom and Harmann, 2005; Conway et al., 2018). The size of the moraine-like ridges may vary with the size of the upslope alcove, which suggests that some component of the material within the moraine-like ridge originates from the crater wall (Arfstrom and Harmann, 2005). On crater walls, the moraine-like ridges do not always correspond to an upslope alcove and shallower crater wall slopes, so Conway et al. (2018) interpreted similar features as analogous to unconstrained piedmont-style glaciers. However, since the moraine-like ridges do seem to correspond to upslope alcoves here, we suggest that the moraine-like ridges in Fig. 14b reflect the initiation of cirque-style glaciation before the alcove headwalls and sidewalls are increasingly eroded and steepened.




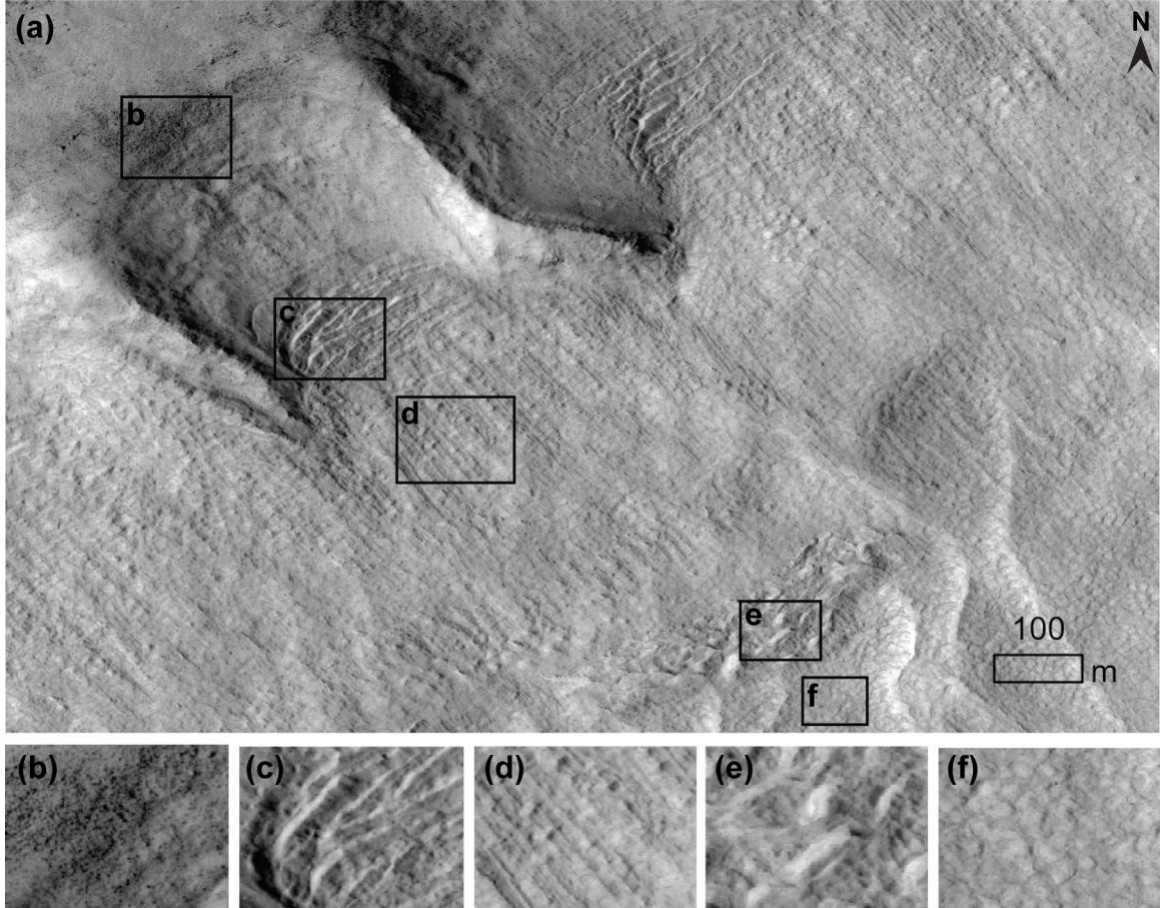

**Figure 12: a) Simple alcove with evidence for remnant ice centered at 22.12°E, 46.57°N in HiRISE image ESP_019214_2270_RED.
b) Boulders near the top of the headwall indicating erosion. c) Crevasse-like features, or sometimes referred to as washboard terrain
(Hubbard et al., 2014; Jawin and Head, 2021). d) Surface lineations are a potential indicator for subglacial erosion, similar to pasted-
on terrain in Conway et al. (2018). Both e) moraine-like ridges resembling brain terrain and f) polygonized terrain correspond to
where ice-loss has occurred (e.g., Levy et al., 2009a; Levy et al., 2009b; Jawin and Head, 2021).**




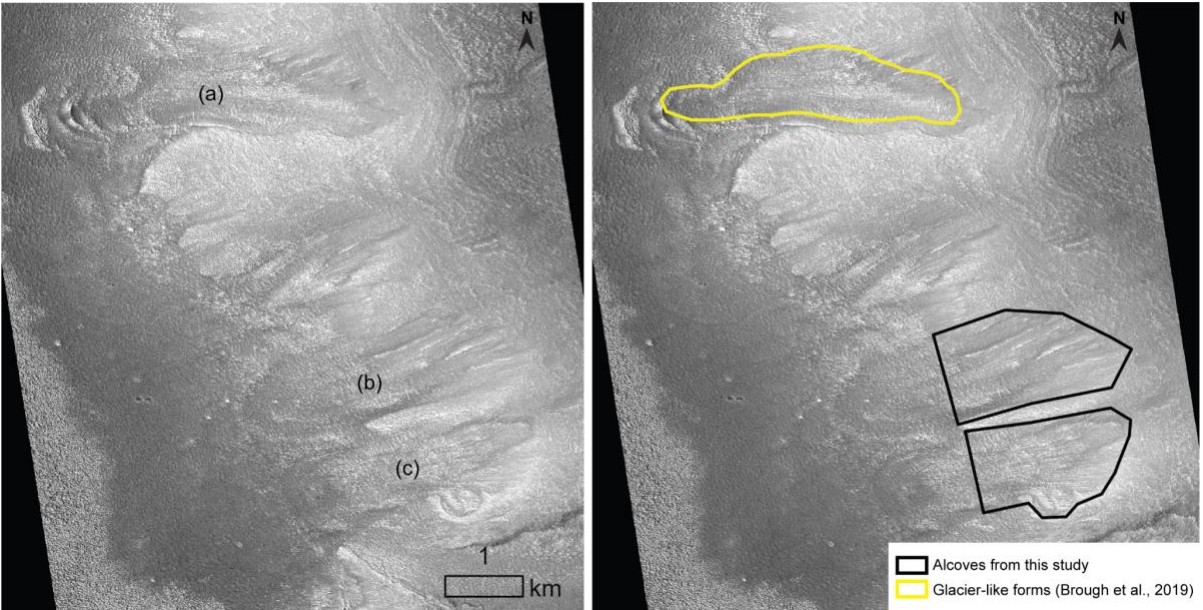

**Figure 13: (a) Previously mapped glacier-like form (Brough et al., 2019). (b) and (c) represent previously unmapped alcoves that lack a clear moraine-like ridge with a topographic high at the terminus, but still contain surface foliation suggesting down-slope flow. Alcove mapping only extends to where the sidewalls end. This HiRISE image ESP_025873_2230_RED is centered at 42.63°N, 25.02°E. HiRISE data credit: NASA/JPL/University of Arizona.**

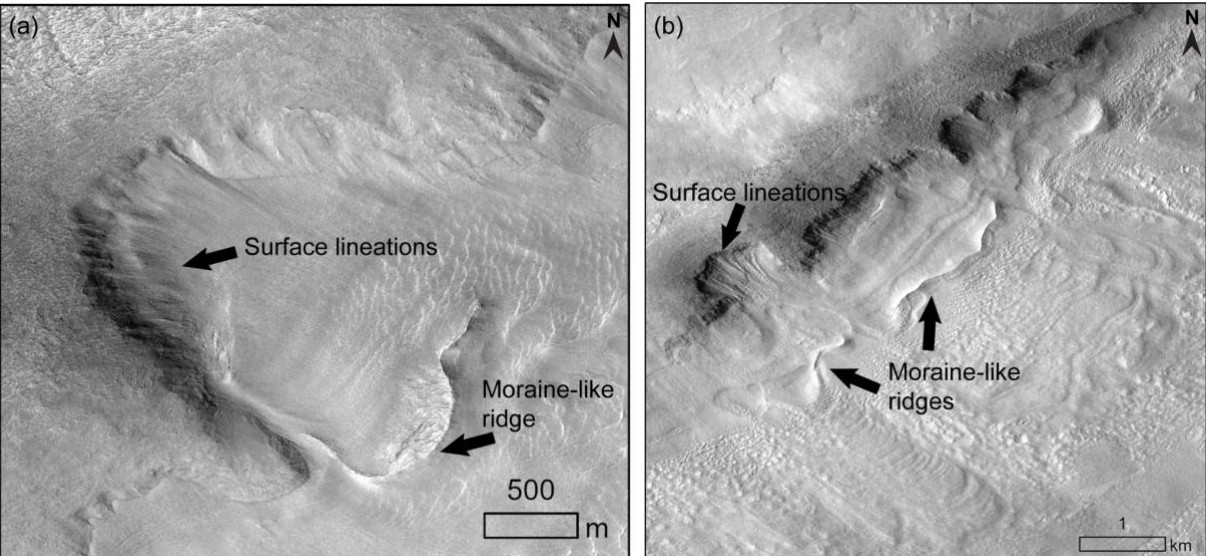

**Figure 14: (a) Potential icy form with a clear moraine-like ridge observed in a cirque-like alcove. HiRISE image ESP_033745_2270_RED is centered at 46.64°N, 29.83°E. (b) Possible unconstrained (known as piedmont) glaciation and moraine-like ridges. Surface lineations, crevasse-like features, and polygonal terrain are all present, though not all alcoves are cirque-like as**



**defined in Section 3. HiRISE image ESP_026941_2275 is centered at 47.09°N, 26.74°E. HiRISE data credit: NASA/JPL/University of Arizona.**

### 5.2.2 Evidence for different stages of cirque-like alcove evolution

Different stages of alcove evolution, likely linked to different histories of glacial occupation and erosion, can be seen in and near mapped cirque-like alcoves. For example, in Fig. 15a and 15b, notches (feature #1 in Fig. 15b) may indicate initial glacial erosion of the mesa sidewall. Stratigraphically, these notches predate the slab of detached mesa sidewall since the notches on the slab can be traced, but are now offset, from the notches above the slab (Fig. 15a). The notches resemble gullies on Mars, where previous work has shown that gully formation may occur during glacier retreat on Mars during paraglacial stages (Jawin and Head, 2021), where degrading ice no longer provides structural support for slopes of sediment. Gully incision may initiate through sediment flow assisted by either liquid water or $CO_2$, or dry mass wasting (e.g., Conway et al., 2019; Dundas et al., 2022; Dickson et al., 2023b). Since the slabs formed after the notches, this is consistent with increased mass wasting of the mesa sidewall during deglaciation. In the middle panel of Fig. 15a, for feature #2, there is evidence that the notches undergo further erosion and begin to connect, eventually forming feature #3 where the outlet between the larger notch head is overridden and enlarged. We suggest that an icy mantling deposit is responsible for this erosion since we see surface lineations that are consistent with pasted-on terrain. Fig. 15a feature #4 in the middle panel demonstrates continued erosion and enlargement of these alcoves as they grow and connect with neighboring alcoves until they lose internal ridges as glacial erosion smoothes the interior of the alcove. In Fig. 15b, the alcoves are smoother, appear to be more U-shaped (though CTX DEMs did not have high enough resolution for profiles), have more arcuate headwalls, and have narrower ridges between alcoves. We suggest that this represents a later, more developed stage of cirque-like alcove evolution, perhaps after multiple cycles of glaciation, where ice could erode repeatedly over time into the mesa sidewalls so that the alcove basin becomes smoother and the sidewalls develop into narrow ridges like in Fig. 12. In Fig. 15b, as in Fig. 15a, we see downslope debris and deposits indicating mesa sidewall erosion. In the middle panel, we see examples of deposits of mesa material that are pushed outwards from the alcoves (Fig. 15b). While it is likely that multiple processes contributed to the incipient form of a cirque-like alcove, we suggest that the morphometrics and conditions observed eventually require glacial erosion in comparison to alternative dominant formation mechanisms discussed in Section 6.4.



(a)(i)

(a)(ii)

(a)(iii)

(b)(i)

(b)(ii)

(b)(iii)



**Figure 15: (a) Top panel is centered at 41.06°N, 17.88°E in CTX image D04_0288880_2193_XI_39N342W. Notches may indicate glacial erosion of the mesa sidewall. Slabs and deposits suggesting active mass wasting from the slopes. Flow lines indicate the downslope direction of flow. Middle panel is centered at 40.02°N, 23.20°E in CTX image D21_035499_2203_XN_40N336W. Feature #1 represents initial notches, #2 represents the initial notches undergoing further erosion and beginning to connect, #3 demonstrates an outlet being overridden and enlarged, as indicated by flow lines, and #4 demonstrates the continual enlargement of these alcoves. Bottom panel is centered at 41.60°N, 18.46°E in CTX image N01_062743_2222_XI_42N341W. The slab indicates the unstable slopes. We see an alcove with surface lineations and leading to deposits, adjacent to a nearby glacier-like form on the same mesa sidewall. (b) Top panel is centered at 46.67°N, 26.13°E in HiRISE image ESP_016247_2270. Debris label in the top panel points to a possible detached block that is ~160x80m. Middle panel is centered at 46.57°N, 26.02°E in CTX image P13_006160_2252_XN_45N334W. Flow lines indicate a flow toward the top of the image (north direction), consistent with the deposits detaching from the mesa in the south and being transported northwards. Bottom panel is centered at 46.56°N, 22.13°E in HiRISE image ESP_019214_2270. These alcoves represent the most mature cirque-like forms due to their defined ridges. A deposit has an outline similar to the mesa sidewall, suggesting downslope flow. HiRISE data credit: NASA/JPL/University of Arizona. CTX data credit: Caltech/NASA/JPL/MSSS.**

## 5.3 Possible origin of remnant material in cirque-like alcoves

In Section 5.1, we presented evidence suggesting that at least some of our mapped cirque-like alcoves in Deuteronilus Mensae retain glacial remnants. We consider rock glaciers on Earth as a potential analog for the glacial remnants observed in these cirque-like alcoves. Glacier-like forms have been noted as likely having more in common with rock glaciers on Earth than typical ice-dominated glaciers (Hubbard et al., 2011). Rock glaciers are commonly observed at the base of cirque headwalls where there is ample debris (e.g., Lillquist and Weidenaar, 2021) and extend from cirques on Earth (e.g., Janke et al., 2013; Munroe, 2018). Cirque glaciers may also evolve into rock glaciers (e.g., Berger et al., 2004; Oliva et al., 2023). Since rock glaciers can form by different processes (e.g., Knight et al., 2019; Anderson et al., 2018), these glacial remnants may represent a late stage of glacier-like forms or a rock glacier evolving from freeze-thaw cycling.

## 5.4 Estimating the timescales for cirque-like alcove erosion on Mars

Using terrestrial understanding of glacier erosion we can calculate approximately how long the erosion of a median cirque-like alcove would take based on different assumptions. First, we take into account that the surface gravity of Mars is about one third of Earth's at 3.71 m/s$^2$. Second, in order to calculate erosion rates in the past we estimate the former ice-surface velocity to derive the basal sliding velocity, keeping in mind that sliding velocities are notably higher for wet-based glaciers than for cold-based glaciers (Table 1). Third, we estimate the total occupation time of active (flowing) glaciers in the alcoves, which on Mars is likely a function of orbital forcing (e.g., Head et al., 2003). In order to evaluate the possibility that alcoves were eroded by wet-based glaciers, we calculate erosion rates for both wet- and cold-based ice. We find erosion rates consistent with previous estimates of erosion rates (Levy et al., 2016; Conway et al., 2018) and timescales (Berman et al., 2015; Hepburn et al., 2020), indicating that a cold-based glacier would take an order of magnitude longer than a wet-based glacier to erode the alcoves.



Our erosion rate estimates are derived from basal sliding velocities ($U_S$) which are in turn derived from glacier surface
velocities, using an empirical relationship from a terrestrial global dataset of 38 glaciers (Cook et al., 2020):
$U_s = U_{surf} - \frac{2A}{n+2}(\rho g sin\alpha)^n h^{n+1}$,                                    (1)
where $A$ is a temperature-dependent ice softness parameter (for warm ice, $A = 24 \times 10^{-25} s^{-1} Pa^{-3}$; for cold ice, $A =$
$3.5 \times 10^{-25} s^{-1} Pa^{-3}$; Cuffey and Paterson, 2010), $n$ is a flow-law exponent that is typically 3, $\rho$ is ice density, $g$ is
gravitational acceleration, $\alpha$ is ice surface slope, $h$ is ice thickness, and $U_{surf}$ is glacier surface velocity. For glacier-like forms
on Mars, average $h$ is 130 m and, since $\alpha$ ranges from 2 to 8° (Brough et al., 2019), we use 5° here to represent an order-of-
magnitude estimate. Since surface velocities of glacier-like forms on Mars are unknown (Brough et al., 2019; Hubbard et al.,
2014), for the warm-based case, we use a surface velocity of 2 m/yr (Cook et al., 2020) and for the cold-based case we use a
surface velocity of $8 \times 10^{-3}$ m/yr, which was measured for Meserve Glacier, a cold-based glacier in the Antarctic Dry Valleys
(Cuffey et al., 1999b) that represents a low glacier flow speed on Earth. Erosion rate $E$ was calculated as:
$E = K_G U_s^l$,                                    (2)
where $K_G$ is a bedrock erodibility constant and $l$ is an erosion exponent. While $K_G$ and $l$ can vary depending on the bedrock
type, $K_G$ is commonly $10^{-4}$ (Cook et al., 2020). Cook et al. (2020) empirically estimated $l$ to be 0.69 based on the relationship
between the erosion rate and glacier sliding velocity of 38 glaciers, including Meserve Glacier, and we use that value in our
estimates. We consider erosion rate as vertical erosion rate relative to the height of the median alcove, which is 353 m, and the
height of the average alcove which is 454, so rounding up from the median, we use 400 m.
While this study does not provide exact age constraints for cirque-like alcoves, our erosion rate estimates help
constrain the minimum length of time required for their development. For the warm-based glaciation scenario, we estimate an
erosion rate of ~160 m/Myr, which is close to the upper end of the 0.08 to 181 m/Myr range estimated by Conway et al. (2018)
for glaciated crater walls on Mars. For continuous glacial occupation and ignoring that glaciers may have only been active
(and eroding their bed) during certain obliquity periods, this would suggest that a total of ~2.5 Myr would be required for the
cirque-like alcoves to form. However, accounting for obliquity changes when conditions may not have always been optimal
for active glaciation would extend this time. As an estimate, over the last 10 Myr, there were 100 kyr orbital cycles, with
periods of high obliquity lasting 20-40 kyr (Head et al., 2003). On Earth, cirques are presumed to be mostly eroded at the
beginning and the end of glaciations (e.g., Barr et al., 2019), so assuming that the cirque-like alcoves only have 20 kyr of
erosion time during every 100 kyr period, or 20% of the total time passing by, it would take ~13 Myr total time to erode a
median height cirque-like alcove. This timescale is consistent with previous estimates of the age of certain populations of
glacier-like forms (Hepburn et al., 2020), which means that at least some of the glacier-like forms could have eroded the
alcoves which they currently occupy and that at least some of the empty alcoves could have hosted glaciers in the past tens of
millions of years.





On the other hand, if we assume cold-based conditions for glaciers that occupied the cirque-like alcoves, then the erosion rate estimated from the median values for cirque-like alcoves is ~3.6 m/Myr, which is consistent with the wide-ranging estimate of 0.1-10 m/Myr for cold-based viscous flow features on Mars (Levy et al., 2016) but is lower than the Conway et al. (2018) estimates for glaciated crater walls. Thus, for a cold-based glacier, a total glacier occupation time of ~110 Myr would be required for the cirque-like alcoves to form, but accounting for obliquity variations a median height cirque-like alcove would require ~560 Myr to erode with only cold-based glaciation. If the glaciers were cold-based during their entire evolution, the erosion timescale is much longer and therefore the alcoves must be much older than if they evolved with periods of warm-based glaciation. A timescale of hundreds of millions to a billion years is in the range of when lobate debris aprons were thought to have formed, where in Deuteronilus Mensae the lobate debris aprons are estimated to be as old as 1.1 Gyr (Berman et al., 2015).

The supraglacial debris covering the lobate debris aprons averages ~25 meters in thickness and a major fraction of the debris was sourced as rockfall from the mesas (Baker et al., 2019). Thus, it is reasonable to expect that the present state of the mesa sidewalls, including the cirque-like alcoves, formed either concurrently with or after the lobate debris aprons evolved and became covered with debris. Otherwise, the erosional process sourcing the supraglacial debris would likely have erased the cirque-like alcoves. Although the material in the lobate debris aprons in the present-day does not exhibit a connection to cirque-like alcoves, here we use the maximum age estimate of lobate debris aprons of 1.1 Gyr as the earliest time that glaciers could have began the erosion process that led to the formation of the cirque-like alcoves. By also including the consideration of obliquity that only around 20% of the 1.1 Gyr would have been conducive for ice accumulation, the maximum erosion depth achievable by cold-based glacier erosion would be ~790 m. Since ~14% of the cirque-like alcoves are larger than 790 m, we conclude that at least some of the cirque-like alcoves could have required a faster erosion rate than the ~3.6 m/Myr suggested for cold-based glaciers. However, some have suggested cold-based glaciers erosion rate on Mars up to 10 m/Myr (Table 1; Levy et al., 2016). If this upper-end rate is applied, then all of the cirque-like alcoves could have formed via cold-based glacier erosion.

On Earth, the chronology of cirque formation is difficult to constrain (e.g., Turbull and Davies, 2006), and estimates for total glacial cirque erosion time range from 125 Kyr (Larsen and Mangerud, 1981) to a few million years (Andrews and Dugdale, 1971; Anderson, 1978; Sanders et al., 2013). Our estimates here find that a median height cirque-like alcove in Deuteronilus Mensae would take in the range of ~13 Myr to form if occupied by a wet-based glacier, and ~560 Myr if occupied by a cold-based glacier.

## 5.5 Argument for wet-based glacial erosion of cirque-like alcoves

Since the alcoves are located in the fretted terrain at the dichotomy boundary, it has been posited that tectonic and fluvial activity may have generated the initial alcove notches that allowed for later accumulation of ice (Morgan et al., 2009).



Although previous research agrees that the alcove walls were likely widened by erosion from past glaciations (Head et al.,
2006; Morgan et al., 2009), which is similar to glacial cirque erosion on Earth, this is inconsistent with the typically accepted
hypothesis that these glaciers have always been cold-based, as cold-based glaciers are minimally to not erosive at all of the
subglacial terrain (Table 1). While present-day glacier-like forms on Mars show little influence from liquid water, previous
work has also suggested that former glacier-like forms or their predecessor ice masses may have been wet-based from
identification of megascale glacial lineations and eskers (Hubbard et al., 2014). Our work supports that the cirque-like alcoves
were eroded by former wet-based glacier-like forms, or predecessor ice masses that were wet-based. For the alcoves to start
as notches and evolve to their later stage (Fig. 15a) erosion is necessary beyond that which initiated the alcoves (e.g., gullies,
landslide scars, small craters, etc.). To maintain the tall, steep headwall and concave shape of the cirque-like alcoves, wet-
based glacial erosion is necessary to keep pace with the talus accumulating from erosion and weathering of the headwall, just
like with cirques on Earth (Evans, 2020). In addition, the type of overdeepening observed in the profiles (Fig. 6) is consistent
with wet-based glacial erosion on Earth. Finally, as described in Section 5.2, the surface lineations that appear similar to pasted-
on terrain (Fig. 12) could also be indicative of wet-based subglacial erosion.

While cirques on Earth can be efficiently eroded by wet-based glaciers, the time required for wet-based glacial erosion
of cirque-like alcoves on Mars remains an open question. Estimates of present-day erosion within cirques on Earth often lead
to relatively slow rates, for example: vertical incision ~0.5-0.9 mm/yr and headward retreat ~1.2 mm/yr (Sanders et al., 2013)
and the total erosion time ranges from hundreds of thousands of years (Larsen and Mangerud, 1981) to a few million years
(Andrews and Dugdale, 1971; Anderson, 1978; Sanders et al., 2013). Previous work limits active erosion of cirque basins only
to the onset and termination of glacial cycles (Kleman and Stroeven, 1997). As a result, it is likely that for numerous cirques
to fully develop in a given region, they must undergo multiple glacial/interglacial cycles (Rudberg, 1992), during which they
may also be glacier-free for periods of time (Barr et al., 2019). However, measured or hypothesized erosion rates typical of
cirque glaciers are high enough to indicate that cirques on Earth might have attained most of their size at the beginning of the
Quaternary, within the initial glacial cycles, and evolved very little since (Barr et al., 2019). This occurs as a "least-resistance"
shape is reached in which subglacial sediment accumulates at the bedrock interface and slows down subsequent erosion (Barr
et al., 2019). Given this, we note that the present-day glacial features within a cirque-like alcove did not necessarily contribute
to the erosion of most of the cirque's current form and that instead prior glacial cycles may have contributed to a higher
proportion of alcove erosion. However, the glacial geomorphic features (Fig. 12) demonstrate that ice accumulation is
favorable in the alcoves, thus strengthening the idea that the alcoves may have hosted previous glaciers over different glacial
cycles. Since the pressure for ice to melt is favorable at high obliquities on Mars (Dickson et al., 2023), we find warm-based
erosion to be a more likely candidate than cold-based erosion for cirque formation on Mars.

**5.6 Possible alternative mechanisms for alcove formation using examples from Earth**

While in Section 5.2.2 we discussed examples of erosional processes that may initiate a hollow for ice to later fill,
here we discuss other possible mechanisms that may generate the circular bowl shape of the cirque-like alcoves. Fig. 16a



provides examples of landslide scars resulting from active-layer detachments due to periglacial processes on Ellesmere Island,
Canada (e.g., Lewkowicz, 1990; Lewkowicz, 2007). Active-layer detachments are rapid mass-wasting processes or landslides
that result from the thawed active layer sliding over the underlying frozen soil of permafrost (Lewkowicz, 1990). The notches
and gullies in Deuteronilus Mensae, a potential incipient phase of cirque-like alcoves, (Fig. 15a) have some similarities to the
morphology of active-layer detachments (Fig. 16a). In addition, active-layer detachments with cracks leading into the headwall
scarp (Fig. 16a) seem to resemble the channels leading into the cirque-like alcoves (Fig. 4f). However, issues remain for an
entirely periglacial origin to explain the cirque-like alcoves on Mars. In Fig. 16a, we see that almost all of the landslide scars
have a visible crack leading into the headwall scarp, which we do not observe to be common for the cirque-like alcoves on
Mars. In addition, the scale of these headwall scarps for active-layer detachments are only on the order of tens of meters in
length and width (Fig. 16a). Without an additional type of landscape erosion, it is difficult to scale from the size of these
features to the cirque-like alcoves that can be kilometers in size $(LWH)^{1/3}$ on Mars (Fig. 3).
In some locations on Mars, previous work has invoked groundwater sapping or seepage erosion to explain
amphitheater-headed valleys (e.g., Harrison and Grimm, 2005). However, initial stages of the cirque-like alcoves (Fig. 15a)
do not appear to have morphologies consistent with amphitheater-headed valleys known to have been initiated by groundwater
sapping on Earth, such as those in Canyonlands National Park, Utah (Fig. 16b). In addition, these types of channel networks
are uncommon on Earth and the volumes of water required to carve bedrock canyons on Mars are unlikely (Lamb et al., 2006).
In place of groundwater seepage, catastrophic outburst floods may incise the steep headwall canyons on Mars (e,g., Lamb et
al., 2006; Lapôtre et al., 2016; Lapôtre et al., 2018). Such floods may be initiated by basal melting beneath ice sheets, pooling
in hydraulic lows, and subglacial floods (Baker and Milton, 1974; Baker, 2001; Evatt et al., 2006; Buffo et al., 2022). A past
ice sheet has been proposed to cover the northern mid-latitudes, including Deuteronilus Mensae (Madeleine et al., 2009),
however, it is modeled for cold-based glaciation during the late Amazonian (Fastook et al., 2011). In order for the branching
channel networks to form prior to the smaller alcoves incised within their sidewalls (Fig. 2, 4), this would require outburst
flooding to occur during an earlier cycle of glaciation, such as from melting ice sheets during a Late Noachian Icy Highlands
scenario (Buffo et al., 2022). In addition, since any erosional process sourcing ~25 m of debris over the lobate debris aprons
likely would have erased pre-existing alcoves (Section 5.4), we find it unlikely that the alcove shapes were primarily preserved
from the late Noachian. Further studies could model the interaction between outburst flooding as a source of initial alcove
formation and later glacial occupation eroding the cirque-like alcove into its current form.
Using Earth analogs we qualitatively discussed two possible erosional mechanisms, but in order to further
differentiate between possible erosional mechanisms for features similar to the cirque-like alcoves we compared
morphometrics of our alcoves against known erosional mechanisms (Table 6). We see that the H/L ratio of a glacial cirque is
expected to be deeper than any of the other features with known morphometrics, which we find to be consistent with our
population of simple cirque-like alcoves in Deuteronilus Mensae.





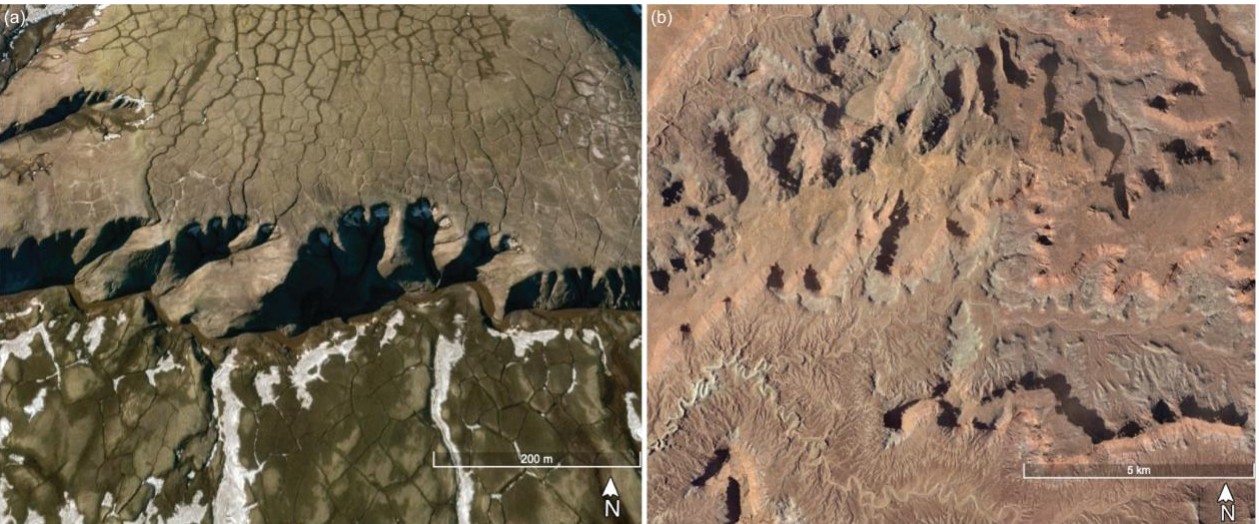


**Figure 16: Examples for alternative processes that can generate alcove forms: alcoves resulting from (a) active-layer detachment**
**on Ellesmere Island, Canada, and (b) groundwater sapping in Canyonlands National Park, Utah. Images are from © Google Earth;**
**note that they are at different scales. Imagery is from © Google Earth including Airbus coverage.**

**Table 6:** Morphometrics consistent with different erosional mechanisms.

| Formation mechanism/Landform | L/W | H/L | Aspect | Related geology | Typical scale (m) |
|---|---|---|---|---|---|
| Glacial cirque on Earth | ~1, generally ranges from 0.5-4.25 (Barr and Spagnolo, 2015) | ~0.67 (Barr and Spagnolo, 2015) | All directions; poleward is favorable (Barr and Spagnolo, 2015) | Overdeepening, moraines | $10^2$-$10^3$ (Barr and Spagnolo, 2015) |
| Periglacial landslides/active-layer detachments on Earth[n.s.] | ~2-3 (Lewkowicz, 1990) | Not available*, though typically H < 1 m deep (French, 2018) | Not available* | Cracks leading up to alcoves; Landslide deposit | $10^1$-$10^2$ (Lewkowicz, 2007) |
| Deep-seated landslide on Earth[n.s.] | >2.5 (Fran et al., 2006) | 0.1-0.35 (LaHusen et al., 2016; landslide scars | Not available* | Hummocky landslide deposits | $10^1$-$10^2$ (LaHusen et al., 2016) |




|  |  | from glacial sediment) |  |  |  |
| --- | --- | --- | --- | --- | --- |
| Impact crater on Mars | ~1 | 0.1-0.3 (Robbins et al., 2017) | N/A | Ejecta blanket | $10^1$-$10^3$ (Palucis et al., 2020) |
| Groundwater sapping or outburst flooding theater-headed valley on Mars | 1-10 (Laity, 1988) | Not available* | Not available* | Sandstone, not basalt bedrock (Lapotre and Lamb, 2018) | $10^1$-$10^2$ for canyon heads, up to $10^3$ for the main channel (Lapotre et al., 2016) |

Not available*: As of writing this paper, focused studies on the morphometrics of these landforms on the population scale are not widely available for other landforms.

[n.s.]: "n.s." stands for "not scarp" since landslide morphometrics do not usually include measurements of the morphometrics of just the headscarp and sidewalls of where the landslides initiated.

**6 Conclusions**

This is the first in-depth, population scale study of the morphometrics and geomorphic evidence associated with cirque-like alcoves as indicators of wet-based glaciation on Mars. By mapping ~2000 alcoves in Deuteronilus Mensae that did not contain previously mapped glacier-like forms, grouping them into seven classes, and then downselecting to only simple alcoves with length/width (L/W) between 0.5 to 4.25, length/height (L/H) of 1.5 to 4.0, and width/height (W/H) of 1.5 to 4.0, which are consistent with terrestrial cirques, we are able to identify a population of 386 "cirque-like alcoves." Using HiRISE imagery available for ~12% of these cirque-like alcoves, we find evidence of associated cryogenic features, including crevasse-like/washboard terrain, surface lineations, polygonal terrain, and rectilinear terrain/moraine-like ridges (Fig. 12). All of these features have been found in association with glacier-like forms in previous work (e.g., Arfstrom and Hartmann, 2005; Morgan et al., 2009; Hubbard et al., 2011; Hubbard et al., 2014), suggesting that the cirque-like alcoves represent a stage of glacier-like form evolution, potentially as the glacier-like forms degrade to a landform similar to rock glaciers and contain less ice. In addition, we observe notches which enlarge and join to form larger alcoves (Fig. 15). The cirque-like alcoves also have a southward aspect bias, signifying that insolation has played a role in regulating where they have developed. Since both the morphology of the notches and the southward aspect bias for the cirque-like alcoves correspond to gullies today, this may suggest that the formation of cirque-like alcoves may be linked to gullies, for example if ice were to accumulate in the gullies.



While other topographic depressions such as landslide scarps and amphitheater-headed valleys eroded by either
groundwater seepage or outburst flooding have similarities to cirques, the cirque-like alcoves have morphometrics and directly
associated glacial landforms which suggest that glacial erosion has been the primary driver in their development (Section 5.6).
As on Earth, future work is necessary to better investigate the interaction between incipient hollow formation and later glacial
cirque development.
Using the surface gravity of Mars, extrapolating model estimates for obliquity excursions to the early Amazonian,
and assessing conditions for both warm- and cold-based glaciation, we provide estimates for the amount of time that the cirque-
like alcoves of median height would take to form in Deuteronilus Mensae (Section 5.4). Assuming a wet-based glacier with
an erosion rate of ~160 m/Myr, it would take ~13 Myr. For a cold-based glacier with an erosion rate of ~3.6 m/Myr, it would
take ~560 Myr. The timescale of a wet-based glacier is consistent with age constraints for glacier-like forms and lobate debris
aprons, while the timescale of a cold-based glacier is consistent with only the older lobate debris aprons.
The cirque-like alcoves in Deuteronilus Mensae have a median length and width  ~30% larger than the average length
and width of cirques on Earth (Table 4), which may suggest that cirques in Deuteronilus Mensae underwent more or longer
episodes of erosive glaciation than cirques on Earth. The largest cirques are in the lower latitudes of the study region at 40-
42.9°N (Fig. 8a). This likely suggests cirque-like alcove formation during a period of high obliquity when conditions were
more favorable for glacier growth at these latitudes.
The cirque-like alcoves in our population have a south to southeastward bias in aspect (Fig. 11). The slight eastward
bias is consistent with both glacier-like forms all across Mars (e.g., Souness et al., 2012; Brough et al., 2019) and climate
modeling of westerly winds in Deuteronilus Mensae (Madeleine et al., 2009). Similarly, terrestrial cirques are also found to
preferentially face eastward due to westerly winds. Future work could help to better understand the atmospheric controls on
cirque-like alcove formation in Deuteronilus Mensae, as well as other locations on Mars. To explain the southward bias in the
aspect of the cirque-like alcoves (Fig. 7), we proposed that this may be due to either poleward facing slopes receiving higher
insolation and warmer summer daytime temperatures during high obliquity (>45°) and/or an association with gully formation,
since gullies also preferentially face the equator for slopes poleward of 40° (Harrison et al., 2015; Conway et al., 2018). Above
46.5°N, more alcoves have a southward bias (Fig. 9a), consistent with alcoves requiring some kind of meltwater to form.
We constrained our dataset to a total of 386 cirque-like alcoves in the study area, where only 74 glacier-like forms
had been previously mapped (Brough et al., 2019). Thus, if cirque-like alcoves are indeed glacially eroded, we greatly extend
what we know about the extent of kilometer-scale glaciation in the region. And, future work could extend the mapping of
cirque-like alcoves to other areas. For both glacier-like forms and cirque-like alcoves, the largest are in the southeast part of
the study region (Fig. 10). This suggests that there may be an interdependent relationship between glacier-like form size and
cirque-like alcove size. In evaluations between volume and aspect, we found that the cirque-like alcoves with the greatest
volumes faced north and southwest (Fig. 11bii), thus volume did not correspond to the most common aspect. Similarly, glacier-
like forms with the greatest volume faced northwest and south (Fig. 11bi), though this may be due to a localized topographic
effect since overall for the northern hemisphere, glacier-like forms flowing northward are larger than those flowing southward.



While the aspect corresponding to the greatest volume was slightly offset between glacier-like forms and cirque-like alcoves,
we note that generally south-facing glacier-like forms and cirque-like alcoves have greater volume. Again this may represent
previous conditions with higher obliquity and/or a requirement for slopes that face the equator and receive more insolation
and therefore experience greater meltwater production.
The presence of glacial geomorphic features, especially overdeepenings (Fig. 6), surface lineations, and moraine-like
ridges (Fig. 12) are all consistent with wet-based erosion. In addition, the presence of icy remnants with surface structures that
appear similar to structures on rock glaciers on Earth suggests a glacial origin for the icy remnants on Mars (Fig. 14). While it
is possible that cold-based glaciation had a role in initiating the hollows, we find that warm-based glaciation is the more likely
surface process for major formation and growth of cirque-like alcoves in Deuteronilus Mensae on Mars.

**Data availability**

We will include the shapefile and spreadsheet of the cirque-like alcoves mapped in this study. For review, the spreadsheet is
available here: https://drive.google.com/drive/folders/1bL3GGsEvHqnbJ_FO1hh-w_HIeObOQwL-?usp=sharing. The HRSC
DEM was mosaicked using the following 29 Level 4 HRSC data frames: h5436, h5418, h5400, h5364, h5339, h5328, h5321,
h5310, h5303, h5285, h5267, h5249, h5231, h5213, h3304, h3293, h3249, h3183, h2191, h1644, h1622, h1571, h1289, h1395,
h1461, h1450, h1428, h1483, and h1201. The Level 4 HRSC data frames can be accessed at the ESA Planetary Science
Archive: http://www.rssd.esa.int/index.php?project=PSA, HRSCview by FU Berlin/DLR: http://hrscview.fu-berlin.de/, or the
NASA Planetary Data Science (PDS) http://pds-geosciences.wustl.edu/missions/mars_express/. The CTX mosaic is available
through ArcGIS Pro by selecting "Portal" and selecting "Mars CTX V01" or for download at the Murray Lab website:
https://murray-lab.caltech.edu/CTX/. HiRISE frames were accessed from the University of Arizona's HiRISE website:
https://www.uahirise.org/hiwish/browse and are also available through the PDS. The CTX DEM used in Figure 6 was made
by Mackenzie Day's GALE lab at UCLA by request and is publicly accessible here: https://github.com/GALE-
Lab/Mars_DEMs.

**Author contribution**

Project conceptualization by MRK and AYL with funding obtained by MRK. Methodology development by AYL, MRK, and
SB. Mapping, classification, initial analyses, and figures were created by AYL. All authors contributed to discussions of the
interpretations. All authors also revised and approved the submitted manuscript.

**Acknowledgments**

AYL and MRK acknowledge funding from NASA SSW 80NSSC20KO747. We thank Anjali Manoj for her work on alcove
classifications. We are very grateful to Mackenzie Day and the GALE lab at UCLA for making CTX DEMs.



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

Availability of subsurface water-ice resources in the northern mid-latitudes of Mars. Nature Astronomy, 5(3),
230-236, 2021.

Mustard, J. F., Christopher, D. C., and Moses, K. R. Evidence for recent climate change on Mars from the identification of
youthful near-surface ground ice. Nature, 412(6845), 411, 2001.

Neukum, G., Jaumann, R., The HRSC Co-Investigator and Experiment Team, HRSC: the high resolution stereo camera of
Mars Express. In: Wilson, A. (Ed.), Mars Express: The Scientific Payload, 1240. European Space Agency Special
Publication, pp. 17–35, 2004.

Oliva, M., Andrés, N., Fernández-Fernández, J. M., and Palacios, D. The evolution of glacial landforms in the Iberian
Mountains during the deglaciation. In European Glacial Landscapes (pp. 201-208), 2023.

Palucis, M. C., Jasper, J., Garczynski, B., and Dietrich, W. E. Quantitative assessment of uncertainties in modeled crater
retention ages on Mars. Icarus, 341, 113623, 2020.

Pilorget, C., and Forget, F. Formation of gullies on Mars by debris flows triggered by CO2 sublimation. Nature Geoscience,
9(1), 65-69, 2016.

Plaut, J. J., Safaeinili, A., Holt, J. W., Phillips, R. J., Head III, J. W., Seu, R., Radar evidence for ice in lobate debris aprons in
the mid-northern latitudes of Mars. Geophysical Research Letters, 36(2), 2009.

Rudberg, S. Multiple glaciation in Scandinavia: seen in gross morphology or not? Geogr. Ann. Ser. A Phys. Geogr. 74,
231–243, 1992.

Sanders, J. W., Cuffey, K. M., MacGregor, K. R., and Collins, B. D. The sediment budget of an alpine cirque. GSA Bulletin,
125(1-2), 229-248, 2013.

Selby, M. J., and Wilson, A. T. Possible Tertiary age for some Antarctic cirques. Nature, 229(5287), 623-624, 1971.
Sharp, R.P., 1973. Mars: Fretted and chaotic terrains. J. Geophys. Res. 78, 4073–4083.
Shean, D. E., Head, J. W., and Marchant, D. R. Origin and evolution of a cold-based tropical mountain glacier on Mars: The
Pavonis Mons fan-shaped deposit. Journal of Geophysical Research: Planets, 110(E5), 2005.

Sholes, S. F., and Rivera-Hernández, F. Constraints on the uncertainty, timing, and magnitude of potential Mars



oceans from topographic deformation models. Icarus, 378, 114934, 2022.

Squyres, S.W. Martian fretted terrain: Flow of erosional debris. Icarus, 34, 600– 613, 1978.
Squyres, S. W. The distribution of lobate debris aprons and similar flows on Mars. Journal of Geophysical Research:

Solid Earth, 84(B14), 8087-8096, 1979.

Souness, C., Hubbard, B., Milliken, R. E., and Quincey, D. An inventory and population-scale analysis of martian

glacier-like forms. Icarus, 217(1), 243-255, 2012.

Souness, C. J., and Hubbard, B. An alternative interpretation of late Amazonian ice flow: Protonilus Mensae,

Mars. Icarus, 225(1), 495-505, 2013.

Spagnolo, M., Pellitero, R., Barr, I. D., Ely, J. C., Pellicer, X. M., and Rea, B. R. ACME, a GIS tool for automated

cirque metric extraction. Geomorphology, 278, 280-286, 2017.

Williams, K. E., Toon, O. B., Heldmann, J. L., McKay, C., and Mellon, M. T. Stability of mid-latitude snowpacks on

Mars. Icarus, 196(2), 565-577, 2008.

Williams, K. E., Toon, O. B., Heldmann, J. L., and Mellon, M. T. Ancient melting of mid-latitude snowpacks on Mars as

a water source for gullies. Icarus, 200(2), 418-425, 2009.

Williams, J. M., Scuderi, L. A., McClanahan, T. P., Banks, M. E., and Baker, D. M. Comparative planetology–Comparing

cirques on Mars and Earth using a CNN. Geomorphology, 440, 108881, 2023.

Woodley, S. Z., Butcher, F. E., Fawdon, P., Clark, C. D., Ng, F. S., Davis, J. M., and Gallagher, C. Multiple sites of

recent wet-based glaciation identified from eskers in western Tempe Terra, Mars. Icarus, 386, 115147, 2022.

Wray, J. J. Contemporary liquid water on Mars?. Annual Review of Earth and Planetary Sciences, 49, 141-171, 2021.