# Peer review of "Cirque-like alcoves in the northern mid-latitudes of Mars as evidence"

_EGUsphere, 2023_

## Referee Comment (RC1)

General Comments.

This manuscript presents geomorphic measurements of alcoves associated with martian glacial terrains and interprets the morphometric data and HiRISE observations of the sites to describe a sequence of glacial erosion processes. The alcove measurements are novel, interesting, and exceptionally important for understanding glacial and permafrost processes on Mars.

However, the manuscript has several critical weaknesses that need to be addressed.

1) One of the foundational assumptions presented in this paper is that glacial cirques require wet-based glaciation to form. While this is true for temperate and mid-latitude cirques, particularly for modern alpine glaciers and for older glacial remnants, it is not universally the case. Antarctic, cold-based glaciers can produce pronounced alcoves with large reliefs, over-deepened regions where bed shear stress is maximized, and downslope export of eroded or rockfall debris—all without any evidence of wet-based or polythermal conditions. See fig below from (Mackay et al., 2014) and (Mackay and Marchant, 2017):

[Figure]

The paper would be greatly strengthened by *evaluating the hypothesis* that morphometric observations can be used to distinguish wet-based vs. cold-based glacial cirques. If wet-based and cold-based glacial cirques can't be distinguished in the Earth-derived measurements were collected as part of this study and in prior work, then there's no basis for thinking that they can be distinguished on the basis of morphometric observations on Mars. If they can be distinguished in the Earth dataset, then it is a powerful tool applying to the Mars data in order to *test the hypothesis* that the morphology of martian cirques is more similar to the morphology of wet-based cirques than to cold-based. If it's impossible to tell the difference between cold- and wet-based glacial cirques on Earth, or to

distinguish "cirque-like" versus non-cirque-like alcoves on the basis of morphometry in the Mars measurements, that's useful to know, too—it means that other tools or measurements need to be applied to figure out the thermal state of martian glacier beds.

My main concern is that from the very start of the manuscript, the wet-based model is accepted as an assumption: "Cirques are expected to form from depressions in mountainsides that fill with snow/ice and over time support active glaciers that deepen the depressions by wet based glacial erosion." If that's the only possible model, then there's nothing being tested by the hard-won measurements the team has made. More needs to be done to critically evaluate the many and interesting measurements the team has made.

2) The paper would be greatly strengthened by a full assessment of the morphometric data collected for the martian alcoves and the example cirques and alcoves on Earth. Analyses are only shown for the "cirque-like" alcoves, which were determined by a combination of morphometry and image interpretation. This begins to feel like cherry-picking of the data to describe alcoves that look right. It would be great to see summary statistics like those shown in table 4 for all 7 of the morphological groupings. A box and whiskers or violin plot with each group on it for L, W, H, size, area, would be helpful to see if the classes are different. Most statistical software packages could tell if there are significant differences between the groups based on the measured properties. It would be neat if there are—but if there aren't significant differences, it just means that either the morphometric properties being measured don't capture the variation or that the resolution of the dataset wasn't up for the task. Scoping the manuscript's conclusions based on the properties of the whole dataset is very important and without it, the results could be construed to be highly cherry-picked.

3) One of the challenges in the paper is distinguishing the alcoves from which small, superposing glacier-like forms emerge versus larger alcove which appear to be associated with large LDA/LVF glaciers. Sometimes a small GLF superposes an LDA/LVF, and they may share an alcove. But without distinguishing these two very different sizes and generations of glacial activity, it's difficult to evaluate the estimates of erosion timescales. Small GLF emerging from small alcoves that superpose LDA/LVF may very well have rapidly eroded their alcoves. But LDA are much larger and much older than GLF, and likely had longer timescales to remove material from the large alcoves they occupy. These two classes of alcoves seem to be lumped together in the median and mean height/volume datasets, and might be giving the impression that small, superposing, younger glacier/flow features have done much more geomorphic work much faster than they actually have.

Specific comments.

Lines 15-17. On Earth, cirques most commonly form due to basal erosion beneath wet-based or polythermal glaciers—but why start this analysis with this assumption? There are many cirque and arete-like landforms in environments where cold-based glaciation dominates, for example, ringing Beacon Valley in the McMurdo Dry Valleys (77.88 S, 160.58 E). Presupposing that cirques form through **wet-based glaciation is an unnecessary assumption and sets out the analysis as having something** to prove, rather than examining the morphology and seeing what conclusions can be drawn from it.

Line 22-23. Is the proposal that the cirques and the lobes are a late stage of glacier evolution, or that the cirques are? The cirques could pre-date the presently occupying lobes, I'd think.

Line 35. "Presumed" is pretty editorial in tone. It has been modeled (Hecht, 2002; Kreslavsky et al., 2008; Schorghofer et al., 2019) to be difficult, but certainly not impossible, that that episodic meltwater occurrences could have happened in the Amazonian, usually during obliquity optima, or on steep slopes, sometimes with the assistance over thermal shielding from overlying $CO_2$ ice. But widespread conditions where ice is present when surface temperatures reach the melting point is not produced in most models, especially on flat surfaces.

Line 38. (Levy et al., 2016) showed that mean erosion rates during Amazonian glaciation are several orders of magnitude higher than average Amazonian martian erosion rates and bracket erosion rates for cold-based glacial environments (which do erode their beds, just not much!)

Line 42. There's definitely evidence supporting englacial debris bands in LDA, e.g., (Butcher et al., 2023; Levy et al., 2021), but the jury is still out whether this debris is alcove-derived from rock fall or somehow entrained at the bed.

Line 57. I think the strongest evidence for the age of CCF is 300-800 Ma from (Fassett et al., 2014). The reason I thing those dates are less likely to be off from others is because they didn't count craters on the surface of glacial deposits themselves, which have been heavily reworked, instead counting on large ejecta blankets from craters.

Line 79. Coordinates are usually reported as lat/long—is there a reason to report location as long/lat here?

Line 83. What does it mean to be analogous here? Formed by wet-based glaciation? Formed by subglacial erosion? Formed by evacuation of rockfall from over-steepen cliffs? There's lots of ways these alcoves and cirques could be similar—in what ways are they being compared?

Lines 85-86. See comment for line 15-17. Strong analog studies require open-mindedness to the idea that physical conditions may be different between alpine, polar, and planetary settings. I think the best planetary science emerges from framing our Earth-bound heuristics as a question: on Earth, many (though not all) cirques form through wet-based erosion at the point of maximum basal shear stress; how does the morphology of martian alcoves from which glaciers emanate compare to the morphology of terrestrial cirques in cold-based and wet-based environments?

Line 87. I think the cirques/alcoves you show in Fig 2, and the many glacial valleys/cirques in the Antarctic argue against the idea that cold based glaciers don't carve alcoves/cirques. It's true that most alpine glaciers are wet-based or polythermal and that the erosion rates produced by these glaciers are higher, but the slower rate does not necessarily mean cold-based glaciers don't carve cirques. Put another way, can you show based on your terrestrial analog examples that there's a substantiative difference between the alcoves that cold-based glaciers flow from versus the cirques that are carved by wet-based glaciers? That would make for a really interesting basis for comparison for understanding what attributes of the martian alcoves are more like wet-based cirques or more like cold-based non-cirque alcoves.

Fig. 2. The DEM served by Google Earth is coarse spatial resolution and is also smoothed in places to improve image draping. It's not the most reliable dataset for geomorphic measurements. For many of your global sites, SRTM (Shuttle Radar Topography Mission) data are available and are certainly not coarser in spatial resolution, but have the advantage of being traceable to a particular

dataset with known positioning uncertainties. For alcoves in the US, the National Elevation Dataset (NED) would be a more traceable and higher resolution dataset for measuring cirques. It is served up for easy access by the USDA Geospatial Gateway (https://datagateway.nrcs.usda.gov/). For the Antarctic examples, the REMA reference elevation model for Antarctica might be a good bet for higher spatial resolution topography (served up by the Polar Geospatial Center at http://pgc.umn.edu). The best way to measure cirque properties in places like Beacon Valley or the high Asgards/Olympus range in the Antarctica would be to use the lidar data for those sites, distributed in DEM format by OpenTopography.

Lines 91-94. Given that cirques are most commonly recognized when they are empty of ice or nearly empty of ice, what measurement challenges exist for mapping martian alcoves that still have ice and/or debris in them? Are there ice-free cirques on Mars in which an over-deepened depression (a Mars tarn?) could be identified?

Lines 100-101. Is it also possible that the alcoves formed slowly during the ~500 My of large-scale (LDA and CCF-producing glaciation) described by (Fassett et al., 2014)? We don't usually have multi-million-year glacial erosion events on Earth—so the timescales on Mars may be different from our terrestrial expectations.

Fig. 3. What projection was used for data analysis in the Deuteronilus Mensae region? The craters in Fig. 3 look foreshortened, like they are being plotted in a geographic coordinate system. For length measurements, locally projected coordinates will produce less projection error, which can be large at middle to high latitudes.

Fig 1 and Fig 3. I'm a little confused by the selection criteria for branching cirques. There's several features that look like they could be mapped as branching cirques in Fig. 1, but they are not tagged as cirques because they have GLF in them. But many of the features marked as candidate cirques in Fig. 1 have LDA or remnant LDA or debris aprons of some kind in them, too. So none of these candidate cirques are empty in the sense that deglaciated cirques in places like the Uintas or the English Lake District are empty. Is it that the GLF-filled alcoves are too small/narrow to map with HRSC DTMs? That's a different issue from what's reported, which is that GLF-filled cirques were not mapped.

Line 149. The classification of the cirques seems very subjective—especially distinguishing crater-like cirques from other rounded alcoves. Do crater-like cirques show ejecta? Is it possible that all the cirques initiated a craters where impact damage broke up rock, allowing for easier down-slope export by the glacier? Craters commonly have internal landslides—could that explain some of the stepped cirques or the features interpreted later to be active layer detachments?

Line 149. Having the classification scheme in the Methods feels a little bit misplaced. Classification is interpretation, so it really could be in the Discussion section. Are there natural breaks in the measured morphometry of the cirques that leads to their classification in certain ways? Are the morphological classes of the cirques that are based on inspection distinguishable from one another in the morphometric measurements?

Line 163. If the description in Table 2 is what is used to define simple alcoves on Mars based on simple cirques on Earth, those are not morphometrics (i.e., things with lengths, volumes, slopes, etc. that are measured)—"having an armchair shape with a defined headwall, two sidewalls, and are open

downslope" is a qualitative description of the landform. I think it is not accurate then to say "simple alcoves have morphometrics consistent with simple cirques on Earth." It would be more accurate to say, "Like simple cirques on Earth, simple alcoves on Mars have an armchair shape with an identifiable headwall, two sidewalls, and an opening downslope."

Line 163-164. This is the methods section—no data have been presented or analyzed—but a conclusion is being reported: "Herein, we use the term "cirque like alcove" for these martian alcoves that are the most likely candidate cirques." That's a clear indication that this should be in the discussion section, after measurements and results have been presented. This is especially concerning because the introduction implies that "cirque-like" in the context used in this paper has a genetic meaning, i.e., carved by wet-based glaciation. At this point in the manuscript, there is not enough evidence to support that claim. If "cirque-like" just means "an eroded alcove that hosts the accumulation zone of a glacier," it's less problematic to use it here, since these are alcoves, many of which host landforms interpreted by debris-covered glaciers. But if an origin is implied by "cirque-like," more evidence needs to be marshalled to show that these shapes are exclusively consistent with wet-based glaciation.

Fig. 4. It looks like the spatial resolution of the DEM profiles is very different between these alcoves. Is it a different dataset for each of them?

Fig. 4. What is the purpose of plotting the alcove profiles? To show that they open downslope? Put another way, if someone presented you with a profile of a random alcove in the collection, would you be able to use just the elevation profile to figure out its classification?

Lines 192-193. The use of the morphological classifications suggests that the morphometry does not discriminate well between the observational morphological classes. How many simple alcoves wouldn't have been classified as cirque-like if you just used the morphometrics? How many other types of alcoves would have been classified as "cirque-like" if you hadn't excluded them based on the classification?

Line 192. How many cirques from cold-based glacier sites are in the Barr & Spagnolo (2015) database? Put another way, could these morphometric means and ranges distinguish alcoves formed in cold-based sites from warm-based sites? Are they diagnostic of one process or another?

Lines 193-195. This seems like it's a result, not a method section sentence.

Line 223. It shows a tremendous eagerness to find similarities between the martian alcoves and the terrestrial cirques by grouping results of the measurements in with comparison to their counterparts on Earth. It might be more clear to report the novel measurements for Mars in the results and then compare them to the properties of cirques on Earth in the Discussion section.

Line 224. Again, no results shown yet, and more conclusions: "Focusing on cirque-like alcoves only because they are the most likely candidate cirques." Inferences, interpretations, and classifications belong in the discussion section, and should be framed as inferences based on the data analysis, not based on assumptions made going into the measurements.

Line 229. It would be best to show all the DEM-based measurements, not just the measurements for the alcoves that were assumed to be most Earth-like. It would be great to see summary statistics

like these for all 7 of your morphological groupings. A box and whiskers or violin plot with each group on it for L, W, H, size, area, would be helpful to see if the classes are different. Most statistical software packages could tell you if there are significant differences between the groups based on the measured properties. It would be neat if there are—but if there aren't significant differences, it just means that either the morphometric properties being measured don't capture the variation or that the resolution of the dataset wasn't up for the task. Scoping your conclusions based on the properties of the whole dataset is very important and without it, the results could be construed to be highly cherry-picked.

Line 236. "Size" is confusing here—it only gets defined later in the paper. Please define it in the main text and not just in the caption.

Lines 235-236. Is this trend in all alcoves or just the narrowed down subset?

Line 243. When doing orientation analyses, it's important to normalize to the availability of host slopes. Are S and SE-facing alcoves abundant because they really form preferentially in that direction, or are there just a lot of E-W slopes in the study area and alcoves will form normal to the slope they are seeded on. Can the rose diagram be normalized based on a sampling of slopes in the study area?

Fig. 8. Binning the data seems like losing resolution. Could these plots be rendered as scatterplots to show the trend more clearly? If the inference is that there's a correlation between these geographic attributes like latitude and elevation, it would be possible to compute a correlation coefficient using the unbinned data.

Fig. 9b. This seems like another case where it's important to check whether the study site increases in elevation towards the north. A random sampling of elevation points in the study area should help normalize to the available elevations for alcoves to see if the alcoves are over- or under-represented at high or low elevations.

9c. Likewise, is the mesa heigh limiting cirque height in the study area? All those fretted terrain blocks are about the same elevation above the surrounding plains. Do alcoves cut all the way down through them to plains level? Is there something else that's limiting incision of the alcove into the mesas like internal layering? The change in alcove height with latitude is neat, but it's important to know if the mesas get shorter/taller with latitude, or if it really is something to do with the alcoves.

Line 269. If preserved GLF are pointed north, but alcoves are pointed south, doesn't that argue that whatever produced alcoves is driven by being equator-facing (warm at low obliquity or cooler at high) and whatever drives GLF formation is driven by being pole-facing (cold at low obliquity and warmer at high)? So, maybe thermal cycling matters more for alcoves and ice accumulation matters more for GLF? It just seems like so many of the alcoves mapped in this study are associated with much larger, much older LDA and LVF, that the connection to GLF seems very tenuous. It's like looking at the McMurdo Dry Valleys and inferring a causal link between the Miocene fjords cut by ice sheet draining outlet glaciers and the small alpine glaciers that occupy the valleys today—except the time scale may be more directly comparable to linking LGM glaciation to Proterozoic snowball Earth glaciation (which is what the age difference between the alcoves and the GLF may be).

Line 307. In most places where gullies and glacier-related alcoves are present and have been mapped (Dickson et al., 2023; Dickson and Head, 2009), they postdate the LDM which postdates the alcoves (based on superposition relationships). The hypothesis that gullying could provide seed points for alcove formation is intriguing and implies cyclicity to ice accumulation and melt, but there's not evidence to support that interpretation that has been presented here.

Line 322. One interesting thing we saw in (Levy et al., 2021) was that the number of boulder bands on LDA (inferred to be internal debris bands) increases with latitude—so further north sites seem to meet threshold conditions to both start and stop glaciation more often than low-latitude sites. So it's interesting that the alcoves get smaller closer to the pole. That suggests that glaciation on Mars is very inefficient at moving rock, especially at higher latitudes, where even though glaciation spins up more over the course of the Amazonian, it doesn't move more rock than low latitude sites.

Line 332-333. This does not follow. Not all alcoves have GLF emanating from them, and not all GLF emerge from mapped alcoves. So how can a correlation be drawn between them?

Line 342. Again, this assumes that GLF and alcoves are contemporary processes, which is an assumption not supported by the measurements here. All these alcoves fall within the zone over which large debris-covered glaciers (LDA, LVF) are found, so there's not a need to expand the area over which glacial activity occurred. There is not a causal or time link shown between alcoves and GLF, so I don't understand the basis for inferring that there is a much larger area over which GLF were active but are somehow vanished now.

All Figures. It would be helpful to show the alcoves that were digitized using hollow polygons and/or dashed lines. In Fig. 5, the opaque polygons cover up details that would help a reader understand the extent and morphological features in the alcove zone. In Fig. 4 the mapped alcoves are absent from the figures. It would be very helpful to see what was being considered.

Line 348. Presenting new geomorphic observations seems like results. Can this be moved from the discussion section to the results section?

Line 382. Frost heave (needle ice formation via cryosuction) is a mechanism to produce sorted patterned ground on Earth, but it is not a mechanism that is needed to create thermal contraction crack polygons of any type.

Lines 367-375 and Line 428. It seems unlikely that the downslope surface lineations shown here atop the mantling terrain that superposes the bedrock of the cirque resulted from subglacial erosion beneath the glacier that carved the alcove. The ridges superpose a younger mantle unit that in turn superposes the alcove. Neither the stratigraphic relationships, nor the relative ages of the deposits support the interpretation that the ridges formed subglacially.

Line 402 and Fig 14. These ridges are really interesting, but also very mysterious. I think we don't know what those ridges, which appear within LDA, near the toes of LDA, and sometimes along the edges really are. They could be drop moraines or medial moraines (Baker and Carter, 2019). They could also be some kind of internal thrusting feature (Stuurman, 2017). They could be eskers (Butcher et al., 2021), or internal debris bands outcropping (Levy et al., 2021). So it's best practice to not give them a genetic name when a descriptive one will do. Unless there's new evidence here suggesting they are moraines, it would be best to call them transverse or flow-parallel ridges and

then to use the observations to help evaluate their origin. What seems most important here is that these ridges are on or in the top-most mantling unit that superposes the bedrock alcoves. So it's stratigraphically some of the most recent material present. Inferring moraine deposition based primarily on ridge sinuosity seems like insufficient evidence. Is there a different in lithology, color, grain size—any other evidence that might help shape the interpretation?

Lines 417-419. Is the area/ridge size relationship something seen in this study? The alcove measurements made in this paper seem very well suited to evaluate this previously reported trend.

Line 483. What does it mean for these blobs to be glacial remnants? Does that imply shrinking of a previously expansive glacier that accumulated and flowed? Or is it possible they are dead ice fields that never grew large enough to deform and flow? How could the two be distinguished?

Lines 492-493. Many assumptions go into using Earth-based erosion rates to estimate the duration of processes on Mars, many of which could be orders of magnitude off in scale. Is it possible to use evidence from Mars to constrain the timescales through crater counting, or published age estimates for mantles, LDA, etc.?

Line 504. At Mars temperatures and ice grain sizes and dust contents, ice can be quite brittle and resistant to flow, see (Milliken et al., 2003). How important is the A parameter in this Earth-derived model?

Line 508. In the mapping, h represents the relief of the alcove—it's maximum minus its minimum elevation. Why is it used here to stand in for ice thickness? In many modern alcoves or cirques, ice thickness is quite a bit thinner than the relief of the headwall. See radar cross sections from (Mackay and Marchant, 2017; Petersen et al., 2020). h is a kind of "bank full" flow, but it's not the maximum possible ice thickness, nor the minimum.

Line 513. $K_g$ seems like an important parameter, too. Are there reported $K_g$ values for basalt? How much do Kg and l values vary? Could the erosion amounts be framed in terms of that range of possible inputs?

Line 538. This ~500 My timescale is pretty consistent with the duration of LDA and CCF-forming glaciation based on the (Fassett et al., 2014) crater counts. That's very neat!

Lines 546-547. Is it possible that erosion of the supra-glacial debris is what carved or evacuated the alcove?

Lines 573-577. Taking a look at the steep, concave shape, and over-deepened basin and threshold in the (Mackay et al., 2014) radargrams, there is not evidence that those formed through wet-based glaciation. I think the heuristic that steepness, over-deepening, and concave shape only form in wet-based glaciers is incorrect and inappropriate to apply here.

Line 600. Active layers are difficult to produce on Mars during the past 10 Ma. While not impossible, especially in steep settings (Kreslavsky et al., 2008), generating saturated conditions at the base of a detachment surface might be expected to produce downslope spring features, channelized erosion, etc. Are these accessory features observed as well?

Baker, D.M.H., Carter, L.M., 2019. Probing supraglacial debris on Mars 1: Sources, thickness, and stratigraphy. Icarus 319, 745–769. https://doi.org/10.1016/j.icarus.2018.09.001

Butcher, F.E.G., Arnold, N.S., Conway, S.J., Berman, D.C., Davis, J.M., Balme, M.R., 2023. The internal structure of a debris-covered glacier on Mars revealed by gully incision. Icarus 115717. https://doi.org/10.1016/j.icarus.2023.115717

Butcher, F.E.G., Balme, M.R., Conway, S.J., Gallagher, C., Arnold, N.S., Storrar, R.D., Lewis, S.R., Hagermann, A., Davis, J.M., 2021. Sinuous ridges in Chukhung crater, Tempe Terra, Mars: Implications for fluvial, glacial, and glaciofluvial activity. Icarus 357, 114131. https://doi.org/10.1016/j.icarus.2020.114131

Dickson, J.L., Head, J.W., 2009. The formation and evolution of youthful gullies on Mars: Gullies as the late-stage phase of Mars' most recent ice age. Icarus 204, 63–86. https://doi.org/10.1016/j.icarus.2009.06.018

Dickson, J.L., Palumbo, A.M., Head, J.W., Kerber, L., Fassett, C.I., Kreslavsky, M.A., 2023. Gullies on Mars could have formed by melting of water ice during periods of high obliquity. Science 380, 1363–1367. https://doi.org/10.1126/science.abk2464

Fassett, C.I., Levy, J.S., Dickson, J.L., Head, J.W., 2014. An extended period of episodic northern mid-latitude glaciation on Mars during the Middle to Late Amazonian: Implications for long-term obliquity history. Geology. https://doi.org/10.1130/G35798.1

Hecht, M., 2002. Metastability of Liquid Water on Mars. Icarus 156, 373–386. https://doi.org/10.1006/icar.2001.6794

Kreslavsky, M.A., Head, J.W., Marchant, D.R., 2008. Periods of active permafrost layer formation during the geological history of Mars: Implications for circum-polar and mid-latitude surface processes. Planetary and Space Science 56, 289–302.

Levy, J.S., Fassett, C.I., Head, J.W., 2016. Enhanced erosion rates on Mars during Amazonian glaciation. Icarus 264, 213–219. https://doi.org/10.1016/j.icarus.2015.09.037

Levy, J.S., Fassett, C.I., Holt, J.W., Parsons, R., Cipolli, W., Goudge, T.A., Tebolt, M., Kuentz, L., Johnson, J., Ishraque, F., Cvijanovich, B., Armstrong, I., 2021. Surface boulder banding indicates Martian debris-covered glaciers formed over multiple glaciations. PNAS 118. https://doi.org/10.1073/pnas.2015971118

Mackay, S.L., Marchant, D.R., 2017. Obliquity-paced climate change recorded in Antarctic debris-covered glaciers. Nature Communications 8, 1–12. https://doi.org/10.1038/ncomms14194

Mackay, S.L., Marchant, D.R., Lamp, J.L., Head, J.W., 2014. Cold-based debris-covered glaciers: Evaluating their potential as climate archives through studies of ground-penetrating radar and surface morphology 119, 2505–2540. https://doi.org/10.1002/2014JF003178

Milliken, R.E., Mustard, J.F., Goldsby, D.I., 2003. Viscous flow features on the surface of Mars: Observations from high-resolution Mars Orbiter Camera (MOC) images 108, 5057. https://doi.org/10.1029/2002JE002005

Petersen, E.I., Levy, J.S., Holt, J.W., Stuurman, C.M., 2020. New insights into ice accumulation at Galena Creek Rock Glacier from radar imaging of its internal structure. Journal of Glaciology 66, 1–10. https://doi.org/10.1017/jog.2019.67

Schorghofer, N., Levy, J.S., Goudge, T.A., 2019. High-Resolution Thermal Environment of Recurring Slope Lineae in Palikir Crater, Mars, and Its Implications for Volatiles. Journal of Geophysical Research Planets 124, 2852–2862. https://doi.org/10.1029/2019JE006083

Stuurman, C., 2017. Ridges on martian debris-covered glaciers : deconvolving structural and climate processes.

---

## Author Comment (AC1)

*This paper explores the formation of cirque glaciers in glaciated terrains, typically forming in bowl-shaped depressions with flat floors and steep headwalls, primarily originating from the accumulation of snow and ice avalanching from upslope areas. The authors apply this concept to identify similar cirques on Mars, suggesting that cirques deepen through wet-based glacial erosion. Their detailed morphometric analysis is commendable, particularly in relating gullies with cirques, aligning with the proposal that gully heads serve as initiation points for cirques on Earth. However, there is one major point that requires clarification to enhance the paper's clarity.*

We thank the reviewer for their comments on the connection between gullies and cirque-like alcoves. We agree with the reviewer that the manuscript would benefit from further clarity in areas. We outline these changes below.

*In lines 314-317, the authors reference works by Costard et al. (2002), Williams et al. (2008, 2009), and Dickson et al. (2023b), suggesting that gully formation on Mars may be driven by meltwater resulting from increased insolation during high obliquity excursions. Later, in lines 450-452, they propose that gully incision might also be influenced by CO2 or dry mass wasting. It is unclear which gully formation mechanism the authors prefer in their work to relate to the formation of cirque-like alcoves. Can gullies formed by CO2 ice sublimation or dry mass wasting be considered initiation points for cirque-like alcoves, or is this specific to gullies formed by meltwater generation linked to higher obliquity excursions?*

The model for gullies to act as initiation points for cirque-like alcoves is not related to whether the gullies formed by CO2 ice sublimation, dry mass wasting, or meltwater generation. Cirque glaciers on Earth require some kind of topographic depression to form, possibly at the initial depth of 50-100 m (e.g., Barr et al., 2019). Cirque-like alcoves on Mars likely require an initial hollow to form, though the exact mechanism of how the hollow or gully formed does not matter. We mention gullies as one example, where the gullies could act as cold traps for snow (e.g., Dickson et al., 2023) to accumulate into glaciers.

We edited these lines in Section 5.1.2 to read as follows:

"To explain the southward bias of cirque-like alcoves, we propose that this is consistent with periods of higher obliquity >45° on Mars, when poleward facing slopes received higher insolation and summer day temperatures (Costard et al., 2002) and equator facing slopes received less insolation. As a result, southward facing alcoves in the northern mid-latitudes were more favorable for ice accumulation during periods of

high obliquity. Similarly, for regions poleward of 40° like Deuteronilus Mensae, gullies were primarily observed on on equator-facing slopes (Harrison et al., 2015; Conway et al., 2018), possibly due to the melting of ground ice during periods of high obliquity (Costard et al., 2022), though the exact formation mechanism of gullies remains unclear (e.g., Conway et al., 2019; Dundas et al.. 2022). Regardless of how gullies are initiated, they may act as a local depression in a location where water-ice precipitation could later accumulate for cirque-like alcove formation, such as if the gullies acted as a cold trap for snow (e.g., Dickson et al., 2023). For example, gullies could provide the initial concavity for a later cirque-like alcove to develop when glaciation occurs (Section 5.2.2), which is consistent with gully heads that have been proposed as initiation points for cirques on Earth (Derbyshire and Evans; 1976). However, in the case of meltwater, we note that cirque-like alcoves may prefer to reside on equator-facing slopes because this would allow for increased insolation and the chance for meltwater as temperatures increase (e.g., Pilorget and Forget; 2016; Dundas et al., 2022; Dickson et al., 2023b). We explore this potential association between gullies and cirque-like alcoves further in Section 5.2.2."

See response below to the comment "*Currently, the concepts appear mixed…*" for the edited text in Section 5.2.2 to address the reviewer's comments.

*This distinction is crucial if the authors use it to link the early stages of cirque-like alcoves to gully activity, explaining the southward bias in the aspect of cirque-like alcoves, and gully heads as initiation points for cirque-like alcoves. The authors must address the two distinct thoughts on gully formation on Mars: one linking gully formation to meltwater and the other proposing a dry origin. For the sake of clarity and the readers' understanding, the authors should make this part clear in the paper.*

One reason that the dry mass wasting origin has been deemed unlikely is because the mean gradient of gullies is lower than the angle of repose of dry material on Mars (e.g., Noblet et al. 2024). The exact mechanism that formed the gullies is not necessarily (and on Earth typically not expected to be) the same way that the cirque-like alcoves formed. We added the following sentence to reflect this in the manuscript: "We suggest that the notches are gullies and may act as initiation points for cirque-like alcoves, though the formation of cirque-like alcoves are not determined by whether the gullies formed by CO2 ice sublimation, dry mass wasting, or meltwater generation."

*Currently, the concepts appear mixed. Initially, the authors relate slope orientation preferences based on latitude, higher obliquity excursions, and insolation-driven meltwater generation with both gullies and cirque-like alcoves (e.g., lines 673-681).*

*They also suggest that gullies can initiate through sediment flow assisted by either liquid water or CO2 ice sublimation. These are two different concepts, and the authors have mixed them. Please clarify this distinction and accordingly revise the paper.*

Currently, it does not seem like the literature on gullies definitively agrees on one origin or another. Similarly, the trend that gullies shift from pole-facing to equator-facing around 40° in both hemispheres (e.g., Noblet et al., 2024) does not seem tied to a specific hypothesis. While overall equator-facing slopes are typically discussed in connection to liquid water, $CO_2$ cannot be completely ruled out either. We added the following paragraphs to Section 5.2.2 to address this comment:

"We suggest that the notches are gullies and would be able to act as necessary initiation points for ice accumulation that would later support glaciation and erosion to form cirque-like alcoves. However, the formation of cirque-like alcoves is not dependent on how the gullies are formed. Gully formation hypotheses currently include $CO_2$ ice sublimation, dry mass wasting, meltwater generation, and a combination of these factors. For example, meltwater generation is more commonly invoked for older, inactive gullies during periods of higher obliquity (e.g., Dickson et al., 2023; Noblet et al., 2024), while gullies that have been observed to be recently active invoke $CO_2$ frost, as well as dry mass wasting during frost-free seasons (e.g., Dundas et al. 2022).

While determining how gullies formed is outside the scope of this paper, we include a discussion of the current hypotheses. Dry mass wasting alone for gully formation has recently been challenged since the mean gradient of gullies is lower than the angle of repose of dry material on Mars (e.g., Noblet et al., 2024). Gullies that are either in 1) the northern hemisphere at latitudes lower than ~50° or 2) non-polar regions and are equator-facing are modeled to be inactive gullies (Roelofs et al., 2024). These inactive gullies are inconsistent with where $CO_2$ frost deposition is expected to occur on pole-facing slopes (e.g., Lange et al. 2023). For example, in the Southern hemisphere, $CO_2$ frost is only observed on pole facing slopes between 30-50°S and is not expected on equator-facing gullied slopes between 40°S and 50°S during current obliquity conditions (Noblet et al., 2024). Nevertheless, we note that $CO_2$ sublimation cannot be completely ruled out for equator-facing slopes since seasonal deposition of $CO_2$ frost at these latitudes could have been more prevalent in the past (Noblet et al., 2024). For present-day gully activity, rather than inactive gullies, sublimation of $CO_2$ is typically invoked (e.g., Dundas et al., 2022), though $H_2O$ ice melt has been suggested to occur with dusty ice (e.g., Khuller et al., 2021).

Gullies are preferentially found on terrains that have subsurface water ice (Noblet et al., 2024). It is suggested that these inactive gullies are formed by melting of ground ice during past high obliquities (e.g., Noblet et al., 2024; Dickson et al., 2023). In addition, modeling found temperatures above freezing for meltwater and gully formation

are possible during high obliquity excursions in the mid-latitudes (Costard et al., 2002; Williams et al., 2008; Williams et al., 2009; Dickson et al., 2023b). According to Dickson et al. (2023), at high obliquities of 35° in the past, meltwater is possible during the Amazonian because pressures exceed the triple point of water. Since increased surface meltwater has been linked to increased subglacial flow at the bed of polythermal glaciers (e.g., Bingham et al., 2006; Thomson and Copland, 2017), increased meltwater for glaciers would also likely lead to more wet-based glacial conditions and erosion (see Section 5.4 for arguments supporting wet-based glacier erosion). Future work is necessary to elucidate the potential relationship between gullying as initiation points for alcove formation and how that is tied to cyclicity in ice accumulation and melt."

*My other concern, which would come alongside papers' revision, is that the authors have to present evidence that can substantiate the relationship between ice meltwater and gully formation on Mars. A lot of papers published during the last decade have provide vital inputs regarding the role of CO2 ice sublimation as the main driver for gully formation on Mars. So the authors would have to present evidence for the role of water if they wish to relate meltwater driven gullies as the initiation point for cirque-like alcoves.*

The connection between cirque-like alcoves and gullies does not necessarily depend on the formation mechanism of each. Regardless of how the gullies formed, we use the southward bias, prevalence in the mid-latitudes, and geomorphic signatures for both features to propose that cirque-like alcove generation may be connected to where gullies previously existed. We agree that much of the recent literature points to $CO_2$ sublimation for active gully processes, with a few exceptions (e.g., Khuller et al., 2021). However, gullies that are either in 1) the northern hemisphere at latitudes lower than ~50° or 2) non-polar regions and are equator-facing are modeled to be inactive gullies (Roelofs et al., 2024). It is suggested that these inactive gullies are formed by melting of ground ice during past high obliquities (e.g., Noblet et al., 2024; Dickson et al., 2023). Since the southward-facing trend of the cirque-like alcoves in Deuteronilus Mensae correspond to the equator-facing aspect of these inactive gullies at these latitudes, it is possible that these cirque-like alcoves also formed during periods of high obliquity in association with the melting of water ice. We incorporated these points in the manuscript, which we included in the response to the above comment: "*Currently, the concepts…*"

**References**

Bingham, R. G., Nienow, P. W., Sharp, M. J., and Copland, L. Hydrology and dynamics of a polythermal (mostly cold) High Arctic glacier. Earth Surface Processes and Landforms: The Journal of the British Geomorphological Research Group, 31(12), 1463-1479, 2006.

Costard, F., Forget, F., Mangold, N., & Peulvast, J. P. Formation of recent Martian debris flows by melting of near-surface ground ice at high obliquity. Science, 295(5552), 110–113, 2002.

Derbyshire, E., and Evans, I.S. The climatic factor in cirque variation. In Geomorphology and Climate, 447, 494, 1976.

Dickson, J. L., Palumbo, A. M., Head, J. W., Kerber, L., Fassett, C. I., & Kreslavsky, M. A. Gullies on Mars could have formed by melting of water ice during periods of high obliquity. Science, 380(6652), 1363-1367, 2023.

Dundas, C. M., Conway, S. J., & Cushing, G. E. Martian gully activity and the gully sediment transport system. Icarus, 386, 115133, 2022.

Harrison, T. N., Osinski, G. R., Tornabene, L. L., & Jones, E. Global documentation of gullies with the Mars Reconnaissance Orbiter Context Camera and implications for their formation. Icarus, 252, 236–254, 2015.

Khuller, A. R., & Christensen, P. R. Evidence of exposed dusty water ice within Martian gullies. Journal of Geophysical Research: Planets, 126, e2020JE006539. https://doi.org/10.1029/2020JE006539, 2021.

Lange, L., Forget, F., Dupont, E., Vandemeulebrouck, R., Spiga, A., Millour, E., et al. Modeling slope microclimates in the Mars Planetary Climate Model. Journal of Geophysical Research: Planets, 128, e2023JE007915. https://doi.org/10.1029/2023JE007915, 2023.

Noblet, A., Conway, S. J., & Osinski, G. R. 2024. A global map of gullied hillslopes on Mars. Icarus, 116147, 2024.

Roelofs, L. et al. How, when and where current mass flows in Martian gullies are driven by CO2 sublimation. Commun Earth Environ 5, 1–9, 2024.

Thomson, L. I., & Copland, L. Multi-decadal reduction in glacier velocities and mechanisms driving deceleration at polythermal White Glacier, Arctic Canada. Journal of Glaciology, 63(239), 450-463, 2017.

Williams, K. E., Toon, O. B., Heldmann, J. L., McKay, C., and Mellon, M. T. Stability of mid-latitude snowpacks on Mars. Icarus, 196(2), 565-577, 2008.

Williams, K. E., Toon, O. B., Heldmann, J. L., and Mellon, M. T. Ancient melting of mid-latitude snowpacks on Mars as a water source for gullies. Icarus, 200(2), 418-425, 2009.

---

## Author Comment (AC2)

*General Comments.*

*This manuscript presents geomorphic measurements of alcoves associated with martian glacial terrains and interprets the morphometric data and HiRISE observations of the sites to describe a sequence of glacial erosion processes. The alcove measurements are novel, interesting, and exceptionally important for understanding glacial and permafrost processes on Mars.*

*However, the manuscript has several critical weaknesses that need to be addressed.*

We thank the reviewer for their in-depth comments, which we think have helped improve the manuscript, and provide responses in black text below.

*1) One of the foundational assumptions presented in this paper is that glacial cirques require wet based glaciation to form. While this is true for temperate and mid-latitude cirques, particularly for modern alpine glaciers and for older glacial remnants, it is not universally the case. Antarctic, cold based glaciers can produce pronounced alcoves with large reliefs, over-deepened regions where bed shear stress is maximized, and downslope export of eroded or rockfall debris—all without any evidence of wet-based or polythermal conditions. See fig below from (Mackay et al., 2014) and (Mackay and Marchant, 2017):*

[Figure]

*The paper would be greatly strengthened by evaluating the hypothesis that morphometric observations can be used to distinguish wet-based vs. cold-based glacial cirques. If wet-based and cold-based glacial cirques can't be distinguished in the Earth-derived measurements were collected as part of this study and in prior work, then there's no basis for thinking that they can be distinguished on the basis of morphometric observations on Mars. If they can be distinguished in the Earth dataset, then it is a powerful tool applying to the Mars data in order to test the hypothesis*

*that the morphology of martian cirques is more similar to the morphology of wet-based cirques than to cold-based. If it's impossible to tell the difference between cold- and wet-based glacial cirques on Earth, or to distinguish "cirque-like" versus non-cirque-like alcoves on the basis of morphometry in the Mars measurements, that's useful to know, too—it means that other tools or measurements need to be applied to figure out the thermal state of martian glacier beds.*

*My main concern is that from the very start of the manuscript, the wet-based model is accepted as an assumption: "Cirques are expected to form from depressions in mountainsides that fill with snow/ice and over time support active glaciers that deepen the depressions by wet based glacial erosion." If that's the only possible model, then there's nothing being tested by the hard-won measurements the team has made. More needs to be done to critically evaluate the many and interesting measurements the team has made.*

While we agree that there are cold-based glaciers in Antarctica that have an overdeepened basin, as shown in Mackay et al. (2014), we disagree that this means the overdeepened basin was eroded by that cold-based glacier rather than by an earlier warm-based phase of glaciation. Previous work has shown that most of the landscape of the Dry Valleys "had largely achieved its form in the middle Miocene" (Sugden and Denton, 2004). This is consistent with the idea that the cold-based glaciation that occurred after the middle Miocene contributed minimally to subsurface erosion (e.g., Sugden et al., 1995; Sugden and Denton, 2004; Lewis et al., 2007).

Mackay et al. (2014) proposed that the rockfall origin for the inclined debris layer (IDL) is the most likely candidate, rather than basal shearing, entrainment, or erosion, which would all likely require a polythermal regime (Mackay et al., 2014). In addition, Mackay et al. (2014) state that where there are smooth basal returns that are interpreted as bedrock, "they appear to lie well below the onset of mapped IDL, suggesting individual IDL are most probably sourced above the bed." This would mean that the entrained IDL could not subsequently erode the bed and act as the source of the overdeepening either. And, while the IDL may correspond to back or sidewall cirque retreat, they do not provide evidence for overdeepening and subglacial erosion. There is evidence from an older study on cirque morphometrics of glaciers in the Dry Valleys that indicates headwall erosion continues after the primary episode(s) of cirque basin erosion (Aniya and Welch, 1981).

In terms of addressing the hypothesis that morphometric observations can be used to distinguish warm-based vs. cold-based glacial cirques, there is currently no strong evidence that glacial cirques can be eroded mainly/only by cold-based glaciers. As mentioned above, while some contribution to erosion is expected, significant subglacial erosion requires at least a polythermal basal condition. Currently, there is no literature on morphometrics of glacial cirques eroded by warm-based vs. cold-based glaciers because glacial cirques are expected to involve some kind of warm-based or polythermal erosion. Glaciers might be eroding cirques over the timescale of over a million years from multiple phases of glaciation. However, cirques most likely experience most erosion and attain much of their present-day size during the first glacial occupation(s) (Barr et al., 2019).

The reviewer's concern is further discussed below and changes to the manuscript have been made to provide additional clarity, as specified below.

*2) The paper would be greatly strengthened by a full assessment of the morphometric data collected for the martian alcoves and the example cirques and alcoves on Earth. Analyses are only shown for the "cirque-like" alcoves, which were determined by a combination of morphometry and image interpretation. This begins to feel like cherry-picking of the data to describe alcoves that look right. It would be great to see summary statistics like those shown in table 4 for all 7 of the morphological groupings. A box and whiskers or violin plot with each group on it for L, W, H, size, area, would be helpful to see if the classes are different. Most statistical software packages could tell if there are significant differences between the groups based on the measured properties. It would be neat if there are—but if there aren't significant differences, it just means that either the morphometric properties being measured don't capture the variation or that the resolution of the dataset wasn't up for the task. Scoping the manuscript's conclusions based on the properties of the whole dataset is very important and without it, the results could be construed to be highly cherry-picked.*

Below we have text for a possible section 4.x.x and possible Figure Y to address this comment, and we appreciate the input and consideration of including all alcoves in the results. Note that the crater-like alcove would be removed due to the reviewer's later comment, but we include it here as a reference point. A similar comment related to the plot style also came up in the editor's comments prior to accepting the paper for Discussion. However, after working up this text and figures we felt that including the section and figure describing all of the alcove classifications did not advance the main goal of our study, which is to evaluate cirque-like alcoves in relation to terrestrial cirques in order to constrain their evolution. We therefore suggest we do not include the text and figure below, unless the reviewer and editor feel that this is important to the paper.

It is interesting that there appear to be different types of alcoves but we do not have a way to evaluate alcove classes against terrestrial counterparts other than the simple alcoves, unless we digitize all of the different classes as individual alcoves. And, the goal of the study is to investigate alcoves that could be candidate cirques, so reducing our initial set of all alcove forms is consistent with our goal in the paper. The classification and calculating morphometrics were designed to be two separate steps that were used together to define a "cirque-like alcove." We have made this more clear in the text as it is an important part of our analysis.

The cirque-like alcoves on Mars are the ones that we can compare to terrestrial cirques on Earth. In addition, the ACME tool that we apply to calculate the morphometrics was designed for a simple terrestrial cirque, which we mention in Section 3.2.2: "However, we note that the ACME tool is designed for classic cirques on Earth and while the tool works with complex shapes, it should not be relied on for curving, elongated features (Spagnolo et al., 2017)." We also added this sentence: "As a result, we only include morphometrics for simple alcoves from the ACME output and not the other alcove classifications." Thus, ACME is not designed for the more complex shapes we see on Mars. In addition, there are no obvious trends in Figure Y that significantly add to the content of this manuscript.

**4.x.x Morphometric observations of all mapped alcoves in Deuteronilus Mensae**

Here we provide morphometric observations of the different alcove classes. However, we note that the ACME tool is designed for classic cirques on Earth and while the tool works with complex shapes, it should not be relied on for

curving, elongated features (Spagnolo et al., 2017). For completeness we present key morphometrics for all mapped alcove classes in this section. However, the rest of the paper focuses only on cirque-like alcoves. As seen in Fig. *Ya*, L/W for all alcove classes have medians around 1, except for branching, which is closer to 1.5. The L/H medians approach 5, except channel-related and branching classes both have higher values and ranges (Fig. *y*b). W/H medians approach 5, except for the median for channel-related alcoves is closer to 10. Aspect medians are all close to 150°, though crater-like extends to a median of 210. Median slopes are mostly around 15, except channel-related and branching which are both less than 10.

[Figure]

[Figure]

**Figure Y**. **Box plots for each alcove class based on: a) L/W, b) L/H, c) W/H, d) aspect, and e) slope. Alcoves that belong in multiple classes were included in all corresponding categories.**

*3) One of the challenges in the paper is distinguishing the alcoves from which small, superposing glacier-like forms emerge versus larger alcove which appear to be associated with large LDA/LVF glaciers. Sometimes a small GLF superposes an LDA/LVF, and they may share an alcove. But without distinguishing these two very different sizes and generations of glacial activity, it's difficult to evaluate the estimates of erosion timescales. Small GLF emerging from small alcoves that superpose LDA/LVF may very well have rapidly eroded their alcoves. But LDA are much larger and much older than GLF, and likely had longer timescales to remove material from the large alcoves they occupy. These two classes of alcoves seem to be lumped together in the median and mean height/volume datasets, and might be giving the impression that small, superposing, younger glacier/flow features have done much more geomorphic work much faster than they actually have.*

While it is true that lobate debris aprons/lineated valley fill (LDA/LVF) and glacier-like forms (GLFs) are different scales and these landforms likely relate to different episodes of glaciation, the alcove mapping done here does not connect directly to glacier-like forms. For mapping the alcoves, we excluded alcoves that contained previously mapped glacier-like forms that extended out of the alcoves, and the morphometrics of GLFs have been previously evaluated (Brough et al., 2019). However, alcove mapping did include overlap with some mapped lobate debris aprons from Baker and Head (2015). We did this so that we could evaluate the morphometrics of the alcoves that do not currently contain glacier-like forms.

*Specific comments.*

*Lines 15-17. On Earth, cirques most commonly form due to basal erosion beneath wet-based or polythermal glaciers—but why start this analysis with this assumption? There are many cirque and arete-like landforms in environments where cold-based glaciation dominates, for example, ringing Beacon Valley in the McMurdo Dry Valleys (77.88 S, 160.58 E). Presupposing that cirques form through **wet-based glaciation is an unnecessary assumption and sets out the analysis as having something** to prove, rather than examining the morphology and seeing what conclusions can be drawn from it.*

To address this concern, we added this sentence after lines 16-17: "Cold-based glaciers may also contribute to headward and sidewall retreat in cirques, though there is limited evidence for their contribution to subglacial erosion." As mentioned above, the example from Mackay et al. (2014) provides evidence for back and sidewall erosion, but not subglacial erosion. This is because the inclined debris layer likely originated from a rockwall rather than basal erosion and is sourced from above the bed (Mackay et al., 2014).

*Line 22-23. Is the proposal that the cirques and the lobes are a late stage of glacier evolution, or that the cirques are? The cirques could pre-date the presently occupying lobes, I'd think.*

The proposal is that the icy remnants or "lobes" residing within the cirque basins represent a late stage of glacier evolution (the cirque basins likely predated this ice). We rephrased the abstract for clarity to the following: "We propose that the icy relicts present in some cirque-like alcoves are similar to debris-covered ice on Earth and represent a late stage of glacier-like form evolution."

*Line 35. "Presumed" is pretty editorial in tone. It has been modeled (Hecht, 2002; Kreslavsky et al., 2008; Schorghofer et al., 2019) to be difficult, but certainly not impossible, that that episodic meltwater occurrences could have happened in the Amazonian, usually during obliquity optima, or on steep slopes, sometimes with the assistance over thermal shielding from overlying CO2 ice. But widespread conditions where ice is present when surface temperatures reach the melting point is not produced in most models, especially on flat surfaces.*

We removed the word "presumed" and rephrased to the following: "Extending from the cold and dry conditions of present-day Mars, the climate of the past 3 Gyr (Amazonian Epoch; Michael, 2013) has limited to no widespread liquid water on the planet's surface (e.g., Kite, 2019)."

*Line 38. (Levy et al., 2016) showed that mean erosion rates during Amazonian glaciation are several orders of magnitude higher than average Amazonian martian erosion rates and bracket erosion rates for cold-based glacial environments (which do erode their beds, just not much!)*

We have a citation of Levy et al., 2016 in Table 1, row 1. We also added this sentence on line 39: "Previous work has estimated cold-based erosion rates of 0.1-10 m/Myr during Amazonian glaciation on Mars (Levy et al., 2016; Table 1)."

*Line 42. There's definitely evidence supporting englacial debris bands in LDA, e.g., (Butcher et al., 2023; Levy et al., 2021), but **the jury is still out whether this debris is alcove-derived from rock fall or somehow entrained at the bed.***

We added the following references in the last sentence of the first paragraph on line 43: "In addition, englacial debris bands have been found in viscous flow features, though it is not yet known if the debris is predominantly from rockfall at the headwall or entrained from the subglacial bed (e.g., Butcher et al., 2023; Levy et al., 2021)."

*Line 57. I think the strongest evidence for the age of CCF is 300-800 Ma from (Fassett et al., 2014). The reason I think those dates are less likely to be off from others is because they didn't count craters on the surface of glacial*

*deposits themselves, which have been heavily reworked, instead counting on large ejecta blankets from craters.*

We rephrased the manuscript to address this comment: "Lobate debris aprons, lineated valley fills, and concentric crater fills are estimated to range from ~10 Myr to 1.2 Gyr in age (Morgan et al., 2009; Berman et al., 2015) or when excluding potentially reworked craters between 300 and 800 Myr (Fassett et al., 2014). Glacier-like forms can superpose lobate debris aprons or lineated valley fills, suggesting polyphase glaciation (Levy et al., 2007; Brough et al., 2015), with age clusters estimated to be around 2-20 Myr and 45-65 Myr (Hepburn et al., 2020)."

*Line 79. Coordinates are usually reported as lat/long—is there a reason to report location as long/lat here?*

We changed the coordinates to lat/long.

*Line 83. What does it mean to be analogous here? Formed by wet-based glaciation? Formed by subglacial erosion? Formed by evacuation of rockfall from over-steepen cliffs? There's lots of ways these alcoves and cirques could be similar—in what ways are they being compared?*

We rephrased this sentence since Hubbard et al. (2014) did not specify if they were referring to cirques in the context of wet-based glaciation, subglacial erosion, or evacuation of rockfall from over-steepened cliffs. The context in which Hubbard et al. (2014) mention cirques is as follows: "GLFs thereby generally form in small cirque-like alcoves or valleys, appear to flow downslope between bounding side-walls, and terminate in a distinctive tongue which may or may not feed into a higher-order ice-rich terrain type."

However, for the GLFs, they do comment that the "the thermal regime of former GLFs is unknown, and the possibility of partial wet-based conditions remains unproven and their extent unevaluated." They refer to their previous work in Hubbard et al. (2011), which classified bedforms associated with a GLF as 'mound and tail' terrain and 'linear' terrain and were likened to terrestrial drumlins and MSGL, respectively. Since both of these landforms are predominantly associated with wet-based glacial conditions on Earth, these authors proposed that such conditions may have prevailed beneath GLF #948 at a time in the past when it had expanded and thickened to fill its moraine-bounded basin. This interpretation, however, was considered side by side with an alternative – not involving wet-based glaciation – based on the mound and tail and linear terrains representing degraded supra-GLF forms, in this case wind-blown dune deposits and exposed longitudinal foliation, respectively." In the Summary at the end of Hubbard et al. (2014), they state "More extensive former GLFs, and/or their predecessor ice masses, may have been partially wet-based."

Since it is an open question whether former GLFs and/or their predecessor ice masses were warm-based, we edited the line so that it now reads: "Previous work suggested that glacier-like forms are initiated in and extend out of 'cirque-like alcoves' and that it is unknown whether the glacier-like forms and/or any ice masses that came before them were wet-based (Hubbard et al., 2014)."

*Lines 85-86. See comment for line 15-17. Strong analog studies require open-mindedness to the idea that physical conditions may be different between alpine, polar, and planetary settings. I think the best planetary science emerges from framing our Earth-bound heuristics as a question: on Earth, many (though not all) cirques form through wet-based erosion at the point of maximum basal shear stress; how does the morphology of martian alcoves from which glaciers emanate compare to the morphology of terrestrial cirques in cold-based and wet-based environments?*

Thank you for the comment and we agree the environmental setting is an important part of evaluating processes responsible for landform generation. However, on Earth there is no strong evidence that cold-based glacial erosion alone is responsible for cirque erosion. While there are cirques that contain cold-based glaciers in present-day Antarctica, it is unclear how much of the cirque erosion resulted from these cold-based glaciers. Since inclined debris layers (IDL) likely originate from rockfall and not the glacier bed, the IDLs show that cold-based glaciers contribute to cirque headwall and sidewall erosion, but there is no direct evidence of their contribution to subglacial erosion. Specifically, cirques in Antarctica might have primarily developed during earlier (warm-based) phases of glaciation, as suggested for cirques in general (Barr et al., 2019). As a result, there is no clear morphological/morphometrical evidence in the literature for cirques eroded mostly/only by cold-based ice.

*Line 87. I think the cirques/alcoves you show in Fig 2, and the many glacial valleys/cirques in the Antarctic argue against the idea that cold based glaciers don't carve alcoves/cirques. It's true that most alpine glaciers are wet-based or polythermal and that the erosion rates produced by these glaciers are higher, but the slower rate does not necessarily mean cold-based glaciers don't carve cirques. Put another way, can you show based on your terrestrial analog examples that there's a substantiative difference between the alcoves that cold-based glaciers flow from versus the cirques that are carved by wet-based glaciers? That would make for a really interesting basis for comparison for understanding what attributes of the martian alcoves are more like wet-based cirques or more like cold-based non-cirque alcoves.*

We added the following text to address this comment: "In locations such as Antarctica where cold-based glaciers currently reside within cirques, it is possible that much of the cirque was eroded during earlier, warmer, phases of glaciation, i.e., in the Miocene (e.g., Sugden and Denton, 2004) or even Tertiary (Selby and Wilson, 1971). One way in which an otherwise cold-based glacier might be erosive at its base is when meltwater from the surface of the glacier refreezes at the glacier's base and incorporates loose debris into the ice (e.g., Andrews and LeMasurier, 1973). This might mean there are localized areas of basal water, so that the glacier is not completely frozen to its bed, though how much of a cirque can be eroded downwards through this process is an open question."

*Fig. 2. The DEM served by Google Earth is coarse spatial resolution and is also smoothed in places to improve image draping. It's not the most reliable dataset for geomorphic measurements. For many of your global sites, SRTM (Shuttle Radar Topography Mission) data are available and are certainly not coarser in spatial resolution, but have the advantage of being traceable to a particular
dataset with known positioning uncertainties. For alcoves in the US, the National Elevation Dataset (NED) would be a more traceable and higher resolution dataset for measuring cirques. It is served up for easy access by the USDA Geospatial Gateway ([https://datagateway.nrcs.usda.gov/](https://datagateway.nrcs.usda.gov/)). For the Antarctic examples, the REMA reference elevation model for Antarctica might be a good bet for higher spatial resolution topography (served up by the Polar*

*Geospatial Center at [http://pgc.umn.edu](http://pgc.umn.edu)). The best way to measure cirque properties in places like Beacon Valley or the high Asgards/Olympus range in the Antarctica would be to use the lidar data for those sites, distributed in DEM format by OpenTopography.*

We updated Fig. 2 to use non-Google Earth DEMs based on the references recommended by the reviewer. Figure 2 and its caption are now as follows:

[Figure]

[Figure]

**Figure 2: (a)(i)** Example of a cirque-like alcove on Mars (40.24°N, 34.48°E) (CTX mosaic; Dickson et al., 2023a) **(a)(ii)** a cirque on Earth in the Uinta Mountains (40.712°N, 110.114°W). CTX data credit: Caltech/NASA/JPL/MSSS. Earth imagery is from Google Earth including Landsat/Copernicus/U.S. Geological Survey coverage. North is toward the top of the page, unless otherwise indicated. **(b)-(d)** Examples of cirques on Earth incised into mesa topography, along with an example of a cirque profile in each. Part (i) of (b)-(d) provides an overview of the cirques in that location with an inset of

the location of part (ii). Part (ii) of (b)-(d) offers a zoomed-in view of an individual cirque. Part (iii) of (b)-(d) shows the profile of the individual cirques in part (ii). (b) Kamchatka Peninsula, Russa (58.48°N, 160.70°E). DEM data: Shuttle Radar Topography Mission (NASA SRTM, 2013), access via EarthExplorer. (c) Uinta Mountains, Utah, USA (40.74°N, 110.05°W). DEM data: National Elevation Dataset (Howat et al., 2022), access via The National Map.; (d) Transantarctic Mountains, Antarctica (80.01°S, 156.35°E). DEM data: Reference Elevation Model of Antarctica, access via the Polar Geospatial Center.

*Lines 91-94. Given that cirques are most commonly recognized when they are empty of ice or nearly empty of ice, what measurement challenges exist for mapping martian alcoves that still have ice and/or debris in them? Are there ice-free cirques on Mars in which an over-deepened depression (a Mars tarn?) could be identified?*

We added the following text to address the concern of ice and/or debris: "For both cirques on Earth (e.g., Barr and Spagnolo, 2015) and alcoves on Mars, infilling ice and debris may affect their topographic profiles and obscure any overdeepening resulting from subglacial erosion." There is a small lip in the example of the profile from Fig. 4aii) of about ~50 m at a distance of about 3 km. However, we recommend future work to obtain and apply higher resolution DEMs than HRSC to have more confidence in the results.

*Lines 100-101. Is it also possible that the alcoves formed slowly during the ~500 My of large-scale (LDA and CCF-producing glaciation) described by (Fassett et al., 2014)? We don't usually have multi-million-year glacial erosion events on Earth—so the timescales on Mars may be different from our terrestrial expectations.*

While it is possible that the alcoves could have formed slowly during large-scale glaciation at the spatial extent of LDAs, it is not known whether erosion rates on Mars would have been consistent throughout the last ~500 Myr. On Earth, Barr et al. (2019) proposed that cirques reach most of their size during the first glacial occupations and while the cirques may be occupied for over a million years on Earth, cirque growth is expected to slow during subsequent occupations. To address this comment, we edited the sentence to read as follows: "If these martian alcoves are analogous to terrestrial glacial cirques, then they may have formed either during an earlier wet-based phase of glacier-like form activity, or formed during a prior glacial cycle separate from the episodes that developed what we observe today as glacier-like forms, for example during episodes when lobate debris aprons formed." We leave further discussion of this point in Section 5.4 Estimating the timescales for cirque-like alcove erosion on Mars.

*Fig. 3. What projection was used for data analysis in the Deuteronilus Mensae region? The craters in Fig. 3 look foreshortened, like they are being plotted in a geographic coordinate system. For length measurements, locally projected coordinates will produce less projection error, which can be large at middle to high latitudes.*

The projection used for the HRSC DEM and mapping the shapefiles was Sinusoidal_Mars and the projection from the CTX mosaic was GCS_Mars_2000_Sphere.

*Fig 1 and Fig 3. I'm a little confused by the selection criteria for branching cirques. There's several features that look like they could be mapped as branching cirques in Fig. 1, but they are not tagged as cirques because they have GLF in them. But many of the features marked as candidate cirques in Fig. 1 have LDA or remnant LDA or debris*

*aprons of some kind in them, too. So none of these candidate cirques are empty in the sense that deglaciated cirques in places like the Uintas or the English Lake District are empty. Is it that the GLF-filled alcoves are too small/narrow to map with HRSC DTMs? That's a different issue from what's reported, which is that GLF-filled cirques were not mapped.*

For the most part, Levy et al. (2014) mapping of LDA didn't include the alcoves. While Baker and Head (2015) did include more of the debris entering the alcoves, they did not consistently map the debris in all alcoves. As a result, since it's not always clear why some alcoves were included and some weren't in the overall LDA mapping, we mapped alcoves individually even if previous work included LDA mapping within some of the alcoves. We excluded any cirque basins with GLFs to focus our study on alcoves without constrained valley glaciers. It is not the case that the GLF-filled alcoves were excluded because they were too small/narrow for HRSC DTMs.

*Line 149. The classification of the cirques seems very subjective—especially distinguishing crater like cirques from other rounded alcoves. Do crater-like cirques show ejecta? Is it possible that all the cirques initiated a craters where impact damage broke up rock, allowing for easier down-slope export by the glacier? Craters commonly have internal landslides—could that explain some of the stepped cirques or the features interpreted later to be active layer detachments?*

We agree that the crater-like category might seem subjective. While all crater-like alcoves had height to width ratios of 0.1-0.3, which is consistent with typical depth-to-diameter ratios on Mars (e.g., Robbins et al., 2017), this is something we can exclude in our morphometrics step. As a result, we removed the category of crater-like alcoves in the revised manuscript and put those alcoves under the simple alcove category before it is filtered for morphometrics corresponding to cirques on Earth. Most of the crater-like cirques did not show ejecta and were very degraded. While it is possible that some cirques were initiated by craters, we do not expect that all of them were, since the distribution of cirques adjacent to each other on the edges of the mesas would signify a disproportionately large number of cirques crowded together compared to the typical crater distribution on the mesas.

*Line 149. Having the classification scheme in the Methods feels a little bit misplaced. Classification is interpretation, so it really could be in the Discussion section. Are there natural breaks in the measured morphometry of the cirques that leads to their classification in certain ways? Are the morphological classes of the cirques that are based on inspection distinguishable from one another in the morphometric measurements?*

Thank you for the feedback, though having a classification scheme in the Methods section is a common practice that has been incorporated in other papers as well (e.g., Fig. 3 in Cesar et al., 2022; Table 1 in Orgel et al., 2018). We are happy to take editorial guidance if this is against the journal style.

We think there are not obvious natural breaks in the morphometrics of the cirques (see above in the response to point 2 at the beginning of the comments). We included the morphometrics as a separate step from the classification system. Since the morphometrics are derived from simple cirques on Earth and not from complex cirque shapes, we did not include the morphometrics of other classes (though we do include the example plots in the response to comment 2 above). As an

example, since the branching alcove includes alcoves offshooting from one main trunk, it becomes difficult to acquire accurate morphometrics that represent the entire shape rather than the individual parts.

*Line 163. If the description in Table 2 is what is used to define simple alcoves on Mars based on  simple cirques on Earth, those are not morphometrics (i.e., things with lengths, volumes, slopes, etc.  that are measured)—"having an armchair shape with a defined headwall, two sidewalls, and are open*
*downslope" is a qualitative description of the landform. I think it is not accurate then to say "simple  alcoves have morphometrics consistent with simple cirques on Earth." It would be more accurate to  say, "Like simple cirques on Earth, simple alcoves on Mars have an armchair shape with an  identifiable headwall, two sidewalls, and an opening downslope."*

We edited the sentence to read as follows: "Like simple cirques on Earth, simple alcoves on Mars have an armchair shape with an  identifiable headwall, two sidewalls, and an opening downslope."

*Line 163-164. This is the methods section—no data have been presented or analyzed—but a  conclusion is being reported: "Herein, we use the term "cirque like alcove" for these martian alcoves  that are the most likely candidate cirques." That's a clear indication that this should be in the  discussion section, after measurements and results have been presented. This is especially  concerning because the introduction implies that "cirque-like" in the context used in this paper has a  genetic meaning, i.e., carved by wet-based glaciation. At this point in the manuscript, there is not  enough evidence to support that claim. If "cirque-like" just means "an eroded alcove that hosts the  accumulation zone of a glacier," it's less problematic to use it here, since these are alcoves, many of  which host landforms interpreted by debris-covered glaciers. But if an origin is implied by "cirque like," more evidence needs to be marshalled to show that these shapes are exclusively consistent  with wet-based glaciation.*

On Earth, subglacial erosion of cirques is expected to be associated with either warm-based or polythermal glaciation (see comments responding to point 1 above). When we say that the cirque-like alcoves are the most likely candidate cirques, what we mean is that after narrowing down our dataset, we focus on this subset as the most likely to be cirques and focus the rest of the study on evaluating whether the cirque-like alcoves are indeed cirques based on the distribution of their morphometrics. We rephrased our sentence to address this comment: "Herein, we use the term 'cirque-like alcove' for the martian alcoves that we will evaluate in this study as candidate cirques." We also moved this sentence to the end of section 3.2.2. We are open to changing the term "cirque-like alcove" to "armchair alcoves" throughout the manuscript if this is recommended by both the reviewer and editor.

*Fig. 4. It looks like the spatial resolution of the DEM profiles is very different between these  alcoves. Is it a different dataset for each of them?*

All of the profiles should have been using the HRSC DEM. However, we redid them and found that the profile for (a) does look smoother than the earlier version.

[Figure]

*Fig. 4. What is the purpose of plotting the alcove profiles? To show that they open downslope? Put another way, if someone presented you with a profile of a random alcove in the collection, would you be able to use just the elevation profile to figure out its classification?*

The purpose of plotting the alcove profiles is to provide them as comparison points with the profiles of cirques on Earth in Figure 2. Unfortunately, at the HRSC resolution, it is not possible to use just the elevation profile to inform their classification based on the categories we have put forward from this study of alcoves in Deuteronilus Mensae. While Fig. 4aii) does have a small lip of ~50 m at about a distance of 3 km, this is close to the resolution of the HRSC DEM so we suggest that future work create and analyze higher resolution DEMs for more overdeepenings. Other slight examples of overdeepenings are also present in Fig. 4d and 4f, but at this resolution, it is also difficult to ascertain how much is a result from icy debris such as the LDA.

*Lines 192-193. The use of the morphological classifications suggests that the morphometry does not discriminate well between the observational morphological classes. How many simple alcoves wouldn't have been classified as cirque-like*

*if you just used the morphometrics? How many other types of alcoves would have been classified as "cirque-like" if you hadn't excluded them based on the classification?*

We're not sure what the reviewer's first question is referring to since simple alcoves were a morphological class and not a morphometric class. For the second question, using only the morphometrics, 229 more alcoves would have been classified as "cirque-like" for a total of 615 alcoves instead of 386.

*Line 192. How many cirques from cold-based glacier sites are in the Barr & Spagnolo (2015) database? Put another way, could these morphometric means and ranges distinguish alcoves formed in cold-based sites from warm-based sites? Are they diagnostic of one process or another?*

While the database in Barr and Spagnolo (2015) includes 56 cirques in Antarctica, which likely contain present-day cold-based glaciers, this does not mean that these cirques were formed by cold-based glaciers; it means that erosion of these cirque basins predominantly occurred during earlier phases of warm-based glaciation (see responses above). In a different case, for Northern Sweden in the Rassepautasjtjåkka massif, cirques were underneath a cold-based ice sheet and show little evidence of being modified (Barr and Spagnolo, 2015; Jansson et al., 1999). In addition, cirque size and length of glacier occupation do not seem to increase proportionally to each other (e.g., Barr et al., 2024), and currently, for terrestrial cirques, to our knowledge, there isn't work comparing the morphometrics of cirques eroded by warm-based versus cirques that may have potentially been eroded by cold-based glaciers (especially because it is difficult to measure the erosion rates of past glaciations). And, it is an open question on Earth how much of a cirque basin can be eroded by a cold-based glacier, especially since substantial subglacial erosion by a cold-based glacier has not been observed.

*Lines 193-195. This seems like it's a result, not a method section sentence.*

We edited the sentence to read as follows: "Herein, we use the term 'cirque-like alcove' for the martian alcoves that we will evaluate in this study as candidate cirques. By applying these constraints, we narrow down our dataset to 386 cirque-like alcoves from our initial mapping and classification of 2018 alcoves."

*Line 223. It shows a tremendous eagerness to find similarities between the martian alcoves and the terrestrial cirques by grouping results of the measurements in with comparison to their counterparts on Earth. It might be more clear to report the novel measurements for Mars in the results and then compare them to the properties of cirques on Earth in the Discussion section.*

To follow the reviewer's comment, we moved what was previously Section "4.1 Comparison of length, width, height of cirque-like alcoves on Mars with cirques on Earth" to the Discussion section 5.1.

*Line 224. Again, no results shown yet, and more conclusions: "Focusing on cirque-like alcoves only because they are the most likely candidate cirques." Inferences, interpretations, and classifications belong in the discussion section, and should be framed as inferences based on the data analysis, not based on assumptions made going into the*

*measurements.*

We moved this sentence to the Discussion section 5.1.

*Line 229. It would be best to show all the DEM-based measurements, not just the measurements for the alcoves that were assumed to be most Earth-like. It would be great to see summary statistics like these for all 7 of your morphological groupings. A box and whiskers or violin plot with each group on it for L, W, H, size, area, would be helpful to see if the classes are different. Most statistical software packages could tell you if there are significant differences between the groups based on the measured properties. It would be neat if there are—but if there aren't significant differences, it just means that either the morphometric properties being measured don't capture the variation or that the resolution of the dataset wasn't up for the task. Scoping your conclusions based on the properties of the whole dataset is very important and without it, the results could be construed to be highly cherry-picked.*

We included examples of box plots for the morphological groupings in response to point 2 at the beginning of this review. There are no significant differences, likely because the resolution of the dataset wasn't up to the task as mentioned by the reviewer. As mentioned in Spagnolo et al. (2017):

> "The tool is designed to work with features that agree with the classic definition of cirques, i.e., features with a sub-circular or semi-circular plan shape, that encompasses an arcuate headwall and an open down-valley extent (Evans and Cox, 1974). The tool will work with more complex shapes, but should not be relied upon for curving, elongated features, though given the definition above, such features are unlikely to be cirques."

We chose not to include this in the manuscript because the ACME tool is designed for simple alcoves and complications arise for applying it to more complex shapes, such as the "joined" and "branching" classes. For example, both of these classes have more than one alcove, but ACME would require that the alcoves are drawn individually. As a result, this would lead to an inaccurate portrayal of the morphometrics of these classes. This is also an issue on Earth, where simple terrestrial cirques are more appropriate for this automated style of analysis. We are open to further discussion on this topic in case the Editor and/or Reviewer have additional concerns.

*Line 236. "Size" is confusing here—it only gets defined later in the paper. Please define it in the main text and not just in the caption.*

We modified the main text to read as follows: "The largest mean size, defined as $(LWH)^{\frac{1}{3}}$, and area for cirque-like alcoves correspond to lower latitudes (Fig. 8a)." We also added this sentence in Methods section 3.2.2: "We also calculate the alcove size by multiplying the length, width, and height as follows: $\sqrt[3]{LWH}$."

*Lines 235-236. Is this trend in all alcoves or just the narrowed down subset?*

The trend of an average of 156.33° is for the narrowed down subset of 386 cirque-like alcoves. However, the trend for all alcoves is very similar with a mean aspect of 157.97°. In the paper, we apply the term "cirque-like alcoves" only to the narrowed down subset: "By examining the aspect of the population of 386 cirque-like alcoves, we observe a south to southeast bias with an average of 156.33° (Fig. 7)."

*Line 243. When doing orientation analyses, it's important to normalize to the availability of host slopes. Are S and SE-facing alcoves abundant because they really form preferentially in that direction, or are there just a lot of E-W slopes in the study area and alcoves will form normal to the slope they are seeded on. Can the rose diagram be normalized based on a sampling of slopes in the study area?*

Using the HRSC DEM, we found the aspect for each point and calculated the land surface percent that belonged to each aspect bin. We then divided the percent of cirque-like alcoves in each aspect bin by the land surface percent bins and got the normalized percentages. After normalizing, we found that the same southeastward trend persisted.

Below is the table of the numbers.

| Aspect Bin | Land Surface Percent | Cirque-Like Alcove Aspect Percent | Normalized Percent |
|---|---|---|---|
| Northwest | 12.01 | 3.37 | 3.41 |
| North | 15.95 | 6.48 | 4.94 |
| Northeast | 12.92 | 15.03 | 14.15 |
| East | 12.43 | 13.47 | 13.18 |
| Southeast | 10.95 | 16.06 | 17.85 |
| South | 12.21 | 25.65 | 25.56 |
| Southwest | 11.29 | 12.95 | 13.96 |
| West | 12.24 | 6.99 | 6.95 |

We also added part b) to Figure 7 and edited the caption to read as follows:

[Figure]

**Figure 7: (a) Rose diagram showing the aspect of cirque-like alcoves. Cirque-like alcove aspect averages 156.33° between the south and southeast directions. (b) Rose diagram showing the relative percentages of cirque-like alcoves in each aspect bin after normalizing by the percent of the total land surface in each aspect bin. Aspect bins are as follows: N, 337.5–22.5°; NE, 22.5–67.5°; E, 67.5–112.5°; SE, 112.5–157.5°; S, 157.5–202.5°; SW, 202.5–247.5°; W, 247.5–292.5°; NW, 292.5–337.5.**

*Fig. 8. Binning the data seems like losing resolution. Could these plots be rendered as scatterplots to show the trend more clearly? If the inference is that there's a correlation between these geographic attributes like latitude and elevation, it would be possible to compute a correlation coefficient using the unbinned data.*

We did include the plots as scatterplots in an early draft of the manuscript. We decided these were less easily interpretable and therefore did not include in the submitted (and now revised) manuscript. Below we include examples of the scatterplots.

[Figure]

*Fig. 9b. This seems like another case where it's important to check whether the study site increases in elevation towards the north. A random sampling of elevation points in the study area should help normalize to the available elevations for alcoves to see if the alcoves are over- or under-represented at high or low elevations.*

Previously, in Section 5.1.3, we mention that this is due to the topography: "At lower latitudes, the mesas are at a higher elevation relative to the basin at lower latitude, and the overall elevation decreases toward the north" (Fig. 12). We added Fig. 12 (the plots below) to the manuscript. We used the raster to point tool in ArcGIS Pro to get 41,618,659 points from the HRSC DEM. Then we found the average at each half latitude in our study region, and found the below plot Fig. 12a , which demonstrates this same trend.

[Figure]

Figure 12: Using the raster to point tool in ArcGIS Pro led to 41,618,659 points from the HRSC DEM. (a) Each point on the plot represents the mean elevation value of all the points at each half latitude in the study region. (b) The elevation difference was calculated based on the mean of the highest 10,000 points and the mean of the lowest 10,000 points at each latitude.

*9c. Likewise, is the mesa heigh limiting cirque height in the study area? All those fretted terrain blocks are about the same elevation above the surrounding plains. Do alcoves cut all the way down through them to plains level? Is there something else that's limiting incision of the alcove into the mesas like internal layering? The change in alcove height with latitude is neat, but it's important to know if the mesas get shorter/taller with latitude, or if it really is something to do with the alcoves.*

As mentioned above: "At lower latitudes, the mesas are at a higher elevation relative to the basin at lower latitude." We used the raster to point tool in ArcGIS Pro to get 41,618,659 points from the HRSC DEM. Then we rounded the points by half latitude values. The representative height of the mesas at each latitude value was calculated as the difference between the average of the highest 10,000 points and the average of the lowest 10,000 points. This led to the plot in Fig. 12b below, which shows that the mesa height does in fact limit cirque height because mesa height decreases as latitude increases.

[Figure]

Figure 12: Using the raster to point tool in ArcGIS Pro led to 41,618,659 points from the HRSC DEM. (a) Each point on the plot represents the mean elevation value of all the points at each half latitude in the study region. (b) The elevation difference was calculated based on the mean of the highest 10,000 points and the mean of the lowest 10,000 points at each latitude.

*Line 269. If preserved GLF are pointed north, but alcoves are pointed south, doesn't that argue that whatever produced alcoves is driven by being equator-facing (warm at low obliquity or cooler at high) and whatever drives GLF formation is driven by being pole-facing (cold at low obliquity and warmer at high)? So, maybe thermal cycling matters more for alcoves and ice accumulation matters more for GLF? It just seems like so many of the alcoves mapped in this study are associated with much larger, much older LDA and LVF, that the connection to GLF seems very tenuous. It's like looking at the McMurdo Dry Valleys and inferring a causal link between the Miocene fjords cut by ice sheet draining outlet glaciers and the small alpine glaciers that occupy the valleys today—except the time scale may be more directly comparable to linking LGM glaciation to Proterozoic snowball Earth glaciation (which is what the age difference between the alcoves and the GLF may be).*

We agree that it seems like the alcoves are driven by being equator-facing. On the other hand, GLFs are mostly pole-facing, which correspond to present-day conditions that are favorable for ice accumulation. This is a good point for why it is likely that the cirque-like alcoves were generated during an earlier phase of glaciation rather than by the current GLFs. Whether this earlier phase of glaciation was on the scale of valley glaciers like GLFs (a GLF predecessor) versus larger scales like the LDA is unclear. It is also possible that valley glaciers in the alcoves eventually connected with larger ice bodies like the LDA and LVF.

*Line 307. In most places where gullies and glacier-related alcoves are present and have been mapped (Dickson et al., 2023; Dickson and Head, 2009), they postdate the LDM which postdates the alcoves (based on superposition relationships). The hypothesis that gullying could provide seed points for alcove formation is intriguing and implies cyclicity to ice accumulation and melt, but there's not evidence to support that interpretation that has been presented here.*

Later in the paper in Figure 15, we further evaluate potential geomorphic links between gully and cirque-like alcove generation, though we agree that future work is necessary to establish whether there is an actual link between gullies as seed points for alcove formation and how that connects to cyclicity in ice accumulation and melt. Here, we merely mention it as a possibility based on the

similar aspects of gullies and cirque-like alcoves. We added the following sentence to address this concern: "Future work is necessary to elucidate the potential relationship between gullying as initiation points for alcove formation and how that is tied to cyclicity in ice accumulation and melt."

*Line 322. One interesting thing we saw in (Levy et al., 2021) was that the number of boulder bands on LDA (inferred to be internal debris bands) increases with latitude—so further north sites seem to meet threshold conditions to both start and stop glaciation more often than low-latitude sites. So it's interesting that the alcoves get smaller closer to the pole. That suggests that glaciation on Mars is very inefficient at moving rock, especially at higher latitudes, where even though glaciation spins up more over the course of the Amazonian, it doesn't move more rock than low latitude sites.*

That is an interesting point. We wonder though if this may also be due to less availability of material to move in the northern latitudes. At least in Deuteronilus Mensae, the mesas in the north are smaller than the mesas in the south. Again, this would be an interesting avenue for future work.

*Line 332-333. This does not follow. Not all alcoves have GLF emanating from them, and not all GLF emerge from mapped alcoves. So how can a correlation be drawn between them?*

We edited this paragraph to address the comment: "Since both the largest cirque-like alcoves and glacier-like forms are located in southeast Deuteronilus Mensae, this may indicate that there is a local factor impacting both glacier-like form size and cirque-like alcove size, or that the two are connected in how they form. For the first case where local factors have influence, this may be because of local topography that enhances the conditions for precipitation and snow accumulation. If we assume that the cirque-like alcoves were eroded by the same phase of glaciation as the glacier-like forms, then the cirque-like alcoves may not contain glacier-like forms today because their preservation became unfavorable in current obliquity conditions. In that case, conditions in the southeast of this region resulted in both the largest glacier-like forms and cirque-like alcoves. In the second case, if we instead assume that all alcoves had reached most of their current size before the glaciation cycle that brought the glacier-like forms, then the size of glacier-like forms may be limited to the initial size of the alcove that it occupies. However, the paucity of craters in the cirque-like alcoves suggests that the cirque-like alcoves have recently been eroded, which makes it less likely that the cirque-like alcoves had already existed in their present form before glacier-like forms. While we do not distinguish between these two cases in this study, we recommend future work to investigate reasons for finding the larger glacier-like forms and the larger cirque-like alcoves in the southeast part of Deuteronilus Mensae."

*Line 342. Again, this assumes that GLF and alcoves are contemporary processes, which is an assumption not supported by the measurements here. All these alcoves fall within the zone over which large debris-covered glaciers (LDA, LVF) are found, so there's not a need to expand the area over which glacial activity occurred. There is not a causal or time link shown between alcoves and GLF, so I don't understand the basis for inferring that there is a much larger area over which GLF were active but are somehow vanished now.*

We added the following text to Section 3.1: "~10% of the cirque-like alcoves included some partial coverage from lobate debris apron and lineated valley fill mapping using Baker and Head (2015) and

much less with Levy et al. (2014). It wasn't always clear how the cutoffs for the lobate debris aprons and lineated valley fills were decided on relative to where the alcoves and mesa sidewalls were, which is why alcoves were mapped regardless of where the boundaries for the lobate debris aprons and lineated valley fills were drawn."

The alcove also has lobe deposits extending out of it that superpose the LDA (e.g., Fig. 12), which means glaciation periods of alcove formation and LDA formation are not necessarily contemporaneous. We are not requiring that cirque-like alcoves were formed by GLFs, however, we are instead suggesting that they were eroded by a similar size of glaciation as the GLFs, rather than something at the scale of LDA. On Earth, most of the glacial erosion is expected to occur during small scale cirque glaciation (e.g., Barr and Spagnolo, 2015). Once the glacier extends beyond the cirque, then more subglacial erosion occurs further downvalley and and the cirque may be occupied by a cold-based, minimally erosive glacier (Barr and Spagnolo, 2015).

*All Figures. It would be helpful to show the alcoves that were digitized using hollow polygons and/or dashed lines. In Fig. 5, the opaque polygons cover up details that would help a reader understand the extent and morphological features in the alcove zone. In Fig. 4 the mapped alcoves are absent from the figures. It would be very helpful to see what was being considered.*

Fig. 4 has been updated to include the mapped alcoves:

**(a) Simple**

[Figure]

[Figure]

**(b) Joined**

[Figure]

[Figure]

**(c) Interiorly ridged**

[Figure]

[Figure]

**(d) Staircase**

[Figure]

**(e) Channel-related**

**(f) Branching**

We updated Fig. 5 with an updated hollow polygon:

[Figure]

*Line 348. Presenting new geomorphic observations seems like results. Can this be moved from the discussion section to the results section?*

The geomorphic observations were previously in the results section, however, upon the editor's comments, we moved the observations into the Discussion section. While we agree with the reviewer's comments, at this point, we will leave the geomorphic observations in the Discussion section for now since there are interpretations that are interwoven with the results.

*Line 382. Frost heave (needle ice formation via cryosuction) is a mechanism to produce sorted patterned ground on Earth, but it is not a mechanism that is needed to create thermal contraction crack polygons of any type.*

We modified the statement and removed "frost heave."

*Lines 367-375 and Line 428. It seems unlikely that the downslope surface lineations shown here atop the mantling terrain that superposes the bedrock of the cirque resulted from subglacial erosion beneath the glacier that carved the alcove. The ridges superpose a younger mantle unit that in turn superposes the alcove. Neither the stratigraphic relationships, nor the relative ages of the deposits support the interpretation that the ridges formed subglacially.*

We added the following text to address this comment: "Conway et al. (2018) interpret the surface lineations as glacial till deposits (possibly icy sediments) sourced from glacial erosion of the headwall. One interpretation of how these types of downslope surface lineations could still result from subglacial erosion despite superposing a mantle unit is by using the model that layers of dust and snow build up in the mantle over multiple obliquity cycles (e.g., Khuller et al., 2021). Applied here, this would mean that the ridges were subglacially eroded but from another layer of ice (compacted from dust and snow) that formally sat on top of the rest of what is left of the mantle unit now."

*Line 402 and Fig 14. These ridges are really interesting, but also very mysterious. I think we don't know what those ridges, which appear within LDA, near the toes of LDA, and sometimes along the edges really are. They could be drop moraines or medial moraines (Baker and Carter, 2019). They could also be some kind of internal thrusting feature (Stuurman, 2017). They could be eskers (Butcher et al., 2021), or internal debris bands outcropping (Levy et al., 2021). So it's best practice to not give them a genetic name when a descriptive one will do. Unless there's new evidence here suggesting they are moraines, it would be best to call them transverse or flow-parallel ridges and then to use the observations to help evaluate their origin. What seems most important here is that these ridges are on or in the top-most mantling unit that superposes the bedrock alcoves. So it's stratigraphically some of the most recent material present. Inferring moraine deposition based primarily on ridge sinuosity seems like insufficient evidence. Is there a different in lithology, color, grain size—any other evidence that might help shape the interpretation?*

We edited the terminology from "moraine-like ridges" to "transverse" and "flow-parallel" ridges throughout the manuscript, except in cases where we are referencing another paper that specifically used the terminology "moraine-like ridges." Unfortunately, we did not find THEMIS imagery with fine enough resolution to capture a difference in lithology or grain size for comparing the ridges to the mesa bedrock.

*Lines 417-419. Is the area/ridge size relationship something seen in this study? The alcove measurements made in this paper seem very well suited to evaluate this previously reported trend.*

This study did not examine the area/ridge size relationship. However, this would be an interesting component for a future study. The main issue is having DEMs that are high enough resolution to resolve the height of the ridges across all of the alcoves in question, which is not possible with the HRSC DEM and the HiRISE and CTX DEMs do not have coverage of all the cirque-like alcoves.

*Line 483. What does it mean for these blobs to be glacial remnants? Does that imply shrinking of a previously expansive glacier that accumulated and flowed? Or is it possible they are dead ice fields that never grew large enough to deform and flow? How could the two be distinguished?*

We are not sure what the reviewer means regarding the difference between the shrinking of a previously expansive glacier versus dead ice fields. Typically, dead ice fields in literature refer to "stagnant glacial ice where movement by glacier flow have ceased" (Schomacker 2008). There are also "glacierets" that result from disconnections of existing glaciers, which usually occurs at icefalls or steeply crevassed regions of ice (Davies et al. 2022), though the term has also been applied to perennial snowfields (e.g., Ødegård et al. 2017). "Ice patches" is a term that has been applied to both (e.g., Ødegård et al. 2017) glacierets and perennial snowfields, though these wouldn't be expected to have lobate shapes (e.g., Tussetschläger et al., 2020) like in Figure 12 and what rock glaciers typically have (e.g., Janke et al. 2013). There are also protalus lobes and ramparts, though these are not usually found in front of cirque headwalls (e.g., Harrison et al., 2007). We apply the idea that the glacial remnants are similar to rock glaciers. While there may be different pathways for rock glaciers to form, one hypothesis is that they did evolve from a clean-ice glacier that accumulated and flowed and then progressively degraded with rock/debris cover and reduced flow (Anderson et al., 2018). We added the following sentence to include the consideration of dead ice fields: "The lobes of the glacial remnants resemble a rock glacier more than the hummocky structures of dead-ice, which typically form from stagnant debris-covered glaciers (Schomacker 2008)."

*Lines 492-493. Many assumptions go into using Earth-based erosion rates to estimate the duration of processes on Mars, many of which could be orders of magnitude off in scale. Is it possible to use evidence from Mars to constrain the timescales through crater counting, or published age estimates for mantles, LDA, etc.?*

Unfortunately, likely due to both latitude dependent mantling and the slopes of the alcoves, there is a limited number of craters available. Out of the 386 cirque-like alcoves, with an area of 856 km,$^2$ there were ~100 craters, many of which are on the latitude dependent mantle. Using crater counting, we would have gotten a more accurate age for latitude dependent mantle, which has been dated before, than the alcoves. Since both the mantles superpose the cirque-like alcoves, and the LDA are seen to at least partially superpose ~10% of the cirque-like alcoves, it is possible that the more developed cirque-like alcoves attained much of their size prior to both the mantles and the LDA. However, as noted in Section 5.2.2, while the cirque-like alcoves focused on more fully eroded landforms similar to cirques on Earth, we also propose an incipient form for the depression that could host a glacier that would then go on to erode the cirque-like alcove, and that this

depression may be connected to gully formation. In addition, debris currently residing within the cirque-like alcoves and superposing the LDA (e.g., Fig. 12) and previous work showing debris sourced from the alcove headwalls (e.g., Baker and Carter, 2019) indicates that the alcoves are likely still actively eroding. As a result, we propose that the more fully developed cirque-like alcoves versus other types of alcoves reached much of their size before LDA and mantles, but now they are covered by young mantles and continue to erode, thus making it difficult to estimate their exact formation age.

*Line 504. At Mars temperatures and ice grain sizes and dust contents, ice can be quite brittle and resistant to flow, see (Milliken et al., 2003). How important is the A parameter in this Earth-derived model?*

To address this comment, we added the following text in the manuscript: "For the A parameter, since it is unknown how much the temperature of ice on Mars has fluctuated throughout the Amazonian, we use both a warm and cold ice scenario. For cold ice, while values for $A$ exist down to -50 ℃ (Cuffey and Paterson, 2010), compared to -20 ℃, we found approximately the same results for erosion rate when keeping the other factors the same. Here, we use the $A$ parameter corresponding to -20 ℃ to match the temperature near the surface of the rock glacier in Beacon Valley (Rignot et al., 2002), which is the glacier that the surface velocity we use is based on. We also do not know how the Mars temperatures have fluctuated throughout the Amazonian epoch."

In addition, to match the slowest rock glaciers on Earth, we edited the surface velocity from 8 mm/yr to 1 mm/yr: "for the cold-based case we use a surface velocity of $18 \times 10^{-3}$ m/yr, which was measured for a rock glacier in the Beacon Valley sector of the McMurdo Dry Valleys of Antarctica, (Rignot et al., 2002) which represents a low glacier flow speed on Earth."

The A parameter has a negligible effect on the ultimate value of time required for erosion, as seen in the plot below. The color represents the continuous time of erosion, which assumes constant glaciation. It is more likely that due to obliquity changes, only around 20% of the total time would include glaciation. As a result, the surface sliding velocities slower than 1e-3 m/yr become unrealistic relative to the age of Mars. The plot includes surface sliding velocities as low as 5e-5 m/yr, however, measured values on Earth have only reached as low as 1e-3 m/yr. Currently, this plot is not included in the manuscript, but we are open to including it if the reviewer suggests so.

[Figure]

*Line 508. In the mapping, h represents the relief of the alcove—it's maximum minus its minimum elevation. Why is it used here to stand in for ice thickness? In many modern alcoves or cirques, ice thickness is quite a bit thinner than the relief of the headwall. See radar cross sections from (Mackay and Marchant, 2017; Petersen et al., 2020). h is a kind of "bank full" flow, but it's not the maximum possible ice thickness, nor the minimum.*

In the mapping, we used uppercase "H" for the height of the alcove. Here, lower case "h" is for ice thickness, which is what the equation calls for (Cook et al., 2020). The equation uses ice or glacier thickness because the equation is to calculate the surface velocity of the glacier.

*Line 513. $K_g$ seems like an important parameter, too. Are there reported $K_g$ values for basalt? How much do Kg and l values vary? Could the erosion amounts be framed in terms of that range of possible inputs?*

To our knowledge, there is not a $K_G$ for basalt. $K_G$ and l are empirically derived and a range of values of the two together can produce accurate erosion rates. We used an l value of 0.69, which is lower than previous estimates that have used l = 1 or greater (Cook et al., 2020). The value of l = 0.69 or < 1, calculated based on a global dataset, implies that the erosion rate is slower than the rate of increase in sliding (Cook et al., 2020). Meserve Glacier, a cold-based glacier in the Antarctic Dry Valleys, fits within these global values. As such, we believe that the lower l value used by Cook et al., 2020 fits for the type of cold-based glacier we might expect on Mars.

The plot below demonstrates the effect of $K_G$, which is bedrock erodibility constant, and l, which is the erosion exponent, on the continuous time needed for erosion of a median cirque-like alcove. These calculations were done using the lowest A parameter for -50 $^\circ$C and lowest observed surface velocity of 1 mm/yr for rock glaciers on Earth. Values in the shaded region are all possible given a 4.5 Gyr time restraint on erosion. The blue line represents the minimum K and maximum l required. However, they are not all likely given the typical values for K and l found in previous work mentioned above. Currently, this plot is not included in the manuscript, but we are open to including it if the reviewer suggests so.

[Figure]

*Line 538. This ~500 My timescale is pretty consistent with the duration of LDA and CCF-forming glaciation based on the (Fassett et al., 2014) crater counts. That's very neat!*

This will be an interesting point to continue considering, especially if we had better constraints on the climate within the last 500 Myr. To represent an even lower cold-based erosion rate estimate, we used a different surface velocity of 1 mm/yr (previously 8 mm/yr), which led to a timescale closer to 2.3 Gyr. We added the paragraph to read as follows: "On the other hand, if we assume cold-based conditions for glaciers that occupied the cirque-like alcoves, then the erosion rate estimated from the median values for cirque-like alcoves is ~0.85 m/Myr, which is consistent with the wide-ranging estimate of 0.1-10 m/Myr for cold-based viscous flow features on Mars (Levy et al., 2016) but is lower than the Conway et al. (2018) estimates for glaciated crater walls. Thus, for a cold-based glacier, a total glacier occupation time of ~470 Myr would be required for the cirque-like alcoves to form, which is consistent with the ~500 Myr timescale of LDA- and CCF-forming glaciation based on crater counts (Fassett et al., 2014). However, accounting for obliquity variations, a median height cirque-like alcove would require ~2.3 Gyr to erode with only cold-based glaciation. If the glaciers were cold-based during their entire evolution, the erosion timescale is longer and therefore the alcoves must be much older than if they evolved during warm-based glaciation. A timescale of hundreds of millions to a billion years is in the range of when lobate debris aprons were thought to have formed, such as in Deuteronilus Mensae, the lobate debris aprons are estimated to be as old as 1.1 Gyr (Berman et al., 2015). Thus, using the slowest estimated erosion rates corresponding to cold-based glaciers in Antarctica, the alcoves would predate the lobate debris aprons. While ~ % of the cirque-like alcoves do have partially overlapping LDA, there are also cases where the debris on the LDAs are clearly sourced from the alcoves (e.g., Baker and Carter, 2019)."

*Lines 546-547. Is it possible that erosion of the supra-glacial debris is what carved or evacuated the alcove?*

It is possible that erosion of the supra-glacial debris contributed to what carved the alcove, though the amount of the contribution remains unclear. For example, Baker and Carter (2019) used THEMIS images to find trails of debris going from the mesas onto the lobate debris aprons, indicating that there is a contribution from rockfall for the debris on the lobate debris aprons. However, it remains an open question how much of the alcove erosion was created by this process versus an earlier, separate process due to glacial erosion. On Earth, Barr et al. (2019) propose that cirques attain most of their size during the initial onset of glaciation of a previously non-glacially sculpted landscape. Similarly, Bickerdicke et al. (2017) conclude that cirques exist where the land system is at the threshold of glaciation and the volume of debris in the cirques can indicate cirque modification associated with a glaciation period rather than their creation. As such, it remains ambiguous in Deuteronilus Mensae whether the cirque-like alcoves attained much of their size from earlier glaciations prior to any erosion that might be associated with lobate debris apron or later glaciations.

*Lines 573-577. Taking a look at the steep, concave shape, and over-deepened basin and threshold in the (Mackay et al., 2014) radargrams, there is not evidence that those formed through wet-based glaciation. I think the heuristic that steepness, over-deepening, and concave shape only form in wet based glaciers is incorrect and inappropriate to apply*

*here.*

Radargrams in Mackay et al. (2014) demonstrate overdeepening in the basin, however, as mentioned at the beginning of the response to reviewer, this does not necessarily mean cold-based glaciation was responsible for this overdeepening and subglacial erosion. Previous work has stated that most of the landscape of the Dry Valleys "had largely achieved its form in the middle Miocene" (Sugden and Denton, 2004). This is consistent with the survival of 8 Ma ashes on buried glacier ice in Beacon Valley (Sugden et al., 1995) and cold-based glaciation for millions of years after the middle Miocene that relatively contributed to much less erosion. In addition, Mackay et al. (2014) propose that the rockfall origin for the inclined debris layer (IDL) is the most likely candidate, rather than basal shearing, entrainment, or erosion, which would all require a polythermal regime (Mackay et al., 2014). In addition, Mackay et al. (2014) state that where there are smooth basal returns that are interpreted as bedrock, "they appear to lie well below the onset of mapped IDL, suggesting individual IDL are most probably sourced above the bed." This would mean that the entrained IDL could not subsequently erode the bed and act as the source of the overdeepening either.

We added the following sentence so that this point now reads as follows: "In addition, the type of overdeepening observed in the profiles (Fig. 6) is consistent with wet-based glacial erosion on Earth. While cold-based glaciation has been observed to contribute to headward and sidewall erosion, such as through rockfall at the headwall, a polythermal regime rather than a pure cold-based regime is likely necessary for any substantial basal erosion (e.g., Mackay et al., 2014)."

*Line 600. Active layers are difficult to produce on Mars during the past 10 Ma. While not impossible, especially in steep settings (Kreslavsky et al., 2008), generating saturated conditions at the base of a detachment surface might be expected to produce downslope spring features, channelized erosion, etc. Are these accessory features observed as well?*

A subset of the alcoves did have channels nearby or feeding into the alcove. However, these constituted a small percentage of the total alcoves mapped (0.3%) and weren't included in the cirque-like alcove analyses.

**References:**
Aniya, M., & Welch, R. (1981). Morphometric analyses of Antarctic cirques from photogrammetric measurements. Geografiska Annaler: Series A, Physical Geography, 63(1-2), 41-53.
Barr, I. D., Spagnolo, M., & Tomkins, M. D. (2024). Cirques in the Transantarctic Mountains reveal controls on glacier formation and landscape evolution. Geomorphology, 445, 108970.
Bickerdike, H. L., Ó Cofaigh, C., Evans, D. J., & Stokes, C. R. (2018). Glacial landsystems, retreat dynamics and controls on Loch Lomond Stadial (Younger Dryas) glaciation in Britain. Boreas, 47(1), 202-224.
Cesar, C., Pommerol, A., Thomas, N., Portyankina, G., Hansen, C.J., Tornabene, L.L., Munaretto, G. and Cremonese, G., (2022). Seasonal southern circum-polar spots and araneiforms observed with the colour and stereo surface imaging system (CaSSIS). Planetary and Space Science, 224, 105593.
Davies, B., Bendle, J., Carrivick, J., McNabb, R., McNeil, C., Pelto, M., Campbell, S., Holt, T., Ely, J.,

& Markle, B. (2022). Topographic controls on ice flow and recession for Juneau Icefield (Alaska/British Columbia). Earth Surface Processes and Landforms, 47(9), 2357-2390.

Harrison, S., Whalley, B., & Anderson, E. (2008). Relict rock glaciers and protalus lobes in the British Isles: implications for Late Pleistocene mountain geomorphology and palaeoclimate. Journal of Quaternary Science: Published for the Quaternary Research Association, 23(3), 287-304.

Jansson, P., Richardson, C., & Jonsson, S. (1999). Assessment of requirements for cirque formation in northern Sweden. Annals of Glaciology, 28, 16-22.

Lewis, A. R., Marchant, D. R., Ashworth, A. C., Hemming, S. R., & Machlus, M. L. (2007). Major middle Miocene global climate change: Evidence from East Antarctica and the Transantarctic Mountains. Geological Society of America Bulletin, 119(11-12), 1449-1461.

Ødegård, R. S., Nesje, A., Isaksen, K., Andreassen, L. M., Eiken, T., Schwikowski, M., & Uglietti, C. (2017). Climate change threatens archaeologically significant ice patches: insights into their age, internal structure, mass balance and climate sensitivity. *The Cryosphere*, *11*(1), 17-32.

Orgel, C., Hauber, E., van Gasselt, S., Reiss, D., Johnsson, A., Ramsdale, J.D., Smith, I, Swirad, Z.M., Séjourné, A., Wilson, J.T. and Balme, M.R., (2019). Grid mapping the northern plains of Mars: A new overview of recent water‐and ice‐related landforms in Acidalia Planitia. Journal of Geophysical Research: Planets, 124(2), 454-482.

Rignot, E., Hallet, B., & Fountain, A. (2002). Rock glacier surface motion in Beacon Valley, Antarctica, from synthetic‐aperture radar interferometry. Geophysical research letters, 29(12), 48-1.

Schomacker, A., & Kjær, K. H. (2008). Quantification of dead‐ice melting in ice‐cored moraines at the high‐Arctic glacier Holmströmbreen, Svalbard. Boreas, 37(2), 211-225.

Sugden, D.E., Marchant, D.R., Potter, N., Jr., Souchez, R.A., Denton, G.H., Swisher, C.C., and Tison, J.-L. (1995). Preservation of Miocene glacier ice in East Antarctica. Nature, v. 376p. 412–414.

Sugden, D., & Denton, G. (2004). Cenozoic landscape evolution of the Convoy Range to Mackay Glacier area, Transantarctic Mountains: Onshore to offshore synthesis. Geological Society of America Bulletin, 116(7-8), 840-857.

Tussetschläger, H., Brynjólfsson, S., Brynjólfsson, S., Nagler, T., Sailer, R., Stötter, J., & Wuite, J. (2020). Perennial snow patch detection based on remote sensing data on Tröllaskagi Peninsula, northern Iceland. *Jökull*, *69*, 103-128.

*Baker, D.M.H., Carter, L.M., 2019. Probing supraglacial debris on Mars 1: Sources, thickness, and stratigraphy. Icarus 319, 745–769. https://doi.org/10.1016/j.icarus.2018.09.001 Butcher, F.E.G.,*

*Arnold, N.S., Conway, S.J., Berman, D.C., Davis, J.M., Balme, M.R., 2023. The internal structure of a debris-covered glacier on Mars revealed by gully incision. Icarus 115717. https://doi.org/10.1016/j.icarus.2023.115717*

*Butcher, F.E.G., Balme, M.R., Conway, S.J., Gallagher, C., Arnold, N.S., Storrar, R.D., Lewis, S.R., Hagermann, A., Davis, J.M., 2021. Sinuous ridges in Chukhung crater, Tempe Terra, Mars: Implications for fluvial, glacial, and glaciofluvial activity. Icarus 357, 114131. https://doi.org/10.1016/j.icarus.2020.114131*

Dickson, J.L., Head, J.W., 2009. The formation and evolution of youthful gullies on Mars: Gullies as  the late-stage phase of Mars' most recent ice age. Icarus 204, 63–86. https://doi.org/10.1016/j.icarus.2009.06.018

Dickson, J.L., Palumbo, A.M., Head, J.W., Kerber, L., Fassett, C.I., Kreslavsky, M.A., 2023. Gullies  on Mars could have formed by melting of water ice during periods of high obliquity. Science  380, 1363–1367. https://doi.org/10.1126/science.abk2464

Fassett, C.I., Levy, J.S., Dickson, J.L., Head, J.W., 2014. An extended period of episodic northern  mid-latitude glaciation on Mars during the Middle to Late Amazonian: Implications for long term obliquity history. Geology. https://doi.org/10.1130/G35798.1

Hecht, M., 2002. Metastability of Liquid Water on Mars. Icarus 156, 373–386. https://doi.org/10.1006/icar.2001.6794

Kreslavsky, M.A., Head, J.W., Marchant, D.R., 2008. Periods of active permafrost layer formation  during the geological history of Mars: Implications for circum-polar and mid-latitude surface  processes. Planetary and Space Science 56, 289–302.

Levy, J.S., Fassett, C.I., Head, J.W., 2016. Enhanced erosion rates on Mars during Amazonian glaciation. Icarus 264, 213–219. https://doi.org/10.1016/j.icarus.2015.09.037 Levy, J.S., Fassett,

C.I., Holt, J.W., Parsons, R., Cipolli, W., Goudge, T.A., Tebolt, M., Kuentz, L.,  Johnson, J., Ishraque, F., Cvijanovich, B., Armstrong, I., 2021. Surface boulder banding  indicates Martian debris-covered glaciers formed over multiple glaciations. PNAS 118. https://doi.org/10.1073/pnas.2015971118

Mackay, S.L., Marchant, D.R., 2017. Obliquity-paced climate change recorded in Antarctic debris covered glaciers. Nature Communications 8, 1–12. https://doi.org/10.1038/ncomms14194 Mackay,

S.L., Marchant, D.R., Lamp, J.L., Head, J.W., 2014. Cold-based debris-covered glaciers: Evaluating their potential as climate archives through studies of ground-penetrating radar and surface morphology 119, 2505–2540. https://doi.org/10.1002/2014JF003178 Milliken, R.E.,

Mustard, J.F., Goldsby, D.I., 2003. Viscous flow features on the surface of Mars: Observations from high-resolution Mars Orbiter Camera (MOC) images 108, 5057. https://doi.org/10.1029/2002JE002005

Petersen, E.I., Levy, J.S., Holt, J.W., Stuurman, C.M., 2020. New insights into ice accumulation at  Galena Creek Rock Glacier from radar imaging of its internal structure. Journal of  Glaciology 66, 1–10. https://doi.org/10.1017/jog.2019.67

Schorghofer, N., Levy, J.S., Goudge, T.A., 2019. High-Resolution Thermal Environment of  Recurring Slope Lineae in Palikir Crater, Mars, and Its Implications for Volatiles. Journal of  Geophysical Research Planets 124, 2852–2862. https://doi.org/10.1029/2019JE006083

Stuurman, C., 2017. Ridges on martian debris-covered glaciers : deconvolving structural and climate  processes.

---

## Author Response (AR2)

Review by Susan Conway

The manuscript contains novel results pertaining to alcoves in the glaciated mid-latitudes of Mars that the authors argue are carved by glacial erosion akin to cirques on Earth. The contribution is worthy of publication, but the paper (and most notably, the discussion) needs to be shortened and the methods (and some of the results) expanded to better represent their work. Only the conclusions that are robustly supported by the author's collected data should be presented and the paper shortened via that mechanism. I have included detailed comments on the attached PDF and reproduced below those that require a response and either a change to the manuscript or a reason why the manuscript has not been changed (I note from the previous review that the authors made many replies to the review and yet did not necessarily change the manuscript – the reviewer's question about map projection of Fig 3 is an example, where a response was provided as a reply, but the information should also have been added to the text, so I have had to raise a similar comment in this review).

We thank the reviewer for taking the time to provide very detailed and helpful comments.

In Section 3.1, we added the following sentence about the map projection: "Measurements from all of the imagery and DEMs used a Sinusoidal projection centered on longitude 25.5 degrees East, and were based on the IAU Mars 2000 Sphere datum."

In general, the paper reads as if the authors were arguing strongly that these cirque-like alcoves were formed by the erosion of wet-based glaciers in the first version and after a round of reviews they were forced to acknowledge that their data do not allow them to conclude this (I wrote this text before looking at the previous reviews). So even though the paper does acknowledge that these cirque-like alcoves could be formed by the erosion of cold-based glaciers, it is somewhat begrudging. I encourage them to take a fresh look at the paper and try to "clean it up" by focussing on the conclusions that are best supported by the data collection effort they have undertaken. Briefly these would be in my opinion: that the alcoves they investigate are likely caused by significant glacial erosion, they are bigger than cirques on Earth, their orientation/size/distribution shows climate signal also seen in gullies and GLF globally, the timescales for formation under wet and cold based glaciation are realistic, but different (with different implications). An effort should be made to shorten the text substantially. Hence, I feel that the main concern in the previous round of reviews "My main concern is that from the very start of the manuscript, the wet-based model is accepted as an assumption" has not been fully addressed.

We have removed paragraphs of both the Introduction and Conclusion to address this concern, and deleted what was previously Section "5.5 Discussion of wet-based versus cold-based glacial erosion of cirque-like alcoves." While we have softened our language that the cirque-like alcoves must point to wet-based glaciers, we maintain the position that it remains a possibility alongside cold-based glaciation.

In summary, the main points that need addressing before publication are:
1. The discussion is very speculative and not focussed so the reader gets lost as to how the authors' work even pertains to what is being discussed. I think the authors should focus their paper to present 4-5 solid conclusions which have a direct link to the data they have collected and remove the other conclusions with associated discussion to make the paper easier to understand and read.
Following the reviewer's recommendation, we deleted sections 5.5 "Discussion of wet-based versus cold-based glacial erosion of cirque-like alcoves" and 5.6 "Possible alternative mechanisms for alcove formation using examples from Earth" so that the discussion and relevant conclusions are more straightforward.

2. That cirques on Mars indicate wet-based glaciation is going too far. Even that these alcoves are probably cirques is already enough of a finding without the authors needing to go further. The wet-based glaciation can appear where the authors consider rates of erosion and timescales, but should not form part of the abstract, the introduction and conclusions. The other parts of the discussion dealing with this are speculative and unfounded, so should be reduced/removed. See my detailed comments. This was the main concern of one of the previous reviewers.
Following the reviewer's recommendation, we greatly reduced the discussion on wet-based glaciation and instead focused on glacial erosion generally. This included deleted sections related to wet-based erosion in the introduction, section 5.5, and conclusion.

3. The comparison in the discussion to rock glaciers (and its recurrence in the abstract and conclusions) is speculative and no comparison data from Earth are directly presented to support this point. Personally, I do not see the resemblance, yet I should not need to make a personal judgement if this is a conclusion of the paper, I should have the data presented to me and be convinced by the authors' arguments. This is not the case – please see my detailed comments – so I suggest removing this comparison as it distracts from the more robust conclusions in the paper.

We deleted this section in the discussion, however, we did want to note below that rock glaciers are considered to be analogs for GLFs as well.

In addition to being compared to debris-covered glaciers, GLFs/VFFs are also compared to rock glaciers on Earth (e.g., Hubbard et al., 2011; Hubbard et al., 2014). Hubbard et al. (2014) also notes that the distinction between debris-covered glaciers and rock glaciers isn't always clear on Earth. For example, one complex glacier system can include components that are mapped as both a rock glacier and a debris-covered glacier (e.g., Janke et al., 2015; Tanarro et al. 2021).

While rock glaciers oftentimes have furrows, similar to the glacier-like form studied by Hubbard et al. (2011), the furrows are not a requirement for a rock glacier. For example, Janke et al. (2015) identifies Class 6 rock glaciers as glaciers that have deflated and contain <10% ice content. These Class 6 rock glaciers lose their defined furrows and their sharp transition from their toe to the front slope (Janke et al., 2015).

References:
Hubbard, B., Souness, C., & Brough, S. (2014). Glacier-like forms on Mars. The Cryosphere, 8(6), 2047-2061.

Hubbard, B., Milliken, R. E., Kargel, J. S., Limaye, A., & Souness, C. (2011). Geomorphological characterisation and interpretation of a mid-latitude glacier-like form: Hellas Planitia, Mars. Icarus, 211(1), 330-346.

Janke, J. R., Bellisario, A. C., & Ferrando, F. A. (2015). Classification of debris-covered glaciers and rock glaciers in the Andes of central Chile. Geomorphology, 241, 98-121.

Tanarro, L. M., Palacios, D., Fernández-Fernández, J. M., Andrés, N., Oliva, M., Rodríguez-Mena, M., Schimmelpfennig, I., Brynjólfsson, S., Sæmundsson, þ., Zamorano, J.J., Úbeda, J. and ASTER Team. (2021). Origins of the divergent evolution of mountain glaciers during deglaciation: Hofsdalur cirques, Northern Iceland. Quaternary Science Reviews, 273, 107248.

4. The methods need some clarification so the reader fully understands the data and methods used. Namely:
a) Demonstrate the difference to other alcove forming processes on Earth up front by integrating Table 6 on page 15 and removing section 5.6. This provides additional justification for the down-selection using the alcove morphometrics and frees up space in the discussion.
Following the reviewer's recommendation, we moved Table 6 to page 15, so that it is now Table 2. In addition, we deleted section 5.6. We added the following text to page 15 as well: "By using morphometrics, we also exclude other types of mechanisms for alcove formation, including active-layer detachments, deep-seated landslides, and theater-headed valleys (Table 2). This is because the H/L ratio of a terrestrial glacial cirque is expected to be deeper than any of the other alcove landforms with known morphometrics on Earth (Table 2)."

b) Clarify the data used to make the initial alcove classification (e.g., simple, joined, staircase, etc). Longitudinal profiles are included in Figure 4 which presents the classification, yet it is not explicitly stated that they have been used to inform the classification and what attributes of them were used. If they were indeed used then section

3.2.3 which speaks to the effect on the long profiles in the uncertainty of the elevation data used, makes more sense. The authors should make sure to update this section to incorporate the effect of the elevation uncertainty on each attribute they list in the methods as being critical for the classification, as well as comment c below. A similar point was raised in the previous review, but the manuscript not changed in response.
Longitudinal profiles were not used to classify the alcoves. Following the reviewer's #5 comment, we moved Figure 4 to the Supplementary Material Section as Figure S1.

c) when considering the uncertainties in the elevation data in section 3.2.3 please address how these may also affect the ACME data collection, specifically consider the noise in the HRSC product (clearly visible as step-artefacts on Figs 4 and 6), and how well the CTX and HRSC data were co-registered. Noise is accentuated in topographic derivatives such as slope, which is amongst the parameters extracted. Presumably the position of the long profile was determined based on the CTX image data (if this or is not the case then it should be described in the methods as mentioned in point b), hence co-registration is critical to have reliable and representative elevation data. Please state what projection system was used for the morphometric analyses and consider whether this introduced any uncertainties/distortion (including the slope calculation from the HRSC DTM).
We added the following text to what is now section 3.2.3 "Uncertainties in elevation and alcove longitudinal profile": We mapped the cirque-like alcove and identified the mid-threshold point using the CTX imagery. As mentioned in Section 3.1, both the CTX imagery and HRSC DEM were aligned to a Sinusoidal projection centered on longitude 25.5 degrees East, and were based on the IAU Mars 2000 Sphere datum. Any misalignment of up to 100 m between the image and the DTM is of little concern when it translates into metrics made by ACME2 since most metrics rely on multiple pixel measurements. This is certainly the case for slope, aspect and average elevation along the cirque length or the entire cirque area. Any misalignment might affect minimum and maximum elevation, but this is not a concern when using a large sample size to evaluate population-scale metrics."

d) How the different ice-related-morphologies were recognised in the HiRISE images should be explained in the methods, with references to support their ice-related-origin. The results of this work should be in the results and then this frees up space in the discussion.
We added Table 3 in the methods section 3.3 to explain ice-related-morphologies and Table S1 in the Supplementary Material Section to demonstrate the associated HiRISE frames with these features. We also provide Table 5 to demonstrate the percentage of HiRISE and CTX imagery with each type of icy geomorphic feature. The tables are included below in response to the specific comments.

5. Further, the inclusion of all the alcoves types in the methods and in the first part of the results makes the paper bulky and are unnecessary as these results are not used to support the main conclusions. I strongly suggest omitting them. In the methods it can simply be

stated that “alcoves that show any of the following morphologies [bulleted list of properties of joined, staircase, channel, etc], were not included in our database”. I understand that work was done by the authors to map these landforms, but this is not a masters’ thesis where one has to demonstrate how much work was done, and so I do not think this is adequate justification to include them in the paper.
An earlier version didn’t include these in the results but they were added back in by request from other reviewers. However, we agree with this reviewer’s suggestion, and what was previously section 4.1 describing all alcoves was deleted, along with the associated figure.

6. The objective distinction between what was previously mapped as GFL and the alcoves mapped by the authors is not clear to me. It seems that many of the alcoves mapped by the authors contain GLF missed in this previous survey and as demonstrated in the HiRISE survey many of the alcoves contain deposits that have one or more ice-related morphologies (these need to be tabulated somewhere, as noted in my detailed comments) which could be the extension of the VFF up into the alcove (I am talking about the visible extension of VFF as can be seen on images and not mapped outlines, which are never totally reliable as they are usually made using low resolution image bases suitable for global studies – not meant to be looked at “in detail”). Global surveys are often incomplete so this statement is not a criticism of the previous work. I think the back and forth discussion when comparing the GLF and cirque-like alcove distribution would become clearer if these landforms were treated as a continuum. This is a similar comment to that raised in previous review point 3 and was not addressed by the authors by a change to the manuscript.
We addressed these comments by splitting the discussion section up into three sections in the methods (3.3), results (4.3), and discussion (5.2.1). The associated text, tables, and figures are included in the response to comments for page 33.

Detailed comments (please refer to PDF for placement as no line numbers were included)

Page2:
***please include references for each of these types of ice, especially because subsurface ice captures debris covered ice, so the distinction that is trying to be made is not clear
We edited the sentence on lines 36-37 to read as follows: “The surface morphology of the mid-latitudes of Mars (especially between 30 and 60°, north and south) is characterized by glacial remnants in the form of subsurface ice (Fig. 1; e.g., Brough et al., 2019; Levy et al., 2014), and icy mantling deposits (Mustard et al., 2001).”

***it is not clear for a general reader why this is "in addition" to the previous sentences that point to evidence of wet-based glaciation, so make it clearer this is also being used to make that case or remove
We removed “in addition” on line 47.

***suggest being more specific and saying "where ground penetrating radar data are exploitable"

We edited the sentence on lines 56-57 to read as follows: “In the cases where subsurface radar sounding data are available, lobate debris aprons consist of up to ~90% of ice…”

***this is vague, be more specific

The sentence now reads as follows on lines 61-62: “All mid-latitude viscous flow features are believed to have been deposited during orbital excursions of ≥45° in the Amazonian (Madeleine et al., 2009) and to have been cold-based (e.g., Head and Marchant, 2003).”

Page3

***not a good citation for the LDM and not in reference list

The Conway et al. 2018 reference was removed from that sentence and added to the reference list since it is mentioned later.

***this is vague, be more specific

More specificity was added on lines 69-70 as follows: “The latitude-dependent mantle consists of different layers rich in water ice and dust (Schon et al., 2009). The ice was deposited during high obliquity excursions and the dust formed during low obliquities when the ice sublimated and left behind a dusty lag (Schon et al., 2009).”

***not in reference list

We added Conway and Balme (2014) to the reference list.

***not good refs for the age, better: Schon et al PSS 10.1016/j.pss.2012.03.015 Willmes et al PSS 10.1016/j.pss.2011.08.006

We changed the references for age to the recommended papers on line 76.

***this text should appear after the first sentence as it applies to both panels a and c.

Following the reviewer’s suggestion, we moved the text to after the first sentence on line 80.

***place with text describing legend items, above

Following the reviewer’s suggestion, we moved the text about “Green filled polygons” to follow the third sentence of the caption.

***missing space

We added a space between at and 41.5 on line 86.

***This section is not convincing. There is no good evidence presented that cold based glaciation cannot create cirques. We do not observe them on Earth because currently cold based glaciers hide them and any currently exposed cirques have experienced warm-based conditions at some point confusing the signal. On Mars there is a lot of time to do geomorphic work because of the lack of plate tectonics and active hydrosphere/biosphere, so "it takes too long" is not a good reason to throw out cold based glaciation. I think simply ignoring the uncertainty is dishonest to the reader. It also sets a precedent for future works to use cirques as "proof" of wet-based glaciation on Mars or other planets.
I don't disagree that terrestrial cirques are generally associated with wet-based glaciation, but this does not prove that cold-based glaciers cannot make them. In order to use these as evidence of wet-based glaciation on Mars or even suggestive of wet-based glaciation on Mars there needs to be solid proof that water is needed to form cirques on Earth, which to my knowledge does not currently exist in the literature.
I think all this paragraph should be in the discussion and not the introduction
We moved this paragraph to discussion section 5.4.

***it would be useful to mention briefly what processes contribute to cirque growth/formation, see the nice summary in the intro of this paper:
https://journals.openedition.org/geomorphologie/13057
NB: this paper also highlights that not everyone thinks that cirques are principally glacial, I am not saying the weight of evidence is on their side (e.g. Evans ESurf 2020), I am just saying it is better to acknowledge that cirque origin is not a completely "solved problem"
Following the reviewer's recommendations, we edited the sentence to read as follows: "Cirques develop from incipient depressions in mountain and plateau sides that fill with snow/ice and over time support active, wet-based glaciers that deepen the depressions by glacial erosion (Evans and Cox, 1974; Glasser and Bennett, 2004) via a combination of plucking, abrasion (e.g., White, 1970), and frost weathering (e.g., Sanders et al., 2012), though it is debated whether non-glacial processes such as rock-slope failures may have a substantial contribution to erosion as well (e.g., Turnball and Davies, 2006; Coquin et al., 2019; Evans, 2020)."

***move to discussion
We moved the following sentences to Section 5.4 starting on line 626: "If these martian alcoves are analogous to terrestrial glacial cirques, then they may have formed either during an earlier wet-based phase at the scale of an active glacier-like form, or formed during a prior cold-based glacial cycle separate from the glacier-like forms, such as when lobate debris aprons formed."

***the colour-keyed DEM is not very useful to show the form, I suggest using the air image (as this approximates the CTX most closely) with contour lines. Or use the hillshaded+colourised DTM and another panel to show the actual image of the cirque

We edited the figure to use contour lines, following the reviewer’s suggestion:

[Figure]

***all panels have a north arrow, just delete this text

This text was deleted following the reviewer's suggestion.

***Russia
We corrected the text to Russia on line 144.

***please add longitude labels on bottom axis because the longitude lines do not run straight up-down so the top labels cannot easily be linked to the bottom
We added longitude labels on the bottom axis and latitude labels to the right. We also added gridlines. The figure now looks as follows:

[Figure]

***can be deleted, redundant with legend
The text was deleted.

***white rectangles are extremely hard to see, make more visible
We edited the text to read as follows: “The white sections on the top left show where the CTX beta01 mosaic does not have coverage and grayscale areas of the map show where the mosaicked HRSC DEM does not have coverage.”

***please state the version of the CTX mosaic that was used

We included the version as follows: “We mapped ~2000 alcoves at a 1:30,000 scale using the ~6 m/pixel Context Camera imagery beta01 mosaic (Malin et al., 2007; Dickson et al., 2023a). ”

***please state how this mosaicing was done and how the alignment between HRSC and CTX was managed (or not) and if it was not please say how much the mismatch was and therefore the inaccuracy on the placement of your profiles. to assess the mismatch it is easiest to use the orthorectified ND4 image and the CTX mosaic. Note the CTX mosaic is not controlled, so is unlikely to align properly with the HRSC which is controlled at level 4.
We added the following sentence at the end of Section 3.1 to explain how mosaicking was done in ArcGIS Pro: “The HRSC DEMs were mosaicked together using the Mosaic to New Raster tool in ArcGIS Pro.” We addressed any mismatch between the HRSC and CTX in our response to 4c) above.

***please list the images used in your data availability statement
We edited the statement to read as follows: Where available, we used ~25 cm/pixel High Resolution Imaging Science Experiment (HiRISE; McEwen et al., 2007) images to examine glacial geomorphic features within and next to the alcoves, which are listed in the Data Availability section.

In the Data Availability section, we added this sentence: “The HiRISE frames that we examined for icy geomorphic features included the following: ESP_041934_2265, ESP_040853_2275, ESP_036844_2225, ESP_036580_2260, ESP_036514_2210, ESP_026941_2275, ESP_025873_2230, ESP_025781_2220, ESP_025477_2280, ESP_025253_2245, ESP_023618_2270, ESP_023605_2205, ESP_019768_2220, ESP_019214_2270, ESP_016748_2255, ESP_016471_2260, ESP_016247_2270, ESP_016194_2260, ESP_067108_2240, ESP_060013_2250, ESP_057877_2245, ESP_056004_2255, ESP_055872_2270, ESP_055661_2230, ESP_054527_2225, ESP_053762_2280, ESP_052826_2240, ESP_052681_2240, ESP_052417_2220, ESP_050558_2245, ESP_048949_2230, ESP_046853_2200, ESP_046220_2235, ESP_046075_2200, ESP_046022_2265, ESP_043688_2245, ESP_025319_2240, ESP_016959_2240, ESP_027574_2245, ESP_035011_2240, PSP_006147_2250, ESP_068441_2230, ESP_033745_2270, ESP_035156_2220, and ESP_028418_2240.”

***these classes were determined using the image data only? if so explicitly say so. If not say what other data was used and how.
Yes, these classes were determined using the image data only. We edited the sentence to read as follows: “Based on their kilometer-scale physical characteristics including shape,

size, and associated landforms as seen in CTX imagery,, we classified our population of mapped alcoves into seven broad classes: a) simple, b) joined, c) interiorly ridged, d) staircase, e) channel-related, f) branching (Fig. 4).”

***In the figure the longitudinal profiles are included, but in the table the longitudinal profile characteristics are not cited.
The longitudinal profiles are included in the figure for the reader to reference, but are not actually used for distinguishing the different classes in the table. We removed this figure from the Methods and moved it to the Supplementary Material Section following the reviewer’s #5 comment.

***please state what the i and ii panels mean before getting into the descriptions
The following sentence was added in the caption on line 201: “For each alcove class, panels (i) on the left correspond to an image example, and panels (ii) on the right correspond to an example of the profile.”

***there is no b-ii in my version
The sentence was deleted.

***ridges not visible in elevation data
The ridges are too narrow (<50 m) for the resolution of the elevation data (resolution ~50-100 m).

***colourised elevation
Following the reviewer’s suggestion, the word colorized was added to the sentence.

***elevation values
Following the reviewer’s suggestion, the word elevation was added to the sentence.

***make clear if this is in in addition to the number above it, or subsampled from it (perhaps in the caption as *)
The parenthesis was edited to read as follows: “(and subsampled number that fit in multiple classes).”

***this is the first time craters are mentioned as possible origins for these features. However, craters on the lip of the slope are highly unlikely to have the same morphology as craters on the plains, so this comparison is invalid and does not rule out that these are craters. This

perhaps should be something in the discussion, at least should be mentioned whenever this comparison is brought up

We edited the third paragraph of the Section 5.1.2 to include a discussion of impact cratering for alcove initiation as follows: “While it is likely that multiple processes contributed to the incipient form of a cirque-like alcove like those mentioned in Table 2, we suggest that the morphometrics and conditions observed eventually require substantial glacial erosion. For example, for impact cratering, while glacial geomorphic features may override any signature of impact ejecta, it is very unlikely that similarly sized impacts all happened to occur along mesa edges. Instead, it is more likely that cratering would occur in stochastic sizes and locations. Ultimately, we acknowledge that these other processes likely contributed to at least some erosion of cirque-like alcoves, but the prevalent glacial geomorphic features and consistently sized features are what correspond most to glacial erosion.”

***specify you downsampled from (1266 - 81) which were considered similar to cirques based on image analysis only
We followed the reviewer’s recommendation and changed the sentence to read as follows: “By applying these constraints, we were able to identify 456 most cirque-like alcoves after downsampling from our initial mapping and classification based on only image analysis of 1991 alcoves.”

***Fig 5 really doesn't show this
We deleted the reference to Fig 5.

***simply state "by using standard circular statistics calculation methods"?
We changed the statement to read as follows: “Mean of all pixel aspects across the entire surface of alcove by using standard statistical calculation methods for circular features.”

***Fig 5 the "H" line is misleading in planview, use a different line colour or use points.
We changed the “H” line so that it is dashed instead of solid.

include the contour lines and labels so it can be seen that "H" is the difference between the max and min height, and the min is not necessarily at the point where L starts, i.e. the midpoint

We added contour lines and labels to the figure:

[Figure]

***Figure 6 seems to show a lot more uncertainties than discussed here. Please add discussion on the noise of the HRSC DTM and the resolving power as demonstrated with the comparison with CTX (I note here that you must make sure these two datasets are spatially aligned to make this comparison valid)

We added a paragraph to section 3.2.3, which we provided in our response to the first comment on page 15.

***the longitudinal profile is not used to decide on which alcoves are included in the study, as far as I understand from the text (which if I am mistaken please correct the text), but the elevation data are, so please instead discuss the effects of the elevation data uncertainty on the measurements made by ACME2, which are used to downselect the data further

We added this paragraph to section 3.2.3: “We mapped the cirque-like alcove and identified the mid-threshold point using the CTX imagery. As mentioned in Section 3.1, the CTX imagery and HRSC DEM were aligned to a Sinusoidal projection centered on longitude 25.5 degrees East, and were based on the IAU Mars 2000 Sphere datum. Any misalignment of up to 100 m between the image and the DTM is of little concern when it translates into metrics made by ACME2 since most metrics rely on multiple pixel measurements. This is certainly the case for slope, aspect and average elevation along the cirque length or the entire cirque area. It might affect minimum and maximum elevation, although any effect should be evened out by the large sample size.”

***this should be recorded as an attribute and presented as a result.

We changed the sentence to read as follows: “However, in some cases, we do not see the threshold because of low DEM resolution or because the feature may be covered by other material (Section 4.3).” Section 4.3 now reports the percentage of cirque-like alcoves that were observed to have icy geomorphic features.

***save interpretation for after the results are presented
We deleted the text “from glacial erosion.”

***given the steps in this profile that clearly do not exist in the images, it casts doubt on the "overdeepening" being a real signal or an artefact like the steps.
We added the word “potential” in front of overdeepening and added that it is hard to discern with the current resolution: “Example of the longitudinal profile of a mapped alcove using the HRSC DEM that includes the potential overdeepening (difficult to discern at this resolution).”

***briefly say what methods were used and how it was controlled to the HRSC data in sect 3.2.3
This sentence was added to the end of Section 3.1: “Measurements from all of the imagery and DEMs used a Sinusoidal projection centered on longitude 25.5 degrees East, and were based on the IAU Mars 2000 Sphere datum.”

In addition, we added the following sentences in Section 3.2.3: “For comparison to the HRSC DEM, we include a CTX DEM generated by the GALE lab at UCLA using the Ames Stereo Pipeline (Beyer et al., 2018; Fig. 6).”

***did you do circular statistics to calculate this? in essence this means the alcoves are in all orientations? Aspect would be better shown as a rose diagram or histogram
Following the reviewer’s suggestion 5), this section was deleted.

***15°
Following the reviewer’s suggestion 5), this section was deleted.

***this would have to be done using circular statistics
Following the reviewer’s suggestion 5), this section was deleted.

***explain the symbology, i.e. what is the blue box, the organe line, what are the circles, what are the whiskers…

Following the reviewer's suggestion 5), this section was deleted.

***a first result should really be a map of where they are , i.e. fig 11

We replaced what was previously fig 11 with what is now Fig. 6:

[Figure]

**Figure 6. Distribution of 456 cirque-like alcoves and 74 glacier-like forms in the study region Deuteronilus Mensae. Note that while some glacier-like forms (Brough et al., 2019) exist outside of the teal boundary lines, they are not included in the analyses reported in this study.**

***are located at

We accepted this change and changed it as suggested.

***throughout the first paragraph of section 4.2 (and in the remaining sections) make sure that the term "alcoves" is not used to mean cirque-like alcoves because otherwise the reader becomes confused as to which group of data is being discussed.

We address this concern by adding "cirque-like" in section 4.2 and other sections to clarify when we are referring to cirque-like alcoves.

***please mention how this was calculated (in methods if it takes more than one sentence to explain, then refer to methods from here)

The following sentences were added in Table 1 of the Methods section: “We also found the relative percentages of cirque-like alcoves in each aspect bin after normalizing by the percent of the total land surface in each aspect bin. We did this by converting the HRSC DEM raster to points, finding the aspect for each point, and calculating the land surface percent that belonged to each aspect bin. We then divided the percent of cirque-like alcoves in each aspect bin by the land surface percent bins and got the normalized percentages.”

In the caption for Figure 6, we edited part (b) to read as follows: “Rose diagram showing the relative percentages of cirque-like alcoves in each aspect bin after normalizing by the percent of the total land surface in each aspect bin (we explain the method in Table 1). After normalizing, we found that the same southeastward trend persisted.”

***Fig 11 is not super-easy to understand without the topography/image mosaic for context. It also needs to be a lot bigger (fill page width)

We replaced what was previously Fig 11 with Fig 6, which now includes both the topography and image mosaic for context. The figure is in the response to the second comment on page 21.

***stacked bar charts are really hard to interpret, put bars side-by-side?

We edited the bar charts so that they are now side-by-side:

[Figure]

***this section contains a mix of methods, observations, results and interpretation, please add a methods section to describe the morphologies, then results to summarise them, and then this section can be limited to interpretation

We are grouping our response to this comment with the next one below.

***In general, this section is long and hard to follow. There is a lot of speculation and discussion about each landform and it is hard to follow what the authors are arguing for. Each landform has many possible interpretations and discussing every one in turn in detail makes it really hard to understand why the authors are even focussing on these "details". I strongly suggest cutting back this text and organising it differently. If the only point that is trying to be made in this section is that the 9% of cirque-like alcoves with HiRISE coverage a large percentage (how many total is not clear) of the alcoves are filled with some kind of icy materials then this can be a lot shorter (which I interpret is the main message).

A section should be added to methods to describe how the following features were recognised and citing previous literature to say BRIEFLY how they are known to be icy materials:

1. crevasses/washboard, 2. Lineations 3. polygons etc

The Methods now include the following section:

**3.3 Criteria for identification of icy geomorphic features**

In addition to mapping and calculating the morphometrics of alcoves in Deuteronilus Mensae, we also evaluate the presence of icy geomorphic features in the alcoves where HiRISE imagery is available. While we designed the study so that none of the cirque-like alcoves that we mapped included mapped glacier-like forms, using the available inventory of HiRISE images we observed other features associated with the cirque-like alcoves that appear consistent with the presence of ice or ice loss. The icy geomorphic features that we evaluate for in HiRISE images include flow features, linear terrain, mantle, moraine-like ridges, mound-and-tail terrain, polygonal terrain, moraine-like ridges, rectilinear-ridge terrain, and washboard terrain. We identify these features using the criteria listed in Table 3. Other icy geomorphic features that were observed nearby alcoves but not categorized in this study because they were not directly in or connected to features coming out of alcoves included brain terrain (Levy et al., 2009a) and pitted terrain (Jawin et al., 2018). We note that the icy geomorphic features that we identify may correspond to some of the criteria defined by Souness et al. (2012) for mapping glacier-like forms, which include: 1) surrounded by topography indicative of flow around obstacles, 2) distinct from the surrounding landscape in texture or color, 3) surface foliation indicative of down-slope flow, 4) L/W ratio > 1, 5) discernible head or terminus, 6) appear to contain a volume of ice. However, the icy geomorphic features noted here do not include all of the criteria and were not mapped as glacier-like forms. For example, an icy feature within an alcove might appear to have a terminus, but no convexity from existing ice volume that differentiates it from surrounding topography (Fig. 6).

**Table 3: Icy geomorphic features with their descriptions, proposed formation, and references.**

| Icy Geomorphic Feature | Additional Names | Description | Terrestrial Analog | Proposed Formation Mechanism | Select References |
|---|---|---|---|---|---|

| | | | | | |
|---|---|---|---|---|---|
| Flow features | N/A | Troughs and ridges | Same | Formed by downslope flow and deformation | Hubbard et al., 2011; Souness et al. 2013 |
| Linear terrain | If supraglacial: longitudinal foliation | Parallel raised ridges, bumpy in appearance | If supraglacial: flow stripes, longitudinal foliation | If supraglacial: Caused by deformation of ice as it flows; can be due to compressed, accelerating flow | Hubbard et al., 2011; Conway et al., 2018 |
| | If subglacial: pasted-on terrain | | If subglacial: megalineation,, striations | If subglacial: Ice flow over water-lubricated sediment | |
| Mantle | Latitude-dependent mantle, thicker version is commonly known as pasted-on terrain | "Raised curvilinear edge for the upslope boundary" (Khuller et al. 2021) | N/A | Airfall of ice on dust; sublimation of lag protects ice deposits | Mustard et al. 2001; Christensen et al., 2003; Conway et al., 2018; Khuller et al. 2021 |
| Moraine-like ridge | Moraine ridge | Ridge of debris | Terminal moraine | Dumping, squeezing, and pushing of debris by a glacier | Arfstrom and Hartmann, 2005 |
| Mound-and-tail terrain | N/A, similar to linear terrain | Steep upglacier-facing core with a shallow elongate tail; typically 30-50 m long, 10-30 m across, and 2-4 m high | Closest to drumlins | Subglacial bedforms formed from subglacial sediment moulding and/or deposition beneath wet-based ice masses | Hubbard et al., 2011 |
| Polygonal terrain | Polygonized terrain and scaly terrain (we group the two together here under the term "polygonal" terrain); mantle polygons | Polygonized terrain: ~10° slope, 5-10 m across, tessellating polygons;

Scaly terrain: 12-16° slope, 10-20 m across, tessellating polygons | Periglacial patterned ground | Frost heave and thermal contraction cracking | On Mars: Hubbard et al., 2011; Levy et al., 2009b; Soare et al., 2022

On Earth e.g.,: French, 2018; Marchant and Head, 2007 |

| | | | | | |
|---|---|---|---|---|---|
| Rectilinear-ridge terrain | Push moraines | Series of ridges tens of meters across and 2-3 m high, elongated in an arc parallel to former glacier terminus | Thrust-block moraines, push moraines, moraine-mound complex | Basal debris thrust up from the glacier bed, basal crevasse fills, or ice-contact outwash deposits | On Mars: Hubbard et al., 2011

On Earth e.g.,: Hambrey et al., 1997; Sharp, 1985; Lukas, 2005 |
| Washboard terrain | Crevasse-like features | Transverse scarps, commonly at the base of a steep slope | Crevasses, bergschrunds | Formed from debuttressing and oversteepening of ice on slopes | Hubbard et al., 2011; Jawin et al., 2018; Jawin and Head, 2021 |

[Figure]

**Figure 6: (a) Previously mapped glacier-like form (Brough et al., 2019). (b) and (c) represent previously unmapped cirque-like alcoves no longer appear to contain a volume of ice and raised moraine-like ridge at the terminus. However, they still do contain surface foliations suggesting down-slope flow near the headwall. Cirque-like alcove mapping only extends to where the sidewalls end. This HiRISE image ESP_025873_2230_RED is centered at 42.63°N, 25.02°E. HiRISE data credit: NASA/JPL/University of Arizona.**

Then a section in the results saying the percent of the alcoves that have one or more of these and then the separate percentages (with table containing the alcove ID, HiRISE Id, lat/long and features identified).

The new results section now reads as follows:

**4.3 Icy geomorphic features identified**

In addition to morphometric observations, we identified geomorphic features in association with the cirque-like alcoves as consistent with either remnant or active ice in order to evaluate aspects of the glacial history in the cirque-like alcoves. Using the criteria stated in Table 3, we identified flow features, linear terrain, mantle, moraine-like ridges, mound-and-tail terrain, polygonal terrain, rectilinear-ridge terrain, and washboard terrain in available HiRISE imagery. Out of 435 cirque-like alcoves, there was complete overlap in available HiRISE frames with 26 cirque-like alcoves (8%) and partial overlap with only 10 cirque-like alcoves (1%). In CTX imagery, we were also able to identify flow features, linear terrain, mantle, moraine-like ridges, and washboard terrain. However, at the CTX resolution, it was more difficult to identify features such as mound-and-tail terrain, polygonal terrain, and rectilinear-ridge terrain. For both HiRISE and CTX imagery, the linear terrain and mantle were the two most common features. We provide the percentages of each feature in both HiRISE and CTX imagery in Table 5 and the specific features and HiRISE frames associated with each in Table S1.

**Table 5: Percent of HiRISE and CTX imagery with each type of icy geomorphic feature.**

| Icy Geomorphic Feature | Percent of HiRISE imagery (%) | Percent of CTX imagery (%) |
|---|---|---|
| Flow features | 8 | 9 |
| Linear terrain | 81 | 57 |
| Mantle | 58 | 90 |
| Moraine-like ridges | 14 | 5 |
| Mound-and-tail terrain | 6 | N/A |
| Polygonal terrain | 53 | N/A |
| Rectilinear-ridge terrain | 3 | N/A |
| Washboard terrain | 42 | 2 |

Fig. 12 provides examples of washboard terrain, linear terrain, rectilinear ridges, and polygonal terrain, which all correspond to the presence of ice and/or ice loss, as described in Table 3. In Fig. 12, the linear terrain extends out from the washboard terrain at the base of the cirque-like alcoves (Fig. 12). The rectilinear ridges are downslope of both the washboard terrain and linear terrain. The polygonal terrain is between the two sections of linear terrain (Fig. 12f). In addition, the polygonal terrain is observed farther downslope of the rectilinear ridges (Fig. 12f).

Approximately 14% of cirque-like alcoves with HiRISE imagery coverage have moraine-like ridges. Fig. 13 contains examples of moraine-like ridges. Fig. 13b also shows additional examples of moraine-like ridges downslope of alcoves (that are not all cirque-like), with along-flow linear terrain between the alcove headwall and the moraine-like ridge. As in Fig. 12, washboard terrain, linear terrain, and polygonal terrain are all present.

[Figure]

**Figure 12: a) Cirque-like alcove with evidence for remnant ice centered at 46.57°N, 22.12°E, 46.57°N in HiRISE image ESP_019214_2270_RED. b) Boulders near the top of the headwall indicating erosion. Features corresponding to ice-loss include the following: c) washboard terrain (Jawin and Head, 2021), d) linear terrain, e) rectilinear ridges (Hubbard et al., 2011), and f) polygonal terrain. (e.g., Levy et al., 2009a; Hubbard et al., 2011).**

**Table S1:** Icy geomorphic features identified in cirque-like alcoves using available HiRISE frames.

| Alcove ID | HiRISE ID | Coverage | Latitude, Longitude | Icy geomorphic features identified |
|---|---|---|---|---|
| 50 | PSP_007439_2205 | Partial | 40.18°N, 24.72°E | Linear terrain, mantle, mound-and-tail terrain |
| 56 | ESP_072529_2265 | Partial | 40.29°N, 23.00°E | Mantle |
| 57 | ESP_072529_2265 | Full | 40.26°N, 22.98°E | Mantle |
| 145 | PSP_008810_2225 | Full | 41.85°N, 26.36°E | Polygonal terrain, mantle |
| 572 | ESP_067108_2240 | Partial | 43.70°N, 27.92°E | Mantle |
| 631 | ESP_068441_2230 | Full | 42.63°N, 25.30°E | Linear terrain, mantle, washboard terrain |

| | | | | |
|---|---|---|---|---|
| 637 | ESP_025873_2230 | Partial | 42.76°N, 25.06°E | Linear terrain |
| 650 | ESP_054527_2225 | Partial | 41.97°N, 24.63°E | Linear terrain, mantle |
| 704 | ESP_046220_2235 | Full | 42.94°N, 24.06°E | Linear terrain, polygonal terrain, washboard terrain |
| 705 | ESP_046220_2235 | Full | 42.97°N, 24.05°E | Linear terrain, polygonal terrain, washboard terrain |
| 769 | ESP_052681_2240 | Full | 43.64°N, 24.52°E | Flow features, linear terrain, moraine-like ridges, polygonal terrain |
| 783 | ESP_052826_2240 | Partial | 43.43°N, 26.02°E | Linear terrain, mantle |
| 878 | ESP_025253_2245 | Partial | 44.48°N, 29.82°E | Linear terrain, mound-and-tail terrain, polygonal terrain, washboard terrain |
| 911 | PSP_007162_2250 | Full | 44.60°N, 27.66°E | Linear terrain, mantle |
| 1061 | ESP_046022_2265 | Partial | 46.38°N, 29.00°E | Mantle, polygonal terrain, washboard terrain |
| 1088 | ESP_033745_2270 | Full | 46.66°N, 29.85°E | Linear terrain, mantle, moraine-like ridges, polygonal terrain, washboard terrain |
| 1125 | ESP_043688_2245 | Full | 44.16°N, 25.19°E | Linear terrain, polygonal terrain, washboard terrain |
| 1161 | ESP_053762_2280 | Full | 47.40°N, 27.37°E | Polygonal terrain, linear terrain, broad pit |
| 1170 | EPS_026941_2275 | Full | 47.12°N, 26.71°E | Linear terrain, polygonal terrain, moraine-like ridges, washboard terrain |
| 1171 | ESP_026941_2275 | Full | 47.14°N, 26.75°E | Linear terrain, polygonal terrain, washboard terrain |
| 1218 | ESP_055872_2270 | Full | 46.39°N, 27.09°E | Mantle, linear terrain, washboard terrain |
| 1227 | ESP_056004_2255 | Full | 45.25°N, 24.53°E | Linear terrain, polygonal terrain, mantle |
| 1230 | ESP_056004_2255 | Full | 45.25°N, 24.58°E | Mantle, polygonal, linear terrain |
| 1302 | ESP_057877_2245 | Full | 44.13°N, 23.86°E | Linear terrain, mantle, polygonal terrain |
| 1425 | PSP_002890_2205 | Full | 40.09°N, 22.72°E | Linear terrain, polygonal terrain, washboard terrain |
| 1438 | ESP_046853_2200 | Full | 40.25°N, 22.92°E | Mantle |
| 1487 | ESP_016471_2260 | Full | 45.63°N, 33.47°E | Linear terrain, washboard terrain |

| | | | | |
|---|---|---|---|---|
| 1594 | ESP_019768_2220 | Full | 41.67°N, 18.43°E | Flow features, linear terrain, rectilinear ridge terrain, washboard terrain |
| 1616 | PSP_005857_2225 | Partial | 42.07°N, 19.52°E | Mantle |
| 1802 | ESP_035156_2220 | Full | 41.90°N, 23.90°E | Linear terrain, mantle, moraine-like ridges |
| 1808 | ESP_046075_2200 | Full | 40.29°N, 24.23°E | Linear terrain, mantle, moraine-like ridges, polygonal terrain |
| 1840 | ESP_025781_2220 | Full | 41.63°N, 16.28°E | Flow features, linear terrain, mantle |
| 1842 | ESP_025781_2220 | Partial | 41.64°N, 16.19°E | Linear terrain, mantle |
| 1965 | ESP_019214_2270 | Full | 46.57°N, 22.14°E | Linear terrain, polygonal terrain, washboard terrain |
| 1967 | ESP_019214_2270 | Full | 46.58°N, 22.14°E | Linear terrain, polygonal terrain, washboard terrain |
| 2026 | PSP_006147_2250 | Full | 44.63°N, 21.05°E | Linear terrain, polygonal terrain |

Then perhaps simply a line or two in the discussion about how these observations can be extended to the other filled alcoves in the CTX survey (these filled alcoves need to be identified and reported clearly in the results, which is not currently the case). The details on "possible maybe perhaps" till deposits, glacial dynamics, rock glacier deflation etc etc should be left out and focus maintained on providing evidence pointing to ice as a major component of the fill material.

We edited the discussion so that it now reads as follows:

"**5.2 Geomorphic interpretations of cirque-like alcoves and associated features**

**5.2.1 Icy geomorphic features**

42% of HiRISE images contained washboard terrain, while only 2% of CTX images did, though this is likely due to a resolution issue since thinner fissures cannot be resolved at CTX scale. Except for two exceptions, cirque-like alcoves that contained washboard terrain did not also have an identifiable mantling unit. Similar to its presence at the bottom of crater walls (Jawin et al., 2018; Jawin and Head, 2021), the presence of washboard terrain here at the bottoms of the mesa sidewalls indicates deglaciation.

In both HiRISE (81%) and CTX imagery (57%), a high percentage of images of cirque-like alcoves contained observable linear terrain. In Fig. 12, since the linear terrain extended out from the washboard terrain, which is due to surficial crevasses, this suggests that the linear terrain there may be most similar to supraglacial longitudinal foliation. However, linear terrain could still result from subglacial erosion despite superposing a mantle unit since a mantle

unit consists of layers of dust and snow that build up in the mantle over multiple obliquity cycles (e.g., Khuller et al., 2021). Applied here, this would imply that the ridges could have been subglacially eroded, but from another layer of ice of the mantle unit (compacted from dust and snow) that formerly existed on top of the rest of what is left of the mantle unit today.

At a potentially earlier stage of evolution of the glacier-like forms, moraine-like ridges may lack elongation outside of the alcove (Fig. 13a), potentially similar to a terrestrial cirque glacier sitting within the cirque basin instead of extending into the valley below. In Fig. 13b, the alcoves are not well-developed and do not have morphometrics corresponding to the criteria we set for cirque-like alcoves. Nevertheless, since the moraine-like ridges correspond to upslope alcoves, similar to Arfstrom and Harmann (2005), we suggest that the moraine-like ridges in Fig. 13b reflect the initiation of cirque-style glaciation before the alcove headwalls and sidewalls develop more as they are increasingly eroded and steepened. This is also referred to as unconstrained piedmont glaciation by Conway et al. (2018)."

*** so the definitions of these should be in the methods and then you can report this part as results
Corresponding to this comment, we moved the definitions to Table 3 in the Methods section 3.3.

*** coverage of what we interpet as remnant or active ice
Here we were actually referring to cirque-like alcoves that had overlap with HiRISE frames. We clarified the text to read as follows: "Out of 456 cirque-like alcoves, 6 cirque-like alcoves (1%) had partial overlap, and 38 cirque-like alcoves (8%) had complete overlap within an available HiRISE frame."

*** replace with "some"
We accepted this change and edited it to "some."

*** unfounded speculation, when you restructure this section such unfounded statements should be deleted and replaced with augmented statements.
We deleted this statement.

*** name them or don't speculate on them here
We deleted this statement and saved the discussion for Section 5.3.

*** if this is all the only point that you want to make with this observation, then delete all the following parts, these are unnecessary details

We deleted three sentences but kept the last two remaining sentences: “One interpretation of how these types of downslope surface lineations could still result from subglacial erosion despite superposing a mantle unit is by using the model that layers of dust and snow build up in the mantle over multiple obliquity cycles (e.g., Khuller et al., 2021). Applied here, this would imply that the ridges could have been subglacially eroded, but from another layer of ice of the mantle unit (compacted from dust and snow) that formerly existed on top of the rest of what is left of the mantle unit today.”

*** so instead of doing this, just use ONE of the terms and stick to it please
We edited all instances of “polygonized” terrain to “polygonal” terrain and changed the sentences starting on line 408 to read as follows: “Approximately ~15% of cirque-like alcoves with HiRISE coverage contained “polygonal” or “polygonized” terrain, two terms that are used synonymously (see Fig. 11f). Similar to how it is described in association with the glacier-like forms in Hubbard et al. (2011), we also see this type of polygonal terrain here between the surface lineations (Fig. 113f).”

*** said like the reader already knows what you mean, describe these before you discuss their relation to other features
We deleted the statement.

*** so the interpretation is that these features are related to ground ice? If so say so explicitly
Following the suggestion, we edited the statement to read as follows: “Similarly, on Earth, polygonal terrain results from periglacial processes such as contraction cracking and frost heave (French, 2018), though sublimation-type polygons that arise from thermal contraction and sublimation, indicating the presence of ground ice, have also been observed in the Antarctic Dry Valleys (Marchant and Head, 2007).”

*** what is shown in 13e looks like 13ca and doesn't seem to match the description made in this paragraph
While both features in what was previously Figure 13e and 13a are transverse, 13e has thrust-block plates (also seen for Hubbard et al. 2011) that are more curvilinear in appearance. However, we realized that there were inconsistencies in this paragraph for descriptions of 13e and have separated out rectilinear terrain from moraine-like ridges (previously referred to as transverse ridges). We added definitions for the different features in Table 3. We also note that Figure 13 (now Figure 12) does not have good examples of moraine-like ridges, whereas Figures 6 and 13 do.

*** this is going a little too far I think without showing examples of this morphology. in order to not overload this manuscript I suggest just removing this comparison

We followed the reviewer's comment and deleted this sentence.

*** 13e looks nothing like the arcuate ridges described in this paper, so either you are referring to the wrong image or something else is wrong here
We agree with the reviewer and moved the statements about moraine-like ridges to correspond to the paragraph for Figure 12 instead (note that the prior Fig. 13 is now Fig. 12). We also changed all statements of arcuate ridges to moraine-like ridges. For the new Figure 13, we added the following description: "In Fig. 13b, the alcoves are not well-developed and do not have morphometrics corresponding to the criteria we set for cirque-like alcoves. Nevertheless, since the moraine-like ridges correspond to upslope alcoves, similar to Arfstrom and Hartmann (2005), we suggest that the moraine-like ridges in Fig. 13b reflect the initiation of cirque-style glaciation before the alcove headwalls and sidewalls develop more as they are increasingly eroded and steepened. This is also referred to as unconstrained piedmont glaciation by Conway et al. (2018)."

*** this seems to more accurately reflect fig 13e
We agree and changed the Figure 13e description as mentioned in the response to the first comment on Page 27 above.

*** this is too much of a conceptual jump
We deleted the sentence.

*** unfounded speculation
We deleted this sentence.

*** missing space between words
We added a space corresponding to the reviewer's suggestion.

*** it is impossible to know if what you interpret as "less developed" landforms is how the "more developed ones" looked without a time machine or at least a good knowledge of the process(es) and their rates. I suggest complete removal of this section, it is complete speculation
We find it plausible that in the order of increasing amount of erosion, the mesa edge would go from straight to having shallow depressions then deeper depressions. As such, we kept this section and added the following sentences at the beginning of the second paragraph: "Here, we assume that the side of the mesa evolves from a straight edge to an increasing number and depth of depressions. An alternative interpretation might be that the deeper depressions were subsequently filled up to create a straight edge, however, we do not see

evidence for this amount of infilling.” If the reviewer thinks that further analyses might be beneficial, we are open to any suggestions for additional analyses here.

Following the sentence “We suggest that the observed notches are gullies and would be able to act as necessary initiation points for ice accumulation that would later support glaciation and erosion that could form cirque-like alcoves,” we also added this reference since this idea of gullying tied to alcove formation has been previously proposed in a paraglacial by Jawin et al. 2018 as well: “This is consistent with the mechanism proposed by Jawin et al. (2018).”

*** missing word?
We added the word “processes” so that the phrase now reads “non-glacial processes” on line 519.

*** this section seems like it should be first in the discussion
We agree with the reviewer. Previously, we moved this section in accordance with another reviewer suggestion, but we have moved it back.

*** I did not read any "strong" evidence for glacial erosion in section 5.1, perhaps you meant another section (but I cannot find it)? section 5.1 demonstrates the ice fill of some of the alcoves and speculates based on selected images on the possible development of the alcoves from gullies but doesn't specifically assess the likelihood of glacial erosion processes (simply assumes it must be a given). Rephrase
We deleted that part of the sentence so that it now reads as follows: “We further evaluate whether cirque-like alcoves are candidate cirques by comparing them to cirques on Earth.”

*** I think the comparison between alcoves and GLF orientation is somewhat misleading. You show in your HiRISE study that many alcoves are filled with icy materials that are not mapped as GLF, this suggests they are perhaps occupied by glaciers that for some reason did not get classified as a GLF (no good images there when Colin did his survey? too small? don't have a clear divide between the alcove-fill and the VFF below?)... so I suggest rewriting this section with this caveat in mind
We added a section in the Methods “3.3 Criteria for identification of icy geomorphic features” to demonstrate how glacier-like forms are different from the features that we identified here. We included this section in the response to the third comment of page 25. We also added Table 3 and moved Figure 6 to be in this section, as mentioned in the same response.

*** See Kreslavsky PSS 2008 for a nice diagram of this effect
https://doi.org/10.1016/j.pss.2006.02.010
We added Kreslavksky et al. 2008 as a citation as well.

*** are
We accepted this change and changed the word to “are.”

*** pretty sure this should be 2019 - did you mean this one - doi.org/10.1144/SP467.3
We did mean Conway et al. (2018) *Geomorphology* since it included the following sentences: “They [gullies] are found primarily on pole-facing slopes at latitudes between 30° and 40° and then mostly on equator-facing slopes poleward of 40° (but they can also occur on pole-facing slopes in this latitude interval).”

*** 2023b
We accepted the change.

*** these two papers do not talk about meltwater
We edited the sentence so that the two papers are referenced earlier in the sentence now: “However, in the case of meltwater, we note that cirque-like alcoves may prefer to reside on equator-facing slopes because this would allow for increased insolation (e.g., Pilorget and Forget; 2016; Dundas et al., 2022) and the chance for meltwater as temperatures increase (Dickson et al., 2023b).”

*** corresponds
We changed “correspond” to “corresponds” as suggested.

*** I can't make this make sense, I have read it three times
We deleted part of the sentence to simplify it so that it now reads as follows: “If cirque-like alcoves do in fact correspond to an earlier phase of glaciation, it is unclear if this glaciation was on the scale of glacier-like forms versus larger scales like the lobate debris apron.”

*** delete? totally unconnected with the rest of the section
We deleted this paragraph following the reviewer’s suggestion.

*** section 5.2.2 would be a lot shorter if this is the explanation the authors prefer
Section 5.2.2 (now 5.1.2) was focused on which direction had the most cirque-like alcoves, which was southeastward. Here in section 5.1.4 (previously 5.2.4), the focus is on which aspect has the largest cirque-like alcoves, which is south, in contrast with glacier-like forms, which are largest when facing north in Deuteronilus Mensae. The localized topographic effect was invoked in regards to the largest glacier-like forms facing north.

*** I don't think it was demonstrated that these icy units are more or less remnant than the GLF or indeed the VFF in the area, so either make sure this demonstration takes place or rephrase this statement.
We deleted this section.

*** why? they are morphologically indistinct from the GLF and VFF in the area which are thought to be debris covered glaciers? They do no resemble rock glaciers in my opinion and you do not present any comparisons to defend this interpretation, so I strongly advise not including it, it just distracts from your data
We deleted this section, however, we did want to note that rock glaciers are considered to be analogs for GLFs as well, which we mentioned in our response to #4 above.

*** such hummocky structures occur when there is melting, which is probably not the case on Mars, also why would the upper part of VFF where the alcoves are have dead ice? This seems a nonsensical argument.
This sentence related to dead ice was added in response to another reviewer's comment. We have deleted it. We note for the reviewer that debris-covered glaciers may disconnect or detach (e.g., Pirámide glacier in Janke et al. 2015), though it is true that the part that is most likely to become dead ice would be at the toe of the glacier. Maybe the question is at one point does the deflated glacier become dead ice, but since this example retains glacial geomorphic features, we agree that the term dead ice is likely not representative.

*** the timescale of formation cannot discriminate between cold and wet based because there is no constraint on the age of the alcoves nor on the time taken to form them. delete this sentence
We deleted the highlighted portion of the sentence so that it now only reads as follows: "We estimated erosion rates for both wet- and cold-based ice."

*** their initiation would predate, but they would develop during occupation by the LDA? Rephrase?
We rephrased the sentence as follows: "Using the slowest estimated erosion rates corresponding to cold-based glaciers in Antarctica, the initiation of the cirque-like alcoves likely predated the lobate debris aprons. Then they could continue to develop in size during and/or after when the lobate debris aprons formed."

*** I don't see how this argues against the alcoves forming in concert with the LDA - debris from them should superpose the LDA - in cirques on Earth it is rare that the headwall is always completely covered, cirques form in alpine glaciations and not plateau glaciation

We rephrased the sentence to clarify what we meant: “Since debris from the cirque-like alcoves often superposes the lobate debris aprons (e.g., Baker and Carter, 2019), this means that the cirque-like alcoves have been actively eroding after when the lobate debris aprons formed.”

*** or it is the process contributing to the cirque development - hence a short description of the processes leading to cirque development on Earth in the introduction would be useful
Lines around 120 in the introduction now include the following description of cirque development: “Cirques develop from incipient depressions in mountain and plateau sides that fill with snow which eventually evolves into ice, thus supporting active glaciers. The movement of these lead to glacial erosion which deepens the initial depressions modifying the landscape to create the characteristic valley-head cirque shape that is found in all glaciated mountain regions worldwide (Evans and Cox, 1974; Glasser and Bennett, 2004). Glacial erosion occurs via a combination of quarrying, abrasion (e.g., White, 1970), and frost weathering (e.g., Sanders et al., 2012), which all contribute to enlarging the cirque floor and deepening the cirque (Evans, 2020). Non-glacial processes such as rock-slope failures may play a role in cirque-basin erosion as well, especially in terms of headwall retreat (e.g., Turnball and Davies, 2006; Coquin et al., 2019; Evans, 2020).”

*** not sure what you mean by this, but the VFF certainly extend into the alcoves as shown by your HiRISE observations
We deleted this part of the sentence.

*** this doesn't seem unreasonable
Great, we agree!

In general, the part above reads as if the authors want these alcoves to have formed alongside GLF in a wet-based regime, but every argument they make has an equally convincing counter-argument that they were forced to add in previous reviews, but they are still trying to argue for the wet-based.
We deleted the section titled “5.5 Discussion of wet-based versus cold-based glacial erosion of cirque-like alcoves” and have softened the language to not decisively say cirques require wet-based erosion in the Introduction and Conclusion sections.

*** this section should be removed, it just goes in circles and does not rely on the data in this paper. mostly it is speculation with no firm conclusion at the end of it all
We deleted this section.

*** this seems to mildly contradict what is said in the previous section, which concludes both wet or cold based glaciation are potential candidates over different, yet realistic timescales
We deleted this section.

*** this seems to lead nowhere, so I suggest remove it
We deleted this section.

*** they don't need to keep pace with talus production on Earth, but on Mars, where the rate is likely (a lot) slower as there is not rapid frost-shattering
We deleted this section.

*** strong wording given the noise in the DTM, rephrase or remove
We deleted this section.

*** hard to say require when it totally unknown what processes are going on
We deleted this section.

*** I don't think you need a massive discussion about processes that you can reject based on morphology/morphometry without much effort. I suggest reducing the text to an absolute minimum and simply refer to Table 6 to say these features are unlike those formed by other alcove-forming processes on Earth and most like cirques - this compilation should be presented when you present your selection criteria, page 15 and does not belong here
We moved the table to section 3.2.2 in Methods with condensed text:
"By using morphometrics, we also exclude other types of mechanisms for alcove formation, including active-layer detachments, deep-seated landslides, and theater-headed valleys (Table 2). This is because the H/L ratio of a terrestrial glacial cirque is expected to be deeper than any of the other alcove landforms with known morphometrics on Earth (Table 2)."

**Table 2:** Morphometrics consistent with different alcove-forming erosional mechanisms.

| **Formation mechanism/Landform** | **L/W** | **H/L** | **Aspect** | **Related geology** | **Typical scale (m)** |
|---|---|---|---|---|---|
| Glacial cirque on Earth | ~1, generally ranges from 0.5-4.25 (Barr and Spagnolo, 2015) | ~0.67 (Barr and Spagnolo, 2015) | All directions; poleward is favorable (Barr and Spagnolo, 2015) | Overdeepening, moraines | $10^2$-$10^3$ (Barr and Spagnolo, 2015) |

| | | | | | |
|---|---|---|---|---|---|
| Deep-seated landslide on Earth[n.s.] | >2.5 (Fran et al., 2006) | 0.1-0.35 (LaHusen et al., 2016; landslide scars from glacial sediment) | Not available* | Hummocky landslide deposits | $10^1$-$10^2$ (LaHusen et al., 2016) |
| Impact crater on Mars | ~1 | 0.1-0.2 (Robbins and Hynek, 2012) | N/A | Ejecta blanket | $10^1$-$10^3$ (Palucis et al., 2020) |
| Amphitheater-headed valley on Mars hypothesized to have formed by either groundwater sapping or outburst flooding | 1-10 (Laity, 1988) | Not available* | Not available* | Sandstone, not basalt bedrock (Lapotre and Lamb, 2018) | $10^1$-$10^2$ for canyon heads, up to $10^3$ for the main channel (Lapotre et al., 2016) |

Not available*: As of writing this paper, focused studies on the morphometrics of these landforms on the population scale are not widely available for these other landforms.

n.s.: "n.s." stands for "not scarp" since landslide morphometrics do not usually include measurements of the morphometrics of just the headscarp and sidewalls of where the landslides initiated.

*** these are conditioned on melt and highly unlikely to occur on Mars, also their scale is not big enough and they only affect the active layer and cannot erode into bedrock. I would discourage this kind of hasty comparison based on "looks-like-must-be"

We deleted active layer detachments from the table.

*** if you consider these as "similar" then you have a lot of other features to describe, including rock avalanches, first order fluvial catchments in badlands, debris flow headscarps…

We deleted this section.

*** already this point is debated where there is no ice on Mars, so why open a can of worms? your data don't have anything to contribute here

We deleted this section.

*** this cannot be observed as you do not have a time machine, rephrase. I also do not think this conclusion is justified, and should be removed, along with the associated section.

As mentioned for the above comment, we find it plausible that in the order of increasing amount of erosion, the mesa edge would go from straight to having shallow depressions then deeper depressions. As such, we kept this section and added the following sentences at the beginning of the second paragraph: “Here, we assume that the side of the mesa evolves from a straight edge to an increasing number and depth of depressions. An alternative interpretation might be that the deeper depressions were subsequently filled up to create a straight edge, however, we do not see evidence for this amount of infilling.” If the reviewer thinks that further analyses might be beneficial, we are open to any suggestions for additional analyses here.

We also edited the bullet point in the Conclusions to read as follows: “Headwall notches are observed adjacent to increasing sizes of larger alcoves (Fig. 15). Notches and subsequent stages of their development may act as an initiation point for ice accumulation, similar to what happens on Earth for local-slope glaciation. Larger alcoves may have undergone multiple cycles of glaciation and erosion.”

*** also unjustified, suggest remove
We deleted the statement on rock glaciers.

*** remove, not justified
We removed the sentence.

*** merge with previous paragraph and shorten both to make a shorter snappier conclusion point
The two paragraphs were merged and now read as follows: “The slight eastward bias in aspect aligns with previous studies is consistent with both glacier-like forms on Mars (e.g., Souness et al., 2012; Brough et al., 2019) and climate models of westerly winds in Deuteronilus Mensae (Madeleine et al., 2009). Terrestrial cirques also show a similar pattern due to westerly winds. Future work could help to better understand the atmospheric controls on cirque-like alcove formation in Deuteronilus Mensae, as well as other locations on Mars. There is a dominant southward bias in the aspect of the cirque-like alcoves (Fig. 6), which becomes more pronounced above 46.5°N. Overall, both cirque-like alcoves and glacier-like forms tend to have greater volumes when facing south, which may suggest an interdependent relationship between glacier-like form size and cirque-like alcove size. The southward aspect bias of cirque-like alcoves may result from poleward-facing slopes receiving more insolation during high obliquity, or an association with gully formation, as gullies also tend to face the equator at high latitudes (Harrison et al., 2015; Conway et al., 2018).”

***remove

Following the reviewer’s recommendation, we removed this paragraph.

*** these are extremely tenuous arguments and this paper does not show that this is the case, remove

We edited the sentence to remove “wet-based” so that it now reads as follows on line 888: “The presence of glacial geomorphic features, especially linear terra lineations, moraine-like ridges, mound-and-tail terrain, rectilinear ridges (Fig. 12), and potential overdeepenings (Fig. 5), are all consistent with glacial erosion. In addition, the presence of icy remnants exist in the form of brain terrain, flow features, mantling unit, and polygonal terrain.”

---

## Editor Decision (ED2)

**Minor comments from editor**

*The first digit of line numbers >100 have been cut off in the PDF. So Comments >line 100 are page + 2nd 2 digits of the line number*

L7: Add space in zip code (AB24 3UF)

L15: minimal. Response to editor comments states this has been changed to 'some'. Change has not been made.

L20: 'Evaluate past glaciation on Mars' is vague. Say something like 'to better understand the potential contribution of glaciation to landscape evolution in Deuteronilus Mensae, Mars'.

L21: Here and throughout: avoid the term 'icy geomorphic' as this is ambiguous. Implies the feature itself is icy, but this isn't always the case – some are inferred as having formed by ice that is no longer there. Better 'geomorphic evidence for past glacial occupation of these cirque-like alcoves'

L24-25: These don't really work as alternatives to one-another as currently written - one is posed as a mechanism, and one as a requirement.

Additionally...

If tying to obliquity, then the high-insolation scenario is a different obliquity state to the low-insolation/high-accumulation state. Or it arises from a climate regime that has not yet been well constrained. Can you make this clearer?

L26: 'both glacier-like forms' – 'both the estimated ages of glacier-like forms'

L30: Add 'and the timing of the formation of cirque-like alcoves' to this statement

L42: these aren't 'ice flow regimes' they are thermal regimes.

L43: Could add Gallagher et al. 2021 – they found grooves

L64: space after lag

L65: estimates ranging from > thickness estimates ranging from

L73: As stated in a previous review, the caption first needs to introduce the general purpose of the figure. E.g. 'Examples of viscous flow features and alcoves mapped in this study'. Ensure all figure captions have such a statement at the start.

L75: aprons plural x 2

L81-82: Given the new addition of Figure 7 , and the sample sizes discussed, this isn't correct as 'all alcoves mapped in this study'. It is an example of alcoves mapped in a sub-region of the study area.

L88: These processes don't result in basal slip. Basal slop drives (some of) these processes (quarrying and abrasion, not frost weathering).

L88-89: Delete the detail about meltwater through the bergschrund and randkluft. Too much detail, and inaccurate – meltwater can reach the bed from elsewhere too, e.g. crevasses.

L95: Since we've changed planet, re-state glacier-like forms *on Mars*.

L98: Avoid the term putative. This means generally accepted, which isn't always true particularly for recent papers. Instead use hypothesised, or equivalent.

P4,L00: Should this just be 'alcoves' here? Referring in general to alcoves, not your cirque like ones specifically.

P4,L02: If above change implemented, this can then become 'dedicated to identifying cirque-like alcoves'.

P4,L02: Here and elsewhere in the manuscript this should be regional population scale. Population without qualification implies global.

Figure 2: contour labels need to be larger in ii, and caption to state where panels are oblique views e.g. at least bii, and possibly also cii, as this distorts the shape significantly. It is also not made clear that aii and bii are the same cirque.

P8,L27: not clear what is meant by candidates here. One of the regions on Mars where available data is most highly consistent with…

P8,L30: avoid fretted - this is jargon and not particularly important. the key thing is that the terrain hosts valleys, plateaus and mesas.

Also, I don't think you need to say 'of disputed origin'. It just adds unnecessary confusion. What is important is that it is just the antecedent topography for glaciation.

P8, L31-32: Wouldn't this be better in the intro before you talk about the climate of the last 3Gyr? This also allows you to define Amazonian once only.

P8,L34: Mention glaciers specifically here.

P8,L38: sections on > sections towards

P8, L39: 'including'. These are the only features labelled. Delete

P9,L9: For this extent of study area, this projection will induce distortions towards the longitudinal extremes. The reviewer requested an explanation of the magnitude of these distortions, but this has not been provided in the revisions. For future reference (assuming the calculated uncertainties in sinusoidal are tolerable), a lambert projection, as in Fig 3 would have reduced these distortions.

P9, L65-71: This works well – thanks for sticking with us!

P10,L77: the stated equation doesn't match the words. Need to add 'taking the cube root of the product of'

P10, L94: delete 'to fit a table format'

P11, Altitudinal range row, column 4: delete minus – already say subtract

P11, Elevation row, column 4: Mean elevation of what? All pixels within the polygon?

P11, Aspect row, column 2: delete north (or degrees relative to north, but that isn't typically stated).

Figure 4: Contour labels need to be larger

Also Figure 4: the revisions to panel B have introduced an error - the H line should end at the end of the L line. currently the H line measures outside the alcove boundary.

P13, L08: alcove > alcoves

P13,L09: This isn't 'alignment', it is projection.

P13, L10: is 100m this the typical magnitude of misalignment? Are there any which exceed this?

P15,L26: The lip in this profile appears to correspond to where the transect intersects mantling materials, so I suspect it is not a true lip but the mantling deposit.

P15, L35: see earlier comment re icy geomorphic features. Better would be 'geomorphic features related to ice' - this could then include features *containing* ice and features formed by ice.

P15, L38: Better would be 'mantling deposit' – remember the mantle is a thing!

P16, linear terrain row, column 5: accelerating flow is inherently extensional. Not clear therefore what is meant by compressed accelerating flow.

P16, mantle row, column 3: This isn't the main descriptor of mantling deposits. There should be a preceding statement which is a primary description, then this can supplement it.

P16, mound and tail row, column 3: core of ice? not really possible to tell this at this scale. More likely to be sedimentary. Definitely not a core of ice if equivalent to drumlins.

P16, mound and tail row, column 4: I'd rephrase this as 'Drumlins, but with some morphometric differences'

P18, L53: alcoves *which* no

P18, L53: 'and raised moraine like ridge' this comes after 'no longer appear to', yet I can see arcuate features beyond the alcoves. Should this say that they *have* raised moraine-like ridges at their termini? If not, mention the arcuate features but state that they don't have significant relief - this is evidence that they once did contain glacial ice.

Figure 7: This is a really useful figure - thanks for adding. However, it needlessly repeats elements of Figure 3, and comes too late. Replace Figure 3 with this. You could also then use this figure to point to the locations of the examples shown in other figures.

P22, L83: Is it mean size? Or just size? Individual points are plotted. Please clarify and make clear in text.

P22, Table 4, column 4: Define somewhere how alcove volume was calculated. this isn't in the acme2 output table.

Fig 11: Could simplify this by only showing iii column – it repeats I and ii but is more useful because they are scaled.

P23,L11. 'Average area'. Axis label says volume not area

P23, L16: I would argue that a feature such as a moraine or streamlined bedform does not necessarily mean that there is still ice (be that active or relict). It just means there was once ice there.

P24, Table 5. Change imagery to images. This is a metric of occurrence (yes/no) per image, not according to the area of image in which it occurs. Imagery could be interpreted as the latter. You could make it even clearer by 'Percent of HiRISE images containing feature' (same with CTX column).

P24,L28: presence *or past presence* of ice, or processes of ice loss.

P26,L53: by being called cirque-like, they are inherently *candidate* cirques. Rephrase to 'whether cirque-like alcoves are *indeed* cirques

P27, L63: 'more episodes + lasted a longer amount of time': these are listed as part of the same hypothesis, but should be separated. Then at the end, you can say it could have been a combination of these factors.

P27, L64 'erosion rates on Mars were much more rapid'. I don't think anyone is realistically suggesting this. It is ok to explain which are the least likely explanations. So I recommend counteracting this hypothesis before the future modelling statement.

P27,L69: winds don't have a lee side, slopes do. So more likely to grow on the lee side of slopes crossed by westerly winds?

P27, L72: 'northern winter': under higher obliquity?

P27,L72-73: a bit repetitive given cirque bias is already stated. Simply state here that the observed bias is similar to glacier-like forms.

P27, L74: get rid of northerly, confusing when direction is opposite sense to a northerly wind. Just say poleward facing

P27,L77: what pattern? this reads as if they have an easterly bias (the main topic of the preceding paragraph). But I think you mean to compare to cirques on Earth.

P27,L77 'found glacier-like forms to have a poleward bias' – at global scales

P27,L78-79: grammar here is messy. Cut the sentence after 'glacier-like forms face southwards'. Then: in contrast, cirque-like alcoves in Deuteronilus.

P27,L78: Grammatically, 'this' is referring to the cirques, but you mean to refer to the GLF trend. Revisit the grammar of this entire paragraph - it could be a lot tighter.

P27, L81-82: Move 'for both…' statement before the statement where you talk about the largest cirque-like acloves by volume.

P27,L83-84: again, this is not written very clearly. Revisit this entire paragraph and tighten it up.

P27,L86: should the equator-facing statement be covered by the citations too?

P27, L87-87:. Statement starting 'as a result'. This sentence lacks a citation. I would also qualify southward-facing with (equator-facing), as you do in the conclusions - that is clearer

P28,L90: 'remains unclear'... …and alternative mechnisms which do not require liquid water have been demonstrated as possible mechansims for gully formation on Mars (citation).

P28, L91-92: 'water ice precipitation'…otherwise known as snow
[Figure]

P28, L96: 'would allow for increased insolation' need to explain under what climate/obliquity regime, since the earlier passage says that equator-facing slopes received less insolation. Separate these hypotheses out (there are currently contradictary statements in the same paragraph), and make clear when things are alternatives.

P28, L97-98: we explored this potential association in Section 5.1.2 – this is 5.1.2, delete.

P28, L98-99, sentence beginning 'on the other hand'. move this up amongst the explanation of DM GLF distributions at the start of this paragraph. You say the largest ones face southward, but it is an important point that most are pole-facing.

Figure 14: Captions need to have an introductory statement about what the figure shows overall.

P29, L21: It would make more sense for the GLF comparisons in the aspect section to move here.

P30,L56: space in may lack

P30, L56-57: Not sure what you mean by elongation out of the alcove

Section 5.2.2: I still agree with the reviewer that this section is too speculative. I suggest deleting, to focus on the actual results of the data collection.

P31,L75-76: I don't understand the need to invoke this alternative interpretation. Seems very unlikely.

P34,L42: millions of years > million years.

P35, L76-78: this should explain how the temperature at Beacon valley differs from the current mean annual temperature on Mars. The mention of the Butcher et al. velocities right at the end of this section - which are under Mars' current mean annual temperature comes too late, and contradicts the preceding statement that mars glacier surface velocities are unknown.

P36, L96-97: but this should be qualified by the fact that this is based on surface velocities for terrestrial glaciers, which (as the butcher et al study shows) may not be realistic for the typical temperatures at Mars' mid latitudes. Can note that the temperature history is not well understood and could have been warmer at times, but the temperature assumption in the calculations needs to be addressed.

P37, L27: This should move up to qualify the statement that surface velocities of glaciers are unknown. You should also note that this was for a thin VFF (<100m thick), *but* on a steep slope (these two effects will somewhat counter eachother, but perhaps thicker ice flowed faster, though perhaps also a thin vff is an ok analogue for a cirque glacier).

P38, L48: 'Kilometer-scale glaciation' odd phrasing. I don't think you really need to suggest you've extended the knowledge of glacial extent. The identification of cirque-like alcoves is enough - e.g., extended knowledge about the landscape imprints of formerly more extensive glaciation in Dueteronius Mensae in the past, and potential landscape evolution processes.

P38,L54: Make the first conclusion: 435 alcoves in Deuteronilus Mensae are morphometrically consistent with origins as glacially-eroded cirques.

P38,L57-58: does this explicitly explain the latitudinal gradient *within* the mid latitudes (as opposed to favourable ice accumulation in the mid-lats relative to the poles?)

P39, L80 'glacial conditions'. arguably using terrestrial velocities is not 'using Mars glacial conditions'.

P39,L83-84: Qualify this with 'however, these estimates are highly dependent upon the past flow velocity chosen, which is poorly constrained for past climate regimes on Mars.

Data availability statement: Previous versions of the manuscript stated that the alcove data would be released. Now it reads that it will be available by email. Following ESurf policy, if data will not be released, a robust justification must be given in the statement. I recommend releasing the data as previously stated.

---

## Author Response (AR3)

**L7:** Add space in zip code (AB24 3UF)

Corrected.

**L15:** minimal. Response to editor comments states this has been changed to 'some'. Change has not been made.

Corrected from minimal to some.

**L20:** 'Evaluate past glaciation on Mars' is vague. Say something like 'to better understand the potential contribution of glaciation to landscape evolution in Deuteronilus Mensae, Mars'.

Corrected.

**L21:** Here and throughout: avoid the term 'icy geomorphic' as this is ambiguous. Implies the feature itself is icy, but this isn't always the case – some are inferred as having formed by ice that is no longer there. Better 'geomorphic evidence for past glacial occupation of these cirque-like alcoves'

Corrected.

**L24-25:** These don't really work as alternatives to one-another as currently written - one is posed as a mechanism, and one as a requirement.

Additionally…

If tying to obliquity, then the high-insolation scenario is a different obliquity state to the low-insolation/high-accumulation state. Or it arises from a climate regime that has not yet been well constrained. Can you make this clearer?

We agree that this was previously phrased in a confusing way. We have modified it to read as follows: "One possibility to explain this trend is that southward facing cirque-like alcoves in the northern mid-latitudes formed when conditions were more favorable for ice accumulation during periods of high obliquity."

It is true that originally the intent of the sentence was to propose two possibilities for obliquity states (and periods for when cirque-like alcoves formed), but for the sake of simplicity, we only focus on one (which is also more consistent with the southern aspect observed for gullies).

**L26:** 'both glacier-like forms' – 'both the estimated ages of glacier-like forms'

Corrected.

**L30:** Add 'and the timing of the formation of cirque-like alcoves' to this statement

We edited the statement to read as follows: "Future work is needed to specify the timing of the formation of cirque-like alcoves and whether their formation requires warm-based erosion."

**L42:** these aren't 'ice flow regimes' they are thermal regimes.

Corrected to thermal regime.

**L43:** Could add Gallagher et al. 2021 – they found grooves

Corrected to include Gallagher et al., 2021.

**L64:** space after lag

Corrected.

**L65:** estimates ranging from > thickness estimates ranging from

Corrected.

L73: As stated in a previous review, the caption first needs to introduce the general purpose of the figure. E.g. 'Examples of viscous flow features and alcoves mapped in this study'. Ensure all figure captions have such a statement at the start.

This sentence was added for Fig. 1: "Examples of alcoves mapped in this study and viscous flow features mapped in prior studies."

The following sentences were added for other figures:

Fig. 2: "Example of a cirque-like alcove on Mars alongside cirques from different regions on Earth."

Fig. 12: "Examples of geomorphic features corresponding to past glacial occupation of cirque-like alcoves."

Fig. 13: "Examples of mesa slopes with shallow alcoves, larger alcoves, and adjacent ice."

Fig. 14: "Scatterplot of mean elevation versus latitude."

L75: aprons plural x 2

Corrected.

L81-82: Given the new addition of Figure 7 , and the sample sizes discussed, this isn't correct as 'all alcoves mapped in this study'. It is an example of alcoves mapped in a sub-region of the study area.

Corrected to "A zoomed out view of all alcoves (not just cirque-like alcoves) mapped in a sub-region of the study area."

L88: These processes don't result in basal slip. Basal slop drives (some of) these processes (quarrying and abrasion, not frost weathering).

Corrected to read as follows: "This occurs via a combination of basal slip, quarrying, abrasion (e.g., White, 1970), and frost weathering (e.g., Sanders et al., 2012), which all contribute toward a tendency for rotational flow (Evans, 2020)."

L88-89: Delete the detail about meltwater through the bergschrund and randkluft. Too much detail, and inaccurate – meltwater can reach the bed from elsewhere too, e.g. crevasses.

Deleted so that it now reads as follows: "This occurs via a combination of basal slip, quarrying, abrasion (e.g., White, 1970), and frost weathering (e.g., Sanders et al., 2012), which all contribute toward a tendency for rotational flow (Evans, 2020)."

L95: Since we've changed planet, re-state glacier-like forms *on Mars.*

Corrected.

L98: Avoid the term putative. This means generally accepted, which isn't always true particularly for recent papers. Instead use hypothesised, or equivalent.

Corrected and changed to hypothesized.

P4,L00: Should this just be 'alcoves' here? Referring in general to alcoves, not your cirque like ones specifically.

Yes, good catch. Corrected to just alcoves.

P4,L02: If above change implemented, this can then become 'dedicated to identifying cirque-like alcoves'.

Corrected.

P4,L02: Here and elsewhere in the manuscript this should be regional population scale. Population without qualification implies global.

Corrected.

Figure 2: contour labels need to be larger in ii, and caption to state where panels are  oblique views e.g. at least bii, and possibly also cii, as this distorts the shape significantly. It is also not made clear that aii and bii are the same cirque.
The caption now specifies that aii and bii are the same cirque as the caption for b) now states: "(b) Uinta Mountains, Utah, USA (40.74°N, 110.05°W), same location as (a)(ii)."

aii was adjusted so that the image is no longer an oblique view, and contour labels were increased in size:

[Figure]

[Figure]

P8,L27: not clear what is meant by candidates here. One of the regions on Mars where available data is most highly consistent with...
Corrected to read "one of the regions on Mars…"

P8,L30: avoid fretted - this is jargon and not particularly important. the key thing is that the terrain hosts valleys, plateaus and mesas.
Changed to "terrain hosting valleys and mesas."

Also, I don't think you need to say 'of disputed origin'. It just adds unnecessary confusion. What is important is that it is just the antecedent topography for glaciation.
Corrected and deleted "of disputed origin."

P8, L31-32: Wouldn't this be better in the intro before you talk about the climate of the last 3Gyr? This also allows you to define Amazonian once only.
Moved to the Intro.

P8,L34: Mention glaciers specifically here.
We added the following sentences: "Much of the mantling unit also overlies the viscous flow features in the region (Baker and Head, 2015; Baker and Carter, 2019). These viscous flow features likely formed in the Middle to Late Amazonian (Head et al., 2010)."

P8,L38: sections on > sections towards
Corrected.

P8, L39: 'including'. These are the only features labelled. Delete
Corrected to read as follows: "The major surface features Lyot Crater, Sinton Crater, and Mamers Valles are noted."

P9,L9: For this extent of study area, this projection will induce distortions towards the longitudinal extremes. The reviewer requested an explanation of the magnitude of these distortions, but this has not been provided in the revisions. For future reference (assuming the calculated uncertainties in sinusoidal are tolerable), a lambert projection, as in Fig 3 would have reduced these distortions.
Good to know for future reference, thanks!

P9, L65-71: This works well – thanks for sticking with us!
Great to hear, thanks for the comments!

P10,L77: the stated equation doesn't match the words. Need to add 'taking the cube root of the product of'
This was edited to include volume as well. The corrected text now reads: We also calculate the total cavity volume of the alcove by multiplying the length, width, and height (LWH) and find the size by then taking the cube root of volume: $\sqrt[3]{LWH}$, following previous work (e.g., Barr and Spagnolo, 2015)

P10, L94: delete 'to fit a table format'
Deleted.

P11, Altitudinal range row, column 4: delete minus – already say subtract
Corrected to read as follows: "Range of elevations found by subtracting minimum ax elevation from maximum elevation."

P11, Elevation row, column 4: Mean elevation of what? All pixels within the polygon?
Corrected to "Mean elevation of all pixels within the polygon."

P11, Aspect row, column 2: delete north (or degrees relative to north, but that isn't typically stated).
Deleted.

Figure 4: Contour labels need to be larger
Corrected in the figure below:

[Figure]

Also Figure 4: the revisions to panel B have introduced an error - the H line should end at the end of the L line. currently the H line measures outside the alcove boundary.
Corrected, see figure above.
P13, L08: alcove > alcoves
Corrected.
P13,L09: This isn't 'alignment', it is projection.
The word "aligned" was changed to "projected."
P13, L10: is 100m this the typical magnitude of misalignment? Are there any which exceed this?
Following the reviewer's comments, we have deleted that sentence and rephrased for clarification to the following: To evaluate any potential error introduced by misalignment between the CTX imagery and mosaicked HRSC DEM, we compared a CTX DEM to the mosaicked HRSC DEM for nine cirque-like alcoves. The CTX DEM that included coverage of nine cirque-like alcoves had the ID j02_045640_2209_xn_40n339w_j16_050901_2209_xn_40n339w. We found the average percent changes for these nine cirque-like alcoves between the two DEMs to be 2.55% for height, 6.44% for aspect, and 2.35% for slope. Since these percent changes were minimal, all less than 7%, we determine that the mosaicked HRSC DEM is an accurate method to evaluate the cirque-like alcoves.

P15,L26: The lip in this profile appears to correspond to where the transect intersects mantling materials, so I suspect it is not a true lip but the mantling deposit.
This figure was deleted based on the reviewer's comments.
P15, L35: see earlier comment re icy geomorphic features. Better would be 'geomorphic features related to ice' - this could then include features *containing* ice and features formed by ice.
Corrected.

P15, L38: Better would be 'mantling deposit' – remember the mantle is a thing!
Corrected to mantling deposit—though not that as in Table 3 and throughout, we were referring to the latitude-dependent mantle.

P16, linear terrain row, column 5: accelerating flow is inherently extensional. Not clear therefore what is meant by compressed accelerating flow.
Good catch, we deleted that part.

P16, mantle row, column 3: This isn't the main descriptor of mantling deposits. There should be a preceding statement which is a primary description, then this can supplement it.
Edited to include the following description: "A deposit of layers of ice and dust from meters to tens of meters thick; Characterized by a 'raised curvilinear edge for the upslope boundary' (Khuller et al. 2021)."

P16, mound and tail row, column 3: core of ice? not really possible to tell this at this scale. More likely to be sedimentary. Definitely not a core of ice if equivalent to drumlins.
This was a typo, deleted "core of ice."

P16, mound and tail row, column 4: I'd rephrase this as 'Drumlins, but with some morphometric differences'
Corrected.

P18, L53: alcoves *which* no
Corrected.

P18, L53: 'and raised moraine like ridge' this comes after 'no longer appear to', yet I can see arcuate features beyond the alcoves. Should this say that they *have* raised moraine-like ridges at their termini? If not, mention the arcuate features but state that they don't have significant relief - this is evidence that they once did contain glacial ice.
This was rephrased as follows: "(b) and (c) represent previously unmapped cirque-like alcoves which no longer appear to contain a volume of ice. While the unmapped cirque-like alcoves have arcuate features similar to moraine-like ridges that indicate past glacial ice, these arcuate features no longer have significant vertical relief."

Figure 7: This is a really useful figure - thanks for adding. However, it needlessly repeats elements of Figure 3, and comes too late. Replace Figure 3 with this. You could also then use this figure to point to the locations of the examples shown in other figures.
We were initially hesitant to do this because it brings the results early into the work prior to when cirque-like alcoves are introduced in the Methods section, but we have made the suggested change.

P22, L83: Is it mean size? Or just size? Individual points are plotted. Please clarify and make clear in text.
Corrected, it should just be size and area.

P22, Table 4, column 4: Define somewhere how alcove volume was calculated. this isn't in the acme2 output table.
The following sentence was added in 3.2.2: We also calculate the total cavity volume of the alcove by multiplying the length, width, and height ($LWH$) and find the size by then taking the cube root of volume: $\sqrt[3]{LWH}$, following previous work (e.g., Barr and Spagnolo, 2015).

Fig 11: Could simplify this by only showing iii column – it repeats I and ii but is more useful because they are scaled.

Simplified to only show iii column:

[Figure]

Figure 9: a) Bar plots of the aspect compared to the quantity of both cirque-like alcoves and glacier-like forms. b) Bar plots of the aspect compared to the average volume in each aspect direction for both cirque-like alcoves and glacier-like forms.

P23,L11. 'Average area'. Axis label says volume not area
Corrected to average volume.

P23, L16: I would argue that a feature such as a moraine or streamlined bedform does not necessarily mean that there is still ice (be that active or relict). It just means there was once ice there.
We agree and have edited the sentence to read as follows: "In addition to morphometric observations, we identified geomorphic features in association with the cirque-like alcoves as consistent with either past, remnant, or active ice in order to evaluate aspects of the glacial history in the cirque-like alcoves."

P24, Table 5. Change imagery to images. This is a metric of occurrence (yes/no) per image, not according to the area of image in which it occurs. Imagery could be interpreted as the latter. You could make it even clearer by 'Percent of HiRISE images containing feature' (same with CTX column).
Corrected.

P24,L28: presence *or past presence* of ice, or processes of ice loss.
Corrected.

P26,L53: by being called cirque-like, they are inherently *candidate* cirques. Rephrase to 'whether cirque-like alcoves are *indeed* cirques
Corrected.

P27, L63: 'more episodes + lasted a longer amount of time': these are listed as part of the same hypothesis, but should be separated. Then at the end, you can say it could have been a combination of these factors.
Corrected as follows: "This suggests that compared to Earth, the cirque-like alcoves in Deuteronilus Mensae on Mars may have formed due to more frequent glaciation events, longer-lasting glaciation, larger initial hollows for snow accumulation, faster glacial erosion rates (though this is considered unlikely; see Table 7), or a combination of these factors."

P27, L64 'erosion rates on Mars were much more rapid'. I don't think anyone is realistically suggesting this. It is ok to explain which are the least likely explanations. So I recommend counteracting this hypothesis before the future modelling statement.

This component was rephrased to state: "faster glacial erosion rates (though this is considered unlikely; see Table 7)."

P27,L69: winds don't have a lee side, slopes do. So more likely to grow on the lee side of  slopes crossed by westerly winds?

Good point, corrected.

P27, L72: 'northern winter': under higher obliquity?

Yes, corrected to include "under higher obliquity."

P27,L72-73: a bit repetitive given cirque bias is already stated. Simply state here that the  observed bias is similar to glacier-like forms.

The earlier sentence was moved down an combined with this sentence to read as follows: "Similar to cirque-like alcoves in Deuteronilus Mensae, the glacier-like form population in the northern hemisphere also has an eastward bias (Souness et al., 2012)."

P27, L74: get rid of northerly, confusing when direction is opposite sense to a northerly  wind. Just say poleward facing

Corrected to just say poleward facing.

P27,L77: what pattern? this reads as if they have an easterly bias (the main topic of the  preceding paragraph). But I think you mean to compare to cirques on Earth.

We rephrased to "This poleward bias is typically seen…"

P27,L77 'found glacier-like forms to have a poleward bias' – at global scales

Corrected to the following sentence and moved to 5.1.4 per a later comment: "Even though most glacier-like forms face the pole, the glacier-like forms with the largest volume face southwards in Deuteronilus Mensae (Souness et al., 2012; Fig. 9)."

P27,L78-79: grammar here is messy. Cut the sentence after 'glacier-like forms face  southwards'. Then: in contrast, cirque-like alcoves in Deuteronilus.

These sentences were edited to as follows and moved to section 5.1.4 as per the later comment: "Even though most glacier-like forms face the pole, the glacier-like forms with the largest volume face southwards in Deuteronilus Mensae (Souness et al., 2012; Fig. 9). This may be due to a localized topographic effect for glacier-like forms in Deuteronilus Mensae. Glacier-like forms flowing northward are larger than those flowing southward by about 20%, in the northern hemisphere (Brough et al., 2019). The largest cirque-like alcoves by volume face north and west, which may be due to topographic effects as well, since most alcoves face south. For both glacier-like forms and cirque-like alcoves, the aspect with the highest amount of the regional population does not correspond to the aspect with the largest mean volume (Fig. 9)."

P27,L78: Grammatically, 'this' is referring to the cirques, but you mean to refer to the  GLF trend. Revisit the grammar of this entire paragraph - it could be a lot tighter.

We reordered these two sentences as follows and moved them to 5.1.4 as per the later comment: "This may be due to a localized topographic effect for glacier-like forms in Deuteronilus Mensae because overall for the northern hemisphere, glacier-like forms flowing northward are larger than those flowing southward by about 20% (Brough et al., 2019). The largest cirque-like alcoves by volume face north and southwest, which may be due to topographic effects as well, since most alcoves face south."

P27, L81-82: Move 'for both…' statement before the statement where you talk about the  largest cirque-like acloves by volume.

We instead moved the following sentences to section 5.1.4 as per a later comment: "In contrast, the largest cirque-like alcoves by volume face north and southwest. For both glacier-like forms and cirque-like alcoves, the aspect with the highest percentage of the regional population does not correspond to the aspect with the largest mean volume."

P27,L83-84: again, this is not written very clearly. Revisit this entire paragraph and tighten it up.

This sentence was deleted for clarity.

P27,L86: should the equator-facing statement be covered by the citations too?

This isn't really explicitly stated, but we edited it to say "relatively less insolation" and moved the citation to the end of the sentence: "To explain the southward bias of cirque-like alcoves, we propose that this is consistent with periods of higher obliquity >45° on Mars, when poleward facing slopes received higher insolation and summer day temperatures, and equator-facing slopes received relatively less insolation (Costard et al., 2002; Kreslavsky et al., 2008)."

P27, L87-87:. Statement starting 'as a result'. This sentence lacks a citation. I would also qualify southward-facing with (equator-facing), as you do in the conclusions - that is clearer

Corrected to the following sentence, note that a citation was not included because this was a statement from this work: "As a result, during periods of high obliquity, equator-facing cirque-like alcoves in the northern mid-latitudes would have been more favorable for ice accumulation."

P28,L90: 'remains unclear'… ...and alternative mechnisms which do not require liquid water have been demonstrated as possible mechansims for gully formation on Mars (citation).

The following sentences were added instead: "Meltwater generation is more commonly invoked for the formation of older, inactive gullies during periods of higher obliquity (e.g., Dickson et al., 2023; Noblet et al., 2024), while gullies that have been observed to be recently active invoke $CO_2$ frost, well as dry mass wasting during frost-free seasons (e.g., Dundas et al., 2022), though melting within dusty $H_2O$ ice is also a possibility (e.g., Khuller et al., 2021)."

P28, L91-92: 'water ice precipitation'…otherwise known as snow �
Simplified to "snow."

P28, L96: 'would allow for increased insolation' need to explain under what climate/obliquity regime, since the earlier passage says that equator-facing slopes received less insolation. Separate these hypotheses out (there are currently contradictary statements in the same paragraph), and make clear when things are alternatives.

This was referring to present-day conditions (instead of high obliquity), but we have removed this sentence for clarity.

P28, L97-98: we explored this potential association in Section 5.1.2 – this is 5.1.2, delete.

Deleted.

P28, L98-99, sentence beginning 'on the other hand'. move this up amongst the explanation of DM GLF distributions at the start of this paragraph. You say the largest ones face southward, but it is an important point that most are pole-facing.

This paragraph was split into four paragraphs, and the section beginning with "on the other hand" was moved up as the second paragraph before the last two paragraphs on gullies in section 5.1.2.

Figure 14: Captions need to have an introductory statement about what the figure shows overall.

The caption (for what is now figure 13) first reads: "Scatterplot of elevation versus latitude."

P29, L21: It would make more sense for the GLF comparisons in the aspect section to move here.

Corrected so that the earlier two paragraphs are now in this section 5.1.4.

P30,L56: space in may lack

Corrected.

P30, L56-57: Not sure what you mean by elongation out of the alcove

This was rephrased as follows: "At a potentially earlier stage of evolution of the glacier-like forms, moraine-like ridges may reside within the alcove (Fig. 11a) rather than outside of the alcove (e.g., Fig. 11b, Fig. 5), potentially similar to a terrestrial cirque glacier sitting within the cirque basin instead of extending into the valley below and depositing a moraine there."

Section 5.2.2: I still agree with the reviewer that this section is too speculative. I suggest deleting, to focus on the actual results of the data collection.

This section was deleted. Following the reviewer's suggestion, the figure was modified and a portion of the text retained, though altered. This remaining text was moved to the end of section 5.1.2:

"Regardless of how gullies are initiated, they may act as a local depression in a location where snow could later accumulate for cirque-like alcove formation, such as if the gullies acted as a cold trap for snow (e.g., Dickson et al., 2023). For example, gullies could provide the initial concavity for a later cirque-like alcove to develop when glaciation occurs (Fig. 12), which is consistent with gully heads that have been proposed as initiation points for cirques on Earth (Derbyshire and Evans; 1976). In turn, deglaciation may also prime the landscape by exposing unconsolidated sediment for later gullying (e.g., Jawin and Head, 2021). Black arrows in Fig. 12a provides examples of shallow alcoves incised along the mesa slope. Fig. 12b includes similar shallow, elongate alcoves alongside larger alcoves with multiple channels. The shallow alcoves in Fig. 12a may indicate initial erosion of the mesa sidewall, while the larger alcoves in Fig. 12b may represent later stages of alcove development. While outside the scope of this study, additional analyses are necessary to evaluate this potential cycle as shallow alcoves or even gullies could create initiation points for cirque-like alcove formation and then deglaciation acts to prime the landscape for later gullying. Future work is necessary to elucidate this potential cyclical relationship of repeated ice accumulation and melt on the landscape."

P31,L75-76: I don't understand the need to invoke this alternative interpretation. Seems very unlikely.

This section was deleted.

P34,L42: millions of years > million years.

Corrected.

P35, L76-78: this should explain how the temperature at Beacon valley differs from the current mean annual temperature on Mars. The mention of the Butcher et al. velocities right at the end of this section - which are under Mars' current mean annual temperature comes too late, and contradicts the preceding statement that mars glacier surface velocities are unknown.

The text was edited to include the mention earlier, including differences in temperatures: "Both the past temperatures and surface velocities of glacier-like forms on Mars are not well constrained and may have included short, warm periods that allowed for melting (Hubbard et al., 2014). Recent modeling using a mean annual present-day surface temperature of 210 K found a maximum surface velocity of $20 \times 10^{-6}$ m/yr for a thin (<100 m) viscous flow feature on a steep slope (Butcher et al., 2024), which would yield an unrealistic age for the cirque-like alcoves that is older than the age of Mars. For our estimates, for the wet-based case, we use a surface velocity of 2 m/yr (Cook et al., 2020) and for the cold-based case we use a surface velocity of $1 \times 10^{-3}$ m/yr corresponding to a mean annual ground temperature of 250 K, which was measured for a rock glacier in the Beacon Valley sector of the McMurdo Dry Valleys of Antarctica (Rignot et al., 2002). This cold-based case represents a low glacier flow speed on Earth."

P36, L96-97: but this should be qualified by the fact that this is based on surface velocities for terrestrial glaciers, which (as the butcher et al study shows) may not be realistic for the typical temperatures at Mars' mid latitudes. Can note that the temperature history is not well understood and could have been warmer at times, but the temperature assumption in the calculations needs to be addressed.

As mentioned above, the sentence now reads: "Both the past temperatures and surface velocities of glacier-like forms on Mars are not well constrained and may have included short, warm periods that allowed for melting (Hubbard et al., 2014)."

P37, L27: This should move up to qualify the statement that surface velocities of glaciers are unknown. You should also note that this was for a thin VFF (<100m thick), *but* on a steep slope (these two effects will somewhat counter eachother, but perhaps thicker ice flowed faster, though perhaps also a thin vff is an ok analogue for a cirque glacier).

As mentioned above, the sentence now reads: "Recent modeling using a mean annual present-day surface temperature of 210 K found a maximum surface velocity of $20 \times 10^{-6}$ m/yr for a thin (<100 m) viscous flow feature on a steep slope (Butcher et al., 2024), which would yield an unrealistic age for the cirque-like alcoves that is older than the age of Mars."

P38, L48: 'Kilometer-scale glaciation' odd phrasing. I don't think you really need to suggest you've extended the knowledge of glacial extent. The identification of cirque like alcoves is enough - e.g., extended knowledge about the landscape imprints of formerly more extensive glaciation in Dueteronius Mensae in the past, and potential landscape evolution processes.

Following this suggestion, we edited the sentence to now read as follows: "Thus, these cirque-like alcoves expand our knowledge about the landscape imprints of formerly more extensive glaciation in Dueteronius Mensae in the past and potential landscape evolution processes in this region."

P38, L54: Make the first conclusion: 435 alcoves in Deuteronilus Mensae are morphometrically consistent with origins as glacially-eroded cirques.

We accept this change and made it the first concluding point.

P38, L57-58: does this explicitly explain the latitudinal gradient *within* the mid latitudes (as opposed to favourable ice accumulation in the mid-lats relative to the poles?)

We agree that the phrasing here was confusing. We have reworded the last sentence here: "The largest cirques are in the lower latitudes of the study region at 40-42.9°N (Fig. 7a). This is likely the result of local topography because the local mesa height at different latitudes may limit local cirque-like alcove height and size (Section 5.1.3)."

We also moved the previous sentence to a different bullet point: "There is a dominant southward bias in the aspect of the cirque-like alcoves (Fig. 6), which becomes more pronounced above 46.5°N. This likely suggests cirque-like alcove formation during a period of high obliquity when conditions were more favorable for glacier growth at these latitudes because poleward-facing slopes received higher insolation and warmer summer daytime temperatures than equator-facing slopes."

P39, L80 'glacial conditions'. arguably using terrestrial velocities is not 'using Mars glacial conditions'.

We changed "glacial conditions" to "temperatures."

P39,L83-84: Qualify this with 'however, these estimates are highly dependent upon the past flow velocity chosen, which is poorly constrained for past climate regimes on Mars.

We added in the statement "These estimates are highly dependent upon the past flow velocity chosen, which is poorly constrained for past climate regimes on Mars."

Data availability statement: Previous versions of the manuscript stated that the alcove data would be released. Now it reads that it will be available by email. Following ESurf policy, if data will not be released, a robust justification must be given in the statement. I recommend releasing the data as previously stated.

We will release the data as previously stated.

**egusphere-2023-2568 Review by S. Conway**

The manuscript has greatly improved in readability, clarity and robustness since the last review and I congratulate the authors on this. Notably they have sufficiently toned down the message on wet based glaciation, and made a good effort to shorten the manuscript. However, there are two aspects outstanding from my previous review that must be addressed before publication, together with a few other remaining loose ends detailed at the end and in the annotated PDF.

We thank the reviewer for taking the time to provide detailed comments on the manuscript. Below we include responses to the reviewer's main comments, and responses for shorter comments are in the attached annotated PDF.

1) the authors have to acknowledge and assess the impact of the uncertainties and errors in the elevation data they use for their analysis (see below for further details).

We have addressed the impact of uncertainties and errors below using three methods to demonstrate that any error introduced by mosaicking the HRSC DEM data is negligible. To summarize, this included negligible differences between CTX and HRSC DEM values, negligible differences in HRSC profiles across seamlines, and consistency in point locations in overlaps between HRSC DEM frames.

2) the authors should remove the highly speculative section 5.2.2 in the discussion, which detracts from the solid data they present in the results by making highly speculative interpretations based on image data and analysis techniques that are not reported in their methods/results. The discussion in section 5.2.2 is not needed as a basis of the main conclusions because the authors already make the point that former-gullies could be the initiation points for cirque growth when discussing the aspect trends. The existence of notches that resemble gully alcoves, if retained, needs to be demonstrated by showing some gully alcoves for comparison.

We have removed section 5.2.2. Figure 12 is a modified version of a prior figure from that section, and the associated text are now in section 5.1.2.

Associated text and figure (also provided below):

"In turn, deglaciation may also prime the landscape by exposing unconsolidated sediment for later gullying (e.g., Jawin and Head, 2021). Fig. 12a provides examples of shallow alcoves incised along the mesa slope. Fig. 12b includes similar shallow, elongate alcoves alongside larger alcoves with multiple channels. The shallow alcoves in Fig. 12a may indicate initial erosion of the mesa sidewall, while the larger alcoves in Fig. 12b may represent later stages of alcove development. Flow features indicate downslope flow of ice away from the mesa sidewalls. While outside the scope of this study, additional analyses are necessary to evaluate this potential cycle as shallow alcoves or even gullies could create initiation points for cirque-like alcove formation and then deglaciation acts to prime the landscape for later gullying. Future work is necessary to elucidate this potential cyclical relationship of repeated ice accumulation and melt on the landscape."

[Figure]

**Figure 12: Examples of mesa slopes with shallow alcoves, larger alcoves, and adjacent ice. (a) Shallow alcoves may indicate ice-associated erosion all along the mesa sidewall. Flow features indicate the downslope direction of ice flow. Centered at 41.06°N, 17.88°E in CTX image D04_0288880_2193_XI_39N342W. (b) Shallow alcoves may indicate ice-associated erosion while larger alcoves with multiple channels may represent a later stage of development. Flow features indicate the downslope direction of ice flow. Centered at 40.02°N, 23.20°E in CTX image D21_035499_2203_XN_40N336W. HiRISE data credit: NASA/JPL/University of Arizona. CTX data credit: Caltech/NASA/JPL/MSSS.**

**General comments from previous review**

Original comment (c) when considering the uncertainties in the elevation data in section 3.2.3 please address how these may also affect the ACME data collection, specifically consider the noise in the HRSC product (clearly visible as step-artefacts on Figs 4 and 6), and how well the CTX and HRSC data were co-registered. Noise is accentuated in topographic derivatives such as slope, which is amongst the parameters extracted. Presumably the position of the long profile was determined based on the CTX image data (if this or is not the case then it should be described in the methods as mentioned in point b), hence co-registration is critical to have reliable and representative elevation data. Please state what projection system was used for the morphometric analyses and consider whether this introduced any uncertainties/distortion (including the slope calculation from the HRSC DTM). RESPONSE: We added the following text to what is now section 3.2.3 "Uncertainties in elevation and alcove longitudinal profile": We mapped the cirque-like alcove and identified the mid-threshold point using the CTX imagery. As mentioned in Section 3.1, both the CTX imagery and HRSC DEM were aligned to a Sinusoidal projection centered on longitude 25.5 degrees East, and were based on the IAU Mars 2000 Sphere datum. Any misalignment of up to 100 m between the image and the DTM is of little concern when it translates into metrics made by ACME2 since most metrics rely on multiple pixel measurements. This is certainly the case for slope, aspect and average elevation along the cirque length or the entire cirque area. Any misalignment might affect minimum and maximum elevation, but this is not a concern when using a large sample size to evaluate population-scale metrics."

I don't think this is sufficient and I have included annotations in the PDF, but the main points are coped here:
- Line 53 "Mosaic to Raster" - describe what this tool does for those who might want to use another software, for example it is important to know what was done at the seamlines as this can cause big jumps in topography. Also what was the pixel size of the resulting DEM? Presumably the data were all reprojected to the sinusoidal projection of the measurements -
this should also be specified because reprojection can introduce artefacts when resampling the raster data.

We added the following lines: "The HRSC DEMs were mosaicked together using the Mosaic to New Raster tool in ArcGIS Pro using the method of "Last", which is where the output cell value of the overlapping areas will be the value from the last raster dataset mosaicked into that location. The pixel size of the resulting DEM was 100 m." The last sentence of the paragraph was also edited to be stated earlier in the paragraph and mention the projection: "Measurements from all of the imagery and DEMs were projected to use a Sinusoidal projection centered on longitude 25.5 degrees East, and were based on the IAU Mars 2000 Sphere datum."

To evaluate consistency across DEM frames, we found the elevation values of 20 random points that were at overlap locations and found the average percent change to be negligible at only 0.86%. The values for each point are provided in the table below. Note that DEM A and DEM B are different DEMs in each row in the table.

| Point | DEM A (meters) | DEM B (meters) | Percent Change |
|---|---|---|---|
| 1 | -3447 | -3403 | 1.293 |
| 2 | -3355 | -3330 | 0.751 |
| 3 | -3214 | -3233 | 0.588 |
| 4 | -3558 | -3525 | 0.936 |
| 5 | -3877 | -3882 | 0.129 |
| 6 | -1294 | -1276 | 1.411 |
| 7 | -1452 | -1439 | 0.903 |
| 8 | -767 | -783 | 2.043 |
| 9 | -2620 | -2548 | 2.826 |
| 10 | -2843 | -2809 | 1.210 |
| 11 | -1024 | -1023 | 0.098 |
| 12 | -3048 | -3065 | 0.555 |
| 13 | -3214 | -3233 | 0.588 |
| 14 | -1043 | -1027 | 1.558 |
| 15 | -2989 | -3004 | 0.499 |

| | | | |
|---|---|---|---|
| 16 | -2956 | -2957 | 0.034 |
| 17 | -3214 | -3233 | 0.588 |
| 18 | -1785 | -1791 | 0.335 |
| 19 | -2227 | -2230 | 0.135 |
| Average | | | 0.867 |

In addition, we evaluated the profiles across five random seamline locations of HRSC DEMs. Screenshots are provided below with points at the seams. Four of the profiles definitively demonstrate that there are no jumps between seams (pictured above the profile) in the mosaicked HRSC profile (used for the DEM profile). In one of the profiles, it is unclear if there is a jump or if it is the nature of the terrain. Either way, the difference is ~40 m, which is within the 100 m pixel resolution of the HRSC mosaic.

[Figure]

[Figure]

In the example below, there is a dip in elevation by ~40 m where the seamline is. This seems to be indicative of the actual terrain rather than a seamline jump. However, even if this is caused by the seamline between the two DEMs, this difference is within the 100 m pixel size for the DEM mosaic.

[Figure]

[Figure]

- Line 107 onwards: This section should evaluate the influence of the poor resolution and noisy  quality of the HRSC data for making measurements with the ACME2. As the authors have  compared to CTX an obvious way of doing this is to apply the software to alcoves and CTX
and HRSC DEMs and then compare the results. But at the VERY LEAST the fact that noise  exists in the HRSC data and seams from the mosaicking process should be acknowledged and  some attempt to evaluate how this noise/artefacts influence the results. Notably the noise  can bias the results for aspect and slope, e.g. lots of noise means lots of steep pixels that  don't really exist so the slopes will be systematically too high. For aspect there may be over or under representation of certain orientations due to the shadows in the original images that make up the HRSC stereopair.
We have investigated the potential error introduced by using HRSC as follows:

1) There is limited CTX DEM coverage, however, we found one CTX DEM j02_045640_2209_xn_40n339w_j16_050901_2209_xn_40n339w publicly available that includes nine cirque-like alcoves. We reran ACME2 for the nine cirque-like alcoves using the CTX DEM to compare with the results from the HRSC DEM using percent change, calculated by using the following equation: |HRSC value - CTX value| / CTX value * 100%. As expected, length, width, and 2D area were exactly the same. The average percent changes for these nine alcoves were 2.55% for height, 6.44% for aspect, and 2.35% for slope. Since these percent changes were minimal, all less than 7%, we determine that the HRSC DEM is an accurate method to evaluate the cirque-like alcoves. We added the following text to section 3.2.3: "To evaluate any potential error introduced by misalignment between the CTX imagery and mosaicked HRSC DEM, we compared a CTX DEM to the HRSC DEM mosaic for nine cirque-like alcoves. The CTX DEM that included coverage of nine cirque-like alcoves had the ID j02_045640_2209_xn_40n339w_j16_050901_2209_xn_40n339w. We found the average percent changes for these nine cirque-like alcoves between the two DEMs to be 2.55% for height, 6.44% for aspect, and 2.35% for slope. Since these percent changes were minimal, all less than 7%, we determine that the mosaicked HRSC DEM is an accurate method to evaluate the cirque-like alcoves."

2) To evaluate any issues in mosaicking between HRSC frames, we evaluated 20 random points that are located on overlapping frames and compared their elevation values. We found the percent change to be negligible, at only 0.86%. The exact elevation values at each point are provided in the table above.

3) We also provide five examples of profiles from the mosaicked HRSC DEM across where there were seamlines. Four of these demonstrate that there are no obvious jumps. In one case, it is unclear if there is a potential jump of ~40 m, but regardless, it is within the 100 m pixel size of the DEM.

- Line 109: in Section 3.1 there was no mention of "alignment" just "Measurements from all of the imagery and DEMs used a Sinusoidal projection centered on longitude 25.5 degrees East", so please detail the alignment performed in section 3.1 (or say if no alignment was performed)
We added the following sentence in section 3.1: "No alignment was performed, however, we assessed the alignment by examining 20 random points across boundaries of various HRSC frames and found the difference to be negligible because the average percent change was only 0.86%."

- Line 111 "of little concern" - this is a subjective statement, please instead elucidate why it is of little concern and the quantitative data to support this statement
We have deleted this statement and added the following: "To evaluate any potential error introduced by misalignment between the CTX imagery and mosaicked HRSC DEM, we compared a CTX DEM to the HRSC DEM mosaic for nine cirque-like alcoves. The CTX DEM that included coverage of nine cirque-like alcoves had the ID j02_045640_2209_xn_40n339w_j16_050901_2209_xn_40n339w. We found the average percent changes for these nine cirque-like alcoves between the two DEMs to be 2.55% for height, 6.44% for aspect, and 2.35% for slope. Since these percent changes were minimal, all less than 7%, we determine that the mosaicked HRSC DEM is an accurate method to evaluate the cirque-like alcoves."

- Line 111: in what ways do multiple pixels help? If for example the offset CTX-HRSC was always 100m to the west, aspect would be systematically wrong.

We apologize for poor phrasing. What we meant to say here was not that there is a misalignment of 100 m between the image and the DTM, but rather that there is a 100 m native resolution in the mosaicked DEM. This means that any time a mapped polygon touches a DEM pixel, it will be included in the analysis. This can result in up to a 100 m difference. However, over a 100 m length scale, the topography doesn't change much for these cirque-like alcoves. We have rewritten this section as follows: "To evaluate any potential error introduced by misalignment between the CTX imagery and mosaicked HRSC DEM, we compared a CTX DEM to the HRSC DEM mosaic for nine cirque-like alcoves. The CTX DEM that included coverage of nine cirque-like alcoves had the ID j02_045640_2209_xn_40n339w_j16_050901_2209_xn_40n339w. We found the average percent changes for these nine cirque-like alcoves between the two DEMs to be 2.55% for height, 6.44% for aspect, and 2.35% for slope. Since these percent changes were minimal, all less than 7%, we determine that the mosaicked HRSC DEM is an accurate method to evaluate the cirque-like alcoves."

To illustrate this issue of noise in the HRSC DEM data (here at its native 75 m/pix), I show below CTX beta01 compared to a portion of hillshaded HRSC ND4 h5213_0000 and its slope map. Clearly the red pixels which appear in bands across the smooth hillslope and exceed 30° are erroneous – such bands would have a strong influence on a mean value of slope across these hillslopes. The black box is an object approximately the size of the alcoves studied here. Seams between the HRSC DEMs would create significant E-W steps, rather than across slope steps, depending on the mosaic method utilised.

[Figure]

As mentioned above, we use three methods to demonstrate that any error introduced by mosaicking the HRSC DEM data is negligible. To summarize, this included negligible differences between CTX and HRSC DEM values, negligible differences in HRSC profiles across seamlines, and consistency in point locations in overlaps between HRSC DEM frames.

Original Comment on Page 30 *** it is impossible to know if what you interpret as "less developed" landforms is how the "more developed ones" looked without a time machine or at least a good knowledge of the process(es) and their rates. I suggest complete removal of this section, it is complete speculation.
RESPONSE: We find it plausible that in the order of increasing amount of erosion, the mesa edge would go from straight to having shallow depressions then deeper depressions. As such, we kept this section and added the following sentences at the beginning of the second paragraph:
"Here, we assume that the side of the mesa evolves from a straight edge to an increasing number and depth of depressions. An alternative interpretation might be that the deeper depressions were subsequently filled up to create a straight edge, however, we do not see evidence for this amount of infilling." If the reviewer thinks that further analyses might be beneficial, we are open to any suggestions for additional analyses here.
Following the sentence "We suggest that the observed notches are gullies and would be able to act as necessary initiation points for ice accumulation that would later support glaciation and erosion that could form cirque-like alcoves," we also added this reference since this idea of gullying tied to alcove formation has been previously proposed in a paraglacial by Jawin et al. 2018 as well: "This is consistent with the mechanism proposed by Jawin et al. (2018)."
This response does not address my main concern that this section is too speculative – notably the figure includes interpretations that are not sufficiently substantiated with insufficient evidence to support them. For example, to demonstrate convincingly that the scarps mid-hillslope are "detached slabs" rather than just tectonic features would require detailed topographic and image analysis beyond the scope of this work (and even how relevant these observations are to the point that is being made is potentially debatable). I suggest the authors could briefly show small alcoves ("notches") that didn't reach their cirque threshold in a figure, compare them to gully alcoves and briefly suggest that there may be a continuum of features, but a whole section should not be devoted to this point. The connection to gullies extends way beyond what results can actually support in the current presentation.

We agree that this section was previously a bit speculative. As suggested, we have removed the section and retained a portion of it that was added to the last paragraph of 5.1.2: "In turn, deglaciation may also prime the landscape by exposing unconsolidated sediment for later gullying (e.g., Jawin and Head, 2021). Fig. 12a provides examples of shallow alcoves incised along the mesa slope. Fig. 12b includes similar shallow, elongate alcoves alongside larger alcoves with multiple channels. The shallow alcoves in Fig. 12a may indicate initial erosion of the mesa sidewall, while the larger alcoves in Fig. 12b may represent later stages of alcove development. Flow features indicate downslope flow of ice away from the mesa sidewalls. While outside the scope of this study, additional analyses are necessary to evaluate this potential cycle as shallow alcoves or even gullies could create initiation points for cirque-like alcove formation and then deglaciation acts to prime the landscape for later gullying. Future work is necessary to elucidate this potential cyclical relationship of repeated ice accumulation and melt on the landscape."

Figure 12 (previously Fig. 14) has been updated as well:

[Figure]

**Figure 12: Examples of mesa slopes with shallow alcoves, larger alcoves, and adjacent ice. (a) Centered at 41.06°N, 17.88°E in CTX image D04_0288880_2193_XI_39N342W. Shallow alcoves may indicate ice-associated erosion all along the mesa sidewall. Flow features indicate the downslope direction of ice flow. (b) Centered at 40.02°N, 23.20°E in CTX image D21_035499_2203_XN_40N336W. Shallow alcoves may indicate ice-associated erosion while larger alcoves with multiple channels may represent a later stage of development. Flow features indicate the downslope direction of ice flow. HiRISE data credit: NASA/JPL/University of Arizona. CTX data credit: Caltech/NASA/JPL/MSSS.**

Further points
- The supplementary data and/or data availability statement needs to be improved. Notably these is confusion as to which version of the CTX mosaic was used. It would be best-practise to include the datatable and shapefiles with the paper. Further details on the HRSC data are needed.

The shapefiles and spreadsheet of the cirque-like alcoves mapped in this study are provided. The HRSC DEM is also provided and was mosaicked using the following 29 Level 4 HRSC data frames: h5436_0000_da4, h5418_0000_da4, h5400_0000_da4, h5364_0000_da4, h5339_0000_da4, h5328_0000_da4, h5321_0000_da4, h5310_0000_da4, h5303_0000_da4, h5285_0000_da4, h5267_0000_da4, h5249_0000_da4, h5231_0000_da4, h5213_0000_da4, h3304_0000_da4, h3293_0000_da4, h3249_0000_da4, h3183_0000_da4, h2191_0000_da4, h1644_0000_da4, h1622_0000_da4, h1571_0000_da4, h1289_0000_da4, h1395_0000_da4, h1461_0000_da4, h1450_0000_da4, h1428_0000_da4, h1483_0000_da4, and h1201_0000_da4. The Level 4 HRSC data frames can be accessed at the ESA Planetary Science Archive: http://www.rssd.esa.int/index.php?project=PSA, HRSCview by FU Berlin/DLR: http://hrscview.fu-berlin.de/, or the NASA Planetary Data Science (PDS) http://pds-geosciences.wustl.edu/missions/mars_express/. This paper was prepared using the beta01 CTX mosaic: https://murray-lab.caltech.edu/CTX/beta01.html (Dickson et al., 2018).

- The methods to calculate the alcove volumes presented in section 5.1.1 need to be presented and the results should also be first presented in the results before appearing in the discussion.

We think that based on additional comments on section 4.2 in the results section, the reviewer meant 4.2 not 5.1.1 in the discussion section (which is about size). We updated the following text in section 3.2.2 in methods to include the volume calculation: "We also calculate the alcove volume by multiplying the length, width, and height (LWH) and size by then taking the cube root of volume: $\sqrt[3]{LWH}$." In addition, in the caption in Table 4, we added the following: "Area and volume for glacier-like forms are found by Brough et al. (2019), where glacier-like form volume, including both debris and ice, is calculated using a volume-area scaling approach. Statistics for the cirque-like alcoves come from the topographic cavity of the alcove. We use total cavity volume as an approximation here, though cirque-like alcoves in their present state are not completely full of ice."

**Cirque-like alcoves in the northern mid-latitudes of Mars as evidence of glacial erosion**

An Y. Li[1], Michelle R. Koutnik[1], Stephen Brough[2], Matteo Spagnolo[3], Iestyn Barr[4]

[1]Department of Earth and Space Sciences and Astrobiology Program, University of Washington, Seattle, 98195, USA
[2]Department of Geography and Planning, School of Environmental Sciences, University of Liverpool, Liverpool, L69 7ZT, UK
[3]School of Geosciences, University of Aberdeen, Aberdeen, AB243UF, UK
[4]Department of Natural Sciences, Manchester Metropolitan University, Manchester, M15 6BH UK, UK

*Correspondence to*: An Y. Li (anli7@uw.edu)

**Abstract.** Viscous flow features known as glacier-like forms on Mars have been observed emerging from alcoves that resemble cirques on Earth. However, many alcoves exist without associated glacier-like forms, and these features have never been studied or categorized at a population scale. On Earth, cirques form when depressions on mountain slopes accumulate snow, which gradually compacts into glacial ice. As the glacier flows downhill, it deepens the depression through erosion. Most of this erosion is driven by wet-based glaciers, although cold-based glaciers can also contribute to minimal headward and sidewall retreat in some cases. Here, we present evidence that cirque-like alcoves on Mars, similar to terrestrial cirques, are shaped by glacial erosion. To assess which alcoves on Mars are most "cirque-like", we mapped a population of ~2000 alcoves in Deuteronilus Mensae, a region in the mid-latitudes of Mars characterized by mesas surrounded by glacial remnants. Based on visual characteristics and morphometrics, we refined our dataset to 435 "cirque-like alcoves"–nearly six times the amount of glacier-like forms in the region–and used this to evaluate past glaciation on Mars. High-resolution imagery reveals icy geomorphic evidence of glacial occupation within these cirque-like alcoves, including flow features, linear terrain, mantle, moraine-like ridges, mound-and-tail terrain, polygonal terrain, moraine-like ridges, rectilinear-ridge terrain, and washboard terrain. Most cirque-like alcoves face south to southeast, similar to gullies poleward of 40°. Two possibilities to explain this trend are that southward facing cirque-like alcoves in the northern mid-latitudes were more favorable for ice accumulation during periods of high obliquity, or alternatively, increased insolation and meltwater is necessary for cirque-like alcove erosion. Using wet-based glacial erosion rates, the timescales for martian cirque-like alcoves align with both glacier-like forms (millions to tens of millions of years) and other viscous flow features such as lobate debris aprons (hundreds of millions of years). In contrast, cold-based erosion rates are only consistent with the older ages of lobate debris aprons. By mapping cirque-like alcoves at a large scale for the first time, we expand the catalog of features attributed to glacial erosion on Mars. Future work is needed to determine whether their formation requires warm-based erosion.

**1 Introduction**

The surface morphology of the mid-latitudes of Mars (especially between 30 and 60°, north and south) is characterized by glacial remnants in the form of subsurface ice (Fig. 1; e.g., Brough et al., 2019; Levy et al., 2014), and icy mantling deposits (Mustard et al., 2001). Observations and modeling suggest that the climate of the past 3 Gyr was unlikely to have permitted widespread liquid water on the planet's surface, though spatially-limited liquid may have occurred under some conditions (e.g., Kite, 2019). Glacial remnants in the mid-latitudes of Mars, referred to as viscous flow features, are typically considered to have always been frozen to their beds (cold-based) with limited subglacial erosion and ice flow only by internal deformation and gravity-driven viscous creep throughout their evolution (e.g., Mangold and Allemand, 2001; Head and Marchant, 2003; Shean et al., 2005; Mackay et al., 2014). Previous work has estimated cold-based erosion rates of 0.1-10 m/Myr during Amazonian glaciation on Mars (Levy et al., 2016). However, the presence of glacial landforms such as moraines and lineations observed in tandem with at least select viscous flow features suggests subglacial erosion could have occurred and that the ice-flow regime over the evolution of these landforms could have been formerly wet-based or at least a mixed thermal state as polythermal (e.g., Arfstrom and Hartmann, 2005; Morgan et al., 2009; Hubbard et al., 2011; Hubbard et al., 2014). In addition, recent work proposed that some depositional and erosional evidence of wet-based glaciation within the last 1 Gyr (Middle to Late Amazonian) exists, especially in the form of eskers, which would indicate warmer subglacial conditions at these sites (e.g., Gallagher and Balme, 2015; Butcher et al., 2017; Butcher et al., 2021; Gallagher et al., 2021, Woodley et al., 2022). Englacial debris bands have been found in viscous flow features, though it is unclear if the debris is from rockfall at the headwall or eroded and then entrained from the subglacial bed (e.g., Butcher et al., 2024; Levy et al., 2021).

Viscous flow features include the landform classifications of glacier-like forms (Souness et al., 2012), lobate debris aprons, lineated valley fill, and concentric crater fill (e.g., Squyres, 1979; Milliken et al., 2003; Levy et al., 2014). In the cases where subsurface radar sounding data are available, lobate debris aprons consist of up to ~90% ice (Holt et al., 2008; Plaut et al., 2009), and they account for ~63% of the total volume of ice contained within all viscous flow features (Levy et al., 2014). Lobate debris aprons can be a few to tens of kilometers long and up to one kilometer thick (Holt et al., 2008; Plaut et al., 2009). In comparison, glacier-like forms are generally smaller, on average ~4.66 km long, ~1.27 km wide (Souness et al., 2012), and ~130 m thick (Brough et al., 2019). All mid-latitude viscous flow features are believed to have been deposited during orbital excursions of ≥45° in the Amazonian (Madeleine et al., 2009). Lobate debris aprons, lineated valley fills, and concentric crater fills are estimated to range from ~10 Myr to 1.2 Gyr in age (Morgan et al., 2009; Berman et al., 2015), with most age estimations on the order of hundreds of millions of years (e.g., 300-800 Myr from Fassett et al. 2014). Glacier-like forms can superpose lobate debris aprons or lineated valley fills, suggesting polyphase glaciation with age clusters estimated to be around 2-20 Myr and 45-65 Myr (Hepburn et al., 2020). In addition to viscous flow features, there is a separate icy mantling deposit over the mid-latitudes originating from airfall deposits of ice nucleated on dust, known as the latitude-dependent mantle (e.g., Mustard et al., 2001; Kreslavsky and Head, 2002; Schon et al., 2009; Conway et al., 2018). The latitude-dependent mantle consists of different layers rich in water ice and dust (Schon et al., 2009). The ice was deposited during high obliquity excursions and the dust formed during low obliquities when the ice sublimated and left behind a dusty lag(Schon et al., 2009). Although the mantling unit covers >23% of the surface of Mars (Kreslavsky and Head, 2002), with estimates ranging from 1-30 m thick (Mustard et al., 2001; Conway and Balme, 2014), the mantling unit represents a small contribution ($10^3$-$10^4$ km$^3$) to the overall ice volume (Conway and Balme, 2014). In some locations the mantling unit has been mapped as "pasted-on terrain", though it remains unclear whether the pasted-on terrain is a thicker mantling layer that is separate from an overlying mantle (Conway
et al., 2018). The icy mantling unit is estimated to be younger than glacier-like forms at 0.15 to 10 Myr in age (Willmes et al.,
2012; Schon et al., 2012).

[Figure]

**Figure 1: Legend in the bottom left applies to panels (a) and (c). Black-filled polygons are alcoves mapped as part of this study, yellow-filled polygons are previously mapped glacier-like forms (Brough et al., 2019), blue represents the previously mapped lobate debris apron (Levy et al., 2014), and pink represents an updated map of lobate debris apron (Baker and Head, 2015). Note that there is some overlap between what previous studies generally classified as lobate debris apron and Brough et al. (2019) later specifically defined as glacier-like forms. Green-filled polygons represent lineated valley fill mapped by Levy et al. (2014). (a) Standalone mesa in Deuteronilus Mensae, centered at 45.5°N, 26.3°E. (b) The same mesa as in (a) but without mapped units delineated. The basemap is the CTX mosaic (Dickson et al., 2023a) overlaid on a High Resolution Stereo Camera (HRSC) (Neukum et al., 2004) digital elevation model (DEM)**

was mosaicked from 29 frames. (c) A zoomed out view of all alcoves (not just cirque-like alcoves) mapped in this study. The area is centered at 41.5°N, 34.0°E. (d) Same area as in (c) but without mapped units delineated. CTX data credit: Caltech/NASA/JPL/MSSS. HRSC data credit: ESA/DLR/FU Berlin.

Glacial cirques on Earth are characterized by a concave basin connected to a steep headwall, often with a threshold or lip of higher topography at the lower end of the basin (Fig. 2biii). Cirques develop from incipient depressions in mountain and plateau sides that fill with snow/ice and over time support active glaciers that deepen the depressions by glacial erosion (Evans and Cox, 1974; Glasser and Bennett, 2004). This occurs via a combination of quarrying, abrasion (e.g., White, 1970), and frost weathering (e.g., Sanders et al., 2012) resulting in basal slip that is facilitated by rain and meltwater that goes through the bergschrund and randkluft, which all contribute toward a tendency for rotational flow (Evans, 2020). However, it is debated whether non-glacial processes such as rock-slope failures may have a substantial contribution to erosion as well (e.g., Turnball and Davies, 2006; Coquin et al., 2019; Evans, 2020). Due to their presence at high topographic locations on Earth and due to their concave shape, cirques trap snow and ice and are often the first sites to glaciate and the last sites to deglaciate (Graf, 1976). On Earth, with over 10,000 glacial cirques mapped globally, landform morphometrics are used to reveal regional climatic trends and the extent of glaciation in the past (e.g., Mîndrescu et al., 2010; Evans, 2006; Barr and Spagnolo, 2015).

Previous work described that glacier-like forms are formed in and extend out of "cirque-like alcoves" and that it is unknown whether the glacier-like forms and/or any ice masses that came before them were wet-based (Hubbard et al., 2014). Herein, we use the term "alcove" loosely to describe any hollow with an arcuate headwall and opening downslope, on the scale of hundreds of meters to a few kilometers in width and length. Putative cirques on Mars have been identified in the mid-latitudes (Gallagher et al., 2021) and equatorial regions (Davila et al., 2013; Bouquety, et al., 2019; Williams et al., 2023). While the cirque-like alcoves in areas such as Deuteronilus Mensae have been interpreted as potentially connected to past glaciation (e.g., Head et al., 2006; Morgan et al., 2009, Hubbard et al., 2011; Souness and Hubbard, 2013), there have not been any in-depth studies dedicated to these cirque-like alcoves on a population scale. In this study, we mapped 1991 alcoves in Deuteronilus Mensae in the northern mid-latitudes of Mars and conducted a morphometric analysis to narrow down 435 cirque-like alcoves. We evaluate the presence of icy geomorphic features in these cirque-like alcoves, their aspect as a population, and how they compare to other features such as glacier-like forms and gullies. Through mapping cirque-like alcoves at a large scale for the first time, we expand the extent of features attributed to glacial erosion on Mars.

[Figure]

[Figure]

[Figure]

**Figure 2: (a)(i) Example of a cirque-like alcove on Mars (40.24°N, 34.48°E) (CTX mosaic; Dickson et al., 2023a) (a)(ii) a cirque on Earth in the Uinta Mountains (40.712°N, 110.114°W). CTX data credit: Caltech/NASA/JPL/MSSS. Earth imagery is from Google Earth including Landsat/Copernicus/U.S. Geological Survey coverage. (b)-(d) Examples of cirques on Earth incised into mesa topography, along with an example of a cirque profile in each. Part (i) of (b)-(d) provides an overview of the cirques in that location with an inset of the location of part (ii). Part (ii) of (b)-(d) offers a zoomed-in view of an individual cirque. Part (iii) of (b)-(d) shows the profile of the individual cirques in part (ii). (b) Uinta Mountains, Utah, USA (40.74°N, 110.05°W). DEM data: National Elevation Dataset, access via The National Map. (c) Kamchatka Peninsula, Russia (58.48°N, 160.70°E). DEM data: Shuttle Radar Topography Mission, access via EarthExplorer. (d) Transantarctic Mountains, Antarctica (80.01°S, 156.35°E). DEM data: Reference Elevation Model of Antarctica (Howat et al., 2022), access via the Polar Geospatial Center.**

**2 Study Area: Deuteronilus Mensae**

Our study region covers ~600,000 km² of Deuteronilus Mensae in the northern mid-latitudes of Mars (40-48°N, 16-35°E) (Fig. 3). While we also observe alcoves in other regions in the mid-latitudes of Mars, we focus on Deuteronilus Mensae as a study region for identifying cirque candidates due to its high density of icy viscous flow features (e.g., Levy et al., 2014; Baker and Carter, 2019; Brough et al., 2019). In addition to lobate debris aprons observed by the Mars Reconnaissance Orbiter (MRO) SHAllow RADar (SHARAD) instrument (e.g., Plaut et al., 2009; Baker and Carter, 2017), recently, the Mars
Subsurface Water Ice Mapping (SWIM) project identified Deuteronilus Mensae as one of the candidates where available data
is highly consistent with the presence of subsurface ice at the regional scale (Morgan et al., 2021), Thus, Deuteronilus Mensae
is a location of interest for future human missions to Mars (e.g., Morgan et al., 2021). Deuteronilus Mensae is characterized
by fretted mesa terrain of disputed origin encompassed by remnants from previous glaciations (Sharp, 1973; Squyres, 1978;
Carr, 2001; Morgan et al., 2009). The geologic history of Mars is divided into three main epochs: the Noachian around 4.0 to
3.85 Ga, the Hesperian around 3.56 to 3.24 Ga, and the Amazonian around 3.24 Ga to present-day (e.g., Hartmann, 2005;
Michael, 2013; Kite, 2019). Previous geomorphic mapping estimated that the mesas date back to the ancient Noachian and the
plains to the Hesperian, while younger Amazonian sedimentary deposits and a mantling unit overlay the mesas (e.g., Baker
and Carter, 2019).

[Figure]

**Figure 3: The study region Deuteronilus Mensae in the northern mid-latitudes of Mars is within the teal box (40-48°N, 16-35°E). Colors represent HRSC DEM data. The white sections on the top left show where the CTX beta01 mosaic does not have coverage and grayscale areas of the map show where the mosaicked HRSC DEM does not have coverage. Major surface features including Lyot Crater, Sinton Crater, and Mamers Valles are noted. CTX data credit: Caltech/NASA/JPL/MSSS. HRSC data credit: ESA/DLR/FU Berlin.**

**3 Methods and Data**

**3.1 Alcove Mapping**

We mapped 1991 alcoves at a 1:30,000 scale using the ~6 m/pixel Context Camera imagery beta01 mosaic (Malin et al., 2007; Dickson et al., 2023a). We only map alcoves that do not contain previously mapped glacier-like forms (Fig. 1). We did allow overlap with previous mapping of lobate debris apron and lineated valley fill because boundaries for these lobate debris aprons and lineated valley fill were different between Baker and Head (2015) and Levy et al. (2014). In addition, it was not always clear how the mapped boundaries for the lobate debris aprons and lineated valley fills were decided on relative to where the alcoves and mesa sidewalls were located, which is why alcoves were mapped regardless of where the boundaries for the lobate debris aprons and lineated valley fills were drawn. We digitized the outlines of the alcoves using ArcGIS software. For morphometric analyses, we used a High Resolution Stereo Camera (HRSC; Neukum et al., 2004) digital elevation model (DEM) of 29 frames mosaicked together that comprises a mix of true resolution between 50-100 m/pixel (see Data Availability section for exact frames). The HRSC DEMs were mosaicked together using the Mosaic to New Raster tool in ArcGIS Pro. Where available, we used ~25 cm/pixel High Resolution Imaging Science Experiment (HiRISE; McEwen et al., 2007) images to examine glacial geomorphic features within and related to features coming out of the alcoves, which are listed in the Data Availability section. Measurements from all of the imagery and DEMs used a Sinusoidal projection centered on longitude 25.5 degrees East, and were based on the IAU Mars 2000 Sphere datum.

**3.2 Criteria for Identification of Cirque-like Alcoves**

**3.2.1 Alcove Classes**

Cirques on Earth are categorized into five grades ranging from a "classic" cirque that contains "textbook" attributes to a "marginal" cirque, where the cirque status is doubtful (Evans and Cox 1995). In addition, there are also numerous cirque types including simple cirques, compound cirques, cirque complexes, staircase cirques, and cirque troughs (Benn and Evans 2010), which we drew upon to inform our preliminary alcove classes for our Deuteronilus Mensae analyses. Based on their kilometer-scale physical characteristics including shape, size, and associated landforms as seen in CTX imagery, alcoves that show any of the following morphologies: a) joined, b) interiorly ridged, c) staircase, d) channel-related, and e) branching were not included in our database. Descriptions and interpretations of these morphologies are provided in the Supplementary Material section. Although terrestrial glacial cirques may also fall into different categories, for our study of martian alcoves that are considered most analogous to terrestrial cirques, we focus on the alcoves classified as simple alcoves (and that do not have any of the morphologies listed above). Like simple cirques on Earth, simple alcoves on Mars are characterized by an armchair shape with an identifiable headwall, two sidewalls, and an opening downslope.

**3.2.2 Alcove Morphometric Calculations**

We applied the Automated Cirque Metric Extraction (ACME2; Spagnolo et al., 2017; Li et al., 2024) tool in ArcGIS

Pro to calculate alcove morphometrics, specifically, length (L), width (W), elongation (L/W), altitudinal range or height (H; difference between maximum and minimum elevation), area, slope, elevation, and aspect (Table 1). We also calculate the alcove size by multiplying the length, width, and height as follows: $\sqrt[3]{LWH}$. To use the ACME2 tool, we provided the mapped shape of the alcove, a threshold midpoint, which is defined as the midpoint of the down-valley lip of the cirque, and the HRSC

DEM as inputs (Fig. 4). ACME2 outputs the morphometrics into the attribute table of the feature class for alcoves. On Earth, typical L/W ratios are 0.5-4.25 (Derbyshire and Evans, 1976), and based on 10,362 globally distributed cirques, both L/H and

W/H ratios typically range between 1.5 to 4.0 (Barr and Spagnolo, 2015).

Note that by only including W/H ratios between 1.5 to 4.0, we expect that craters are excluded since craters typically have depth-to-diameter ratios of 0.1-0.2 (e.g., Robbins and Hynek, 2012), i.e., W/H ratios of 5:1 and 10:1. By using morphometrics, we also exclude other types of mechanisms for alcove formation, including active-layer detachments, deep- seated landslides, and theater-headed valleys (Table 2). This is because the H/L ratio of a terrestrial glacial cirque is expected to be deeper than any of the other alcove landforms with known morphometrics on Earth (Table 2). We focus only on the simple alcoves (based on morphology) that have these morphometric ranges for L/W, L/H, and W/H to constrain the most

"cirque-like" alcoves on Mars. By applying these constraints, we were able to identify 435 cirque-like alcoves after downsampling from our initial mapping and classification based on only image analysis of 1991 alcoves. Herein, we use the term "cirque-like alcove" for the martian alcoves that we will evaluate in this study as candidate cirques shaped by glacial erosion.

**Table 1:** Alcove morphometrics as outputted by the Automated Cirque Metric Extraction (ACME2) tool. The content was modified from Spagnolo et al. (2017) to fit a table format.

| Name | Unit | ACME2's output name | Definition |
|---|---|---|---|
| Length | Meters | L | Length of the line within the alcove polygon that intersects the alcove threshold midpoint and splits the polygon into two equal halves (Fig. 4) |
| Width | Meters | W | Length of the line perpendicular to the length line and intersecting the length line midpoint (Fig. 4) |
| Elongation | Dimensionless | L/W | Derived from dividing length by width |
| Altitudinal range or height | Meters | Z_range (H in this paper) | Range of elevations found |

| | | | by subtracting max elevation minus minimum elevation |
|---|---|---|---|
| Elevation | Meters | Z_mean | Mean elevation |
| Area | Meters$^2$ | Area_2D | Area of the polygon |
| Slope | Degrees | Slope_mean | Mean value of slope for all DEM pixels included in the alcove polygon. |
| Aspect | Degrees north, within the 0-360° interval | Aspect_mean | Mean of all pixel aspects across the entire surface of alcove by using standard statistical calculation methods for circular features. We also found the relative percentages of cirque-like alcoves in each aspect bin after normalizing by the percent of the total land surface in each aspect bin. We did this by converting the HRSC DEM raster to points, finding the aspect for each point, and calculating the land surface percent that belonged to each aspect bin. We then divided the percent of cirque-like alcoves in each aspect bin by the land surface percent bins and got the normalized percentages. |

**Table 2:** Morphometrics consistent with different alcove-forming erosional mechanisms on Earth and Mars.

| Formation mechanism/Landform | L/W | H/L | Aspect | Related geology | Typical scale (m) |
|---|---|---|---|---|---|
| Glacial cirque on Earth | ~1, generally ranges from 0.5-4.25 (Barr and Spagnolo, 2015) | ~0.67 (Barr and Spagnolo, 2015) | All directions; poleward is favorable (Barr and Spagnolo, 2015) | Overdeepening, moraines | $10^2$-$10^3$ (Barr and Spagnolo, 2015) |

| | | | | | |
|---|---|---|---|---|---|
| Deep-seated landslide on Earth[n.s.] | >2.5 (Fran et al., 2006) | 0.1-0.35 (LaHusen et al., 2016; landslide scars from glacial sediment) | Not available* | Hummocky landslide deposits | $10^1$-$10^2$ (LaHusen et al., 2016) |
| Impact crater on Mars | ~1 | 0.1-0.2 (Robbins and Hynek, 2012) | N/A | Ejecta blanket | $10^1$-$10^3$ (Palucis et al., 2020) |
| Amphitheater-headed valley on Mars hypothesized to have formed by either groundwater sapping or outburst flooding | 1-10 (Laity, 1988) | Not available* | Not available* | Sandstone, not basalt bedrock (Lapotre and Lamb, 2018) | $10^1$-$10^2$ for canyon heads, up to $10^3$ for the main channel (Lapotre et al., 2016) |

Not available*:  As of writing this paper, focused studies on the morphometrics of these landforms on the population scale are
not widely available for these other landforms.

[n.s.]: "n.s." stands for "not scarp" since landslide morphometrics do not usually include measurements of the morphometrics of
just the headscarp and sidewalls of where the landslides initiated.

[Figure]

**Figure 4: (a) Inputs for the Automatic Cirque Metric Extraction (ACME2) tool (Spagnolo et al., 2017; Li et al., 2024) include a shapefile for the alcove, a point for the alcove threshold, as pictured here, and a DEM. (b) Outputs from ACME2 include morphometrics such as the length, width, and height of the alcove (see Table 1). CTX data credit: Caltech/NASA/JPL/MSSS.**

**3.2.3 Uncertainties in elevation and alcove longitudinal profile**

We mapped the cirque-like alcove and identified the mid-threshold point using the CTX imagery. As mentioned in Section 3.1, both the CTX imagery and HRSC DEM were aligned to a Sinusoidal projection centered on longitude 25.5 degrees East, and were based on the IAU Mars 2000 Sphere datum. Any misalignment of up to 100 m between the image and the DTM is of little concern when it translates into metrics made by ACME2 since most metrics rely on multiple pixel measurements. This is certainly the case for slope, aspect and average elevation along the cirque length or the entire cirque area. It might affect minimum and maximum elevation, although any effect should be evened out by the large sample size.

Longitudinal profiles of cirques on Earth are typically characterized by a concave bowl-shape with a steep headwall, flatter floor, and a lip or threshold at the end of the profile that separates the cirque from the valley below (e.g., Barr and Spagnolo, 2015). In Deuteronilus Mensae, an alcove threshold may be visible with the HRSC DEM resolution of 50-100 m/pixel (Fig. 5). However, in some cases, we do not see the threshold because of low DEM resolution or because the feature may be covered by other material (Section 4.3). For comparison to the HRSC DEM, we include a CTX DEM generated by the GALE lab at UCLA using the Ames Stereo Pipeline (Beyer et al., 2018; Fig. 5). Not all glacial cirques on Earth have thresholds either (e.g., Fig. 2), nor is having a threshold a definitional requirement of terrestrial cirques (Barr and Spagnolo, 2015). As a result, we do not use the existence of an observable threshold as a requirement for identification as a cirque-like alcove on Mars.

[Figure]

Figure 5: Examples of HRSC DEM profiles to compare with a CTX DEM profile of an alcove. Arrows represent the path of the profiles from higher to lower elevations for both (a) and (b). (a) Example of the longitudinal profile of a mapped alcove using the HRSC DEM that includes the potential overdeepening (difficult to discern at this resolution) and threshold or lip in the profile. The alcove is centered at 40.22°N, 34.57°E. in the CTX mosaic (Dickson et al., 2023a). (b) Comparison between the HRSC and CTX DEMs. The alcove is centered at 46.55°N, 22.08°E in HiRISE image ESP_019214_2270_RED. The CTX longitudinal profile contains a threshold, but the same feature in the HRSC DEM is not resolvable. CTX data credit: Caltech/NASA/JPL/MSSS. The CTX DEM was constructed by Dr. Mackenzie Day's GALE lab at UCLA. HRSC data credit: ESA/DLR/FU Berlin.

**3.3 Criteria for identification of icy geomorphic features**

In addition to mapping and calculating the morphometrics of alcoves in Deuteronilus Mensae, we also evaluate the presence of icy geomorphic features in the alcoves where HiRISE imagery is available. While we designed the study so that none of the cirque-like alcoves that we mapped included mapped glacier-like forms, using the available inventory of HiRISE images we observed other features associated with the cirque-like alcoves that appear consistent with the presence of ice or ice loss. The icy geomorphic features that we evaluate for in HiRISE images include flow features, linear terrain, mantle, moraine-like ridges, mound-and-tail terrain, polygonal terrain, moraine-like ridges, rectilinear-ridge terrain, and washboard terrain. We identify these features using the criteria listed in Table 3. Other icy geomorphic features that were observed nearby alcoves but not categorized in this study because they were not directly in or connected to features coming out of alcoves included brain terrain (Levy et al., 2009a) and pitted terrain (Jawin et al., 2018). We note that the icy geomorphic features that we identify may correspond to some of the criteria defined by Souness et al. (2012) for mapping glacier-like forms, which include: 1) surrounded by topography indicative of flow around obstacles, 2) distinct from the surrounding landscape in texture or color, 3) surface foliation indicative of down-slope flow, 4) L/W ratio > 1, 5) discernible head or terminus, 6) appear to contain a volume of ice. However, the icy geomorphic features noted here do not include all of the criteria and were not mapped as glacier-like forms. For example, an icy feature within an alcove might appear to have a terminus, but no convexity from existing ice volume that differentiates it from surrounding topography (Fig. 6).

**Table 3:** Icy geomorphic features with their descriptions, proposed formation, and references.

| Icy Geomorphic Feature | Additional Names | Description | Terrestrial Analog | Proposed Formation Mechanism | Select References |
|---|---|---|---|---|---|
| Flow features | N/A | Longitudinal troughs and ridges | Same | Formed by downslope ice flow and deformation | Hubbard et al., 2011; Souness et al., 2012

On Earth:
Benn and Evans, 2010 |

| | | | | | |
|---|---|---|---|---|---|
| Linear terrain | If supraglacial: longitudinal foliation | Parallel raised ridges, bumpy in appearance | If supraglacial: flow stripes, longitudinal foliation | If supraglacial: Caused by deformation of ice as it flows; can be due to compressed, accelerating flow | Hubbard et al., 2011; Conway et al., 2018 |
| | If subglacial: pasted-on terrain | | If subglacial: megalineations, striations | If subglacial: Ice flow over water-lubricated sediment | |
| Mantle | Latitude-dependent mantle, thicker version is commonly known as pasted-on terrain | "Raised curvilinear edge for the upslope boundary" (Khuller et al. 2021) | N/A | Airfall of ice on dust; sublimation of ice generates a lag that protects underlying ice deposit | Mustard et al. 2001; Christensen et al., 2003; Conway et al., 2018; Khuller et al. 2021 |
| Moraine-like ridge | Moraine ridge | Ridge of debris at the terminus (end) of the ice mass | Terminal moraine | Dumping, squeezing, and pushing of debris by a glacier; left behind after ice recedes | Arfstrom and Hartmann, 2005 |
| Mound-and-tail terrain | N/A, similar to linear terrain | Steep upglacier-facing core of ice with a shallow elongate tail; typically 30-50 m long, 10-30 m across, and 2-4 m high | Closest to drumlins | Subglacial bedforms formed from subglacial sediment moulding and/or deposition beneath wet-based ice masses | Hubbard et al., 2011 |
| Polygonal terrain | Polygonized terrain and scaly terrain (we group the two together here under the term "polygonal" terrain); mantle polygons | Polygonized terrain: ~10° slope, 5-10 m across, tessellating polygons; Scaly terrain: 12-16° slope, 10-20 m across, tessellating polygons | Periglacial patterned ground | Frost heave and thermal contraction cracking | On Mars: Hubbard et al., 2011; Levy et al., 2009b; Soare et al., 2022  On Earth e.g.,: French, 2018; Marchant and Head, 2007 |

| | | | | | |
|---|---|---|---|---|---|
| | | | | | |
| Rectilinear-ridge terrain | Push moraines | Series of ridges tens of meters across and 2-3 m high, elongated in an arc parallel to former glacier terminus | Thrust-block moraines, push moraines, moraine-mound complex | Basal debris thrust up from the glacier bed, basal crevasse fills, or ice-contact outwash deposits | On Mars: Hubbard et al., 2011

On Earth e.g.,: Hambrey et al., 1997; Sharp, 1985; Lukas, 2005 |
| Washboard terrain | Crevasse-like features | Transverse scarps, commonly at the base of a steep slope | Crevasses, bergschrunds | Formed from debuttressing and oversteepening of ice on slopes | Hubbard et al., 2011; Jawin et al., 2018; Jawin and Head, 2021 |

[Figure]

**Figure 6: (a) Previously mapped glacier-like form (Brough et al., 2019). (b) and (c) represent previously unmapped cirque-like alcoves no longer appear to contain a volume of ice and raised moraine-like ridge at the terminus. However, they still do contain surface foliations suggesting down-slope flow near the headwall. Cirque-like alcove mapping only extends to where the sidewalls end. This HiRISE image ESP_025873_2230_RED is centered at 42.63°N, 25.02°E. HiRISE data credit: NASA/JPL/University of Arizona.**

**4 Results**

**4.1 Trends in aspect, size, area, latitude, slope, and elevation of cirque-like alcoves**

By examining the aspect of the population of 435 cirque-like alcoves, we observe a south to southeast bias with an average of 153.09° (Fig. 8). The largest binned median size, defined as $\sqrt[3]{LWH}$, and area for cirque-like alcoves are located at lower latitudes (Fig. 9a). Most of the largest binned median size and area cirque-like alcoves correspond to slopes of 20-25° (Fig. 9b), and cirque-like alcoves have an average slope of ~21.1°. The binned median size and area of the cirque-like alcoves increase with elevation (Fig. 9c). Above 46° in latitude, most cirque-like alcoves cluster between 100-275° in aspect (Fig. 10a). We also notice a lower density of cirque-like alcoves facing 250°-360° at all latitudes (Fig. 10a). Cirque-like alcove elevation decreases as latitude increases (Fig. 10b). Similarly, cirque-like alcove height also decreases as latitude increases (Fig. 10c).

[Figure]

**Figure 7. Distribution of 435 cirque-like alcoves and 74 glacier-like forms in the study region Deuteronilus Mensae. Note that while glacier-like forms (Brough et al., 2019) and cirque-like alcoves exist outside of the teal boundary lines, they are not included in the analyses reported in this study.**

[Figure]

Figure 8: (a) Rose diagram showing the aspect of cirque-like alcoves. Cirque-like alcove aspect averages 153.09°
between the south and southeast directions. (b) Rose diagram showing the relative percentages of cirque-like alcoves
in each aspect bin after normalizing by the percent of the total land surface in each aspect bin (we explain the method
in Table 1). After normalizing, we found that the same southeastward trend persisted. Aspect bins are as follows: 337.5
≤ N < 22.5°; 22.5 ≤ NE < 67.5°; 67.5 ≤ E < 112.5°; 112.5 ≤ SE < 157.5°; 157.5 ≤  S < 202.5°; 202.5 ≤ SW < 247.5°; 247.5
≤ W < 292.5°; 292.5 ≤ NW < 337.5.

[Figure]

Figure 9: Mean cirque-like alcove size $\sqrt[3]{LWH}$ and mean area vs. a) latitude, b) slope, and c) elevation for only cirque-like alcoves in Deuteronilus Mensae. Medians are displayed as black bars for each interval.

[Figure]

Figure 10: Cirque-like alcoves in Deuteronilus Mensae plotted by latitude versus a) aspect, b) elevation, and c) height.

**4.2 Comparison between cirque-like alcoves and glacier-like forms mapped in Deuteronilus Mensae**

Both the largest cirque-like alcoves and glacier-like forms are located in the southeast part of the study region (Fig. 7). By comparing the measurements of glacier-like forms (Brough et al., 2019) with cirque-like alcoves, we find that while the average area of an alcove is smaller than a glacier-like form, the total area of all the cirque-like alcoves is larger than the total area of the glacier-like forms (Table 4). There are 74 mapped glacier-like forms in Deuteronilus Mensae (Brough et al., 2019), which is only about 16% of the total 435 cirque-like alcoves in this study area. As a result, the aggregate total area and aggregate total volume for the cirque-like alcoves are larger than for the glacier-like forms. In addition, the average volume of an alcove is larger than that for a glacier-like form because the cirque-like alcoves have a greater height than the typical estimated thickness of a glacier-like form. ~70% of all cirque-like alcoves are within 10 km of a glacier-like form.

Table 4: Area and volume statistics of cirque-like alcoves versus glacier-like forms. Statistics for the cirque-like alcoves come from the topographic expression of the alcove, whereas the statistics for the glacier-like forms are from the present-day ice-rich form. We use total volume as an approximation here, though values are rarely completely full of ice.

| | Average Area (km²) | Total Area (km²) | Average Volume (km³) | Total Volume (km³) |
|---|---|---|---|---|
| Cirque-like alcoves (435) | 1.95 | 848.98 | 2.03 | 881.42 |
| Glacier-like forms (74) | 7.79 | 576.82 | 1.14 | 84.01 |

While the highest percentage (22%) of glacier-like forms have a northeast orientation (based on data from Brough et al., 2019), the highest percentage (20%) of cirque-like alcoves have a southward orientation (Fig. 11a). For both glacier-like forms and cirque-like alcoves, the west and northwest aspects have relatively low numbers ranging from 2% to 9% of the entire population, though unlike glacier-like forms (15%), cirque-like alcoves also have a low proportion of 7% for the north- facing aspect. For mean glacier-like form volume grouped by aspect, the largest glacier-like forms face southwards in

Deuteronilus Mensae, whereas the largest cirque-like alcoves by volume face the north (Fig. 11b).

[Figure]

**Figure 11: a) Bar plots of the aspect compared to the quantity of i) cirque-like alcoves, ii) glacier-like forms, and iii) both cirque-like**

**alcoves and glacier-like forms. b) Bar plots of the aspect compared to the average area in each aspect direction for i) cirque-like**

**alcoves, ii) glacier-like forms, and iii) both cirque-like alcoves and glacier-like forms.**

**4.3 Icy geomorphic features identified**

In addition to morphometric observations, we identified geomorphic features in association with the cirque-like alcoves as consistent with either remnant or active ice in order to evaluate aspects of the glacial history in the cirque-like alcoves. Using the criteria stated in Table 3, we identified flow features, linear terrain, mantle, moraine-like ridges, mound- and-tail terrain, polygonal terrain, rectilinear-ridge terrain, and washboard terrain in available HiRISE imagery. Out of 435

cirque-like alcoves, there was complete overlap in available HiRISE frames with 26 cirque-like alcoves (8%) and partial overlap with only 10 cirque-like alcoves (1%). In CTX imagery, we were also able to identify flow features, linear terrain, mantle, moraine-like ridges, and washboard terrain. However, at the CTX resolution, it was more difficult to identify features such as mound-and-tail terrain, polygonal terrain, and rectilinear-ridge terrain. For both HiRISE and CTX imagery, the linear terrain and mantle were the two most common features. We provide the percentages of each feature in both HiRISE and CTX

imagery in Table 5.

**Table 5: Percent of HiRISE and CTX imagery with each type of icy geomorphic feature.**

| Icy Geomorphic Feature | Percent of HiRISE imagery (%) | Percent of CTX imagery (%) |
|---|---|---|
| Flow features | 8 | 9 |
| Linear terrain | 81 | 57 |
| Mantle | 58 | 90 |
| Moraine-like ridges | 14 | 5 |
| Mound-and-tail terrain | 6 | N/A |
| Polygonal terrain | 53 | N/A |
| Rectilinear-ridge terrain | 3 | N/A |
| Washboard terrain | 42 | 2 |

27   Fig. 12 provides examples of washboard terrain, linear terrain, rectilinear ridges, and polygonal terrain, which all

28 correspond to the presence of ice and/or ice loss, as described in Table 3. In Fig. 12, the linear terrain extends out from the

29 washboard terrain at the base of the cirque-like alcoves (Fig. 12). The rectilinear ridges are downslope of both the washboard

30 terrain and linear terrain. The polygonal terrain is between the two sections of linear terrain (Fig. 12f). In addition, the

31 polygonal terrain is observed farther downslope of the rectilinear ridges (Fig. 12f).

32   Approximately 14% of cirque-like alcoves with HiRISE imagery coverage have moraine-like ridges. Fig. 13 contains

33 examples of moraine-like ridges. Fig. 13b also shows additional examples of moraine-like ridges downslope of alcoves (that

34 are not all cirque-like), with along-flow linear terrain between the alcove headwall and the moraine-like ridge. As in Fig. 12,

35 washboard terrain, linear terrain, and polygonal terrain are all present.

[Figure]

**Figure 12: a) Cirque-like alcove with evidence for remnant ice centered at 46.57°N, 22.12°E, 46.57°N in HiRISE image**
**ESP_019214_2270_RED. b) Boulders near the top of the headwall indicating erosion. Features corresponding to ice-loss include the**
**following: c) washboard terrain (Jawin and Head, 2021), d) linear terrain, e) rectilinear ridges (Hubbard et al., 2011), and f)**
**polygonal terrain. (e.g., Hubbard et al., 2011).**

[revised manuscript text omitted]

high obliquity. Similarly, for regions poleward of 40° like Deuteronilus Mensae, gullies are primarily observed on on equator-facing slopes (Harrison et al., 2015; Conway et al., 2018), possibly due to the melting of ground ice during periods of high obliquity (Costard et al., 2022), though the exact formation mechanism of gullies remains unclear (e.g., Conway et al., 2019; Dundas et al.. 2022). Regardless of how gullies are initiated, they may act as a local depression in a location where water-ice precipitation could later accumulate for cirque-like alcove formation, such as if the gullies acted as a cold trap for snow (e.g., Dickson et al., 2023b). For example, gullies could provide the initial concavity for a later cirque-like alcove to develop when glaciation occurs (Section 5.2.2), which is consistent with gully heads that have been proposed as initiation points for cirques on Earth (Derbyshire and Evans; 1976). However, in the case of meltwater, we note that cirque-like alcoves may prefer to reside on equator-facing slopes because this would allow for increased insolation (e.g., Pilorget and Forget; 2016; Dundas et al., 2022) and the chance for meltwater as temperatures increase (Dickson et al., 2023b). We explored this potential association between gullies and cirque-like alcoves in Section 5.1.2. On the other hand, glacier-like forms are mostly pole-facing in Deuteronilus Mensae, which corresponds to present-day conditions that are favorable for ice preservation. This may indicate that cirque-like alcoves were generated during an earlier phase of glaciation before the glacier-like forms or this may be due to preservation bias as poleward facing glacier-like forms may have outlasted other directions. If cirque-like alcoves do in fact correspond to an earlier phase of glaciation, it is unclear if this glaciation was on the scale of glacier-like forms versus larger scales like the lobate debris apron. It is also possible that valley glaciers in the cirque-like alcoves eventually connected with larger ice bodies like the lobate debris apron and lineated valley fill.

**5.1.3  Trends between size, area, latitude, slope, and elevation**

Relationships between size, area, latitude, mean elevation, and height of the cirque-like alcoves (Fig. 10) are likely due to the nature of the topography in this region. The mean elevation in Deuteronilus Mensae decreases toward the north (Fig. 14) and at lower latitudes, the mesas are at a higher elevation relative to the basin than at lower latitudes (Fig. 3, Fig. 14). These two factors combined mean that at lower latitudes, the cirque-like alcoves have a higher mean value for elevation and height due to the topography. Since size is calculated as $\sqrt[3]{LWH}$, the larger height at the lower latitudes corresponds to a larger cirque-like alcove size. Height also scales with length and width for cirque-like alcoves (Section 4.1), which is why both larger sizes and areas of cirque-like alcoves correspond to lower latitudes (Fig. 10b). Thus, the local elevation and local mesa height limit the local cirque-like alcove height at different latitudes in Deuteronilus Mensae.

[Figure]

Figure 14: Using the raster to point tool in ArcGIS Pro led to 41,618,659 points from the HRSC DEM. The binned mean elevation represents the mean elevation value of all the points at each half latitude in the study region. The binned elevation difference was calculated based on the difference between the mean of the highest 10,000 points and the mean of the lowest 10,000 points at each half latitude.

**5.1.4 Comparison between cirque-like alcoves and glacier-like forms mapped in Deuteronilus Mensae**

Since both the largest cirque-like alcoves and glacier-like forms are located in southeast Deuteronilus Mensae, this may indicate that there is a local factor impacting both glacier-like form size and cirque-like alcove size or that the two are linked in how they form. For the first option, this may be because of local topography that enhances the conditions for precipitation and snow accumulation. If we assume that the cirque-like alcoves were eroded by the same phase of glaciation as the glacier-like forms, then the cirque-like alcoves may now be empty of glacier-like forms because their preservation became unfavorable in current obliquity conditions. In that case, conditions in the southeast of this region resulted in both the largest glacier-like forms and cirque-like alcoves. If we instead assume that all cirque-like alcoves had reached most of their current size before the glaciation cycle that brought the glacier-like forms, then the size of glacier-like forms may be limited to the initial size of the cirque-like alcove that it occupies. While we do not distinguish between these hypotheses in this study, we recommend future work to investigate the direct cause of the larger glacier-like forms and cirque-like alcoves in the southeast part of Deuteronilus Mensae.

Overall, the average glacier-like form has a larger area than the average cirque-like alcove because glacier-like forms typically extend beyond the cirque-like alcoves that they emerge from. Although the average area of a cirque-like alcove is smaller than a glacier-like form, the total area of all the cirque-like alcoves is larger than the total area of the mapped glacier- like forms in our study area (Table 4). If the simple cirque-like alcoves that we identify here are in fact representative of glacial erosion, then we extend our previous knowledge of the areal extent of past glaciation in Deuteronilus Mensae by at least 48%.

While the largest glacier-like forms face southwards in Deuteronilus Mensae and the largest cirque-like alcoves face the north, this may be due to a localized topographic effect for glacier-like forms in Deuteronilus Mensae because overall for the northern hemisphere, glacier-like forms flowing northward are larger than those flowing southward by about 20% (Brough et al., 2019).

**5.2 Geomorphic interpretations of cirque-like alcoves and associated features**

**5.2.1 Icy geomorphic features**

We find that 42% of available HiRISE images contained washboard terrain, while only 2% of CTX images did, though this is likely due to a resolution issue since finer textures cannot be resolved at CTX scale. Except for two exceptions, cirque-like alcoves that contained washboard terrain did not also have an identifiable mantling unit. Similar to its presence at the bottom of crater walls (Jawin et al., 2018; Jawin and Head, 2021), the presence of washboard terrain here at the bottoms of the mesa sidewalls indicates deglaciation.

In both HiRISE (81%) and CTX imagery (57%), a high percentage of images of cirque-like alcoves contained observable linear terrain. In Fig. 12, since the linear terrain extended out from the washboard terrain, which is due to surficial crevasses, this suggests that the linear terrain there may be most similar to supraglacial longitudinal foliation.  However, linear terrain could still result from subglacial erosion despite superposing a mantle unit since a mantle unit consists of layers of dust and snow that build up in the mantle over multiple obliquity cycles (e.g., Khuller et al., 2021). Applied here, this would imply that the ridges could have been subglacially eroded, but from another layer of ice of the mantle unit (compacted from dust and snow) that formerly existed on top of the rest of what is left of the mantle unit today.

At a potentially earlier stage of evolution of the glacier-like forms, moraine-like ridges lack elongation outside of the alcove (Fig. 13a), potentially similar to a terrestrial cirque glacier sitting within the cirque basin instead of extending into the valley below. In Fig. 13b, the alcoves are not well-developed and do not have morphometrics corresponding to the criteria we set for cirque-like alcoves. Nevertheless, since the moraine-like ridges correspond to upslope alcoves, similar to

Arfstrom and Hartmann (2005), we suggest that the moraine-like ridges in Fig. 13b reflect the initiation of cirque-style glaciation before the alcove headwalls and sidewalls develop more as they are increasingly eroded and steepened. This is also referred to as unconstrained piedmont glaciation by Conway et al. (2018).

**5.2.2 Evidence for different stages of cirque-like alcove generation linked with gully evolution**

Different stages of cirque-like alcove evolution, likely linked to different histories of glacial occupation and erosion, can be seen in and near mapped cirque-like alcoves. For example, in Fig. 15a and 15b, notches (feature #1 in Fig. 15b) may indicate initial ice-associated erosion of the mesa sidewall. Stratigraphically, these notches predate the slab of detached mesa sidewall since the notches on the slab can be traced, but are now offset, from the notches above the slab (Fig. 15a). The notches resemble gullies elsewhere on Mars, which is relevant in this work because previous work has shown that gully formation may occur during glacier retreat on Mars during paraglacial stages (Jawin and Head, 2021), where degrading ice no longer provides structural support for slopes of sediment. Gully incision may initiate through sediment flow assisted by either liquid water or $CO_2$, or dry mass wasting (e.g., Conway et al., 2019; Dundas et al., 2022; Dickson et al., 2023b). Since the slabs formed after the notches, this is consistent with increased mass wasting of the mesa sidewall during deglaciation.

Here, we assume that the side of the mesa evolves from a straight edge to an increasing number and depth of depressions. An alternative interpretation might be that the deeper depressions were subsequently filled up to create a straight edge, however, we do not see evidence for this amount of infilling. In the middle panel of Fig. 15a, for feature #2, there is evidence that the notches undergo further erosion and begin to connect, eventually forming feature #3 where the outlet between the larger notch head is overridden and enlarged. We suggest that an icy mantling deposit is responsible for this erosion since we see linear terrain that are consistent with pasted-on terrain. Fig. 15a feature #4 in the middle panel demonstrates continued erosion and enlargement of these alcoves as they grow and connect with neighboring alcoves until they lose internal ridges as glacial erosion smoothes the interior of the alcove.

In Fig. 15b, the alcoves are smoother, appear to be more U-shaped (though CTX DEMs did not have high enough resolution for profiles), have more arcuate headwalls, and have narrower ridges between alcoves. We suggest that this represents a later, more advanced stage of cirque-like alcove evolution, perhaps after multiple cycles of glaciation, where ice could erode repeatedly over time into the mesa sidewalls so that the cirque-like alcove basin becomes smoother and the sidewalls develop into narrow ridges as in Fig. 12. In Fig. 15b, as in Fig. 15a, we see downslope debris and deposits indicating mesa sidewall erosion. In the middle panel, we see examples of deposits of mesa material that are pushed outwards from the cirque-like alcoves (Fig. 15b). While it is likely that multiple processes contributed to the incipient form of a cirque-like alcove like those mentioned in Table 2, we suggest that the morphometrics and conditions observed eventually require substantial glacial erosion. For example, for impact cratering, while glacial geomorphic features may override any signature of impact ejecta, it is very unlikely that similarly sized impacts all happened to occur along mesa edges. Ultimately, we acknowledge that these other processes likely contributed to at least some erosion of cirque-like alcoves, but the prevalent glacial geomorphic features and consistently sized features correspond most to glacial erosion.

We suggest that the observed notches are gullies and would be able to act as necessary initiation points for ice accumulation that would later support glaciation and erosion that could form cirque-like alcoves. This is consistent with the mechanism proposed by Jawin et al. (2018). However, the formation of cirque-like alcoves is not dependent on how the gullies are formed. Gully formation hypotheses currently include $CO_2$ ice sublimation, dry mass wasting, meltwater generation, and a combination of these factors. For example, meltwater generation is more commonly invoked for older, inactive gullies during periods of higher obliquity (e.g., Dickson et al., 2023; Noblet et al., 2024), while gullies that have been observed to be recently active invoke $CO_2$ frost, as well as dry mass wasting during frost-free seasons (e.g., Dundas et al. 2022).

While determining how these gullies formed is outside the scope of this work, we include a discussion of the current hypotheses. Dry mass wasting alone for gully formation has recently been challenged since the mean gradient of gullies is lower than the angle of repose of dry material on Mars (e.g., Noblet et al., 2024). Gullies that are either in 1) the northern hemisphere at latitudes lower than ~50° or 2) non-polar regions and are equator-facing are modeled to be inactive gullies
(Roelofs et al., 2024). These inactive gullies are inconsistent with where $CO_2$ frost deposition is expected to occur on pole-
facing slopes (e.g., Lange et al. 2023). For example, in the southern hemisphere, $CO_2$ frost is only observed on pole facing
slopes between 30-50°S and is not expected on equator-facing gullied slopes between 40°S and 50°S during current obliquity
conditions (Noblet et al., 2024). Nevertheless, we note that $CO_2$ sublimation cannot be completely ruled out for equator-facing
slopes since seasonal deposition of $CO_2$ frost at these latitudes could have been more prevalent in the past (Noblet et al., 2024).
For present-day gully activity, rather than inactive gullies, sublimation of $CO_2$ is typically invoked (e.g., Dundas et al., 2022),
though $H_2O$ ice melt has been suggested to occur within dusty ice (e.g., Khuller et al., 2021).

Gullies are preferentially found on terrains that have subsurface water ice (Noblet et al., 2024). It is suggested that
these inactive gullies are formed from the meltwater of ground ice during past high obliquities (e.g., Noblet et al., 2024;
Dickson et al., 2023). This may be possible because modeling found temperatures above freezing for meltwater and gully
formation during high obliquity excursions in the mid-latitudes (Costard et al., 2002; Williams et al., 2008; Williams et al.,
2009; Dickson et al., 2023b). According to Dickson et al. (2023), at high obliquities of 35° in the past, meltwater was possible
during the Amazonian because pressures exceed the triple point of water, and transient melting may be possible in the present
as well (Hecht et al., 2002). Future work is necessary to elucidate the potential relationship between gullying as initiation
points for cirque-like alcove formation and how that is tied to cyclicity in ice accumulation and melt.

[Figure]

**Figure 15: (a) Top panel is centered at 41.06°N, 17.88°E in CTX image D04_0288880_2193_XI_39N342W. Notches may indicate ice-associated erosion of the mesa sidewall. Slabs and deposits suggesting active mass wasting from the slopes. Flow lines indicate the downslope direction of flow. Middle panel is centered at 40.02°N, 23.20°E in CTX image D21_035499_2203_XN_40N336W. Feature #1 represents initial notches, #2 represents the initial notches undergoing further erosion and beginning to connect, #3 demonstrates an outlet being overridden and enlarged, as indicated by flow lines, and #4 demonstrates the continual enlargement of these alcoves. Bottom panel is centered at 41.60°N, 18.46°E in CTX image N01_062743_2222_XI_42N341W. The slab indicates the unstable slopes. We see an alcove with linear terrain and leading to deposits, adjacent to a nearby glacier-like form on the same mesa sidewall. (b) Top panel is centered at 46.67°N, 26.13°E in HiRISE image ESP_016247_2270. Debris label in the top panel points to a possible detached block that is ~160x80m. Middle panel is centered at 46.57°N, 26.02°E in CTX image P13_006160_2252_XN_45N334W. Flow lines indicate a flow toward the top of the image (north direction), consistent with the deposits detaching from the mesa in the south and being transported northwards. Bottom panel is centered at 46.56°N, 22.13°E in HiRISE image ESP_019214_2270. These alcoves represent the most mature cirque-like forms due to their defined ridges. A deposit has an outline similar to the mesa sidewall, suggesting downslope flow. HiRISE data credit: NASA/JPL/University of Arizona. CTX data credit: Caltech/NASA/JPL/MSSS.**

**5.3 Estimating the timescales for cirque-like alcove erosion on Mars**

From terrestrial studies we know that cirques are formed by glacial erosion, which generally requires liquid water at the base of a wet-based glacier (Glasser and Bennett, 2004). On the other hand, cold-based glaciers on Earth are minimally erosive (Table 7) and are therefore not typically associated with large-scale glacial erosion features such as glacial valleys, troughs or cirques, though it cannot be excluded that minimally erosive cold-based glaciers operating over orders of magnitude larger timescales than a few glaciation cycles might still contribute to the erosion of large features. For example, measurements at cold-based Meserve Glacier in Antarctica find an erosion rate of $9 \times 10^{-7}$ to $3 \times 10^{-6}$ myr$^{-1}$ (Cuffey et al. 2000), meaning that it would take 100-330 millions of years of continuous glacial erosion to produce a 300 m deep cirque on Earth. In locations such as the Dry Valleys of Antarctica where cold-based glaciers currently reside within cirques, it is likely that much of the erosion of the cirque occurred during an earlier more temperate phase of glaciation during the Miocene (e.g., Selby and Wilson, 1971; Sugden and Denton, 2004; Clinger et al., 2020). If these martian cirque-like alcoves are analogous to terrestrial glacial cirques, then they may have formed either during an earlier wet-based phase at the scale of an active glacier-like form, or formed during a prior cold-based glacial cycle separate from the glacier-like forms, such as when lobate debris aprons formed.

[revised manuscript text omitted]

~350 Myr would be required for the cirque-like alcoves to form, which is consistent with the ~500 Myr timescale of LDA- and CCF-forming glaciation based on crater counts (Fassett et al., 2014). However, accounting for obliquity variations, a median height cirque-like alcove would require ~1.8 Gyr to erode from only cold-based glaciation during periods of high obliquity with erosion rates of ~0.85 m/Myr. If the glaciers were cold-based during their entire evolution, the erosion timescale is longer and therefore the alcoves must be much older than if they evolved with periods of wet-based glaciation. A timescale of hundreds of millions to a billion years is in the range of when lobate debris aprons were thought to have formed, such as in

Deuteronilus Mensae, the lobate debris aprons are estimated to be as old as 1.1 Gyr (Berman et al., 2015).

Using the slowest estimated erosion rate corresponding to cold-based glaciers in Antarctica, the initiation of the cirque-like alcoves likely predated the lobate debris aprons and involved erosion rates faster than cold-based glaciation. Then they could continue to develop in size during and/or after when the lobate debris aprons formed. However, since debris from the cirque-like alcoves often superposes the lobate debris aprons (e.g., Baker and Carter, 2019), this means that the process eroding the cirque-like alcoves have been actively eroding after/when the lobate debris aprons formed. . The supraglacial debris covering the lobate debris aprons averages ~25 meters in thickness and a major fraction of the debris was sourced as rockfall from the mesas (Baker et al., 2019). Thus, it is reasonable to expect that the present state of the mesa sidewalls, including the cirque-like alcoves, formed either concurrently with or after the lobate debris aprons evolved and became covered with debris. Otherwise, the erosional process sourcing the supraglacial debris would likely have erased the cirque-like alcoves. Here we use the maximum age estimate of lobate debris aprons of 1.1 Gyr as the earliest time that glaciers could have begun the erosion process that led to the formation of the cirque-like alcoves. By also including the consideration of obliquity that only around 20% of the 1.1 Gyr would have been conducive for ice accumulation, the maximum erosion depth achievable by cold-based glacier erosion would be ~790 m. Since ~20% of the cirque-like alcoves are larger than 790 m, we conclude that at least some of the cirque-like alcoves could have required a faster erosion rate than the ~0.85 m/Myr suggested for cold-based glaciers. Some have suggested cold-based glaciers erosion rate on Mars up to 10 m/Myr (Table 7; Levy et al., 2016). If this upper-end rate of 10 m/Myr is applied, then all heights of the cirque-like alcoves could have formed via cold-based glacier erosion within ~930 Myr. Given these timescales, it is more likely that the cirque-like alcoves had erosion rates higher than 0.85 m/Myr, corresponding to glaciers with surface velocities faster than 1 mm/yr. This surface velocity is also faster than what recent modeling found for a viscous flow feature with a maximum surface velocity of $20 \times 10^{-6}$ m/yr (Butcher et al., 2024), which would yield an unrealistic age for the cirque-like alcoves that is older than the age of Mars.

On Earth, the chronology of cirque formation is difficult to constrain (e.g., Turbull and Davies, 2006), and estimates for total glacial cirque erosion time range from 125 Kyr (Larsen and Mangerud, 1981) to a few million years (Andrews and Dugdale, 1971; Anderson, 1978; Sanders et al., 2013). Our estimates here find that a median height cirque-like alcove in Deuteronilus Mensae would take in the range of ~9.4 Myr to form if occupied by a wet-based glacier with an erosion rate of 160 m/Myr, and ~1.8 Gyr if occupied by a cold-based glacier with an erosion rate of 0.85 m/Myr. However, since cold-based erosion rates may vary by up to two orders of magnitude, maximum cold-based erosion rates closer to 10 m/Myr would allow for timescales of hundreds of millions of years. Whether the cold-based erosion rates on Mars were more similar to what has been observed at Meserve Glacier in Antarctica at ≤3 m/Myr (Cuffey et al., 2000) or at much higher values closer to 10 m/Myr (Levy et al., 2014) remains unknown.

**6 Conclusions**

This is the first in-depth, regional population scale study of the morphometrics and geomorphic evidence of previous ice occupation associated with cirque-like alcoves on Mars, that uses terrestrial knowledge to make a case that a sub-population of the mapped alcoves were likely eroded by past glaciation. By mapping ~2000 alcoves in Deuteronilus Mensae that did not contain previously mapped glacier-like forms, grouping them into six classes, and then downselecting to only simple alcoves with length/width (L/W) between 0.5 to 4.25, length/height (L/H) of 1.5 to 4.0, and width/height (W/H) of 1.5 to 4.0, which are consistent with terrestrial cirques, we are able to identify a population of 435 "cirque-like alcoves." We constrained our dataset to a total of 435 cirque-like alcoves in the study area, where only 74 glacier-like forms had been previously mapped (Brough et al., 2019). Thus, if cirque-like alcoves are indeed glacially eroded, we greatly extend what we know about the extent of kilometer-scale glaciation in the region. Using HiRISE imagery that was available for ~9% of these cirque-like alcoves, we find evidence of associated icy geomorphic features, including flow features, linear terrain, mantle, moraine-like ridges, mound-and-tail terrain, polygonal terrain, moraine-like ridges, rectilinear-ridge terrain, and washboard terrain (Figs. 12-13). All of these features have been found in association with glacier-like forms in previous work (e.g., Arfstrom and Hartmann, 2005; Morgan et al., 2009; Hubbard et al., 2011; Hubbard et al., 2014). This data analysis leads us to draw the following conclusions:

- The cirque-like alcoves in Deuteronilus Mensae have a median size ~11% larger than the average size of cirques on Earth (Section 5.1), which may suggest that cirques in Deuteronilus Mensae underwent more or longer episodes of erosive glaciation than cirques on Earth. The largest cirques are in the lower latitudes of the study region at 40-42.9°N (Fig. 9a). This likely suggests cirque-like alcove formation during a period of high obliquity when conditions were more favorable for glacier growth at these latitudes.

- Cirque-like alcoves contain icy deposits, as signified by the presence of flow features, linear terrain, mantling unit, washboard terrain, rectilinear-ridge terrain, moraine-like ridges, and polygonal terrain. While these icy deposits do not have all of the criteria to correspond to glacier-like forms, the features in many cirque-like alcoves represent a continuum of ice evolution, potentially as ice in viscous flow features such as glacier-like forms degrade and contain less ice volume.

- There is a dominant southward bias in the aspect of the cirque-like alcoves (Fig. 8), which becomes more pronounced above 46.5°N. We proposed this may be due to either poleward facing slopes receiving higher insolation and warmer summer daytime temperatures during high obliquity (>45°) and/or an association with gully formation, since gullies also preferentially face the equator for slopes poleward of 40° (Harrison et al., 2015; Conway et al., 2018). Overall, both cirque-like alcoves and glacier-like forms tend to have greater volumes when facing south, which may suggest a relationship between glacier-like form size and cirque-like alcove size.

- In addition to a southward bias, a slight eastward bias in aspect aligns with previous studies of both glacier-like forms on Mars (e.g., Souness et al., 2012; Brough et al., 2019) and climate models of westerly winds in Deuteronilus Mensae (Madeleine et al., 2009). Terrestrial cirques also show a similar pattern due to westerly winds. Future work could help to better understand the atmospheric controls on cirque-like alcove formation in Deuteronilus Mensae, as well as other locations on Mars.

- Headwall notches (similar to gullies) are observed adjacent to increasing sizes of larger alcoves (Fig. 15). Notches and subsequent stages of their development may act as an initiation point for ice accumulation, similar to what happens on Earth for local-slope glaciation. Larger alcoves may have undergone multiple cycles of glaciation and erosion. This process is consistent with previous work by Jawin et al. (2018).

- We estimate the time required for cirque-like alcove formation in Deuteronilus Mensae using Mars' surface gravity, obliquity models, and glacial conditions. Assuming wet-based glacial erosion (~160 m/Myr), formation would take ~9.4 Myr, consistent with age constraints for glacier-like forms and lobate debris aprons. Assuming cold-based glacial erosion (~0.85 m/Myr), formation would take ~1.8 Gyr—older than all known viscous flow features. However, cold-based erosion rates on Mars may have been higher in the past, with some estimates reaching ~10 m/Myr, which could reduce formation time to ~930 Myr. Further research is needed to evaluate the potential for cold-based glaciers to erode cirques on Mars, even though this process is minimal on Earth.

Here we show that cirque-like alcoves are consistent with the morphometrics of terrestrial cirques and retain geomorphic features indicative of ice. In addition, cirque-like alcoves have trends in aspect similar to other features such as gullies on Mars. Future work may evaluate additional regions on Mars and further explore the main factors influencing cirque-like alcove development throughout multiple cycles of glaciation and deglaciation. While we bring forward new evidence and associations using our understanding of geomorphology for Earth and Mars, further work, especially using additional high-resolution imagery and topography that may be available in the future, will be necessary to determine the style and timing of glacial activity on Mars.

**Data availability**

The shapefiles and spreadsheet of the cirque-like alcoves mapped in this study may be obtained by emailing the first author. The HRSC DEM was mosaicked using the following 29 Level 4 HRSC data frames: h5436, h5418, h5400, h5364, h5339, h5328, h5321, h5310, h5303, h5285, h5267, h5249, h5231, h5213, h3304, h3293, h3249, h3183, h2191, h1644, h1622, h1571, h1289, h1395, h1461, h1450, h1428, h1483, and h1201. The Level 4 HRSC data frames can be accessed at the ESA Planetary Science Archive: http://www.rssd.esa.int/index.php?project=PSA, HRSCview by FU Berlin/DLR: http://hrscview.fu-berlin.de/, or the NASA Planetary Data Science (PDS) http://pds-geosciences.wustl.edu/missions/mars_express/. The CTX mosaic is available through ArcGIS Pro by selecting "Portal" and selecting "Mars CTX V01" or for download at the Murray Lab website: https://murray-lab.caltech.edu/CTX/. HiRISE frames were accessed from the University of Arizona's HiRISE website: https://www.uahirise.org/hiwish/browse and are also available through the PDS. The HiRISE frames that we examined for geomorphic features included the following: ESP_041934_2265, ESP_040853_2275, ESP_036844_2225, ESP_036580_2260, ESP_036514_2210, ESP_026941_2275, ESP_025873_2230, ESP_025781_2220, ESP_025477_2280, ESP_025253_2245, ESP_023618_2270, ESP_023605_2205, ESP_019768_2220, ESP_019214_2270, ESP_016748_2255, ESP_016471_2260, ESP_016247_2270, ESP_016194_2260, ESP_067108_2240, ESP_060013_2250, ESP_057877_2245, ESP_056004_2255, ESP_055872_2270, ESP_055661_2230, ESP_054527_2225, ESP_053762_2280, ESP_052826_2240, ESP_052681_2240, ESP_052417_2220, ESP_050558_2245, ESP_048949_2230, ESP_046853_2200, ESP_046220_2235, ESP_046075_2200, ESP_046022_2265, ESP_043688_2245, ESP_025319_2240, ESP_016959_2240, ESP_027574_2245, ESP_035011_2240, PSP_006147_2250, ESP_068441_2230, ESP_033745_2270, ESP_035156_2220, and

ESP_028418_2240. The CTX DEM used in Figure 5 was made by Mackenzie Day's GALE lab at UCLA by request and is publicly accessible here: https://github.com/GALE-Lab/Mars_DEMs.

**Author contribution**

Project conceptualization by MRK and AYL with funding obtained by MRK. Methodology development by AYL, MRK, and

SB. Mapping, classification, initial analyses, and figures were created by AYL. All authors contributed to discussions of the interpretations. All authors also revised and approved the submitted manuscript.

**Acknowledgments**

AYL and MRK acknowledge funding from NASA SSW 80NSSC20KO747. We thank Anjali Manoj for her work on alcove classifications. We are very grateful to Mackenzie Day and the GALE lab at UCLA for making CTX DEMs. We also appreciate

Yingkui Li for helping us to apply ACME 2 to datasets for Mars. The reviewers Joseph Levy, Susan Conway, and Rishitosh

Sinha, as well as the editor Frances Butcher, have contributed comments that immensely improved this manuscript.

**Supplementary Material**

**Table S1:** Icy geomorphic features identified in cirque-like alcoves using available HiRISE frames.

| Alcove ID | HiRISE ID | Coverage | Latitude, Longitude | Icy geomorphic features identified |
|---|---|---|---|---|
| 50 | PSP_007439_2205 | Partial | 40.18°N, 24.72°E | Linear terrain, mantle, mound-and-tail terrain |
| 56 | ESP_072529_2265 | Partial | 40.29°N, 23.00°E | Mantle |
| 57 | ESP_072529_2265 | Full | 40.26°N, 22.98°E | Mantle |
| 145 | PSP_008810_2225 | Full | 41.85°N, 26.36°E | polygonal terrain, mantle |
| 572 | ESP_067108_2240 | Partial | 43.70°N, 27.92°E | Mantle |
| 631 | ESP_068441_2230 | Full | 42.63°N, 25.30°E | Linear terrain, mantle, washboard terrain |
| 637 | ESP_025873_2230 | Partial | 42.76°N, 25.06°E | Linear terrain |
| 650 | ESP_054527_2225 | Partial | 41.97°N, 24.63°E | Linear terrain, mantle |
| 704 | ESP_046220_2235 | Full | 42.94°N, 24.06°E | Linear terrain, polygonal terrain, washboard terrain |

| 705 | ESP_046220_2235 | Full | 42.97°N, 24.05°E | Linear terrain, polygonal terrain, washboard terrain |
|---|---|---|---|---|
| 769 | ESP_052681_2240 | Full | 43.64°N, 24.52°E | Flow features, linear terrain, moraine-like ridges, polygonal terrain |
| 783 | ESP_052826_2240 | Partial | 43.43°N, 26.02°E | Linear terrain, mantle |
| 878 | ESP_025253_2245 | Partial | 44.48°N, 29.82°E | Linear terrain, mound-and-tail terrain, polygonal terrain, washboard terrain |
| 911 | PSP_007162_2250 | Full | 44.60°N, 27.66°E | Linear terrain, mantle |
| 1061 | ESP_046022_2265 | Partial | 46.38°N, 29.00°E | Mantle, polygonal terrain, washboard terrain |
| 1088 | ESP_033745_2270 | Full | 46.66°N, 29.85°E | Linear terrain, mantle, moraine-like ridges, polygonal terrain, washboard terrain |
| 1125 | ESP_043688_2245 | Full | 44.16°N, 25.19°E | Linear terrain, polygonal terrain, washboard terrain |
| 1161 | ESP_053762_2280 | Full | 47.40°N, 27.37°E | Polygonal terrain, linear terrain, broad pit |
| 1170 | EPS_026941_2275 | Full | 47.12°N, 26.71°E | Linear terrain, polygonal terrain, moraine-like ridges, washboard terrain |
| 1171 | ESP_026941_2275 | Full | 47.14°N, 26.75°E | Linear terrain, polygonal terrain, washboard terrain |
| 1218 | ESP_055872_2270 | Full | 46.39°N, 27.09°E | Mantle, linear terrain, washboard terrain |
| 1227 | ESP_056004_2255 | Full | 45.25°N, 24.53°E | Linear terrain, polygonal terrain, mantle |
| 1230 | ESP_056004_2255 | Full | 45.25°N, 24.58°E | Mantle, polygonal, linear terrain |
| 1302 | ESP_057877_2245 | Full | 44.13°N, 23.86°E | Linear terrain, mantle, polygonal terrain |
| 1425 | PSP_002890_2205 | Full | 40.09°N, 22.72°E | Linear terrain, polygonal terrain, washboard terrain |
| 1438 | ESP_046853_2200 | Full | 40.25°N, 22.92°E | Mantle |
| 1487 | ESP_016471_2260 | Full | 45.63°N, 33.47°E | Linear terrain, washboard terrain |
| 1594 | ESP_019768_2220 | Full | 41.67°N, 18.43°E | Flow features, linear terrain, rectilinear ridge terrain, washboard terrain |
| 1616 | PSP_005857_2225 | Partial | 42.07°N, 19.52°E | Mantle |

| 1802 | ESP_035156_2220 | Full | 41.90°N, 23.90°E | Linear terrain, mantle, moraine-like ridges |
|------|-----------------|------|------------------|---------------------------------------------|
| 1808 | ESP_046075_2200 | Full | 40.29°N, 24.23°E | Linear terrain, mantle, moraine-like ridges, polygonal terrain |
| 1840 | ESP_025781_2220 | Full | 41.63°N, 16.28°E | Flow features, linear terrain, mantle |
| 1842 | ESP_025781_2220 | Partial | 41.64°N, 16.19°E | Linear terrain, mantle |
| 1965 | ESP_019214_2270 | Full | 46.57°N, 22.14°E | Linear terrain, polygonal terrain, washboard terrain |
| 1967 | ESP_019214_2270 | Full | 46.58°N, 22.14°E | Linear terrain, polygonal terrain, washboard terrain |
| 2026 | PSP_006147_2250 | Full | 44.63°N, 21.05°E | Linear terrain, polygonal terrain |

**S2 Classifications of all alcoves**

We identified six broad classes of alcoves a) simple, b) joined, c) interiorly ridged, d) staircase, e) channel-related, and f) branching were not included in our database (Fig. S1). However, for the purposes of this paper to assess possible glacial erosion, we only focus on simple alcoves. Descriptions and interpretations of each class are in Table S1. Note that the joined and staircase alcoves were mapped as one alcove, but due to their larger scale, branching alcoves offshooting from the same valley were mapped as individual alcoves. As such, smaller simple alcoves that reside within the larger branching alcoves would fall into both classes. ~4% of the alcoves were classified as two or more types. Channel-related alcoves suggest that a different erosional mechanism other than glaciation may have dominated their formation.

The ACME2 tool is designed for classic cirques on Earth and while the tool works with complex shapes, it should not be relied on for curving, elongated features (Spagnolo et al., 2017). As a result, we do not apply ACME2 for all alcove classifications. For the classes including compound, joined, staircase, and branching, the way that each class of alcoves was mapped would affect the subsequent morphometric values. For example, while we mapped branching alcoves as separate alcoves, it is possible that they should instead be considered as one large alcove if the development of individual alcoves are all dependent on the main trunk. This has a significant impact on how morphometrics would be reported because not only will the length greatly vary, but other morphometrics like the aspect will also differ across different alcoves branching off of the same trunk. As a result, there is subjectivity introduced from the mapping decisions that subsequently affect evaluations of each alcove class relative to each other. Thus we do not report on the morphometrics of all classes.

**(a) Simple**

[Figure]

[Figure]

**(b) Joined**

[Figure]

[Figure]

**(c) Interiorly ridged**

[Figure]

[Figure]

[Figure]

**Figure S1: Our preliminary classification of these alcoves assigns six classes. For each alcove class, panels (i) on the left**

**correspond to an image example, and panels (ii) on the right correspond to an example of the profile. (a) Simple:**

[revised manuscript text omitted]

Dickson, J. L., Palumbo, A. M., Head, J. W., Kerber, L., Fassett, C. I., & Kreslavsky, M. A. Gullies on Mars could have formed by melting of water ice during periods of high obliquity. Science, 380(6652), 1363-1367, 2023b.

Dundas, C. M., Conway, S. J., & Cushing, G. E. Martian gully activity and the gully sediment transport system. Icarus, 386, 115133, 2022.

Evans, I. S. Local aspect asymmetry of mountain glaciation: a global survey of consistency of favoured directions for glacier numbers and altitudes. Geomorphology, 73(1-2), 166-184, 2006.

Evans, I. S. Glaciers, rock avalanches and the 'buzzsaw' in cirque development: Why mountain cirques are of mainly glacial

 origin. Earth Surface Processes and Landforms, 46(1), 24-46, 2020.

Evatt, G. W., Fowler, A. C., Clark, C. D., and Hulton, N. R. J. Subglacial floods beneath ice sheets. Philosophical

 Transactions of the Royal Society A: Mathematical, Physical and Engineering Sciences, 364(1844), 1769-1794, 2006.

Fassett, C. I., Levy, J. S., Dickson, J. L., & Head, J. W. An extended period of episodic northern mid-latitude glaciation on

 Mars during the Middle to Late Amazonian: Implications for long-term obliquity history. Geology, 42(9), 763-766,

 2014.

Fastook, J. L., Head, J. W., Forget, F., Madeleine, J. B., and Marchant, D. R. Evidence for Amazonian northern mid-latitude

 regional glacial landsystems on Mars: Glacial flow models using GCM-driven climate results and comparisons to

 geological observations. Icarus, 216(1), 23-39, 2011.

Fran, J. S., and Martin, Y. High spatial resolution satellite imagery, DEM derivatives, and image segmentation for the

 detection of mass wasting processes. Photogrammetric Engineering, 72(6), 687-692, 2006.

French, H. M. The periglacial environment. John Wiley and Sons, 2018.

Gallagher, C., and Balme, M. Eskers in a complete, wet-based glacial system in the Phlegra Montes region, Mars. Earth and

 Planetary Science Letters, 431, 96-109, 2015.

Gallagher, C., Butcher, F. E., Balme, M., Smith, I., and Arnold, N. Landforms indicative of regional warm-based glaciation,

 Phlegra Montes, Mars. Icarus, 355, 114173, 2021.

Glasser, N. F., and Bennett, M. R. Glacial erosional landforms: origins and significance for palaeoglaciology. Progress in

 Physical Geography, 28(1), 43-75, 2004.

Graf, W.L., Cirques as glacier locations. Arct. Alp. Res. 8, 79–90, 1976.

Hallet, B., Hunter, L., and Bogen, J. Rates of erosion and sediment evacuation by glaciers: A review of field data and their

 implications. Global and Planetary Change, 12(1-4), 213-235, 1996.

Hambrey, M.J., Huddart, D., Bennett, M.R., Glasser, N.F., Genesis of 'hummocky moraines' by thrusting in glacier ice:

 Evidence from Svalbard and Britain. J. Geol. Soc. Lond. 154, 623–632, 1997.

Harrison, K. P., and Grimm, R. E. Groundwater-controlled valley networks and the decline of surface runoff on early Mars.

 Journal of Geophysical Research: Planets, 110(E12), 2005.

Harrison, T. N., Osinski, G. R., Tornabene, L. L., and Jones, E. Global documentation of gullies with the Mars

 Reconnaissance Orbiter Context Camera and implications for their formation. Icarus, 252, 236-254, 2015.

Head, J. W., and Marchant, D. R. Cold-based mountain glaciers on Mars: western Arsia Mons. Geology, 31(7), 641-644,

 2003.

Head, J. W., Marchant, D. R., Agnew, M. C., Fassett, C. I., and Kreslavsky, M. A. Extensive valley glacier deposits in the

 northern mid-latitudes of Mars: Evidence for Late Amazonian obliquity-driven climate change. Earth and Planetary

 Science Letters, 241(3-4), 663-671, 2006.

Hecht, M. H. Metastability of liquid water on Mars. Icarus, 156(2), 373-386, 2002.

Hepburn, A. J., Ng, F. S. L., Livingstone, S. J., Holt, T. O., and Hubbard, B. Polyphase mid-latitude glaciation on Mars:
Chronology of the formation of superposed glacier-like forms from crater-count dating. Journal of Geophysical
Research: Planets, 125(2), e2019JE006102, 2020.

Herman, F., Beyssac, O., Brughelli, M., Lane, S.N., Leprince, S., Adatte, T., Lin, J.Y., Avouac, J.P. and Cox, S.C. Erosion by
an Alpine glacier. Science, 350, 193, 2015.

Herman, F., De Doncker, F., Delaney, I., Prasicek, G., & Koppes, M. (2021). The impact of glaciers on mountain erosion.
Nature Reviews Earth & Environment, 2(6), 422-435.

Howat, Ian, et al., "The Reference Elevation Model of Antarctica – Strips, Version 4.1",
https://doi.org/10.7910/DVN/X7NDNY, Harvard Dataverse, V1. Accessed: 2024-05-31. 2022.

Hubbard, B., Milliken, R. E., Kargel, J. S., Limaye, A., and Souness, C. Geomorphological characterisation and
interpretation of a mid-latitude glacier-like form: Hellas Planitia, Mars. Icarus, 211(1), 330-346, 2011.

Hubbard, B., Souness, C., and Brough, S. Glacier-like forms on Mars. The Cryosphere, 8(6), 2047-2061, 2014.

Janke, J. R., Regmi, N. R., Giardino, J. R., and Vitek, J. D. Rock Glaciers. Treatise on Geomorphology (Vol. 8), 2013.

Jawin, E. R., Head, J. W., and Marchant, D. R. Transient post-glacial processes on Mars: geomorphologic evidence for a
paraglacial period. Icarus, 309, 187-206, 2018.

Jawin, E. R., and Head, J. W. Patterns of late Amazonian deglaciation from the distribution of martian paraglacial features.
Icarus, 355, 114117, 2021.

Khuller, A. R., Christensen, P. R., Harrison, T. N., and Diniega, S. The distribution of frosts on Mars: Links to present-day
gully activity. Journal of Geophysical Research: Planets, 126(3), 2021.

Kite, E. S. Geologic constraints on early Mars climate. Space Science Reviews, 215, 1-47, 2019.

Kleman, J., and Stroeven, A.P., Preglacial surface remnants and Quaternary glacial regimes in northwestern Sweden.
Geomorphology 19 (1), 35–54, 1997.

Knight, J., Harrison, S., and Jones, D. B. Rock glaciers and the geomorphological evolution of deglacierizing mountains.
Geomorphology, 324, 14-24, 2019.

Koutnik, M., Butcher, F. E., Soare, R. J., Hepburn, A. J., Hubbard, B., Brough, S., C. Gallagher, L. McKeown, and Pathare,
A. (2024). Glacial deposits, remnants, and landscapes on Amazonian Mars: Using setting, structure, and
stratigraphy to understand ice evolution and climate history. In Ices in the Solar-System (pp. 101-142). Elsevier.

Kreslavsky, M. A., Head, J. W., & Marchant, D. R. Periods of active permafrost layer formation during the
geological history of Mars: Implications for circum-polar and mid-latitude surface processes. Planetary and Space
Science, 56(2), 289-302, 2008.

Laity, J. The Role of Groundwater Sapping in Valley Evolution. Sapping Features of the Colorado Plateau: A Comparative
Planetary Geology Field Guide, 491, 63, 1988.

Lamb, M. P., Howard, A. D., Johnson, J., Whipple, K. X., Dietrich, W. E., and Perron, J. T. Can springs cut canyons into rock?. Journal of Geophysical Research: Planets, 111(E7), 2006.

Lapotre, M. G., Lamb, M. P., and Williams, R. M. Canyon formation constraints on the discharge of catastrophic outburst floods of Earth and Mars. Journal of Geophysical Research: Planets, 121(7), 1232-1263, 2016.

Lapotre, M. G., and Lamb, M. P. Substrate controls on valley formation by groundwater on Earth and Mars. Geology, 46(6), 531-534, 2018.

Larsen, E., Mangerud, J., Erosion rate of a Younger Dryas cirque glacier at Krakenes, western Norway. Ann. Glaciol. 2 (1), 153–158, 1981.

Laskar, J., Levrard, B., & Mustard, J. F. Orbital forcing of the Martian polar layered deposits. Nature, 419(6905), 375-377, 2002.

Lehmann, B., Anderson, R. S., Bodin, X., Cusicanqui, D., Valla, P. G., and Carcaillet, J. Alpine rock glacier activity over Holocene to modern timescales (western French Alps). Earth Surface Dynamics, 10(3), 605-633, 2022.

Levy, J. S., Head, J. W., and Marchant, D. R. Concentric crater fill in Utopia Planitia: History and interaction between glacial "brain terrain" and periglacial mantle processes. Icarus, 202(2), 462-476, 2009a.

Levy, J. S., Head, J., and Marchant, D. Thermal contraction crack polygons on Mars: Classification, distribution, and climate implications from HiRISE observations. Journal of Geophysical Research: Planets, 114(E1), 2009b.

Levy, J. S., Fassett, C.I., Head, J.W., Schwartz, C., Watters, J.L., Sequestered glacial ice contribution to the global Martian water budget: Geometric constraints on the volume of remnant, midlatitude debris-covered glaciers. J. Geophys. Res. Planets 119, doi: 10.1002/2014JE004685, 2014.

Levy, J. S., Fassett, C. I., & Head, J. W. Enhanced erosion rates on Mars during Amazonian glaciation. Icarus, 264, 213-219, 2016.

Levy, J.S., Fassett, C.I., Holt, J.W., Parsons, R., Cipolli, W., Goudge, T.A., Tebolt, M., Kuentz, L., Johnson, J., Ishraque, F. and Cvijanovich, B. Surface boulder banding indicates Martian debris-covered glaciers formed over multiple glaciations. Proceedings of the National Academy of Sciences, 118(4), p.e2015971118, 2021.

Lewkowicz, A. G. Morphology, frequency and magnitude of active-layer detachment slides, Fosheim Peninsula, Ellesmere Island, NWT. In Proceedings of the 5th Canadian Permafrost Conference (Vol. 54, pp. 111-118). Université Laval, Québec: Centre d'études nordiques, 1990.

Lewkowicz, A. G. Dynamics of active-layer detachment failures, Fosheim peninsula, Ellesmere Island, Nunavut, Canada. Permafrost and Periglacial Processes, 18(1), 89-103, 2007.

Li, Y., Evans, I. S., Spagnolo, M., Pellitero, R., Barr, I. D., & Ely, J. C. ACME2: An extended toolbox for automated cirque metrics extraction. Geomorphology, 445, 108982, 2024.

Lillquist, K., and Weidenaar, M. Rock glaciers in the Eastern Cascades, Washington State, USA: Impacts of selected variables on spatial distribution and landform dimensions. Geomorphology, 389, 107839, 2021.

Lukas, S., A test of the englacial thrusting hypothesis of 'hummocky' moraine formation: Case studies from the northwest Highlands, Scotland. Boreas 34, 287–307, 2005.

Mackay, S.L., Marchant, D.R., Lamp, J.L., Head, J.W. Cold-based debris-covered glaciers: Evaluating their potential as climate archives through studies of ground-penetrating radar and surface morphology. J. Geophys. Res. Earth Surf. 119, 2505–2540. https://doi.org/10.1002/2014JF003178, 2014.

Madeleine, J. B., Forget, F., Head, J. W., Levrard, B., Montmessin, F., and Millour, E. Amazonian northern mid-latitude glaciation on Mars: A proposed climate scenario. Icarus, 203(2), 390-405, 2009.

Malin, M.C., Bell, J.F., Cantor, B.A., Caplinger, M.A., Calvin, W.M., Clancy, R.T., Edgett, K.S., Edwards, L., Haberle, R.M., James, P.B., Lee, S.W., Ravine, M.A., Thomas, P.C., Wolff, M.J., Context camera investigation on board the Mars Reconnaissance Orbiter. J. Geophys. Res. 112, E05S04, doi: 10.1029, 2007.

Mangold, N., and Allemand, P. Topographic analysis of features related to ice on Mars. Geophysical Research Letters, 28(3), 407-410, 2001.

Marchant, D. R., and Head III, J. W. Antarctic dry valleys: Microclimate zonation, variable geomorphic processes, and implications for assessing climate change on Mars. Icarus, 192(1), 187-222, 2007.

McEwen, A.S., Eliason, E.M., Bergstrom, J.W., Bridges, N.T., Hansen, C.J., Delamere, W. A., Grant, J.A., Gulick, V.C., Herkenhoff, K.E., Keszthelyi, L., Kirk, R.L., Mellon, M.T., Squyres, S.W., Thomas, N., Weitz, C.M., Mars Reconnaissance Orbiter's High Resolution Imaging Science Experiment (HiRISE). J. Geophys. Res. 112, E05S02, doi: 10.1029/2005JE002605, 2007.

Michael, G. G. Planetary surface dating from crater size-frequency distribution measurements: Multiple resurfacing episodes and differential isochron fitting, Icarus, 226, 885–890, 2013.

Milliken, R. E., Mustard, J. F., and Goldsby, D. L. Viscous flow features on the surface of Mars: Observations from high-resolution Mars Orbiter Camera (MOC) images. Journal of Geophysical Research: Planets, 108(E6), 2003.

Mîndrescu, M., Evans, I. S., and Cox, N. J. Climatic implications of cirque distribution in the Romanian Carpathians: palaeowind directions during glacial periods. Journal of Quaternary Science, 25(6), 875-888, 2010.

Morgan, G. A., Head III, J. W., and Marchant, D. R. Lineated valley fill (LVF) and lobate debris aprons (LDA) in the Deuteronilus Mensae northern dichotomy boundary region, Mars: Constraints on the extent, age and episodicity of Amazonian glacial events. Icarus, 202(1), 22-38, 2009.

Morgan, G. A., Putzig, N. E., Perry, M. R., Sizemore, H. G., Bramson, A. M., Petersen, E. I., ... and Campbell, B. A. Availability of subsurface water-ice resources in the northern mid-latitudes of Mars. Nature Astronomy, 5(3), 230-236, 2021.

Mustard, J. F., Christopher, D. C., and Moses, K. R. Evidence for recent climate change on Mars from the identification of youthful near-surface ground ice. Nature, 412(6845), 411, 2001.

NASA Shuttle Radar Topography Mission (SRTM). Shuttle Radar Topography Mission (SRTM) Global. Distributed by OpenTopography. https://doi.org/10.5069/G9445JDF. Accessed: 2024-08-02. 2013.

Neukum, G., Jaumann, R., The HRSC Co-Investigator and Experiment Team, HRSC: the high resolution stereo camera of Mars Express. In: Wilson, A. (Ed.), Mars Express: The Scientific Payload, 1240. European Space Agency Special

Publication, pp. 17–35, 2004.

Oliva, M., Andrés, N., Fernández-Fernández, J. M., and Palacios, D. The evolution of glacial landforms in the Iberian Mountains during the deglaciation. In European Glacial Landscapes (pp. 201-208), 2023.

Palucis, M. C., Jasper, J., Garczynski, B., and Dietrich, W. E. Quantitative assessment of uncertainties in modeled crater retention ages on Mars. Icarus, 341, 113623, 2020.

Pilorget, C., and Forget, F. Formation of gullies on Mars by debris flows triggered by $CO_2$ sublimation. Nature Geoscience, 9(1), 65-69, 2016.

Plaut, J. J., Safaeinili, A., Holt, J. W., Phillips, R. J., Head III, J. W., Seu, R., Radar evidence for ice in lobate debris aprons in the mid-northern latitudes of Mars. Geophysical Research Letters, 36(2), 2009.

Rignot, E., Hallet, B., & Fountain, A. Rock glacier surface motion in Beacon Valley, Antarctica, from synthetic-aperture radar interferometry. Geophysical Research Letters, 29(12), 48-1, 2002.

Robbins, S. J., & Hynek, B. M. (2012). A new global database of Mars impact craters≥ 1 km: 2. Global crater properties and regional variations of the simple-to-complex transition diameter. Journal of Geophysical Research: Planets, 117(E6).

Rudberg, S. Multiple glaciation in Scandinavia: seen in gross morphology or not? Geogr. Ann. Ser. A Phys. Geogr. 74, 231–243, 1992.

Sanders, J. W., Cuffey, K. M., Moore, J. R., MacGregor, K. R., & Kavanaugh, J. L. Periglacial weathering and headwall erosion in cirque glacier bergschrunds. Geology, 40(9), 779-782, 2012.

Sanders, J. W., Cuffey, K. M., MacGregor, K. R., and Collins, B. D. The sediment budget of an alpine cirque. GSA Bulletin, 125(1-2), 229-248, 2013.

Schon, S. C., Head, J. W., & Fassett, C. I. Unique chronostratigraphic marker in depositional fan stratigraphy on Mars: Evidence for ca. 1.25 Ma gully activity and surficial meltwater origin. Geology, 37(3), 207-210, 2009.

Schon, S. C., Head, J. W., & Fassett, C. I. Recent high-latitude resurfacing by a climate-related latitude-dependent mantle: Constraining age of emplacement from counts of small craters. Planetary and Space Science, 69(1), 49-61, 2012.

Selby, M. J., and Wilson, A. T. Possible Tertiary age for some Antarctic cirques. Nature, 229(5287), 623-624, 1971.

Sharp, R.P., 1973. Mars: Fretted and chaotic terrains. J. Geophys. Res. 78, 4073–4083.

Sharp, M., ''Crevasse-fill'' ridges – a landform type characteristic of surging glaciers? Geografiska Annaler 67A, 213–220, 1985.

Shean, D. E., Head, J. W., and Marchant, D. R. Origin and evolution of a cold-based tropical mountain glacier on Mars: The Pavonis Mons fan-shaped deposit. Journal of Geophysical Research: Planets, 110(E5), 2005.

Sholes, S. F., and Rivera-Hernández, F. Constraints on the uncertainty, timing, and magnitude of potential Mars oceans from topographic deformation models. Icarus, 378, 114934, 2022.

Squyres, S.W. Martian fretted terrain: Flow of erosional debris. Icarus, 34, 600– 613, 1978.

Squyres, S. W. The distribution of lobate debris aprons and similar flows on Mars. Journal of Geophysical Research:

Solid Earth, 84(B14), 8087-8096, 1979.

Soare, R. J., Williams, J. P., Hepburn, A. J., & Butcher, F. E. A billion or more years of possible periglacial/glacial cycling in Protonilus Mensae, Mars. Icarus, 385, 115115, 2022.

Souness, C., Hubbard, B., Milliken, R. E., and Quincey, D. An inventory and population-scale analysis of martian glacier-like forms. Icarus, 217(1), 243-255, 2012.

Souness, C. J., and Hubbard, B. An alternative interpretation of late Amazonian ice flow: Protonilus Mensae, Mars. Icarus, 225(1), 495-505, 2013.

Spagnolo, M., Pellitero, R., Barr, I. D., Ely, J. C., Pellicer, X. M., and Rea, B. R. ACME, a GIS tool for automated cirque metric extraction. Geomorphology, 278, 280-286, 2017.

Thomson, L. I., & Copland, L. Multi-decadal reduction in glacier velocities and mechanisms driving deceleration at polythermal White Glacier, Arctic Canada. Journal of Glaciology, 63(239), 450-463, 2017.

Turnbull J. M., Davies T. R. H. A mass movement origin for cirques. Earth Surface Processes and Landforms 31: 1129–1148, 2006.

White, W. A. Erosion of cirques. The Journal of Geology, 78(1), 123-126, 1970.

Williams, K. E., Toon, O. B., Heldmann, J. L., McKay, C., and Mellon, M. T. Stability of mid-latitude snowpacks on Mars. Icarus, 196(2), 565-577, 2008.

Williams, K. E., Toon, O. B., Heldmann, J. L., and Mellon, M. T. Ancient melting of mid-latitude snowpacks on Mars as a water source for gullies. Icarus, 200(2), 418-425, 2009.

Williams, J. M., Scuderi, L. A., McClanahan, T. P., Banks, M. E., and Baker, D. M. Comparative planetology–Comparing cirques on Mars and Earth using a CNN. Geomorphology, 440, 108881, 2023.

Willmes, M., Reiss, D., Hiesinger, H., & Zanetti, M. Surface age of the ice–dust mantle deposit in Malea Planum, Mars. Planetary and Space Science, 60(1), 199-206, 2012.

Woodley, S. Z., Butcher, F. E., Fawdon, P., Clark, C. D., Ng, F. S., Davis, J. M., and Gallagher, C. Multiple sites of recent wet-based glaciation identified from eskers in western Tempe Terra, Mars. Icarus, 386, 115147, 2022.

---

## Author Response (AR4)

Dear authors,
Thank you for submitting your revised manuscript. The revisions have improved the manuscript. I will be pleased to accept it for publication subject to a small number of minor corrections required. These include typographical/figure errors, and some minor issues with passages that were significantly revised in the last review. In all cases, the latter can be addressed with simple edits to the text, for which I have provided suggestions.

We thank the editor for taking the time to provide another round of detailed and very helpful comments.

Several new errors have arisen with each revised version, which should have been identified with proofreading. Please undertake more careful proofreading before submitting the revised version. This includes checks for typographical/figure errors, that citations are correct (e.g. dual author versus et al.), and that all citations are included in the reference list.

We updated the reference list and proofread again.

My most substantive comment pertains to Section 5.1.2: (containing content about gullies that has been particularly queried by previous reviews)
(1) It would be better to name this section 'Trends in aspect for cirque-like alcoves, and implications for cirque initiation'. The examples you show are not gullies according to the strict Martian definition (having alcove, chute, and depositional fan, and typically being rather young features), but you can use gullies as an example of aspect-dependent hillslope processes to comment on potential mechanisms for cirque initiation. The specific reference to gullies also breaks the flow of the argument, since it comes before the comparisons to GLF distribution which are more central to the analysis.

We have changed the name of the section, though in Fig. 12b, we have also added an example of a gully mapped by Noblet et al., (2024) along the same mesa edge.

[Figure]

**Figure 12: Examples of mesa slopes with shallow alcoves, larger alcoves, and adjacent ice. (a) Shallow alcoves may indicate ice-associated erosion all along the mesa sidewall. Flow features indicate the downslope direction of ice flow. Centered at 41.06°N, 17.88°E in CTX image D04_0288880_2193_XI_39N342W. (b) Shallow alcoves may indicate ice-associated erosion while larger alcoves with multiple smaller alcoves nested within may represent a later stage of development. Gullies (Noblet et al., 2024) may also act as initiation points for alcoves. Flow features indicate the downslope direction of ice flow. While the larger alcoves in this figure were mapped as alcoves, they were not classified as simple alcoves (and as a result, not cirque-like alcoves) because they have interior ridges (Appendix B). Centered at 40.02°N, 23.20°E in the CTX mosaic (Dickson et al., 2018). HiRISE data credit: NASA/JPL/University of Arizona. CTX data credit: Caltech/NASA/JPL/MSSS.**

(2) I agree with the reviewers' earlier concerns that you are not actually showing gullies in the figure (and thereby not explicitly supporting gullies as potential initiation points), which still causes some weakness in the argument here. This is a simple fix, requiring only light modifications to the text to focus more explicitly on the shallow, elongate alcoves you observe and show in the figure as follows:

We thank the editor for taking the time to provide the suggested modifications. We have followed the suggestions, and also updated Fig. 12b to include a previously mapped gully by Noblet et al. (2024).

• Make the suggested section title change in (1).

We have accepted the suggested section title change.

• Before 'Regardless of how gullies are initiated', add a sentence, referring to the figure, which specifically states that shallow elongate alcoves exist on hillslopes adjacent to

the larger alcoves. Mention that they are not necessarily gullies according to the narrow morphological definition of Martian gullies in the literature.

Corrected.

• Then, 'Regardless of how shallow alcoves of any kind are initiated (be they gullies or other types of hillslope depressions), they may act as cold-traps where snow could accumulate (e.g. Dickson et al. 2023) and initiate formation of cirque-like alcoves.

Corrected.

• Then, in what follows, just refer to shallow alcoves as a more general category (I think it will be ok to retain the 'shallow alcoves or even gullies' statement you currently use towards the end).

Corrected to "For example, shallow alcoves and gullies could provide the initial concavity for a later cirque-like alcove to develop when glaciation occurs (Fig. 12), which is consistent with gully heads that have been proposed as initiation points for cirques on Earth (Derbyshire and Evans; 1976)."

Specific comments:

L23: Moraine-like ridge repeats

Corrected.

L24: 'the mantling deposit, an ice-rich unit' > an ice-rich mantling unit

Corrected.

L27: alcoves > alcove formation

Corrected.

L27: After wet-based erosion rates, add 'Assuming a mean annual temperature of -20°C (compared to present-day temperatures of ~-60°C)'

Edited to 0°C (instead of -20°C) for the wet-based erosion rates.

L29: Should also state the same temperature assumption is used for this.

The sentence was edited to read as follows: "In contrast, using a temperature assumption of -50 to -68℃, cold-based erosion rates are only consistent with the older ages of lobate debris aprons."

L61: For the mid-latitudes, models show glaciation at obliquities >35. >45 is associated with equatorial glaciation

Corrected.

Figure 2, panel b(i). I have flagged this issue several times previously but it still remains. The label for the extent indicator in panel b(i) should read b(ii), not c(ii)

The panel has been updated to b(ii).

[Figure]

Corrected.

[Figure]

We edited it to read "approximate…" In addition, we edited the following sentence to include the caveat: "Here we find the total cavity volume of the alcove as a proxy for the maximum amount of ice that an alcove may have contained, which is likely an overestimate since it assumes the alcove is round."

L198: Suggest size > 'size'

Corrected.

L199: full stop after citation.

Period was added.

L199: find > use

Corrected.

L200: of alcove > of the alcoves

Corrected.

L204: typical L/W > typical cirque L/W

Corrected.

L211-212: on Mars > in Deuteronilus Mensae, Mars

Corrected.

L286: 'correspond to slopes of 20-25 degrees': careful here. If you are referring to slopes within cirques, this should be 'have slopes of'. Correspond to implies they occur *on* slopes of this angle (i..e that you measured the generalised slope around the cirque)

Corrected.

Table 5 Column 3: State in caption what N/A means (doesn't mean 0%, means images likely too low resolution to identify)

We added the following to the caption: "'N/A' in the third column means the CTX images have too low of a resolution to identify the feature."

L384: more frequent > more (or more frequent): Given longer timescales, events could have had same frequency, but there could simply have been more of them.

Corrected.

L421: ice accumulation and melt on the landscape > ice accumulation on the landscape, including the possible role of past melting.

Corrected.

L426: here and in the figure I'd avoid the use of the term channels - immediately invokes flowing water. You could say large alcoves with multiple smaller alcoves nested within?

Edited to "large alcove with smaller alcoves nested within" as shown in the figure at the beginning of the response.

L426: add a label for the simple alcove to the right of the shallow alcoves in b? Presumably this made it into your cirque-like alcove database? If so, point it out. If not, explain whether it fell into one of the more complex categories.

The caption (at the beginning of the responses) now explains why those alcoves were not classified as simple alcoves.

Fig 13: Latitude (rounded) presumably only applies to the plotted lines, not the points. Take out of axis label and put instead in the relevant part of the caption.

The figure was updated as follows:

[Figure]

The caption now reads: "**Figure 13: Scatterplot of mean elevation for the 435 cirque-like alcoves versus latitude (magenta points). The magenta solid line and teal dashed line are binned from 41,618,659 elevation scatter points from the mosaicked HRSC DEM, which were found using the raster to point tool in ArcGIS Pro. The binned mean elevation (magenta line) represents the mean elevation value of all the HRSC DEM scatter points rounded to each half degree of latitude in the study region. The binned elevation difference (teal dashed line) was calculated based on the difference between the mean of the highest 10,000 points and the mean of the lowest 10,000 points rounded to each half degree of latitude.**"

L493-500. This paragraph is somewhat improved, but is still confusing. I suggest deleting. The observed moraines are well outside the alcoves, which doesn't seem to be the best configuration for subsequent cirque erosion (which is driven by rotational slip, and therefore somewhat requires that the glacier front is at least partially constrained at the alcove edge). That doesn't seem to be the case in these examples. 'moraine-like ridges may reside...' is also very speculative, and not actually observed.

We recognize that the way the paragraph was previously phrased was confusing. We have rewritten it for clarity as follows:

"As with moraines on Earth, the position of the moraine-like ridges on Mars reveal the extent of glaciation relative to the cirque-like alcoves. Moraine-like ridges extend either directly from (Fig. 11a) or downslope of cirque-like alcoves (e.g., Fig. 11b, Fig. 5). The moraine-like ridges that extend directly from cirque-like alcoves (Fig. 11a) are akin to terminal moraines formed by terrestrial cirque glaciers that only remain within their cirque basin. Moraine-like ridges further downslope (Fig. 5) reflect a stage where the glaciers grew beyond the cirque-like alcoves.

Similar to Arfstrom and Hartmann (2005), the moraine-like ridges in Fig. 11b reflect the glacial erosion and deposition of mesa material sourced from their corresponding upslope alcoves. In Fig. 11b, only two of the alcoves are well-developed and have morphometrics corresponding to the criteria we set for cirque-like alcoves. The rest of the alcoves remain underdeveloped and require further glacial erosion to qualify as cirque-like alcoves."

L539-542: There are repetitions about knowledge of temperature here. Revise and tighten.

We have deleted these sentences and updated the calculations and temperatures to correspond to -50 to -68℃, as well as using values from Parsons and Holt 2016 instead of a rock glacier from Earth:

"For finding $U_{surf}$, surface velocities of glacier-like forms on Mars are not well constrained and may have included short, warm periods that allowed for melting (Hubbard et al., 2014). For a lower bound surface velocity for cold-based glaciers on Mars, recent modeling using a mean annual present-day surface temperature of -63 ℃ and found a maximum surface velocity of $20 \times 10^{-6}$ m/yr for a thin (<100 m) viscous flow feature on a steep slope (Butcher et al., 2024). On the other hand, Parsons and Holt (2016) find a surface velocities ranging from 0.05-20 mm/yr for ice at -68 ℃, with the lower velocity of 0.05 mm/yr corresponding to 200 Myr ice and 20 mm/yr corresponding to 200 kyr ice. We include 20 mm/yr as an upper bound for cold-based surface velocity (Table 8). For the wet-based case, we use a surface velocity of 2 m/yr (based on a polythermal glacier in Broggerbeen, Svalbard; Table 1 of Cook et al., 2020)."

L539-542: While we don't know the past temperature, we do know the present average temperature for the mid-latitudes, so this should be stated, as comparison to the values you have chosen.

As mentioned above, we have edited the text to correspond to the new temperatures. In addition, we have edited the following about the *A* parameter:

"For the *A* parameter, since it is unknown how much the temperature of ice on Mars has fluctuated throughout the Amazonian, we use both a warm and cold ice scenario. For the warm ice, we use 0 ℃ (Table 8). For the cold ice, -50 ℃ is the coldest temperature for *A* from Cuffey and Paterson (2010) that approaches the mean annual present-day surface temperature of -63 ℃ (e.g., Forget et al., 1999; Butcher et al., 2024)."

L539: 'Approximately the same' – what are the actual values? It is a little odd to pick the warmer one since the colder one might be the more conservative estimate for Mars, but if you can explicitly show that they really are similar (and that any difference doesn't amplify by propagation through to required timescales), this would be more robust.

As mentioned above for an earlier comment, we have edited the text to use only surface velocities modeled for Mars for the cold-based case, as well as *A* corresponding to -50 ℃. In addition, we added the table below to show the values corresponding to the calculations:

**Table 8.** Overview of the constants, values, and results used for age calculations of cirque-like alcoves on Mars. "*" indicates values that were calculated in this work. "-" indicates that the value was not used for that column.

| | Wet-based glaciation | Cold-based glaciation | |
|---|---|---|---|
| *n*, flow-law exponent | 3 | | - |
| *ρ*, ice density | 917 kg/m$^3$ | | - |
| *g*, gravitational acceleration | 3.71 m/s$^2$ | | - |

| | | | | |
|---|---|---|---|---|
| $\alpha$, ice surface slope | 5°
(typically 2-8° for glacier-like forms on Mars; Brough et al., 2019) | | | - |
| $h$, ice thickness | 130 m for glacier-like forms on Mars (Brough et al., 2019) | | | - |
| Temperature | 0 ℃ | -50 to -68 ℃ | | - |
| $A$, temperature-dependent ice softness parameter | $24 \times 10^{-25} s^{-1} Pa^{-3}$ (Cuffey and Paterson, 2010) | $2.6 \times 10^{-27} s^{-1} Pa^{-3}$ (Cuffey and Paterson, 2010) | | - |
| $U_{surf}$, surface velocity | 2 m/yr (polythermal glacier in Broggerbeen, Svalbard; Table 1 of Cook et al., 2020) | $20 \times 10^{-6}$ m/yr (estimate for present-day debris-covered ice on Mars; Butcher et al., 2024) | $20 \times 10^{-3}$ m/yr (estimate for 200 kyr debris-covered ice on Mars; Parsons and Holt, 2016) | - |
| $U_S$, basal sliding velocity* | 2.0 m/yr | $20 \times 10^{-6}$ m/yr | $20 \times 10^{-3}$ m/yr | - |
| $K_G$, bedrock erodibility constant | $10^{-4}$ (Cook et al., 2020) | | | - |
| $l$, erosion exponent | 0.69 (empirical estimate from Cook et al., 2020) | | | - |

| E, erosion rate* | 160 m/Myr | 0.0572 m/Myr | 6.73 m/Myr | | 10 m/Myr (Levy et al., 2016) |
|---|---|---|---|---|---|
| Cirque-like alcove height* | Median: 300 m | | Median: 300 m | Maximum: 1735 m | Maximum: 1735 m |
| Time required to erode* | 1.86 Myr | 5.24 Gyr | 44.6 Myr | 258 Myr | 174 Myr |
| Total time required* (accounting for obliquity and cirque erosion occurring at the beginning/end of glacial periods) | 9.30 Gyr | 26.5 Gyr | 223 Myr | 1.29 Gyr | 868 Myr |

L541: Beacon valley…on Earth.

Corrected.

L549: Cook citation missing from ref list. Please carefully (a) check all citations are in the ref list, and (b) state here what example such a velocity is based on (Earth warm-based glacier? where?)

(a) We have added Cook et al. and checked all citations are in the reference list.

(b) In parentheses for the wet-based case, we added the following: "(based on a glacier in Broggerbeen, Svalbard; Table 1 of Cook et al., 2020)."

L564: could have been eroded *from* a cirque like alcove

Corrected.

We have reiterated as follows:

"On Earth, cirques are presumed to be mostly eroded at the beginning and the end of glaciations (e.g., Barr et al., 2019), so assuming that the cirque-like alcoves only have 20 kyr of erosion time during every 100 kyr period, or 20% of the total time passing by, it would take ~9.3 Myr of total time to erode a median height cirque-like alcove using wet-based glacial erosion rates (Table 8). This timescale is consistent with previous estimates of the age of certain populations of glacier-like forms (Hepburn et al., 2020), which means that at least some of the glacier-like forms could have eroded the cirque-like alcoves which they currently occupy if they had wet-based glacial erosion rates and that at least some of the empty cirque-like alcoves could have hosted glaciers in the past tens of millions of years."

L578: The Conway estimates are interpreted as wet-based rates, so perhaps confusing to state here. Delete.

Deleted.

We have edited to read as follows: "By including obliquity variations and using an erosion rate of ~6.7 m/Myr, a median height cirque-like alcove would require ~220 Myr to erode (Table 8). For a maximum height cirque-like alcove, an erosion rate of ~6.7 m/Myr would yield a total age of ~1.3 Gyr (Table 8), which is slightly longer than the ~500 Myr (Fassett et al., 2014) to 1.1 Gyr (Berman et al., 2015) timescales of when lobate debris aprons and concentric crater fills were estimated to have formed."

L584: Somewhat repeats earlier LDA/CCF points. Tighten up. You only need to make this point once (suggest removing the LDA/CCF statement).

The sentence on LDA/CCF was based specifically on a prior reviewer's comment, but we deleted it.

L586-605: This paragraph is confusing in its structure. It is unclear how the conclusion in the first sentence is drawn, having not yet been justified by the discussion further down the paragraph. I suggest moving this after the quantitative discussion further down the paragraph, and instead starting the para with the discussion of maximum erosion depths. There is some repetition with the previous paragraph – tighten this up. See also comments below about removing reference to erasure of alcoves by hillslope wastage.

The reference to erasure of alcoves was also removed. We reordered the paragraph and tightened it up, as seen below, by moving the quantitative information up to the prior paragraph:

"On the other hand, if we assume cold-based conditions for glaciers that occupied the cirque-like alcoves, then the erosion rate estimated is ~0.057 to 6.7 m/Myr. However, the lower bound of 0.057 m/Myr is unrealistically low because a median cirque-like alcove would take longer than the age of Mars to erode (Table 8). However, the upper bound of ~6.7 m/Myr falls within the previous wide-ranging estimate of 0.1-10 m/Myr for cold-based viscous flow features on Mars (Levy et al., 2016). Using an erosion rate of 6.7 m/Myr, a total glacier occupation time of ~45 Myr would be required for median cirque-like alcoves to form without accounting for obliquity variations. By including obliquity variations and using an erosion rate of ~6.7 m/Myr, a median height cirque-like alcove would require ~220 Myr to erode (Table 8). For a maximum height cirque-like alcove, an erosion rate of ~6.7 m/Myr would yield a total age of ~1.3 Gyr (Table 8), which is slightly longer than the ~500 Myr (Fassett et al., 2014) to 1.1 Gyr (Berman et al., 2015) timescales of when lobate debris aprons and concentric crater fills were estimated to have formed. If instead the upper-end erosion rate of 10 m/Myr for cold-based glaciers is applied from Levy et al. (2016), then all heights of the cirque-like alcoves could have formed via cold-based glacier erosion within ~870 Myr (Table 8). Given the upper timescale limit of ~1.1 Gyr for viscous flow features, it is likely that cirque-like alcoves had erosion rates higher than 6.7 m/Myr.

Since debris from the cirque-like alcoves often superposes the lobate debris aprons (e.g., Baker and Carter, 2019), this means that the process eroding the cirque-like alcoves has continued after when the ice in the lobate debris aprons formed. The supraglacial debris covering the lobate debris aprons is likely <10 m (Holt et al., 2008; Plaut et al., 2009) and a major fraction of the debris was sourced as rockfall from the mesas (Baker and Carter et al., 2019). In order to remove the debris, cirque-like alcove erosion has occurred either concurrently or after lobate debris apron formation. On Earth, the chronology of cirque formation is difficult to constrain (e.g., Turnbull and Davies, 2006), and estimates for total glacial cirque erosion time range from 125 kyr (Larsen and Mangerud, 1981) to a few million years (Andrews and Dugdale, 1971; Anderson, 1978; Sanders et al., 2013)."

L591: 'averages 25m'. Not strictly what that paper says. The debris PLUS superposing ice-rich mantling likely has thickness LESS THAN 25m. Suggest linking to radar papers e.g. Holt, Plaut, Petersen which show the debris layer on VFFs in general is likely <10m

Corrected as follows: "The supraglacial debris covering the lobate debris aprons is likely <10 m (Holt et al., 2008; Plaut et al., 2009) and a major fraction of the debris was sourced as rockfall from the mesas (Baker et al., 2019)."

L592: Baker et al. 2019 > Baker and Carter 2019. Please check all references and citations carefully.

Corrected.

L594: Sentence beginning: 'Otherwise the erosional process'... I'm not sure this is true. The debris layer is only thin, but removal of cirque alcoves of sizes >100s metres would require large volumes to be eroded. Delete, and instead just use the argument stated previously that debris from the alcoves superposes the VFFs

Deleted.

L603: Sentence referring to Butcher et al. velocities. Now that this is incorporated earlier, I think you can delete this statement from here. It refers to present-day climate so isn't a good comparison.

Deleted.

Corrected.

L608-610: restate the assumed surface temperature, and how this differs from the contemporary mean annual surface temp

We edited as follows: "Our estimates here find that a median height cirque-like alcove in Deuteronilus Mensae would take ~9.3 Myr to form if occupied by a wet-based glacier at 0 ℃ with an erosion rate of 160 m/Myr. For the cold-based case, using a very low surface velocity from Butcher et al. (2024) and present-day temperature of -60 ℃ yields an unrealistic age for the cirque-like alcoves that is older than the age of Mars. However, if a faster surface velocity is applied from Parsons and Holt (2016), then a median cirque would require ~220 Myr if occupied by a cold-based glacier at -50 to -68 ℃ with an erosion rate of 6.73 m/Myr. Since cold-based erosion rates may vary by up to two orders of magnitude, maximum cold-based erosion rates closer to 10 m/Myr would allow for timescales of hundreds of millions of years. Thus, if the glaciers were cold-based during their entire evolution, the erosion timescale is longer and therefore the alcoves must be much older than if they evolved during periods of wet-based glaciation. Whether the cold-based erosion rates on Mars were more similar to what has been observed at Meserve Glacier in Antarctica at ≤3 m/Myr (Cuffey et al., 2000) or at much higher values closer to 10 m/Myr (Levy et al., 2014) remains unknown."

Deleted the second "previously."

Corrected.

L624: Delete 'in the past' – covered already with 'formerly'

Deleted.

Deleted the second "moraine-like ridges."

Corrected to "an ice-rich mantling unit."

L643-655: I'll leave this up to your discretion, but personally, I would merge these to discuss a southeastward bias, and then explain south + east in the same paragraph. the reader could interpret the first point as facing due south and the second point as facing due east, which seems contradictory when detached from the results

Corrected and merged into one paragraph.

L661: -'and temperature': 20 degrees is used, which isn't Mars' current mid latitude temperature.

We edited this sentence as follows: "We estimate the time required for cirque-like alcove formation in Deuteronilus Mensae using Mars' surface gravity, obliquity models, and assume an ice temperature of -50 to -68 ℃ (Section 5.3)."

L677: 'are provided'. Where? Please provide either a link to a repository, or a reference to the relevant supplementary material. I cannot see it included presently.

The spreadsheet of the cirque-like alcoves mapped in this study is attached to Appendix C and the description is below. We include only the spreadsheet because it is allowed as a table in the Appendix, but other files can only be included as a repository.

**Appendix C: Spreadsheet of cirque-like alcoves**

The spreadsheet for the 435 cirque-like alcoves identified in this work is attached.

**Table C1:** Parameters and their corresponding descriptions and units for columns in the spreadsheet of 435 cirque-like alcoves.

| Parameter | Description | Unit |
|---|---|---|
| OBJECTID | Unique ID for each cirque-like alcove entry | - |
| lda_lvf_ov | Overlap with LDA or LVF of Levy et al. (2014), 1 = yes, 0 = no. | - |
| Lat | Longitude coordinate in decimal degrees | o |
| Long | Latitude coordinate in decimal degrees | o |
| Type | Classified type (all 1 for cirque-like alcoves) | - |

| | | |
|---|---|---|
| L | Length of median axis of cirque-like alcove | m |
| W | Maximum, at right angles to axis, through cirque centroid | m |
| L/W | Length over width ratio of cirque-like alcove | - |
| Perimeter | Perimeter of cirque-like alcove | km |
| A_2D | 2D (map) area as defined in Li et al. (2024) | km$^2$ |
| A_3D | 3D (surface) area as defined in Li et al. (2024) | km$^2$ |
| Circular | Circularity index, defined in Li et al. (2024) | - |
| DEMresolut | DEM resolution | m |
| Easting | Easting (x coordinate) of the cirque centroid point | km |
| Northing | Northing (y coordinate) of the cirque centroid point | km |
| H | Height: Z_max-Z_min | m |
| CS | Cirque-like alcove size: the cubic root of L×W×H | m |
| L_H | L/H ratio | - |
| L_W | L/W ratio | - |
| A3D_A2D | 3D area/2D area ratio | - |
| Slope_mean | Mean slope of cirque-like alcove | o |
| Slope_max | Maximum slope of cirque-like alcove | o |
| Slope_min | Minimum slope of cirque-like alcove | o |
| H_W | H/W ratio | - |
| Slpgt33 | Percentage of area with steep slopes of >33° | - |
| Slplt20 | Percentage of the area with gentle slopes of <20° | - |
| Slp20to33 | Percentage of the area with slopes between 20° and 33° | - |
| Aspectmean | Vector mean aspect of all points within the cirque-like alcove | o |
| Aps_east | Sine value of Aspectmean | - |
| Asp_north | Cosine value of Aspectmean | - |
| Z_min | Minimum elevation cirque-like alcove | m |

| | | |
|---|---|---|
| Z_max | Maximum elevation of cirque-like alcove | m |
| Z_median | Median elevation of cirque-like alcove | m |
| Z_mid | Middle elevation of the cirque-like alcove: $(Z\_max+Z\_min)/2$ | m |
| Hypsomax | Highest mode of cirque-like alocve elevations | m |
| HI | Hypsometric integral. | - |
| Prof_clos | Difference between maximum and minimum slope within the cirque-like alcove | o |
| SubCat | Sub-category notes: a-boulders, b-layers in mesa, c-lineations/streaks, d-slumping debris, e-ridge-like features, f-unclear debris, g-glacier-like form nearby/neighbor, h-dunes, i-scalloped terrain, j-bumpy texture terrain, k-paintbrush-like headwall, l-rubbly, m-pasted-on terrain, n-flow features, o-alcove doesn't reach top of mesa, q-crusty texture, p-transverse ridges, r-layered deposits, s-moraines, t-polygonal terrain | - |

Please also note the four points that were raised by the review file validation.
We have reviewed the four points and made the corresponding edits to add appendices and remove a reference. All text is in black. Previously, the supplementary material was the only location to upload a response to the editor (in addition to a reviewer) and that has been removed for this round.
I do not think that these revisions will be onerous, so I look forward to receiving the revised manuscript soon.

Many thanks
Frances Butcher
Associate Editor

---

## Author Response (AR5)

Dear authors,

Thank you for submitting your revised manuscript. I have now assessed your revisions and will be pleased to recommend it proceeds to publication subject to a few technical corrections on the added text in Section 5.3, corrections to the References, and an edit to the Data Availability statement. Please do a full and thorough final proofread before final submission.

L684: capital A mid-sentence. Make lowercase

We didn't find this (line numbers were incorrect), but we did do a final proofread with final edits. The capital A may have been referring to an intentional *A* corresponding to the temperature-dependent ice softness parameter.

L690: 'Given upper timescale limit of 1.1 Gyr, it is likely that cirque-like alcoves had erosion rates higher than 6.7 m/Myr'. 1.1Gyr based on crater counting refers to the minimum age of the surface layer, such that the ice beneath could be much older than this. This timescale 'limit' also assumes that the surface ages are the true ages of the underlying ice, and that cirques were eroded wholly by the presently observed glaciers, which may not be the case. I therefore suggest either clarifying this statement with a clear explanation of the assumptions, or removing it.

We removed this statement.

L636: 'In order to remove the debris, cirque-like alcove erosion has occurred either concurrently or after lobate debris apron formation'. Two issues with the precision of the wording here:
• What do you mean by 'remove the debris'? Do you mean to erode it from the alcove walls? In that case, it would be 'produce the debris'. Or do you mean to evacuate it from the alcove altogether, and onto an LDA surface beyond the alcove? It will be clearer if you state whether you are referring to superposition relationships observed beyond alcoves (i.e. at their outlets) versus on ice deposits within alcoves.

We meant the latter here, so evacuated from the alcove and onto an LDA surface beyond the alcove. We deleted these sentences for clarity since it was redundant with the same information as prior sentences.

• As currently written this statement reads as if you are suggesting primary cirque erosion processes (i.e. basal abrasion, plucking etc) occurred after lobate debris

aprons, to produce supraglacial debris on the LDAs. Again, the robustness of this interpretation this depends on whether the LDAs in question are within the alcoves or beyond them (this makes some sense if the debris is being evacuated beyond the alcove onto a more distant LDA, but seems unnecessarily complex if simply trying to explain supraglacial debris within an alcove itself). It also depends on what you mean by 'cirque like alcove erosion' – are you referring to the formation of the entire alcove via primary cirque-forming processes, or are you referring to secondary modification of cirques via paraglacial mass wasting processes once the support of ice was removed? Distinguishing between the two is important. If simply aiming to explain how debris gets onto the surface of extant ice within the alcoves, paraglacial/mass wasting processes would be the simplest and most likely explanation. In this case, it would be clearer, with something like 'secondary erosion of cirque-like alcoves continued via paraglacial mass wasting of the alcove walls'.

The LDAs are beyond the alcoves, so this is not referring to debris on the surface of extant ice within the alcoves. Whether the cirque-like alcove erosion is primary or secondary that occurs after the ice formation of the LDA is unknown, and either one would in effect act to erode the cirque-like alcoves. We added the following text to address this issue: "Whether this stage of erosion of the cirque-like alcoves is primary or secondary modification (e.g., via paraglacial mass wasting) is unclear."

References: There are issues remaining with the reference list, including the omission of reference information. For example, Cuffey and Patterson 2010 is missing Edition number (4th, I believe) and number of pages, and Evans and Cox 1974 is missing volume information. DOIs are also missing. Please do a thorough check of the reference list before submission to ensure they are both accurate and complete.

Cuffey and Paterson now includes an edition number and number of pages: "Cuffey, K. M., and Paterson, W. S. B. The physics of glaciers. Fourth edition. Amsterdam, Academic Press., 704 pp. ISBN 9780123694614, 2010."

Evans and Cox 1974 was updated as follows: "Evans, I. S., and Cox, N. Geomorphometry and the operational definition of cirques. Area, 6(2), 150-153, 1974."

DOIs were also added for papers published after 1997.

Data availability statement: If the shapefiles will be made available, even if via a repository, a statement needs to be added to say 'The shapefiles for cirque-like alcoves are available at: [URL]'.
We have updated the statement under Data availability to read as follows: "The shapefiles, spreadsheet, and corresponding table of column descriptions for the 434 cirque-like alcoves are available at: https://doi.org/10.5281/zenodo.17527279."

Note that we also deleted what was previously Appendix C and its associated table that contained the column descriptions because they are now available at the same link at the data repository.

I think the authors for their hard work and patience through multiple rounds of revision, and for enhancing our understanding of cirque-like alcoves on Mars. I also extend my thanks to the reviewers for their diligent and insightful reviews.
We thank the editor for dedicating the time to improve this manuscript, and we are looking forward to its publication!

Many thanks
Frances Butcher
Associate Editor